# Adaptive Generalization and Optimization of Three-Layer Neural Networks

**Khashayar Gatmiry**
MIT
gatmiry@mit.edu

**Stefanie Jegelka**
MIT
stefje@mit.edu

**Jonathan Kelner**
MIT
kelner@mit.edu

## Abstract

While there has been substantial recent work studying generalization of neural networks, the ability of deep networks in automating the process of feature extraction still evades a thorough mathematical understanding. As a step toward this goal, we analyze learning and generalization of a three-layer neural network with ReLU activations in a regime that goes beyond the linear approximation of the network and is hence not captured by the common Neural Tangent Kernel. We show that despite nonconvexity of the empirical loss, a variant of SGD converges in polynomially many iterations to a good solution that generalizes. In particular, our generalization bounds are adaptive: they automatically optimize over a family of kernels that includes the Neural Tangent Kernel to provide the tightest bound.

## 1 Introduction

The ability of overparameterized neural networks trained by (stochastic) gradient descent to generalize well on test data (Krizhevsky et al., 2012; Silver et al., 2016; Hinton et al., 2012), even if they perfectly fit the the training data, has intrigued theoretical researchers and led to many approaches for generalization bounds (Neyshabur et al., 2015; Bartlett et al., 2017; Neyshabur et al., 2018; Dziugaite & Roy, 2017; Wei et al., 2019; Golowich et al., 2018; Arora et al., 2018b; Zhou et al., 2018; Konstantinos et al., 2017). This generalization ability is tied to the optimization procedure, i.e., the trajectory of the training algorithm in a non-convex loss landscape, and the structure of the data.

Hence, several recent works study the training of neural networks. For instance, Safran & Shamir (2018) address the role of overparametrization in avoiding bad local minima, and Zhang et al. (2016) empirically show that overparametrized networks trained by SGD can even perfectly fit to random labels. Within the popular framework of the *Neural Tangent Kernel (NTK)* (Jacot et al., 2018), which uses a linear approximation of the network at initialization, several works analyze the optimization trajectory and show global convergence of (S)GD to a global optimum of the empirical loss (Allen-Zhu et al., 2019; Li & Liang, 2018; Zou et al., 2018; Du et al., 2018). Extending the viewpoint to generalization, Arora et al. (2019a;b) exploit the kernel-like behaviour of two-layer networks close to their initialization to prove generalization for the final network, showing that two-layer neural networks generalize as well as Kernel Ridgeless Regression (KRLR) with the NTK. Cao & Gu (2019) show a tighter bound with a *Neural Tangent Random Feature Model*. The kernel approach, however, has two main limitations: First, while KRLR can generalize well in specific high dimensional regimes (Liang et al., 2020), there is theoretical and empirical evidence that it can be inconsistent with noise (Rakhlin & Zhai, 2019). *Is there an approach for analyzing neural networks that shows they perform at least as well as KRLR, but is also robust to noise?*

Second, importantly, neural networks are known to outperform traditional statistical methods in many regimes as they are able to automate the process of feature extraction from data, as opposed to kernel methods that work with a fixed feature representation. This poses the question of other, adaptive, regimes beyond the linear network approximation. In this realm, Wu et al. (2018) show generalization bounds that, instead of the NTK norm, scale with respect to another functional norm. This norm corresponds to the minimum RKHS norm of the function among a family of kernels, i.e., their method in a sense picks the best kernel in this family. However, this result ignores the computational aspect of the problem. *Are there particular nonlinear regimes beyond NTK for which a gradient-type polynomial-time algorithm, in a way, adaptively chooses a suitable kernel?*

Going beyond the NTK view, a line of work convexifies the optimization problem via an approximation of SGD dynamics with a continuous time gradient flow in the space of probability measures on the hidden units of the network, equipped with the Wasserstein metric (Mei et al., 2018; Chizat & Bach, 2018; Mei et al., 2019; Wei et al., 2018; Sirignano & Spiliopoulos, 2020; Javanmard et al., 2019; Lu et al., 2020). Taking another perspective, Allen-Zhu et al. (2018) consider a three-layer network model that is not captured by the NTK approximation, and learn an underlying concept class by exploiting saddle-point escape theory for nonconvex SGD (Ge et al., 2015a). However, evaluating the complexity measure of Allen-Zhu et al. (2018) is rather involved, and only aligns well with functions that are described by a particular network. Whether one can recover the NTK bound (e.g. the NTK norm) from these results is not clear. For the NTK setting, in contrast, Arora et al. (2019a) develop a purely data dependent generalization bound. *Going beyond two layers, is it possible to prove a **data-dependent** complexity measure beyond the NTK regime that recovers the NTK result (Arora et al., 2019a) as a special case?*

In this work, we address the above questions:

- We consider a regime for 3-layer neural networks that is not captured by the NTK approximation and show that, despite nonconvexity, a variant of projected SGD finds a good solution, as measured by the regularized empirical loss, importantly, after polynomially many iterations.

- We introduce a new function norm $\|.\|_\varsigma$ as the minimum RKHS norm with respect to a family of kernels $\mathcal{K}$, which is upper bounded by the NTK norm up to constants. We show that for an arbitrary function $f$, the generalization gap of the trained network scales by $\|f\|_\varsigma$. This makes our generalization bound adaptive, in the sense that it scales with the best kernel in $\mathcal{K}$. As a byproduct, our bounds are comparable with kernel regression bounds simultaneously with all kernels in $\mathcal{K}$. We hope that our techniques motivate researchers to prove such adaptive generalization bounds for deeper networks, which can potentially result in stronger depth separation.

- We show generalization bounds with a new data-dependent complexity measure that generalizes the NTK-based complexity in (Arora et al., 2019a). Up to logarithmic factors, our bounds are upper bounded by those NTK-based bounds and hence improve over them (if one substitutes their Lipschitz loss with a smooth one) – see Appendix A.1 for a simple explicit example. Importantly, our bound can also handle noisy distributions as opposed to (Arora et al., 2019a).

**Further Related work.** While the idea of a learning algorithm that combines multiple kernels has been employed for a while in the community (Sonnenburg et al., 2006; Rakotomamonjy et al., 2007; Duan et al., 2012), our understanding of the connections between deep learning and multiple kernel learning is yet in its infancy. Recently, Dou & Liang (2020) define a time-varying kernel based on the network weights and show that the limit of the gradient flow converges to a suitable dynamic kernel, in the sense that the residual of the link function onto its RKHS could be in a smaller ranked space compared to the orthogonal complement of the RKHS. Ghorbani et al. (2019) analyze the difference between training a two layer ReLU network and its NTK or random feature simplifications, for a mixture of Gaussians input distribution and quadratic target functions. Ignoring the computational hardness imposed by nonconvexity, Bach (2017) prove a dimension dependent generalization bound beyond NTK. In another line of work, Chizat & Bach (2020) study gradient flow on losses with exponential tail and its relation to the max margin solution. Wei et al. (2019) show an interesting separation between the learning power of two layer ReLU networks and their NTK approximation, by showing a sample complexity gap for an artificially constructed distribution.

With a different approach, Allen-Zhu & Li (2020) analyze multi-layer networks with quadratic activations, and prove generalization bounds polynomial in the dimension and precision by assuming an underlying teacher network, which shows a remarkable algorithmic depth separation. The problem of depth separation for neural networks and more generally their expressive power has been investigated by several researchers before (Raghu et al., 2017; Daniely, 2017; Barron, 1994; Funahashi, 1989; Safran & Shamir, 2016; Safran et al., 2019). The assumption of an underlying teacher network that one seeks to recover is common, too (Li & Yuan, 2017; Zhong et al., 2017; Brutzkus & Globerson, 2017). Other works focus mainly on the algorithm and use other techniques, such as tensor factorization, to find a global optimum (Tian, 2016; Bakshi et al., 2019; Janzamin et al., 2015; Zhong et al., 2017). Finally, many authors study the loss landscape under various assumptions (Freeman & Bruna, 2016; Nguyen & Hein, 2017; Soudry & Carmon, 2016; Soltanolkotabi et al., 2018; Ge et al., 2017), some of them consider the simplified case of deep linear networks (Arora et al., 2018a; Saxe et al., 2013; Bartlett et al., 2018; Kawaguchi, 2016).

## 2    SETUP AND APPROXIMATION BY KERNELS

We analyze a 3-layer ReLu neural network from inputs $x \in \mathbb{R}^d$ to outputs $y \in \mathbb{R}$ of the form

$$f_{V',W'}(x) = \frac{1}{\sqrt{m_2}} a^T \sigma\Big((V^{(0)} + V')W^s \frac{1}{\sqrt{m_1}} \sigma((W^{(0)} + W')x)\Big), \tag{1}$$

where $a \in \mathbb{R}^{m_2}$ is a vector of random signs, $V^{(0)} \in \mathbb{R}^{m_2 \times m_3}$ and $W^{(0)} \in \mathbb{R}^{m_1 \times d}$ are random weight initializations with i.i.d Gaussian entries $V_{j,k}^{(0)} \sim \mathcal{N}(0, \kappa_2^2), W_{j,k}^{(0)} \sim \mathcal{N}(0, \kappa_1^2)$, and $W^s \in \mathbb{R}^{m_3 \times m_1}$ is a random sign matrix, which is roughly a random projection and change of coordinates into a lower dimensional space. We refer to $W^s \frac{1}{\sqrt{m_1}} \sigma((W^{(0)} + W')x)$ as the first layer and $\frac{1}{\sqrt{m_2}} a^T \sigma((V^{(0)} + V')(.))$ as the second layer. The algorithm trains weight matrices $V'$ and $W'$, and $W^s, a$ are fixed. We assume that the outputs are a.s. bounded by a constant, $|y_i| \leq B$, and $\|x_i\| = 1$. As loss $\ell(.,.)$, we use the squared loss. We denote the training (empirical) loss of a function $f$ on our data $\{x_i\}_{i=1}^n, \{y_i\}_{i=1}^n$ and the expected loss with respect to the data distribution (population loss) by

$$R_n(f) = \frac{1}{n} \sum_{i=1}^n \ell(f(x_i), y_i), \quad \text{and} \quad R(f) = \mathbb{E}\ell(f(x), y),$$

respectively. Sometimes, we refer to the vector of labels $(y_i)_{i=1}^n$ by $y$. Finally, $\mathcal{H}_K$ is the space of functions with bounded RKHS-norm of kernel $K$, and the notation $\tilde{O}$ hides log factors.

### 2.1    KERNEL APPROXIMATIONS, DECOMPOSITION AND ADAPTIVITY

Kernel approximations of neural networks play an important role in our analysis. First, a common approximation is the NTK. The Neural Tangent kernel for a 2-layer ReLu network is

$$H^\infty(x_1, x_2) = \langle x_1, x_2 \rangle \cdot F_2\Big(\langle x_1, x_2 \rangle / (\|x_1\|\|x_2\|)\Big), \quad \text{for } F_2(x) = \tfrac{1}{4} + \arcsin(x)/(2\pi). \tag{2}$$

To introduce adaptivity, a key part of our analysis is to approximate the second layer in the 3-layer network by a product kernel $K^\infty \odot G$ that decomposes into a "fixed" part $K^\infty$ and an "adaptive" part $G$. To define these kernels, for every $i \in [n]$, let $\phi^{(0)}(x_i)$ be the output of the first layer of the network at initialization, $\phi^{(0)}(x_i) = \frac{1}{\sqrt{m_1}} W^s \sigma(W^{(0)} x_i)$, and $\phi^{(0)}(x_i) + \phi'(x_i)$ be that output for weights $W^{(0)} + W'$. The adaptive kernel $G$ captures the dot product between the learned weights:

$$G(x_i, x_j) = \langle \phi'(x_i), \phi'(x_j) \rangle. \tag{3}$$

This form of $G$ motivates the complexity measure we define in the next section, if one thinks of the entries of $\phi'$ as bounded NTK-norm functions of the input. Next, we consider the second layer, where the part $K^\infty$ arises from roughly stable activations. To formalize this stability, let $\text{Sgn}(Vx)$ be the diagonal matrix whose diagonal contains the coordinate-wise signs of the vector $Vx$. If we assume that $\text{Sgn}\big((V^{(0)} + V')(\phi^{(0)}(x_i) + \phi'(x_i))\big) \approx \text{Sgn}\big(V^{(0)}\phi^{(0)}(x_i)\big)$ – we prove a rigorous statement in Appendix A.12 – then

$$f_{W',V'}(x_i) = \frac{1}{\sqrt{m_2}} a^T (V^{(0)} + V')(\phi^{(0)}(x_i) + \phi'(x_i)) \tag{4}$$

$$\approx \Big\langle V^{(0)} + V', \frac{1}{\sqrt{m_2}} a^T \text{Sgn}\big(V^{(0)}\phi^{(0)}(x_i)\big)\big(\phi^{(0)}(x_i) + \phi'(x_i)\big)^T \Big\rangle. \tag{5}$$

Focusing on the adaptive part $\phi'(x)$ of the first layer, we write

$$\Big\langle V^{(0)} + V', \frac{1}{\sqrt{m_2}} a^T \text{Sgn}\big(V^{(0)}\phi^{(0)}(x_i)\big)\phi'(x_i)^T \Big\rangle := \Big\langle V^{(0)} + V', \Upsilon(x_i) \Big\rangle \tag{6}$$

and can then view the second layer as a function in the RKHS of the product kernel $\langle \Upsilon(x_i), \Upsilon(x_j) \rangle =: \tilde{K}^\infty(x_i, x_j)G(x_i, x_j)$. This defines the new kernel $\tilde{K}^\infty$, which we simplify into the kernel $K^\infty$ that is independent of initialization ($K^\infty$ is defined in Equation (8)). To do so, we first observe that $\tilde{K}^\infty$ concentrates around

$$\mathbb{E}_{w \sim \mathcal{N}(0, \kappa_2^2 I)} \mathbf{1}\{w^T \phi^{(0)}(x_i)\} \mathbf{1}\{w^T \phi^{(0)}(x_j)\} = F_2\Big(\frac{\langle \phi^{(0)}(x_i), \phi^{(0)}(x_j) \rangle}{\|\phi^{(0)}(x_i)\|\|\phi^{(0)}(x_j)\|}\Big).$$

Moreover, the Gaussian initialization and assumption $\|x_i\| = 1$ imply that $\langle \phi^{(0)}(x_i), \phi^{(0)}(x_j) \rangle$ concentrates around $m_3 F_3(\langle x_i, x_j \rangle)$, for $F_3 : [-1, 1] \to [0, \frac{1}{2}]$ defined as

$$F_3(x) = \frac{1}{2\pi}\sqrt{1 - x^2} + \frac{1}{4}x + \frac{1}{2\pi}x \arcsin(x), \tag{7}$$

$$\text{so} \quad \tilde{K}^\infty(x_i, x_j) \approx F_2(2F_3(\langle x_i, x_j \rangle)) =: K^\infty(x_i, x_j). \tag{8}$$

For general $x_1, x_2$ not necessarily unit norm, we define $K^\infty(x_1, x_2) = K^\infty(x_1/\|x_1\|, x_2/\|x_2\|)$.

It is easy to check that the coefficients in the Taylor series of $F_2$ and $F_3$ are nonnegative. Combining this with Schur's Product Theorem implies $K^\infty$ is PSD (Appendix A.4). We also denote the data kernel matrix on $(x_i)_{i=1}^n$ by $K^\infty$ and $H^\infty$. Like Arora et al. (2019a), we assume the data distribution is $(\lambda_0, \delta, n)$-non-degenerate with respect to $H^\infty$ and $K^\infty$, i.e., with probability at least $1 - \delta$, the smallest eigenvalues of $H^\infty$ and $K^\infty$ are at least $\lambda_0 > 0$.

## 3 DATA DEPENDENT COMPLEXITY MEASURE AND GENERALIZATION

The emergence of $G \odot K^\infty$ above gives rise to an *adaptive* kernel-like complexity measure that will determine generalization bounds. Intuitively, this complexity measure reflects the two layers. Here, we view $G$ as the Gram matrix of some "ideal" first-layer feature functions $g_k$. We measure the complexity of the prediction function via the RKHS of $A := G \odot K^\infty$, and allow a flexible choice of the features $g_k$. The $g_k$ may be viewed as feature representation of $\phi'$ in Equation (3): $G(x_i, x_j) := \sum g_k(x_i)g_k(x_j)$. To regularize this choice, we penalize the complexity of the features $g_k$ via the NTK norm. Alternatively, the features $g_k$ are flexible but have bounded NTK norm.

For a labeling $f^* \in \mathbb{R}^n$ of the $n$ data points and fixed $G$, this leads to the complexity

$$\zeta(f^*, G) = f^{*T}A^{-1}f^* \cdot \langle H^{\infty-1}, G \rangle = f^{*T}A^{-1}f^* \sum_{k=1}^{m_3} \|g_k\|_{H^\infty}^2; \quad A = G \odot K^\infty, \tag{9}$$

where $m_3$ is the number of intermediate features. The choice of $m_3$ is discussed in Appendix A.13. Our data-dependent complexity measure implicitly selects the $G$ (or equivalently the feature vectors $g_k$) that leads to the tightest bound, trading off data fit and function complexity:

$$\Im = \Im((x_i)_{i=1}^n, (y_i)_{i=1}^n) := \min_{f^* \in \mathbb{R}^n} \left\{ 2nR_n(f^*) + \varpi \min_{G \succ 0} \zeta(f^*, G) \right\}, \tag{10}$$

where we use a log factor $\varpi = O(\log(n)^3 + \log(1/\lambda_0))$.

To make the relation to adaptive kernel spaces even more explicit, assume that the $g_k$ are bounded as $\sum_k \|g_k\|_{H^\infty}^2 \le 1$. Then we define a family $\mathcal{K}$ of corresponding kernels of the form

$$K_{\{g\}}(x_1, x_2) = K^\infty(x_1, x_2)\left(\sum_{g \in \{g\}} g(x_1)g(x_2)\right), \tag{11}$$

i.e., $\mathcal{K} := \{K_{\{g\}} | \{g\} \text{ finite}, \sum_{g \in \{g\}} \|g\|_{H^\infty}^2 \le 1\}$. With this notation, the complexity measure is

$$\Im((x_i), (y_i)) = \min_{f^* \in \mathbb{R}^n} \left\{ 2nR_n(f^*) + \varpi \min_{K \in \mathcal{K}} f^{*T}K^{-1}f^* \right\}. \tag{12}$$

Hence, this measure may be understood as searching for the most efficient and effective feature representation within a family of RKHSs.

We may also relate this complexity measure to the NTK-based complexity measure $y^T H^{\infty-1} y$ (Arora et al., 2019a). For any labeling $f^*$, let $\tilde{f}^* \in \mathcal{H}_{H^\infty}$ be the function with minimum NTK norm that maps $x_i$'s to $f_i^*$'s, so $\|\tilde{f}^*\|_{H^\infty} = f^{*T}H^{\infty-1}f^*$. If we set $\{g\} = \{\tilde{f}^*/\|\tilde{f}^*\|\}$, then one can show (Appendix A.5)

$$f^{*T}K_{\{\tilde{f}^*/\|\tilde{f}^*\|\}}^{-1}f^* \le 4f^{*T}(f^*f^{*T})^{-1}f^* \times \|\tilde{f}^*\|^2 = 4\|\tilde{f}^*\|^2 = 4f^{*T}H^{\infty-1}f^*, \tag{13}$$

which implies

$$\Im \le \min_{f^* \in \mathbb{R}^n} \left\{ 2nR_n(f^*) + (4\varpi)f^{*T}H^{\infty-1}f^* \right\}.$$

One can further set $f^* = y$ above and obtain

$$\Im \le (4\varpi_1)y^T H^{\infty-1} y. \tag{14}$$

## 3.1 GENERALIZATION

With the complexity $\Im$ in hand, we can now state our generalization result. It assumes optimization by a Projected Stochastic Gradient Descent (PSGD), described in detail in Section 4.

**Theorem 1** *Suppose we run Projected Stochastic Gradient Descent (PSGD) on the regularized empirical risk with parameters as in Section 4, and $|y_i| \leq B$ a.s.. Then, with high probability (e.g. 0.99) over the randomness of data, initialization and noise of the gradient steps, PSGD converges in $poly(B \vee 1/B, 1/\lambda_0, n)$ iterations to a solution $(W_{PSGD}, V_{PSGD})$ with population risk bounded as*

$$R(f_{W_{PSGD}, V_{PSGD}}) \leq \frac{\Im((x_i)_{i=1}^n, (y_i)_{i=1}^n)}{n} + \frac{B^2 \varpi}{n}. \tag{15}$$

As a side remark, the factor 2 in front of $R_n(f^*)$ in the definition of $\Im$ in (12) is not special and a similar generalization bound can be obtained for any $\gamma > 1$. Substituting the upper bound on the complexity in Equation (14), one recovers an NTK-based generalization bound that scales with $y^T {H^\infty}^{-1} y/n$ up to log factors, which is roughly the square of the generalization bound presented in (Arora et al., 2019a). The reason for the faster squared rate here is that we are considering smooth losses, while they work with a bounded Lipschitz loss. Indeed, it is not hard to apply a more rigorous uniform convergence analysis from (Srebro et al., 2010) to also obtain a faster squared rate for the approach used in (Arora et al., 2019a).

Since Equation (14) is an upper bound on our complexity, our result generalizes and tightens the NTK bound (Arora et al., 2019a). To illustrate the flexibility of our complexity measure, we show in Appendix A.1 a simple explicit example of functions represented as polynomial series where our bound improves upon the NTK bound. Notably, we only substitute low-rank matrices $G$ in our complexity measure for this construction. We leave further investigation of our complexity measure for arbitrary $G$'s to future work.

## 3.2 UNDERLYING CONCEPT CLASS

Instead of data dependent generalization bounds, one may study the generalization gap with respect to some concept class. The complexity measure $\Im$ implicitly uses the following adaptive norm on the space of functions from $\mathbb{R}^d$, the infimum of the RKHS norms for the family of kernels $\mathcal{K}$:

$$\|f\|_\zeta = \inf_{K_{\{g\}} \in \mathcal{K}} \|f\|_{K_{\{g\}}}. \tag{16}$$

It is not hard to check that $\|.\|_\zeta$ is in fact a norm, and that the inf is achieved by a particular set $\{g\}$. Similar to the derivation of the upper bound on the complexity measure in Equation (14), by setting $\{g\} = \{f^*/\|f^*\|_{H^\infty}\}$, we obtain the following NTK upper bound:

$$\|f\|_\zeta \leq 4\|f\|_{H^\infty}. \tag{17}$$

This leads to a function-dependent generalization bound which bounds the risk of the learned network against an arbitrary function $f$ with $\|f\|_\zeta < \infty$.

**Theorem 2** *For any measurable function $f : \mathbb{R}^d \to \mathbb{R}$, in the same setting as Theorem 1, the population risk of the trained network can be bounded as*

$$R(f_{W_{PSGD}, V_{PSGD}}) \leq 2R(f) + O\left(\varpi \frac{\|f\|_\zeta^2 + B^2}{n}\right). \tag{18}$$

As in the data-dependent case, the factor 2 on $R(f)$ can be reduced to any constant $\gamma > 1$.

## 3.3 INTERACTION OF LAYERS BEYOND THE LINEAR APPROXIMATION

Here, we give a high level intuition on how the adaptivity is achieved in our regime compared to NTK. In the NTK approach, for every input $x$, the neural net $f_{W,V}(x)$ is approximated by its linear approximation at $(W^{(0)}, V^{(0)})$ (the initialized network), $f_{W,V}(x) = \langle \nabla_{W,V} f_{W^{(0)}, V^{(0)}}(x), (W - W^{(0)}, V - V^{(0)}) \rangle$. The NTK approximation works as long as $(W, V)$ are close enough to their

initialization that the linear approximation remains accurate and the interaction of weights between layers is negligible. Specifically, the features $\phi'(x_i)$ behave almost linearly with respect to $W - W^{(0)}$ as $\|W - W^{(0)}\|$ is taken to be small and the sign pattern $\text{Sgn}\big((W^{(0)} + W')x_i\big)$ is proven not to change much compared to $\text{Sgn}\big(W^{(0)}x_i\big)$. Additionally, the NTK-type analysis needs the following two conditions to be satisfied: (1) the sign pattern of $\phi^{(0)}(x_i) + \phi'(x_i)$ with respect to $V^{(0)} + V'$ remains almost the same as the sign pattern of $\phi^{(0)}(x)$ with respect to $V^{(0)}$, and (2) the weight changes $W'$ and $V'$ should not interact, which means the "interaction" term, $\frac{1}{\sqrt{m_2}}a^T\text{Sgn}\Big(V^{(0)}\phi^{(0)}(x_i)\Big)V'\phi'(x_i) \approx 0$, should be negligible. Therefore, the non-negligible terms for the NTK are: (1) $\frac{1}{\sqrt{m_2}}a^T\text{Sgn}\Big(V^{(0)}\phi^{(0)}(x_i)\Big)V^{(0)}\phi'(x_i)$, which is almost linear in $W'$ (recall that $\phi'(x_i)$ depends on $W'$), and (2) $\frac{1}{\sqrt{m_2}}a^T\text{Sgn}\Big(V^{(0)}\phi^{(0)}(x_i)\Big)V'\phi^{(0)}(x_i)$ which is linear in $V'$. This approach has two important implications: (1) it convexifies the optimization (for convex loss), as the approximation is now linear in $W$; and (2) it simplifies proving generalization, as it works with the class of functions in the RKHS space of some fixed kernel. However, this simplification leaves no room for the ability of the neural network to learn intermediate feature representations.

In our regime, in contrast, we enforce the condition $\forall j, i: V'_j \perp \phi^{(0)}(x_i)$ $(\star)$, which implies the second (2) above is zero, while the interaction term is not negligible any more and the network behaves similar to a quadratic function with respect to $(W', V')$ (for fixed $x_i$). Condition $(\star)$ is critical both in proving the convergence of the algorithm as well as bounding the Rademacher complexity of the class of networks with bounded weights. Rather than working with a fixed kernel, the interaction term enables us to use the first layer for representing the input in a suitable feature space, which can be interpreted as picking a suitable kernel, then use the second layer to describe the output based on those features. This is also indirectly encoded in our complexity measure. In addition to enforcing the orthogonality condition $(\star)$ (in the SGD variant), conditions for entering our regime are that the overparameterization $m_1, m_2, m_3$ and $\kappa_1, \kappa_2$ are within a specific range with respect to each other. We listed these relations in Appendix A.3.

To illustrate the benefit of going to this more involved regime, denote the class of neural networks with bounded Frobenius norms $\|W'\| \leq \gamma_1, \|V'\| \leq \gamma_2$ by $\mathcal{G}_{\gamma_1,\gamma_2}$ (and a bit more structure which we elaborate upon in the proofs); it turns out that $\mathcal{G}_{\gamma_1,\gamma_2}$ roughly includes $\mathcal{H}_K(O(\gamma_1\gamma_2))$ for every kernel $K \in \mathcal{K}$, in the sense that each $f \in \mathcal{H}_K(O(\gamma_1\gamma_2))$ is well-approximated within $G_{\gamma_1,\gamma_2}$ to arbitrarily small error on fixed input (the error goes down with the size of the network). On the other hand, we show that the Rademacher Complexity (RC) of $\mathcal{G}_{\gamma_1,\gamma_2}$ behaves similar to the RC of the NTK class $\mathcal{H}_{H^\infty}(O(\gamma_1\gamma_2))$! As our algorithm guarantees finding a network with sufficiently small empirical risk within $\mathcal{G}_{\gamma_1,\gamma_2}$, this phenomenon underlies our adaptive generalization bounds.

Compared to previous work that provides an adaptive kernel analysis still for a two layer model (Dou & Liang, 2020) (although their analysis is for the gradient flow and non-algorithmic), our model requires an additional layer so it can, in a sense, "simulate" the process of feature extraction in one layer to be used in the next layer.

### 3.4 COMPARISON WITH KERNEL FITTING

We compare our generalization bounds with some kernel fitting rates. Given a kernel $K$ with $K(x,x) \leq 1$ for every $x: \|x\| \leq 1$, suppose we want to fit a function from $\mathcal{H}_K(B')$, i.e. having $K$-RKHS norm bounded by $B'$. In the realizable setting, when there is an underlying $f^{**} \in \mathcal{H}_K(B')$ with zero risk, Empirical Risk Minimization (ERM) enjoys a fast rate using the smoothness of the loss (Srebro et al., 2010). The Rademacher Complexity bound $\mathfrak{R}(\mathcal{H}_K(B')) \leq O(\frac{B'}{\sqrt{n}})$ then implies

$$R(f^{\text{ERM}}) \leq \tilde{O}(B'^2/n) \tag{19}$$

for the squared loss, which is minimax optimal up to log factors. To compare to the neural network, we substitute $f^{**}$ into Theorem 2. To relate the $B$ in our bound to $B'$, assume for simplicity of exposition that $\|f\|_\zeta = \|f\|_{K^*}$ for some $K^* \in \mathcal{K}$ (otherwise we can use a convergent sequence). Observing that $K_{\{g\}}(x,x) \leq 1$ for every kernel $K_{\{g\}} \in \mathcal{K}$, we obtain that $|f^{**}(x)| \leq \|f^{**}\|_{K^*} = \|f^{**}\|_\zeta$ (Appendix A.6). Combining this fact with the realizability assumption, we can then upper

bound the parameter $B$ in Theorem 2 by $\|f^{**}\|_\zeta$, and obtain

$$R(f_{W_{\text{PSGD}}, V_{\text{PSGD}}}) = \tilde{O}(\|f^{**}\|_\zeta^2/n). \tag{20}$$

If we further take $K$ to be in $\mathcal{K}$, then Equation (20) combined with $\|f^{**}\|_\zeta \leq \|f^{**}\|_K \leq B'$ implies:

$$R(f_{W_{\text{PSGD}}, V_{\text{PSGD}}}) = \tilde{O}(B'^2/n),$$

that is, for every kernel $K \in \mathcal{K}$, our deep learning approach almost achieves the conventional kernel bound in Equation (19).

Repeating the uniform risk bound stated in Theorem 1 in (Srebro et al., 2010) for $\mathcal{H}_K(B')$ where $B'$ is set to all powers of two, followed by a union bound, one can easily obtain a fast rate of

$$R(f^{KRLR}) \leq \tilde{O}\Big(\frac{y^T K^{-1} y}{n} + \frac{B^2}{n}\Big), \tag{21}$$

for the solution of KRLR in the general case (not realizable) for the squared loss. On the other hand, for a $B$-bounded Lipschitz loss, we instead get a slow rate for KRLR:

$$R(f^{KRLR}) \leq \tilde{O}\Big(\sqrt{\frac{y^T K^{-1} y}{n}} + \frac{B}{\sqrt{n}}\Big),$$

where $B$ is an a.s. bound on $|y|$ as before. This bound is similar to Arora et al. (2019a). Note that our data dependent generalization bound in Theorem 1 already achieves the fast rate for KRLR in (21) for any $K \in \mathcal{K}$. Finally, in the non-realizable case, we still have the following fast rate for ERM regarding the hypothesis class $\mathcal{H}_K(B')$ (Srebro et al., 2010):

$$R(f^{ERM}) \leq \tilde{O}(R(f^{**}) + \tfrac{B'^2 + B^2}{n}),$$

where now $f^{**} := \operatorname{argmin}_{f \in \mathcal{H}_K(B')} R(f)$, while Theorem 2 also implies (again for every $K \in \mathcal{K}$):

$$R(f_{W_{\text{PSGD}}, V_{\text{PSGD}}}) \leq \tilde{O}\big(R(f^{**}) + \tfrac{\|f^{**}\|_\zeta^2 + B^2}{n}\big) = \tilde{O}\big(R(f^{**}) + \tfrac{B'^2 + B^2}{n}\big).$$

## 4 ALGORITHM: PROJECTED STOCHASTIC GRADIENT DESCENT

In this section, we describe our algorithm `PSGD`, presented as pseudocode in Figure 1, which is roughly Stochastic Gradient Descent modified to project out a low-dimensional random subspace from the second-layer weights. `PSGD` approximately runs SGD on a smoothed version of the following loss function ($\psi_1, \psi_2$ are defined in Appendix A.2)

$$L_1(W', V') = R_n(f_{W', V'}) + \psi_1 \|W'\|^2 + \psi_2 \|V'\|^2.$$

Compared to standard SGD, our algorithm makes two modifications: (1) it uses randomized smoothing to alleviate the non-smoothness of the ReLUs, (2) it ensures that the weights in the second layer are orthogonal to the data features $\phi^{(0)}(x)$ computed by the first layer at initialization. This helps to control layer interactions as pointed out in Section 3.3. For smoothing, we add Gaussian smoothing matrices $W^\rho$ and $V^\rho$ to the weights with i.i.d. entries drawn from $\mathcal{N}(0, \beta_1^2/m_1)$ and $\mathcal{N}(0, \beta_2^2/m_2)$ respectively, for $\beta_2 = O_p((\kappa_1\sqrt{m_1})^{-1}(\kappa_2\sqrt{m_2})^{-2/3})$, $\beta_1 = O_p(m_3^2 \kappa_2 \sqrt{m_2}(\kappa_1 \sqrt{m_1})^{-1})$. To simplify the exposition, $O_p(.)$ is hiding the dependencies on the basic parameters $B, n, 1/\lambda_0$ and log factors. Our convergence proof uses the loss with respect to this smoothed network.

For the projection, let $\Phi^\perp \subset \mathbb{R}^{m_2 \times m_3}$ be the subspace of weights of the second layer whose rows are orthogonal to the first-layer data representations $\phi^{(0)}(x_i)$'s $\forall i \in [n]$ at initialization:

$$V' \in \Phi^\perp \leftrightarrow \forall j \in [m_2], \ \forall i \in [n]: \ V'_j, \phi^{(0)}(x_i) = 0. \tag{22}$$

In summary, at point $(W', V')$, the algorithm samples a random $(x_i, y_i)$ from the data, as well as smoothing matrices $W^{\rho,1}, V^{\rho,1}, W^{\rho,2}, V^{\rho,2}$. It then computes an unbiased estimate for the gradient $(\hat{\nabla}_W, \hat{\nabla}_V)$, adds additional normalized Gaussian noise matrices $\Xi_1, \Xi_2$ and moves in this direction with step size $\eta = 1/\operatorname{poly}(n, B \vee 1/B, 1/\lambda_0)$:

$$(W', V') \leftarrow (W', V') + \eta\Big(\hat{\nabla}_W + \Xi_1/(\sqrt{m_1}\|\Xi_1\|), \ \texttt{Proj}_{\Phi^\perp}(\hat{\nabla}_V + \Xi_2/\|\Xi_2\|)\Big). \tag{23}$$

**Parameters.** Our results apply to the overparameterized regime, when the size of the network, i.e. parameters $m_1, m_2, m_3$ are polynomially large in $n, B \vee 1/B, 1/\lambda_0$. This guarantees that the network has suitable function representation capacity, and PSGD is able to find a good local direction at every iteration. The regularization coefficients $\psi_1, \psi_2$ can be set with respect to any candidate $(f^*, G)$ for our complexity measure (9). In Appendix A.2, we introduce a simple doubling trick that handles the case when we do not have access to an optimal candidate solution. With such an $f^*$, as we describe in Remark 1, define $\nu := \max\{R_n(\bar{f}^*)/2, B^2/n\}$, and set $\psi_1 = \nu/4$, $\psi_2 = \nu/(4\zeta(\bar{f}^*, G))$, where $\bar{f}^*$ is the projection of $f^*$ along the span of eigenvectors of $A$ with eigenvalue as large as $\Omega(1/n^2)$. We list the suitable regime for overparameterization in Appendix A.3.

---

**Algorithm 1** PSGD(Projected Stochastic Gradient Descent)

---

**Input:** network architecture $m_1, m_2, m_3$, initialization parameters $\kappa_1, \kappa_2$, smoothing parameters $\beta_1, \beta_2$, training set $(x_i, y_i)_{i=1}^n$, label parameter $B$, $(f^*, G)$ from the complexity measure

1: Gaussian initialization $W_{j,k}^{(0)} \leftarrow \mathcal{N}(0, \kappa_1)$, $V_{j,k}^{(0)} \leftarrow \mathcal{N}(0, \kappa_2)$
2: Define parameters $\psi_1, \psi_2, \nu, \eta$, subspace $\Phi^{\perp}$, and objective $L_1$ as described in Section 4
3: **while** $L_1(W', V') > R_n(f^*) + 2\nu$ **do**
4:     Gaussian matrices $W_{j,k}^{\rho,1}, W_{j,k}^{\rho,2} \leftarrow \mathcal{N}(0, \frac{\beta_1^2}{m_1})$, $V_{j,k}^{\rho,1}, V_{j,k}^{\rho,2} \leftarrow N(0, \frac{\beta_2^2}{m_2})$
5:     Sample data $(x_i, y_i)$ uniformly at random
6:     Compute gradient estimates
7:     $\begin{cases} \hat{\nabla}_W = \dot{\ell}(f_{W'+W^{\rho,1}, V'+V^{\rho,1}}(x_i), y_i) \nabla_W f_{W'+W^{\rho,2}, V'+V^{\rho,2}}(x_i) + 2\psi_1 W', \\ \hat{\nabla}_V = \dot{\ell}(f_{W'+W^{\rho,1}, V'+V^{\rho,1}}(x_i), y_i) \nabla_V f_{W'+W^{\rho,2}, V'+V^{\rho,2}}(x_i) + 2\psi_2 V' \end{cases}$
8:     Move as $(W', V') \leftarrow (W', V') + \eta\left(\hat{\nabla}_W + \Xi_1/(\sqrt{m_1}\|\Xi_1\|), \texttt{Proj}_{\Phi^{\perp}}(\hat{\nabla}_V + \Xi_2/\|\Xi_2\|)\right)$
9: **Return** $(W', V')$

---

## 5 HIGH LEVEL IDEA OF THE PSGD ANALYSIS

The reason for considering a Frobenius norm regularizer in PSGD is that we want the weights to remain close to their initialization so the final network is in the class $\mathcal{G}_{\gamma_1, \gamma_2}$ for suitably chosen $\gamma_1, \gamma_2$; while still reducing the nonconvex empirical loss $R_n(f_{W', V'})$. We prove convergence for PSGD by building on ideas from Allen-Zhu et al. (2018), with a framework based on the classic result that SGD can escape saddle points for nonconvex functions. Compared to them, we take a different approach driven by our purely data-dependent complexity measure. We augment this by a careful Rademacher complexity analysis of the class $\mathcal{G}_{\gamma_1, \gamma_2}$ in Appendix A.11.

**Construction of a good Network** To study the loss landscape, similar to (Allen-Zhu et al., 2018), we show the existence of a good local update at reasonable points $(W', V')$, using the ideal pair $(W^*, V^*)$ that we carefully construct from our complexity measure. Here, we sketch our proof for constructing $(W^*, V^*)$. Let $(W', V')$ be the current weights of the algorithm. Fix a sample $i \in [n]$. In Appendix A.12, we use $G$ to construct $W^*$ for the first layer weights with decomposition $W^* = \sum_{k=1}^{m_3} W_k^*$ and $O(1)$ bounded norm, such that $\phi^*(x_i)_k := \frac{1}{\sqrt{m_1}} W^s \text{Sgn}\left((W^{(0)} + W')x_i\right) W^* x_i$. This decomposition ensures for every $k, k' \in [m_3]$, negating $W_k^*$ only negates $\phi^*(x_i)_{k'}$ when $k' = k$ and has no effect on $\phi^*(x_i)_{k'}$ for $k' \neq k$. This way, we can easily generate any arbitrary sign flip of the entries of $\phi^*(x_i)$. We use this property to generate a suitable random descent direction.

Next, we construct a suitable weight matrix $V^*$ for the second layer which maps the features $\phi^*(x_i)$ into $f_i^*$ (recall the definition of the complexity measure). The key here is that we consider a regime where the norm of $\phi^{(0)}(x_i)$ is typically larger than that of $\phi'(x_i)$ and $\phi^*(x_i)$, so it is very likely that the sign pattern in the second layer is determined by $\phi^{(0)}(x_i)$ in most rows. In such a scenario, the condition $V_j' \perp \phi^{(0)}(x_i)$ becomes vital as the interaction of $V'$ with $\phi^{(0)}(x_i)$ is problematic for both generalization and optimization. From the standpoint of generalization, without excluding this interaction, one can exploit the large size of $\phi^{(0)}(x_i)$ and build a network within the class $\mathcal{G}_{\gamma_1, \gamma_2}$ corresponding to a complex function that overfits the data. Indeed, we utilize the large magnitude of $\phi^{(0)}$ and its orthogonality to the rows of $V'$ in the RC bound. On the other hand, since the weights

of the first layer does not affect $\phi^{(0)}(x_i)$, the interaction of $V'$ and $\phi^{(0)}(x_i)$ is problematic for the algorithm's convergence, particularly in proving the existence of a local descent direction. This is the rationale behind our orthogonality constraint (22).

Finally, the $\phi^*(x_i)$'s, the above control on the signs, and the fact that $\langle\phi^{(0)}(x_{i_1}),\phi^{(0)}(x_{i_2})\rangle$ concentrates around $m_3\mathbb{E}_{w\sim N(0,\kappa_1 I)}[\sigma(w^T x_{i_1})\sigma(w^T x_{i_2})]$ which recovers the structure of the kernel $K^\infty$ (Section 2), give rise to the kernel $G\odot K^\infty$ in the second layer. Using this structure, we construct $V^*$ that maps $\phi^*(x_i)$'s to $f_i^*$'s, which has additional good properties, including $O(f^{*T}(G\odot K^\infty)^{-1}f^*)$-bounded norm, and rows that are orthogonal to $\phi^{(0)}(x_i)$'s. For more details, see Appendix A.12.

**Nonexistence of Bad Saddle Points** Next, we want to exploit $(W^*,V^*)$ to prove the existence of a good direction along which the objective decreases locally. Moving along $(W^*,V^*)$ is the first idea, which fails as the cross terms created between $W',V^*$ and $V',W^*$ cannot be bounded effectively. Instead, we randomly perturb $W^*$ and $V^*$ in a coupled way and prove a reduction in expectation. We elaborate more on this suitable random direction. Multiplying random signs $\Sigma_k$ onto $W_k^*$, we define the sum $W_\Sigma^* = \sum_{k=1}^{m_3}\Sigma_k W_k^*$. We also multiply the same signs to the columns of $V^*$ and project it back onto $\Phi^\perp$ to obtain $V_\Sigma^*$. Then, we move in the random direction $(\sqrt{\eta}W_\Sigma^* - \eta W/2, \sqrt{\eta}V_\Sigma^* - \eta V/2)$; this update creates additional cross terms in the objective that we must bound to prove a local reduction argument. A key point here is that we prove with high probability the norm of the weights is always bounded. This norm restriction enables us to substitute terms that we do not have control over by their worst-case supremum. We refer to Appendix A.13 for similar techniques.

**Convergence of `PSGD`** Finally, we use the fact that SGD escapes good saddle points (Ge et al., 2015b). For proving the existence of a good random direction to escape saddle points above, we use that the norm of weights is uniformly bounded along all iterations; this bound, in fact, is looser than the bound that we show for the final weights of the network. Yet, this additional restriction cannot be addressed by the classical nonconvex theory of SGD. Consequently, we refine and adapt the proof of (Ge et al., 2015b) to incorporate this additional constraint. At a high level, Ge et al. (2015b) work with a supermartingale based on the loss value. To guarantee the additional norm restriction, it is initially tempting to apply Azuma-Hoeffding concentration to bound the upward deviations of this process. However, this fails as the process has a two-fold behavior, depending on how large the gradient is. At the core of our refinement proof here, we instead directly bound the MGF of the martingale using Doobs maximal inequality. We refer to Section A.16 for more details.

ACKNOWLEDGMENTS

The work was supported in part by NSF awards SCALE MoDL 2134108, CCF-2112665 (TILOS AI Research Institute), CCF-1955217, CCF-1565235, and DMS-2022448. We would like to also thank Sasha Rakhlin and Elchanan Mossel for fruitful discussions.

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

# A   APPENDIX

This appendix contains the different main sections of the proof. Lower-level lemmas may be found in Appendix B.

## CONTENTS

A.1 STRONGER GENERALIZATION BOUNDS FOR POLYNOMIALS

In this section, we prove an explicit generalization bound for functions represented as a polynomial sum. Note that the bounds in Arora et al. (2019a) for polynomials assume the monomials with degree larger than one to have even powers, while here we do not impose this restriction. In addition, different from Arora et al. (2019a), our bounds remain meaningful in the noisy case (recall our Theorem 2).

More specifically, we bound the $\zeta$ norm of such functions. Consider the target function $s$ with the following power series formula:

$$y = s(x) = \sum_{p=1}^{\infty} a_p (w_p^T x)^p, \tag{24}$$

where $a_p \in \mathbb{R}$ and $w_p \in \mathbb{R}^d$. We can write

$$s(x) = g_1(x) + \sum_{k=1}^{d} x_k g_2^k(x), \tag{25}$$

where $x_k$ denotes the $k$th entry of vector $x$ here and

$$g_1(x) = \sum_{p \in A_1 := \{p=1 \text{ or } p \text{ even}\}} a_p (w_p^T x)^p,$$

and for all $k \in [d]$:

$$g_2^k(x) = \sum_{p \in A_2 := \{p>2, \ p \text{ odd}\}} w_{p_k} a_p (w_p^T x)^{p-1}.$$

Then, using the Taylor series of $x(\frac{1}{4} + \frac{\arcsin(x)}{2\pi}) = \sum_{p=1}^{\infty} \gamma_p x^p$ for $|x| \le 1$, the RKHS $\mathcal{H}(H^\infty)$ of the NTK can be identified by square-summable sequences of reals $(a_{p'})_{p'=1}^{\infty}$ with dot product

$$\langle (a_{p'})_{p'=1}^{\infty}, (b_{p'})_{p'=1}^{\infty} \rangle = \sum_{p'=1}^{\infty} \gamma_{\lambda(p')} a_{p'} b_{p'},$$

where $\lambda(p') : \mathbb{Z}_{\ge 0} \to \mathbb{Z}_{\ge 0}$ such that it maps zero to zero, the first $d$ positive integers are mapped to one, the next $d^2$ ones are mapped to 2, etc. Moreover, the RKHS mapping $\Psi : \mathbb{R}^d \to \mathcal{H}(H^\infty)$ from the Euclidean space is:

$$\Psi(x) = \|x\| \left( x'_1, ..., x'_d, (x'_{k_1} x'_{k_2})_{k_1, k_2 \in [d]}, \ldots, (x'_{k_1} x'_{k_2} \ldots x'_{k_p})_{k_1, \ldots, k_p \in [d]}, \cdots \right),$$

where $x' = x/\|x\|$ and in the notation above we are presenting a sequence of sequences, by which we mean the inner sequences simply unfold. Using this identification, one can see using the linear representations of $g_1, g_2^k$ in $\mathcal{H}$:

$$\|g_1\|_{H^\infty}^2 = \sum_{p \in A_1} \gamma_p a_p^2 \|w_p\|_2^{2p}, \tag{26}$$

$$\|g_2^k\|_{H^\infty}^2 = \sum_{p \in A_2} \gamma_p w_{p_k}^2 a_p^2 \|w_p\|_2^{2(p-1)}. \tag{27}$$

Summing above and noting the linear representation of $g$:

$$\|g_1\|_{H^\infty}^2 + \sum_{k=1}^{d} \|g_2^k\|_{H^\infty}^2 = \sum_{p \in A_1} \gamma_p a_p^2 \|w_p\|_2^{2p} + \sum_{p \in A_2} \gamma_p a_p^2 \|w_p\|_2^{2p} = \sum_{p=1}^{\infty} \gamma_p a_p^2 \|w_p\|_2^{2p} = \|g\|_{H^\infty}^2. \tag{28}$$

Now for $\{g\} := \{g'_k\}_{k=1}^{d+1} := \{g_1\} \cup \{g_2^k\}_{k=1}^{d}$, we consider the kernel $K_{\{g\}}$. Expanding the tailor series of $F_2(2F_3)(x) = \sum_{p=0}^{\infty} \mu_p x^p$, we find the identification $(h_{p'}^k)_{k \in [d+1], p'=0, \ldots, \infty}$ with dot product

$$\sum_{p'=0}^{\infty} \mu_{\lambda(p')} \sum_{k=1}^{d+1} h_{p'}^k q_{p'}^k,$$

with RKHS map

$$\Psi_2(x) = \Big(g_1'(x), \ldots, g_{d+1}'(x), x_1'g_1'(x), \ldots, x_1'g_{d+1}'(x), \ldots, x_d'g_1'(x), \ldots,$$

$$x_d'g_{d+1}'(x), (x_{k_1}'x_{k_2}'g_k'(x))_{k_1,k_2 \in [d], k \in [d+1]}, \ldots, (x_{k_1}'x_{k_2}' \ldots x_{k_p}'g_k'(x))_{k_1,\ldots,k_p \in [d], k \in [d+1]}, \cdots \Big).$$

Now, we compute the norm of function $s$ with respect to $K_{\{g\}}$, combining the above representation and dot product with Equation (25) and the fact that we work with unit norm $x$, so $x' = x$:

$$\|s\|_{K_{\{g\}}}^2 = \mu_0 + (d+1)\mu_1. \tag{29}$$

Plugging the above and Equation (28) into the definition of $\|.\|_\zeta$ in (16), we conclude

$$\|s\|_\zeta^2 \le \|s\|_{K_{\{g\}}}^2 (\|g_1\|_{H^\infty}^2 + \sum_{k=1}^d \|g_2^k\|_{H^\infty}^2) \le (\mu_0 + (d+1)\mu_1) \sum_{p=1}^\infty \gamma_p a_p^2 \|w_p\|_2^{2p}. \tag{30}$$

Note that if the odd exponents (except possibly one) in the definition of $s$ in (24) are zero, then we could consider only the function $g_1$ and kernel $K_{g_1}$, which would have implied a bound of $\mu_0 \sum_{p=1}^\infty \gamma_p a_p^2 \|w_p\|_2^{2p}$.

## A.2 THE DOUBLING TRICK

For the SGD optimization, we set the regularization coefficients in the loss $L_1$ as

$$\psi_1 = \nu/4, \quad \psi_2 = \nu/(4\zeta(f^*, G)), \tag{31}$$

with $\nu := \max\{R_n(f^*)/2, B^2/n\}$. This assumes we know the $f^*$ and $G$ that minimize the adaptation within the complexity measure (12). To achieve generalization bound in Theorem 1, here, we explain how to use a simple doubling trick to get over the fact that we might not know these optimal solutions $f^*$ and $G$. The proof here is based on the generalization result in Theorem 3.

**Theorem 1** *Without explicitly knowing the exact value of the complexity measure, i.e., the optimal solution of Equation (12), one can still achieve the generalization bound in Theorem 1.*

**Proof of Theorem 1**

Our core generalization result is in Theorem 3. The proof of Theorem 1 is simply adding a doubling trick on top of the argument of Theorem 3. We also prove Theorem 2 as a consequence of Theorem 1 in Appendix 2. In the rest of the proofs, for simplicity, we refer to $(W_{\mathrm{PSGD}}, V_{\mathrm{PSGD}})$ by $(W', V')$. Let $f^{**}, G^*$ be the optimal solution to (10). With a simple rescaling of $G^*$, we can assume $\langle H^{\infty-1}, G^* \rangle = 1$. (Note that the complexity does not change by such rescaling). Now one can exploit the condition $\|y_i\|_\infty \le B$, and consider the setting $f^* = 0$ to get the following trivial upper bound on the complexity measure:

$$\Im((x_i), (y_i)) \le 2nB^2.$$

Therefore,

$$2nR_n(f^{**}) + f^{**}A^{-1}f^{**}(c'\varpi) \le 2nB^2. \tag{32}$$

Using Equation (32) and the optimality of $(f^{**}, G^*)$:

$$R_n(f^{**}) \le B^2,$$
$$\zeta = \zeta(f^{**}, G^*) \le 2nB^2. \tag{33}$$

Combining the first equation above with the definition of $\nu$ in Equation (41), we get

$$B^2/n \le \nu \le B^2. \tag{34}$$

To initialize $\psi_1$ and $\psi_2$, we use Equations (31) for any $f^*$ and $G$, and as a result we get a generalization bound as in Equation (45). However, to achieve the best possible rate characterized by our complexity measure in Theorem 1 without explicitly computing the answer of (12), we use a simple

doubling trick; for every pair $(\zeta', \nu')$ such that $\zeta'$ is a power of 2 between $B^2$ and $2nB^2$, and $\nu'$ is a power of two between $B^2/n$ and $B^2$, we initialize $\psi_1, \psi_2$ as in Equation (31) and run the algorithm, then return the network which minimizes the empirical loss after the required polynomial number of steps. This is to make sure that the value of the loss will go at some point below the PSGDstopping threshold on the loss, since the stopping threshold depends on $R_n(f^{**})$ which we are not aware of. Another way to resolve this issue, to have an early stop when the value of the loss pass the threshold is to again run a doubling trick on the value of $R_n(f^{**})$ for every fixed value of $\nu$ and $\zeta$, and run PSGD with stop threshold $R_n(f^{**}) + 2\nu$ (here, $R_n(f^{**})$ is set using the doubling trick variable). This approach works because our final upper bound on the risk ignores the constants (note that the doubling trick introduce additional constants). Moreover, since $\nu \geq B^2/n$ by definition, we don't need to run $R_n(f^{**})$ over values smaller than $\Omega(B^2/n)$, since it does not change the order of $R_n(f^{**}) + 2\nu$. Particularly, combining this with the upper bound on $R_n(f^{**})$, we only need to run the doubling trick for $R_n(f^{**})$ in the interval $(\Omega(B^2/n), O(B^2))$. Now let $\nu'$ be the power of 2 within $\nu(G^*, f^{**})/2 \leq \nu' < \nu(G^*, f^{**})$. If we are in the case

$$f^{**T}A^{-1}f^{**} < B^2, \tag{35}$$

then for $\zeta'$ equal to the smallest power of two larger than $B^2$, when we run PSGD with pair $(\nu', \zeta')$, by Theorem 3:

$$R(f_{W',V'}) \leq 2R_n(f^*) + c''\varpi \frac{2B^2 + B^2}{n} \leq \frac{\Im((x_i)_{i=1}^n, (y_i)_{i=1}^n)}{n} + c' \frac{B^2\varpi}{n}. \tag{36}$$

Because we return the minimum upper bound on the risk (the tighter lower bound of Equation (44)) among all such powers of two, we certainly achieve the above rate in (36). Otherwise, if $f^{**T}A^{-1}f^{**} \geq B^2$, let $\zeta'$ be the power of two within $f^{*T}A^{-1}f^* \leq \zeta' \leq 2f^{*T}A^{-1}f^*$, then again it is easy to check that conditions of Theorem 3 are satisfied, hence we get the following generalization bound:

$$R(f_{W',V'}) \leq 2R_n(f^{**}) + c''\varpi \frac{(\zeta' + B^2)}{n} \leq 2R_n(f^{**}) + c''\varpi \frac{2\zeta(f^{**}, G^*) + B^2}{n} \tag{37}$$

$$\leq \frac{\Im((x_i)_{i=1}^n, (y_i)_{i=1}^n)}{n} + c' \frac{B^2\varpi}{n}, \tag{38}$$

which proves the bound of Theorem 1.

### A.3 AMOUNT OF OVERPARAMETERIZATION

In this section, to provide high-level insight, we indicate the right order of magnitude that our over-parameterization should be in, with respect to one another. Note that the exact coefficients in these inequalities would depend on the basic parameters $B, 1/\lambda_0, n$, which we have avoided here for sake of simplicity. We refer the reader to our main proof (mostly Appendix parts A.12, A.13) for more details.

$$\kappa_1\kappa_2 m_3 << 1,$$
$$\kappa_2\sqrt{m_2} >> 1,$$
$$\kappa_1\sqrt{m_3} >> \kappa_2\sqrt{m_2},$$
$$m_1 >> m_3^4,$$
$$\kappa_1\sqrt{m_1} >> m_3^{3/2},$$
$$\kappa_2 << 1/\sqrt{m_3}$$
$$\sqrt{m_3}\kappa_2 << 1/\sqrt{m_3}$$
$$\sqrt{m_2} >> m_3^{3/2}\kappa_1\kappa_2$$
$$m_3^3(\kappa_2 m_2) << \kappa_1 m_1$$
$$m_1, m_2, m_3, 1/\kappa_1, 1/\kappa_2 = \mathrm{poly}(n, B \vee 1/B, 1/\lambda_0).$$

In addition, we set the smoothing parameters as

$$\beta_2 := \Theta_p\left((\kappa_1\sqrt{m_3})^{-1}(\sqrt{m_2}\kappa_2)^{-\frac{2}{3}}\right),$$
$$\beta_1 := \Theta_p\left(m_3\sqrt{m_3}/(\kappa_1\sqrt{m_1})\right),$$

where $\Theta_p$ only shows polynomial dependencies on the overparameterization.

## A.4 PSD PROPERTY OF $K^\infty$

The Schur product theorem states that for PSD matrices $A$ and $B$, $A \odot B$ is also PSD. Now given an analytic function $F$ whose tailor series coefficients are all nonnegative, Suppose we apply $F$ on some PSD matrix $A$ entrywise, denoted by $F(A)$, under the condition that the entries of $A$ are in the radius of convergence of $F$, then using Schur product theorem, it is straightforward that $F(A)$ is also PSD.

Using the above property, one can then check that the tailor series of the defined functions $F_2$ and $F_3$ are nonnegative, hence, the application of the function $F_2(2F_3(x))$ on the gram matrix of $(x_i)_{i=1}^n$ is a PSD matrix, ( note that $\left| \langle x_i, x_j \rangle \right| \le 1$ is in the convergence radius of $F_2(2F_3(x))$.) thus $K^\infty$ is indeed a kernel.

## A.5 COMPLEXITY UPPER BOUND

First we mention a simple fact that hadamard product respects matrix orderings. Given PSD matrices $A, B, C$ such that $A \preceq B$, the fact that $A \odot C \preceq B \odot C$ is an easy consequence of the Schur Product Theorem; indeed, $B - A$ is PSD by definition, so $(B - A) \odot C = B \odot C - A \odot C$ is also PSD.

Next, it is easy to check that the tailor series of $\arcsin(x)$ has all nonnegative coefficients. Therefore, for a PSD matrix $X$, as we discussed in Appendix A.4, applying $\arcsin$ entrywise on $X$, namely $\arcsin X$, is also PSD. Setting $X$ equal to the entrywise application of $2F_3$ to the gram matrix of datapoints $(x_i)_{i=1}^n$, we realize the matrix $\arcsin \left( 2F_3\left( \left( \langle x_i, x_j \rangle \right)_{1 \le i,j \le n} \right) \right)$ is also PSD. Noting the definition of $K^\infty$ in Equation (8), we conclude that for the data kernel matrix $K^\infty$ we have

$$K^\infty \ge \frac{1}{4}\mathbb{1}\mathbb{1}^T,$$

where $\mathbb{1}$ is the all ones $n$-dimensional vector.

Combining the two mentioned facts, we can lower bound the matrix $K = K^\infty \odot G$ for any matrix $G$ as

$$K = K^\infty \odot G \ge \frac{1}{4}\mathbb{1}\mathbb{1}^T \odot G = \frac{1}{4}G.$$

Substituting the rank one matrix $f^* f^{*T}$ for the $n$-dimensional vector $f^*$ in Equation (13):

$$K^{-1}_{\{\tilde{f}^*/\|\tilde{f}^*\|\}} = K^\infty \odot f^* f^{*T}/\|\tilde{f}^*\|^2 \ge \frac{1}{4}f^* f^{*T}/\|\tilde{f}^*\|^2. \tag{39}$$

The inequality used in (13) then follows from Equation (39).

## A.6 COMPLEXITY MEASURE AND THE $\zeta$-NORM

This is a brief section regarding some basic properties of $\Im$ and $\|.\|_\zeta$.

First, note that the two versions of the complexity measure in Equations (10) and (11) are equivalent, as for any finite set of functions $\{g\}$, we can define the gram matrix with respect to the feature vectors of these functions on data, and for an arbitrary nonzero PSD $G$ we can consider a Cholesky factorization for $G$ as $G = \bar{X}^T \bar{X}$, then define the functions $\{g_k\}$ as the minimum-NTK norm functions which map the input to the features corresponding to $\bar{X}$. This observation further implies we can suppose the factor matrix $\bar{X}$ is in $\mathbb{R}^{n \times n}$, and there is a set of at most $n$ functions $\{g_k\}_{k=1}^n$ which corresponds to this $G$.

Next, we show that for an arbitrary function $f$, its sup norm over the unit ball is bounded by its $\zeta$ norm:

$$\sup_{\|x\|=1} |f(x)| \le \|f\|_\zeta. \tag{40}$$

Note that for a kernel $K$ which satisfies $K(x, x) \leq 1$, using Cauchy Schwarz we simply obtain

$$f(x) \leq \|f\|_K \sqrt{K(x, x)} \leq \|f\|_K,$$

where recall that $\|.\|_K$ is the norm corresponding to the RKHS space of $K$. Hence, to show (40), it suffice to show that for all kernels $K \in \mathcal{K}$ and unit norm $x$ we have $K(x, x) \leq 1$. To see this fact, note that the norm of each $x \in \mathbb{R}^d$ in the NTK-space is $H^\infty(x, x) = \frac{1}{2}$. Therefore, for each function $g$ with bounded-NTK norm, again using Cauchy Schwarz:

$$|g(x)| \leq \frac{1}{2}\|g\|_{H^\infty}.$$

As a result, for a family of functions $\{g\}$ with $\sum_{g \in \{g\}} \|g\|_{H^\infty}^2 \leq 1$, we have on every unit norm $x$:

$$\sum_{g \in \{g\}} g(x)^2 \leq 1.$$

On the other hand, it is easy to check that for every unit norm $x$, we have $K^\infty(x, x) \leq \frac{1}{2}$, so for every such $\{g\}$, we have by definition

$$K_{\{g\}}(x) \leq 1,$$

which completes the proof of Equation (40).

A.7 CORE GENERALIZATION RESULT

In this section, we prove our core generalization result for the trained network, Theorem 3, which underlies our generalization bounds in Theorems 1 and 2. Recall that in the rest of the proofs, we refer to the solution $(W_{\text{PSGD}}, V_{\text{PSGD}})$ returned by PSGD simply by $(W', V')$.

**Theorem 3** *Suppose we have a good candidate pair* $(f^*, G)$ *regarding our complexity measure in (10) that satisfies* $\langle H^{\infty -1}, G \rangle \leq 1$, $f^{*T} A^{-1} f^* \leq \zeta$ *(recall* $A = G \odot H^\infty$*), and that* $f^*$ *has zero projection onto the directions of eigenvectors of $A$ whose eigenvalues are smaller than $O(1/n^2)$ (the last condition can be relaxed, see the next remark). Then, for*

$$\nu = \max\{R_n(f^*)/4, \ B^2/n\}, \tag{41}$$

*if we are given* $\nu/2 \leq \nu' \leq \nu$, *and we set*

$$\psi_1 = \frac{\nu'}{4}, \tag{42}$$

$$\psi_2 = \frac{\nu'}{4\zeta}, \tag{43}$$

*then for the solution* $(W', V')$ *returned by* PSGD *we have the following generalization bound:*

$$R(f_{W',V'}) \leq \frac{5}{4} R_n(f_{W',V'}) + c''' \varpi \frac{(\zeta + B^2)}{n} \tag{44}$$

$$\leq \frac{5}{3} R_n(f^*) + c'' \varpi \frac{(\zeta + B^2)}{n}, \tag{45}$$

*for constants* $c'', c'''$ *and log factor* $\varpi = \log(n)^3 + \log(1/\lambda_0)$.

**Remark 1** *Given a pair* $(f^*, G)$ *satisfying* $\langle H^{\infty -1}, G \rangle \leq 1$, $f^{*T} A^{-1} f^* \leq \zeta$, *one can project out the directions that are along the eigenvectors of $A$ with eigenvalues smaller than $\Omega(1/n^2)$ to obtain* $\bar{f}^*$, *then use the pair* $(\bar{f}^*, G)$ *in Theorem 3. This way, the third condition mentioned in Theorem 3 also becomes true. As we show in Lemma 42, by switching $f^*$ to $\bar{f}^*$ the quantity $f^{*T} A^{-1} f^*$ does not increase, and the quantity $R_n(f^*)$ is multiplied by a constant $c > 1$ arbitrarily close to one, then adds up with $O(B^2/n)$. This means that the bounds in Theorem 3 for the pair $(\bar{f}^*, G)$ translates into similar bounds for $(f^*, G)$ albeit with a bit worse contants. It is straightforward to see that with small enough choice of $c$ and careful AM-GM inequality that we apply inthe proof of Theorem 3, one can end up with the same constants regarding the pair $(f^*, G)$ as declared in Theorem 3. For a more careful discussion on this, we refer the reader to the proof of Lemma 10.*

**Proof of Theorem 3**

Almost all of our proofs in the rest are in the aim of proving Theorem 3. Crucially, to prove this Theorem, we need to establish two big results:

1. We need to show that the final network has small training loss, and is within the class $\mathcal{G}_{\gamma_1, \gamma_2}$ for some suitable $\gamma_1, \gamma_2$. This is handled by Theorem 4 in Appendix A.10. We define the class $\mathcal{G}_{\gamma_1, \gamma_2}$ roughly as the class of networks with norm bounds $\|W - W^{(0)}\| \leq \gamma_1, \|V - V^{(0)}\| \leq \gamma_2$ where the rows of $V - V^{(0)}$ are orthogonal to the subspace $\Phi$, plus an additional structure defined in Appendix A.11. This task, on its own, has three main steps in our proof:

   (a) we construct a "good" underlying network, Appendix A.12
   (b) we find a "good" random direction and study the landscape of the objective, Appendix A.13
   (c) we prove the convergence, Appendix A.16

2. The Rademacher Complexity of the class $\mathcal{G}_{\gamma_1, \gamma_2}$ needs to be suitably bounded. This is handled by Theorem 5 in Appendix A.11.

With access to these results, here we show how Theorem 3 follows by a simple application of the generalization bound in (Srebro et al., 2010). Specifically, for fixed constants $z_1, z_3$ and every integer

$i \geq 0$, we use Theorem 1 of (Srebro et al., 2010) for the class $\mathcal{G}_{z_1,\gamma_i}$, $\gamma_i = 2^i \times B/z_3$ with confidence probability $1 - 2^{-i}\delta_3$, which, with a union bound, implies that with probability at least $1 - \delta_3$, for every $i$ and $f_{W',V'} \in \mathcal{G}_{z_1,\gamma_i}$:

$$R(f_{W',V'}) \leq R_n(f_{W',V'}) + K\Big(\sqrt{R_n(f_{W',V'})}\Big(\sqrt{\log(n)^{1.5}\mathcal{R}(\mathcal{G}_{z_1,\gamma_i}) + \sqrt{\frac{b\log(1/(2^{-i}\delta_3))}{n}}}\Big) \tag{46}$$

$$+ \log(n)^3\mathcal{R}(\mathcal{G}_{z_1,\gamma_i})^2 + \frac{b\log(1/(2^{-i}\delta_3))}{n}\Big), \tag{47}$$

where $\ell(f_{W',V'}(x),y)$ is a.s. bounded by $b$ for function within the class $\mathcal{G}_{z_1,\gamma_i}$, and $K$ is a universal constant. In the following, we aim to further bound the Rademacher complexity $\mathcal{R}$ and parameter $b$.

Applying the AM-GM inequality with respect to ratio $z_4 > 0$ for the second term:

$$R(f_{W',V'}) \leq (1 + z_4)R_n(f_{W',V'}) + K^2/z_4\Big(\log(n)^{1.5}\mathcal{R}(\mathcal{G}_{z_1,\gamma_i}) + \sqrt{\frac{b\log(1/(2^{-i}\delta_3))}{n}}\Big)^2$$

$$+ \log(n)^3\mathcal{R}(\mathcal{G}_{z_1,\gamma_i})^2 + \frac{b\log(1/(2^{-i}\delta_3))}{n}$$

$$\leq (1 + z_4)R_n(f_{W',V'}) + K^2/z_4\Big(\log(n)^{1.5}\mathcal{R}(\mathcal{G}_{z_1,\gamma_i}) + \sqrt{\frac{b\log(1/(2^{-i}\delta_3))}{n}}\Big)^2$$

$$+ \log(n)^3\mathcal{R}(\mathcal{G}_{z_1,\gamma_i})^2 + \frac{b\log(1/(2^{-i}\delta_3))}{n}$$

$$\leq (1 + z_4)R_n(f_{W',V'}) + (2K^2/z_4 + 1)\log(n)^3\mathcal{R}(\mathcal{G}_{z_1,\gamma_i})^2 + (2K^2/z_4 + 1)\frac{b\log(1/(2^{-i}\delta_3))}{n}. \tag{48}$$

Now let $\gamma^*$ be the smallest number of the form $2^i B/z_3$ (for some $i$) which is not smaller than $z_2\sqrt{\zeta}$. This definition implies

$$\gamma^* \leq \max\{2z_2\sqrt{\zeta}, B/z_3\}. \tag{49}$$

Now Theorem 5 in Appendix A.11 bounds the Rademacher complexity:

$$\mathcal{R}(\mathcal{G}_{z_1,\gamma^*}) \leq \frac{2z_1\gamma^*}{\sqrt{n}}. \tag{50}$$

On the other hand, from Theorem 4 by setting $z_1, z_2 = \sqrt{40}$, we get $f_{W',V'} \in \mathcal{G}_{z_1,\gamma^*}$.

Moreover, from Lemma 34, for $f_{W',V'} \in \mathcal{G}_{z_1,\gamma^*}$, we have for every $\|x\| \leq 1$:

$$|f_{W',V'}(x)| \leq 2z_1\gamma^*, \tag{51}$$

so the loss $\ell(f_{W',V'}(x),y)$ can be bounded by $(B + 2z_1\gamma^*)^2$ using the 1 smoothness property. Therefore, for the class $\mathcal{G}_{z_1,\gamma^*}$ we can set $b = (B + 2z_1\gamma^*)^2$. Combining this with Equation (50) and plugging into Equation (48):

$$R(f_{W',V'}) \leq (1 + z_4)R_n(f_{W',V'}) + (2K^2/z_4 + 1)\log(n)^3\frac{4z_1^2\gamma^{*2}}{n} + (2K^2/z_4 + 1)\frac{(B + 2z_1\gamma^*)^2\log(1/(2^{-i^*}\delta_3))}{n}.$$

Furthermore, by definition of $\gamma^*$, we have $2^i \leq 2z_2z_3\sqrt{\zeta}/B$:

$$R(f_{W',V'}) \leq (1 + z_4)R_n(f_{W',V'}) + (2K^2/z_4 + 1)4z_1^2(\log(n)^3 + 2\log(2z_2z_3\sqrt{\zeta}/B))\frac{4z_1^2\gamma^{*2}}{n} \tag{52}$$

$$+ (2K^2/z_4 + 1)\frac{2B^2\log(2z_2z_3\sqrt{\zeta}/B)}{n}. \tag{53}$$

Now applying the upper bound on $\gamma^*$:

$$R(f_{W',V'}) \leq (1 + z_4)R_n(f_{W',V'}) + (2K^2/z_4 + 1)4z_1^2(\log(n)^3 + 2\log(2z_2z_3\sqrt{\zeta}/B))\frac{4z_1^2(2z_2\sqrt{\zeta} + 2B/z_3)^2}{n} \tag{54}$$

$$+ (2K^2/z_4 + 1)\frac{2B^2\log(2z_2z_3\sqrt{\zeta}/B)}{n}. \tag{55}$$

If $\zeta > B^2$, in the third term above we substitute $B$ by $\sqrt{\zeta}$. Finally, similar to the bound we stated in Equation (14), note that we have the following trivial bound for $\zeta$:

$$\zeta \leq y^T H^{\infty-1} y \leq 4nB^2/\lambda_0, \tag{56}$$

i.e. there is no point in considering larger $\zeta$'s, which implies $\log(2z_2 z_3 \sqrt{\zeta}/B) = O(\log(n) + \log(1/\lambda_0))$. Plugging this above and picking $z_4 = 1/3$ show the proof of Equation (44). Furthermore, applying Equation (68) in Theorem 4 to the $R_n(f_{W',V'})$ term in Equation (44) further gives the second Equation (45).

**Remark 2** *In the same setting of Theorem 3, if we have $\nu/2 \leq \nu'$ but not generally upper bounded by $\nu$, then PSGD leads to the following generalization bound:*

$$R(f_{W',V'}) \leq R_n(f^*) + \nu' + c'''' \varpi \frac{(\zeta + B^2)}{n},$$

*using a similar argument as we did for Theorem 3.*

## A.8 STRUCTURE OF THE PROOF, SETTING $m_3$, AND FURTHER DEFINITIONS

Throughout the proof, $(W', V')$ represents the pair of matrices of the current iteration of PSGD, $(W^*, V^*)$ are the "ideal" matrices that we construct in Appendix A.12, $(W^\rho, V^\rho)$ and refers to the gaussian smoothing matrices. Importantly, note that our squared loss $\ell(f, y)$ is **zero** at $f = y$. We have tried to make the lower level proofs into sub-lemmas and create a manageable hierarchy as much as we could, to make the document more clear and readable.

Similar to the conditions in Theorem 3, through out most of the proofs we assume that we are given a pair $(f^*, G)$ with a slightly more general setting of Theorem 3:

$$f^{*T} A^{-1} f^* \leq \zeta_2, \text{ for } A = G \odot K^\infty,$$
$$\langle G, H^\infty \rangle \leq \zeta_1.$$

Particularly, $\zeta_1, \zeta_2$ appear in Appendix A.13. Because we are allowed to rescale $G$, we do not really gain much by assuming this more general setting, though we pick to work with the general setting as the abstraction makes the proof more straightforward to understand.

We refer to the parameters $B, 1/\lambda_0, n$ as the "basic parameters", $m_1, m_2, m_3, 1/\kappa_1, \kappa_2$ as the "overparameterization", and $\beta_1, \beta_2$ as the "smoothing parameters." By the phrase "having enough overparameterization" we mean it suffices to pick the overparameterization $m_1, m_2, m_3, 1/\kappa_1, 1/\kappa_2$ only **polynomially** large in the basic parameters.

Throughout the proof, we denote the change in the output of the first layer at $W^{(0)} + W' + W^\rho$ compared to the initialization value by $\phi^{(2)}(x_i)$, i.e.

$$\phi^{(2)}(x_i) = \frac{1}{\sqrt{m_1}} W^s \sigma((W^{(0)} + W' + W^\rho)x_i) - \phi^{(0)}(x_i),$$

while recall that $\phi'(x_i)$ has a similar definition except without the smoothing matrix $W^\rho$. Although our model is a three layer network, throughout the proof, we refer to the parts $W^s \frac{1}{\sqrt{m_1}} \sigma((W^{(0)} + W')x)$ and $\frac{1}{\sqrt{m_2}} a^T \sigma\left((V^{(0)} + V')(.)\right)$ as the "first layer" and "second layer," respectively.

Also, we sometimes refer to the binary sign pattern of vector $x$ multiplied to matrix $W$ by $D_{W,x}$ ($D_{W,x} := \text{Sgn}(Wx)$), i.e. the $j$th diagonal entry of $D_{W,x}$ is one if $W_j^T x \geq 0$, and is zero otherwise. To refer to the $j$th row of $W$ as a vector, we sometimes drop the comma in $W_{j,}$ and write it as $W_j$.

For brevity, we denote the Frobenius norm $\|W\|_F$ of matrix by $\|W\|$, and the Euclidean norm of a vector $x$ by $\|x\|$. For matrices $W_1, W_2$ we denote their natural dot product by $\langle W_1, W_2 \rangle := \text{tr}(W_1^T W_2)$. We refer to the smallest eigenvalue of a matrix by $\lambda_{\min}(.)$. We write $\mathcal{R}(.)$ for the Rademacher complexity of a function class. We refer to the smoothed version of the network by $f'_{W',V'}(x)$, defined by

$$f'_{W',V'}(x) = \mathbb{E}_{W^\rho, V^\rho} f_{W'+W^\rho, V'+V^\rho}(x).$$

In the proof, we mainly work with the loss over the smoothed network $f'$, defined as

$$L(W', V') = R_n(f'_{W',V'}) + \psi_1 \|W'\|^2 + \psi_2 \|V'\|^2. \tag{57}$$

Our algorithm, PSGD can be regarded roughly as an SGD over $L$.

Similar to what we discussed in section 3, let the functions $\{g_k\}_{k=1}^{m_3}$ be some feature representation whose gram matrix is equal to $G$ and $\langle H^\infty, G \rangle = \sum_{k=1}^{m_3} \|g_k\|_{H^\infty}^2$. In such setting, it is not hard to check that we can assume each $g_k$ is the minimum norm NTK function which maps $(x_i)_{i=1}^n$ to $(g_k(x_i))_{i=1}^n$'s. Indeed, if this is not the case for some $g_k$, we can project the RKHS representation of $g_k$ onto the span of the representations of $(x_i)_{i=1}^n$, which can only decrease the complexity measure. Hence we can represent $g_k \in \mathcal{H}_{H^\infty}$ as a linear combination of basic functions $H^\infty(x_i, .)$ on data points:

$$\forall k \in [m_3], \ g_k(x) := \sum_{i=1}^n \mathcal{V}_{k,i} H^\infty(x_i, x). \tag{58}$$

Here, the sum of squared-$H^\infty$ norms of $\mathcal{V}_k$ is bounded as

$$\sum_k \|\mathcal{V}_k\|_{H^\infty}^2 = \sum_k \|g_k\|_{H^\infty}^2 = \langle H^\infty, G \rangle \leq \zeta_1. \tag{59}$$

For each $i \in [n]$, we refer to the feature representation vector $(g_k(x_i))_{k=1}^{m_3}$ on $x_i$ as $\bar{x}_i$. Note that we have the relationship

$$\bar{x}_i = (H_{i,}^\infty \mathcal{V}_k)_{k=1}^{m_3}, \tag{60}$$

where $H_{i,}^\infty$ is the $i$th row of $H^\infty$. In the analysis, we work with a bound $\xi$ on the quantity $\max_k \|\mathcal{V}_k\|$ which should be bounded polynomially by other basic parameters; in particular, it is defined in Lemma 10 and is used to bound a cross term in Lemma 14. However, $\max_k \|\mathcal{V}_k\|$ might not be effectively bounded for an arbitrary feature representation. Fortunately, we can remedy this by a simple trick; for every natural number $s$, one can substitute every $g_k$ by $s$ copies of $g_k/\sqrt{s}$, without changing the gram matrix $G$. Therefore, for any $\delta$, one can increase the multiset of functions $(g_k)$ to a bigger set $(\tilde{g}_k)$, by adding at most $O(\zeta_1/\delta)$ functions, making sure of the following for the new functions:

$$\forall k: \ \mathcal{V}_k^T H^\infty \mathcal{V}_k = \|\tilde{g}_k\|_{H^\infty}^2 \leq \delta. \tag{61}$$

(This is because $\sum_k \|\tilde{g}_k\|_{H^\infty}^2 \leq 1$). Furthermore, observe that for each gram matrix $G$, we have an $n$-dimensional feature representation $(g_k)_{k=1}^n$ for $G$ according to the Cholesky factorization. Combining these facts, we conclude, to guarantee Equation (61), in the worst case, we need $m_3$ to be as large as $n + O(\zeta_1/\delta)$.

Finally, observing the following inequality

$$\|\mathcal{V}_k\|^2 \leq \|\mathcal{V}_k\|_{H^\infty}^2/\lambda_0 \leq \|\tilde{g}_k\|_{H^\infty}^2/\lambda_0. \tag{62}$$

in order to guarantee $\max_k \|\mathcal{V}_k\| \leq \xi$ we need to take $m_3$ as large as $n + O(\zeta_1/(\xi^2 \lambda_0))$, which is indeed bounded polynomially by the basic parameters because of the same condition for $1/\xi$. This computation also brings into sight an important point:

"Although each gram matrix $G$ is representable by $n$ features, in order for the algorithm to be able to find a suitable network, $m_3$ might need to be larger than $n$."

Moreover, for every $1 \leq k, i \leq n$, we define the matrices $Z_i^k \in \mathbb{R}^{md}$ as

$$Z_k^i = 1/\sqrt{m_1}\Big(W_{k,j}^s \mathbb{1}\{W_j^{(0)T} x_i\} x_i\Big)_{j=1}^{m_1}, \tag{63}$$

where in the above notation, $j$ is enumerating the columns of the matrix. We also define the following matrices which we use in our construction later:

$$W_k^{+T} = W^{k+T} = \sum_{i=1}^n \mathcal{V}_{k,i} Z_k^i, \tag{64}$$

and $W^+$ as

$$W^+ = \sum_{k=1}^{m_3} W^{k+}.$$

Finally, to avoid unnecessary complication, we often argue **high probability** bounds without an explicit representation of their dependency on the chance of failure (which is a negligible logarithmic factor). We also ignore all constants and log factors, and mainly work with the notation $\lesssim$ which ignores constants; we write $a \lesssim b \pm c$ as a short form for $b - O(c) \leq a \leq b + O(c)$. As there are several hierarchies of new parameters that are defined based on lower-level ones, we rename the new parameters and continue viewing them as black-box. This makes the proofs more readable, since we also do not care about the exact dependency of the underlying parameters most of the time, rather we are interested in their orders of magnitude, for example that a given parameter goes to zero polynomially fast with respect to the overparameterization, etc. Due to the large number of symbols that we have to work with, we might use a symbol more than once, of course when it is clear from the context which one we are refering to.

### A.9 Proof of Theorem 2

In this section, we prove Theorem 2, stated below.

**Theorem 2** *For any function $f : \mathbb{R}^d \to \mathbb{R}$, in the same setting as Theorem 1, the population risk of the trained network $(W', V')$ can be bounded as*

$$R(f_{W',V'}) \leq 2R(f) + O\Big(\alpha\varpi \frac{\|f\|_\zeta^2 + B^2}{n}\Big). \tag{65}$$

**Proof of Theorem 2**

Theorem 2 is a simple consequence of Theorem 1; for the given function $f$, we apply Theorem 2 with the smaller coefficient $\gamma = \frac{4}{3}$ for $R_n(f^*)$, regarding the complexity upper bound, by setting $f^* := (f(x_i))_{i=1}^n$:

$$R(f_{W',V'}) \leq \frac{4}{3} R_n(f^*) + (\alpha\varpi) \min_{K \in \mathcal{K}} \frac{f^{*T} K^{-1} f^*}{n} + \frac{B^2 \alpha\varpi}{n}$$

$$= \frac{4}{3} R_n(f) + (\alpha\varpi) \min_{K \in \mathcal{K}} \frac{f^{*T} K^{-1} f^*}{n} + \frac{B^2 \alpha\varpi}{n}.$$

On the other hand, because $f^{*T} K^{-1} f^*$ is the minimum-RKHS norm of a function with respect to kernel $K$ which maps $x_i$'s to $f^*_i$ and $f$ is one such function, we have $f^{*T} K^{-1} f^* \leq \|f\|_K$. This inequality implies

$$\min_{K \in \mathcal{K}} f^{*T} K^{-1} f^* \leq \|f\|_\zeta,$$

so we obtain

$$R(f_{W',V'}) \leq \frac{4}{3} R_n(f) + (\alpha\varpi) \frac{\|f\|_\zeta^2 + B^2}{n}. \tag{66}$$

Therefore, it remains to bound $R_n(f)$ by $R(f)$.

As we showed in Appendix A.6, for every input $x$ we have $f(x) \leq \|f\|_\zeta$, so for every data $(x, y)$, by the fact that $|y| \leq B$ a.s. and $\alpha$ smoothness of the loss, we have $\ell(f(x), y) \leq \alpha(\|f\|_\zeta + B)^2$. Moreover, note that the random variable $\ell(f(x), y)$ has mean $R(f)$. It is easy to check that in this setting, the variance of $\ell(f(x), y)$ is at most $R(f)\alpha(\|f\|_\zeta + B)^2$. Therefore, an application of the Bernstein inequality, we have with high probability over the dataset

$$R_n(f) \leq R(f) + O\left( \sqrt{\frac{R(f)\alpha(\|f\|_\zeta + B)^2}{n}} + \frac{\alpha(\|f\|_\zeta + B)^2}{n} \right) \leq \frac{3}{2} R(f) + O\left( \frac{\alpha(\|f\|_\zeta + B)^2}{n} \right).$$

Plugging this back to Equation (66) completes the proof. As a result, the learned network can compete with any function that has reasonably small $\|f\|_\zeta$:

$$R(f_{W',V'}) \leq \min_f \Big\{ 2R(f) + O(\alpha\varpi \frac{\|f\|_\zeta^2 + B^2}{n}) \Big\}.$$

A.10 OPTIMIZATION

In this section, we glue together

- the existence of a good random direction that we prove in Appendix A.13
- the convergence analysis of PSGD that we do based on the work Ge et al. (2015b) in Appendix A.16.

**Theorem 4** *In the same setting of Theorem 3, assume the network $(W', V')$ returned by* PSGD, *has sufficient polynomially large "overparameterization". Then, for the solution $(W', V')$ returned by* PSGD *we have*

$$L(W', V') \leq R_n(f^*) + \nu, \tag{67}$$

*which further implies*

$$R_n(f_{W',V'}) \leq R_n(f^*) + 2\nu, \tag{68}$$

$$\|W'\|^2 \leq 40, \; \|V'\|^2 \leq 40\zeta. \tag{69}$$

*Moreover, for every $i \in [n], j \in [m_1], j \notin P$ for $P$ defined in 1, we have that $\mathrm{sign}((W_j^{(0)} + W_j')^T x_i)$ and $\mathrm{sign}(W_j^{(0)^T} x_i)$ are the same.*

**Proof of Theorem 4**

Let $\Upsilon \in \mathbb{R}^{m_2(m_3-n) \times m_2 m_3}$ be a matrix whose rows are an orthonormal basis for the space of matrices whose rows are orthogonal to $\mathrm{span}(\{\phi^{(0)}(x_i)\}_{i=1}^n)$, i.e. $\Phi^\perp$, as defined in (22). Then, we consider a linear change of coordinates for the subspace $\Phi^\perp$, regarding the second layer weights, as $v' = \Upsilon \mathrm{vec}(V')$ where $\mathrm{vec}(.)$ splits out the vectorized version of a matrix. For consistent notation, we also denote $w' = W'$, so we now have a new coordinate system $(w', v') \in \mathbb{R}^{m_2(m_3-n) \times m_1 d}$ for pairs of weights $(W', V')$ such that $V' \in \Phi^\perp$. We also define the loss function

$$L^\Pi(w := (w', v')) = L(W', V'),$$

with respect to the change of coordinate.

Now it is easy to see that running PSGD on $L$ in the normal coordinates is equivalent to running stochastic gradient descent on $L^\Pi$ with respect to $(w', v')$. Moreover, because multiplying to matrix $\Upsilon$ is an orthonormal change of coordinates for $\Phi^\perp$ and because $V'$ is already in $\phi^\perp$ at each step of PSGD, then $\|v'\| = \|V'\|$, so the conditions $\|W'\| \leq C_1, \|V'\| \leq C_2$ are equivalent to $\|w'\| \leq C_1, \|v'\| \leq C_2$. Furthermore, by our construction, the random matrix $V_\Sigma^*$ is in the subspace $\Phi^\perp$, so the norm bounds $\|W^*\| \leq \zeta_1, \|V^*\| \leq \zeta_2$ are equivalent to $\|w^*\| \leq \zeta_1, \|v^*\| \leq \zeta_2$ for $w^* := W_\Sigma^*$ and $v^* := \Upsilon \mathrm{vec}(V^*)$.

Now we apply the result of Theorem 6 on $L^\Pi$ with parameter $\nu$ set as $\nu'$ (recall the definition of $\nu'$ from Theorem 3), $\zeta_2 := \zeta$ and $\zeta_1 := 1$, and $\Delta := R_n(f^*)$, as defined in Theorem 3. More specifically, based on our arguments above regarding the natural isometry in the change of coordinate, any pair $(w', v')$ in the domain $\|w'\| \leq C_1, \|v'\| \leq C_2, L^\Pi(w', v') \geq R_n(f^*) + \nu'$ translates into a pair $(W', V')$ in the domain $\|W'\| \leq C_1, \|V'\| \leq C_2, L(W', V') \geq R_n(f^*) + \nu'$, for which by Theorem 6 there exists $(W_\Sigma^*, V_\Sigma^*)$ such that

$$\mathbb{E}_\Sigma L(W' - \eta/2W' + \sqrt{\eta}W_\Sigma^*, V' - \eta/2V' + \sqrt{\eta}V_\Sigma^*) \leq L(W', V') - \eta\nu'/4. \tag{70}$$

Translating back to the change of coordinates:

$$\mathbb{E}_\Sigma L^\Pi(w' - \eta/2w' + \sqrt{\eta}w_\Sigma^*, v' - \eta/2v' + \sqrt{\eta}v_\Sigma^*) \leq L(w', v') - \eta\nu'/4. \tag{71}$$

Now we apply Lemma 44 to translate this into an argument about the landscape of $L^\Pi$. As a result, applying the bounds in Equations (106) and (126), we obtain that for $(w', v')$ such that

$$L^\Pi(w', v') \geq R_n(f^*) + \nu',$$

we should either have

$$\|\nabla L^{\Pi}(w', v')\| \geq \frac{\nu/4}{4\sqrt{\|w'\|^2 + \|v'\|^2}}$$
$$= \frac{\nu/4}{4\sqrt{\|W'\|^2 + \|V'\|^2}}$$
$$= \frac{\nu}{16\sqrt{C_1^2 + C_2^2}},$$

or

$$\lambda_{min}\left(\nabla^2 L^{\Pi}(w', v')\right) \leq -\frac{\nu/4}{2\min_{\Sigma}(\|w^*\|^2 + \|v^*\|^2)}$$
$$= -\frac{\nu}{2\min_{\Sigma}(\|W_{\Sigma}^*\|^2 + \|V_{\Sigma}^*\|^2)}$$
$$\leq -\frac{\nu}{16(\zeta_1 + \zeta_2)}$$
$$= -\frac{\nu}{16(1 + \zeta)}.$$

Next, we want to apply Theorem 7 by setting

$$\gamma = \frac{\nu}{16(1 + \zeta)},$$
$$\aleph_{\ell} = R_n(f^*) + \nu',$$

and Lipschitz parameters $\rho_1, \rho_2, \rho_3 = poly(B, C_1, C_2, m_1, m_2, m_3)$ set as described in Appendix B.1, Theorem 9. Also, note that as prescribed by Theorem 7, we set

$$C_1 := \frac{\aleph + 4l}{\psi_1},$$
$$C_2 := \frac{\aleph + 4l}{\psi_2}, \tag{72}$$

where $l = O(1)$ depends on our desired chance of success for the algorithm, specified in Theorem 7. Finally, note that Theorem 7 needs to work with a bounded noise on the gradient whose covariance matrix is bounded between two multipliers of identity. The point of injecting extra noise to SGD in `PSGD` is in fact because of this covariance condition that we need in Theorem 7. On the other hand, note that in general, because of the gaussian smoothing that we use, the noise vector is not supported on a bounded domain, which makes it a bit harder to apply Hoeffding type concentration. To remedy this, we introduce a coupling between our unbounded noise vector for $L(W', V')$ and another noise random variable whose support is bounded, which with high probability is equal to the real noise, along all iterations. In Corollary, we further translate this coupling for the objective $L^{\Pi}$ after change of cooridnates, and write down the exact dependencies of the parameters $Q, \sigma_1$ and $\sigma_2$, which are all polynomial in the basic parameters and the overparameterization.

Hence, the conditions of Theorem 7 are satisfied, so we conclude that after at most $poly(\rho_1, \rho_2, \rho_3, Q, \aleph, C_1, C_2, 1/\gamma, \log(\sigma_1/\sigma_2)) = poly(B, m_1, m_2, m_3, C_1, C_2, \zeta_1, \zeta_2) = poly(n, B \vee 1/B, 1/\gamma_0)$ number of iterations, `PSGD` reach a point $w_t$ in some iteration $t$ with $L^{\Pi}(w_t) \leq \aleph_{\ell}$.

Translating back this $w_t = (w'_t, v'_t)$ by multiplying the $v'_t$ part to $\Upsilon^T$, we get a pair $(W'_t, V'_t)$ with objective value bounded as

$$L(W'_t, V'_t) \leq R_n(f^*) + \nu'. \tag{73}$$

But note that we obviously have the condition $\|W'\| \leq C_1, \|V'\| \leq C_2$ through the whole iterations, for the choice of $C_1, C_2$ in Equation (72). Therefore, using Lemma 34, for every $i \in [n]$:

$$|f'_{W', V'}(x_i)| = O(C_1, C_2), \tag{74}$$
$$|f_{W', V'}(x_i)| = O(C_1, C_2) \tag{75}$$

From Equations (75), as also stated in Theorem 9, we know that for all $i \in [n]$, $\ell(., y_i)$ is $O(C_1 C_2) + B^2$-Lipschitz at points $f_{W',V'}(x_i)$ and $f'_{W',V'}(x_i)$, so we can bound the difference $|\ell(f'_{W',V'}(x_i), y_i) - \ell(f_{W',V'}(x_i), y_i)|$ by $(O(C_1 C_2) + B^2)|f'_{W',V'}(x_i), y_i) - f_{W',V'}(x_i)|$, which in turn can become arbitrarily small having enough overparameterization using Lemma 35, in particular, we force it to be smaller than $O(\nu'/(B^2 + C_1 C_2))$ (recall $\nu' \geq \nu/2 \geq B^2/(2n)$). As a result, we get $|\ell(f'_{W',V'}(x_i), y_i) - \ell(f_{W',V'}(x_i), y_i)| = O(\nu)$ for every $i \in [n]$, which in turn implies $|L(W', V') - L_1(W', V')| \leq \nu$ by picking small constants, where recall that the objective $L_1$ is the same as $L$ but without the smoothing. Now applying this bound to Equation (73), we get

$$L_1(W'_t, V'_t) \leq R_n(f^*) + 2\nu'.$$

Therefore, as $\texttt{PSGD}$ check the values of $L_1$ in the loop, it terminates at such pair $(W_t, V_t)$. From this point onward, we refer to the returned $(W'_t, V'_t)$ as just $(W', V')$.

Opening the definition of $L_1(W', V')$, we clearly get

$$R_n(f_{W',V'}) \leq L_1(W', V') \leq R_n(f^*) + 2\nu' \leq R_n(f^*) + 2\nu. \tag{76}$$

Furthermore, noting the setting of $\psi_1, \psi_2$ in Theorem 3 and the fact that $\nu' \geq \nu/2 \geq R_n(f^*)/8$, we get

$$\|W'\|^2 \leq \frac{4(R_n(f^*) + 2\nu')}{\nu'} \leq 40, \tag{77}$$

$$\|V'\|^2 \leq \frac{4\zeta(R_n(f^*) + 2\nu')}{\nu'} \leq 40\zeta, \tag{78}$$

which completes the proof. The fact that for every $i \in [n], j \in [m_1], j \notin P$ we have that $\mathrm{sign}((W_j^{(0)} + W_j')^T x_i)$ and $\mathrm{sign}(W_j^{(0)^T} x_i)$ are the same follows from Lemma 1.

A.11 RADEMACHER COMPLEXITY

In this section we show the proof for our Rademacher Complexity bound, which is used in Theorem 3.

**Theorem 5** *Let* $G_{\gamma_1,\gamma_2}$ *be the class of neural nets with weights* $(W, V)$ *in our three layer setting, such that* $\|W - W^{(0)}\| \leq \gamma_1$, $\|V - V^{(0)}\| \leq \gamma_2$, *where for every* $j \in [m_2], i \in [n]$: $V_j' \perp \phi^{(0)}(x_i)$, *and for every* $i \in [n], j \in [m_1], j \notin P$ *for* $P \subseteq [m_1]$ *defined in Lemma 1, it satisfies* $\mathrm{sign}((W_j^{(0)} + W_j')x_i) = \mathrm{sign}(W^{(0)}x_i)$. *Then, for large enough overparameterization, we have the following bound on the Rademacher complexity:*

$$\mathcal{R}(G_{\gamma_1,\gamma_2}) \leq \frac{2\gamma_1\gamma_2}{\sqrt{n}}.$$

**Proof of Theorem 5**

Here, we do not have the smoothing matrices $W^\rho, V^\rho$ anymore. In this section, unlike the optimization section that we used $\{x_i'\}_{i=1}^n$ to denote the output of the first layer by incorporating also the smoothing matrices, here we define it without them:

$$x_i' = \frac{1}{\sqrt{m_1}} W^s \sigma((W^{(0)} + W')x_i).$$

Now define the matrices

$$Z_i' = 1/\sqrt{m_2}\Big(a_j \mathbb{1}\{V_{j,}^{(0)} x_i' \geq 0\} x_i'\Big)_{j=1}^{m_2},$$

$$Z_i'^+ = 1/\sqrt{m_2}\Big(a_j(\mathbb{1}\{V_{j,}x_i' \geq 0\} - \mathbb{1}\{V_{j,}^{(0)} x_i' \geq 0\})x_i'\Big)_{j=1}^{m_2}.$$

To bound the $Z_i'^+$ part, note that substituting $C_1$ by $\gamma_1$ in lemma 6 and assuming conditions

$$m_1 = \tilde{\Omega}(m_3^4),$$

$$\frac{2C_1^{3/2}}{\sqrt{\kappa_1}}\Big(\frac{n^3 m_3^3}{m_1\lambda_0}\Big)^{1/4} \leq \gamma_1,$$

(we can use this result because we do not have the smoothing matrix $W^\rho$ here), we get with high probability over the initialization for every $i \in [n]$:

$$\|\phi'(x_i)\| = \|x_i' - \phi^{(0)}(x_i)\| \lesssim \gamma_1. \tag{79}$$

Therefore, we can write

$$|\mathrm{trace}(V Z_i'^+)| = \frac{1}{\sqrt{m_2}} |\sum_j a_j \mathbb{1}\{\mathrm{sign}(V_{j,}x_i') \neq \mathrm{sign}(V_{j,}^{(0)} x_i')\} V_{j,}x_i'|$$

$$\leq \frac{1}{\sqrt{m_2}} \sum_j \mathbb{1}\{\mathrm{sign}(V_{j,}x_i') \neq \mathrm{sign}(V_{j,}^{(0)} x_i')\} |V_{j,}x_i'|$$

$$\leq \frac{1}{\sqrt{m_2}} \sum_j \mathbb{1}\{|V_{j,}^{(0)} x_i'| \leq |(V_{j,} - V_{j,}^{(0)})x_i'|\} \Big(|(V_{j,} - V_{j,}^{(0)})x_i'| + |V_{j,}^{(0)} x_i'|\Big)$$

$$\leq \frac{1}{\sqrt{m_2}} \sum_j \mathbb{1}\{|V_{j,}^{(0)} x_i'| \leq |(V_{j,} - V_{j,}^{(0)})x_i'|\} \Big(2|(V_{j,} - V_{j,}^{(0)})x_i'|\Big).$$

$$\leq \frac{1}{\sqrt{m_2}} \sum_j \mathbb{1}\{|V_{j,}^{(0)} x_i'|, |(V_{j,} - V_{j,}^{(0)})x_i'| \leq \gamma_2^{2/3}(\frac{\kappa_2}{m_2})^{1/3} \min(\gamma_1, \|x_i'\|)\} \Big(2|(V_{j,} - V_{j,}^{(0)})x_i'|\Big)$$

$$+ \frac{1}{\sqrt{m_2}} \sum_j \mathbb{1}\{|(V_{j,} - V_{j,}^{(0)})x_i'| \geq \gamma_2^{2/3}(\frac{\kappa_2}{m_2})^{1/3} \min(\gamma_1, \|x_i'\|)\} \Big(2|(V_{j,} - V_{j,}^{(0)})x_i'|\Big).$$

Now using the fact that $V_j - V_j^{(0)}$ is orthogonal to $\phi^{(0)}(x_i)$'s:

$$LHS \leq \frac{2\gamma_2^{2/3}(\frac{\kappa_2}{m_2})^{1/3}\gamma_1}{\sqrt{m_2}} \sum_j \mathbb{1}\{|V_{j,}^{(0)}x_i'| \leq \gamma_2^{2/3}(\frac{\kappa_2}{m_2})^{1/3}\|x_i'\|\}$$

$$+ \frac{2}{\sqrt{m_2}} \sum_j \mathbb{1}\{\|V_{j,} - V_{j,}^{(0)}\|\|x_i' - \phi^{(0)}(x_i)\| \geq \gamma_2^{2/3}(\frac{\kappa_2}{m_2})^{1/3}\gamma_1\}\|V_{j,} - V_{j,}^{(0)}\|\|x_i' - \phi^{(0)}(x_i)\|.$$

Next, using the upper bound on $\|x_i' - \phi^{(0)}(x_i)\|$:

$$LHS \lesssim \frac{2\gamma_2^{2/3}(\frac{\kappa_2}{m_2})^{1/3}\gamma_1}{\sqrt{m_2}} \sum_j (\mathbb{1}\{|V_{j,}^{(0)}x_i'| \leq \gamma_2^{2/3}(\frac{\kappa_2}{m_2})^{1/3}\|x_i'\|\}.$$

$$+ \frac{\gamma_1}{\sqrt{m_2}} \sum_j \mathbb{1}\{\|V_{j,} - V_{j,}^{(0)}\| \gtrsim \gamma_2^{2/3}(\frac{\kappa_2}{m_2})^{1/3}\}\|V_{j,} - V_{j,}^{(0)}\|$$

$$\lesssim \frac{\gamma_2^{2/3}(\frac{\kappa_2}{m_2})^{1/3}\gamma_1}{\sqrt{m_2}} \sum_j (\mathbb{1}\{|V_{j,}^{(0)}x_i'| \leq \gamma_2^{2/3}(\frac{\kappa_2}{m_2})^{1/3}\|x_i'\|\}$$

$$+ \frac{\gamma_1}{\sqrt{m_2}}\sqrt{\sum_j \mathbb{1}\{\|V_{j,} - V_{j,}^{(0)}\|^2 \geq \gamma_2^{4/3}(\frac{\kappa_2}{m_2})^{2/3}\}}\sqrt{\sum_j \|(V_{j,} - V_{j,}^{(0)})\|^2}$$

$$\leq \frac{\gamma_2^{2/3}(\frac{\kappa_2}{m_2})^{1/3}\gamma_1}{\sqrt{m_2}}\Big(\sum_j \mathbb{1}\{|V_{j,}^{(0)}x_i'| \leq \gamma_2^{2/3}(\frac{\kappa_2}{m_2})^{1/3}\|x_i'\|\}\Big) + \frac{\gamma_2^2\gamma_1}{\sqrt{m_2}} \times (\frac{m_2}{\kappa_2})^{1/3}\frac{1}{\gamma_2^{2/3}}.$$

Then, applying the first argument of Lemma 29, we have with high probability over the randomness of $V^{(0)}$:

$$LHS \leq \frac{\gamma_2^{2/3}(\frac{\kappa_2}{m_2})^{1/3}\gamma_1}{\sqrt{m_2}}(\frac{m_2}{\kappa_2}(\frac{\kappa_2}{m_2})^{1/3}\gamma_2^{2/3}) + \frac{\gamma_2^2\gamma_1}{\sqrt{m_2}} \times (\frac{m_2}{\kappa_2})^{1/3}\frac{1}{\gamma_2^{2/3}}$$

$$\leq \frac{\gamma_2^{4/3}\gamma_1}{(\kappa_2\sqrt{m_2})^{1/3}} + \frac{\gamma_2^{4/3}\gamma_1}{(\kappa_2\sqrt{m_2})^{1/3}}$$

$$\lesssim \frac{\gamma_2^{4/3}\gamma_1}{(\kappa_2\sqrt{m_2})^{1/3}}.$$

Therefore, we can write:

$$\mathcal{R}(\mathcal{G}_{\gamma_1,\gamma_2})|_{(x_i),(y_i)} = \frac{1}{n}\mathbb{E}_\epsilon \sup_{V,W\in S}\sum_{t=1}^{n}\epsilon_i f_{V,W}(x_i)$$

$$= \frac{1}{n}\mathbb{E}_\epsilon \sup_{V,W\in S}\sum_{i=1}^{n}\epsilon_i a^T\sigma(1/\sqrt{m_2}VW^s\sigma(1/\sqrt{m_1}Wx_i))$$

$$= \frac{1}{n}\mathbb{E}_\epsilon \sup_{V,W\in S}\sum_{i=1}^{n}\epsilon_i a^T\sigma(1/\sqrt{m_2}Vx_i')$$

$$= \frac{1}{n}\mathbb{E}_\epsilon \sup_{W\in S}\sup_{V\in S}\sum_{i=1}^{n}\epsilon_i \mathrm{trace}(V(Z_i'+Z_i'^+))$$

$$\lesssim \frac{1}{n}\mathbb{E}_\epsilon \sup_{W,V\in S}\sum_{i=1}^{n}\epsilon_i \mathrm{trace}(VZ_i') + \frac{\gamma_2^{4/3}\gamma_1}{(\kappa_2\sqrt{m_2})^{1/3}}$$

$$\leq \frac{1}{n}\mathbb{E}_\epsilon \sup_{W\in S}\sup_{V\in S}\mathrm{trace}(V(\sum_{i=1}^{n}\epsilon_i Z_i')) + \frac{\gamma_2^{4/3}\gamma_1}{(\kappa_2\sqrt{m_2})^{1/3}}$$

$$= \frac{1}{n}\mathbb{E}_\epsilon \sup_{W\in S}\sup_{V\in S}\mathrm{trace}((V-V^{(0)})(\sum_{i=1}^{n}\epsilon_i Z_i'))$$

$$+ \frac{1}{n}\mathbb{E}_\epsilon \sup_{W\in S}\mathrm{trace}(V^{(0)}(\sum_{i=1}^{n}\epsilon_i Z_i')) + \frac{\gamma_2^{4/3}\gamma_1}{(\kappa_2\sqrt{m_2})^{1/3}}. \tag{80}$$

For the first term, for every $j\in[m_2]$, define $H_j$ to be the set of $i$'s in $[n]$ where the $j$th column of $Z_i'$ is non-zero, i.e.

$$H_j = \{i\in[n]: \ V_j^{(0)}x_i'\geq 0\}.$$

Here, we use the crucial assumption that $(V-V^{(0)})_j^T\phi^{(0)}(x_i) = 0$, so we can drop the $\phi^{(0)}(x_i)$ term when $x_i'$ is multiplied to $V-V^{(0)}$. Using this trick and applying Cauchy Schwarz, we bound the first term as:

$$\frac{1}{n}\mathbb{E}_\epsilon \sup_{W,V\in S}\mathrm{trace}((V-V^{(0)})(\sum_{i=1}^{n}\epsilon_i Z_i'))$$

$$\leq \frac{1}{n}\mathbb{E}_\epsilon\|V-V^{(0)}\|\sup_{W\in S}\sqrt{\frac{1}{m_2}\sum_{j=1}^{m_2}\|\sum_{i\in H_j}\epsilon_i\phi^{(2)}(x_i)\|^2}.$$

Further using Jensen's inequality:

$$\frac{1}{n}\mathbb{E}_\epsilon \sup_{W,V\in S}\mathrm{trace}((V-V^{(0)})(\sum_{i=1}^{n}\epsilon_i Z_i'))$$

$$\leq \frac{\|V-V^{(0)}\|}{n}\sqrt{\mathbb{E}_\epsilon\frac{1}{m_2}\sum_{j=1}^{m_2}\sup_{W\in S}\|\sum_{i\in H_j}\epsilon_i\phi^{(2)}(x_i)\|^2}. \tag{81}$$

Using Equation (110) of Lemma 6 (note that we do not have the smoothing matrix $W^\rho$ here, so we are allowed to use this result), we obtain

$$\|\langle W-W^{(0)}, Z_i^k\rangle - \phi'(x_i)\| \lesssim \frac{2C_1^{3/2}}{\sqrt{\kappa_1}}(\frac{n^3 m_3^3}{m_1\lambda_0})^{1/4},$$

where $Z_i^k$'s are defined in Equation (317).

Plugging this back in (81):

$$\mathbb{E}_\epsilon \sup_{W \in S} \| \sum_{i \in H_j} \epsilon_i \phi'(x_i) \|^2$$

$$\leq \mathbb{E}_\epsilon \sup_{W \in S} \Big( \| \sum_{i \in H_j} \epsilon_i \langle W - W^{(0)}, Z_i^k \rangle \| + \frac{2C_1^{3/2}}{\sqrt{\kappa_1}} (\frac{n^3 m_3^3}{m_1 \lambda_0})^{1/4} \Big)^2$$

$$\lesssim \mathbb{E}_\epsilon \sup_{W \in S} \| \sum_{i \in H_j} \epsilon_i \langle W - W^{(0)}, Z_i^k \rangle \|^2 + \frac{C_1^3}{\kappa_1} (\frac{n^3 m_3^3}{m_1 \lambda_0})^{1/2}$$

$$= \mathbb{E}_\epsilon \sup_{W \in S} \sum_{k=1}^{m_3} \Big( \text{trace}((W - W^{(0)})(\sum_{i \in H_j} \epsilon_i Z_i^k)) \Big)^2 + \frac{C_1^3}{\kappa_1} (\frac{n^3 m_3^3}{m_1 \lambda_0})^{1/2}. \tag{82}$$

Now for every fixed dataset, with high probability over the randomness of $W^s$, for every $k_1 \neq k_2$:

$$\Big| \langle \sum_{i \in H_j} \epsilon_i Z_i^{k_1}, \sum_{i \in H_j} \epsilon_i Z_i^{k_2} \rangle \Big| \leq \sum_{i_1, i_2 \in H_j} \Big| \langle Z_{i_1}^{k_1}, Z_{i_2}^{k_2} \rangle \Big|$$

$$= \frac{1}{m_1} \sum_{i_1, i_2 \in H_j} \Big| \sum_{j=1}^{m_1} W_{k_1,j}^s W_{k_2,j}^s \langle x_{i_1}, x_{i_2} \rangle \mathbb{1}\{W_{j,}^{(0)} x_{i_1} \geq 0\} \mathbb{1}\{W_{j,}^{(0)} x_{i_2} \geq 0\} \Big|$$

But note that because $\langle x_{i_1}, x_{i_2} \rangle \leq 1$, the variables $W_{k_1,j}^s W_{k_2,j}^s \langle x_{i_1}, x_{i_2} \rangle \mathbb{1}\{W_{j,}^{(0)} x_{i_1} \geq 0\} \mathbb{1}\{W_{j,}^{(0)} x_{i_2} \geq 0\}$ are subgaussian with parameter one with respect to the randomness of $W^s$. Hence, with high probability over the randomness of $W^s$, we get

$$\Big| \langle \sum_{i \in H_j} \epsilon_i Z_i^{k_1}, \sum_{i \in H_j} \epsilon_i Z_i^{k_2} \rangle \Big| \lesssim \frac{1}{m_1} \sum_{i_1, i_2 \in H_j} \sqrt{m_1} \leq \frac{n^2}{\sqrt{m_1}}. \tag{83}$$

Therefore, with high probability over the randomness of $W^{(0)}$ and $W'$ and the dataset, we get Equation (83). In order to get rid of the high probability argument on the dataset, we use the stronger Equation (318) in Lemma 46 which uniformly bounds $\langle Z_{k_1}(x), Z_{k_2}(x') \rangle$ by $\log(m_1)d/m_1$ for any $x, x'$, which in turn gives

$$\Big| \langle \sum_{i \in H_j} \epsilon_i Z_i^{k_1}, \sum_{i \in H_j} \epsilon_i Z_i^{k_2} \rangle \Big| \leq \sum_{i_1, i_2 \in H_j} \Big| \langle Z_{i_1}^{k_1}, Z_{i_2}^{k_2} \rangle \Big| \lesssim \frac{n^2 d \log(m_1)}{\sqrt{m_1}},$$

with high probability, independent of the choice of dataset. This bounds is slightly worse comapred to (83), but still efficient for our purpose.

Furthermore, a similar bound to Equation (83) can be obtained in a more adversarial situation when we also take maximum against the choice of the dataset.

Note that the entries of $\sum_{i \in H_j} \epsilon_i Z_i^k$ for $1 \leq k \leq m_3$ can differ only in a sign. Therefore, their norms are all equal. Now suppose that $\mathcal{C}_j$ is the random variable of the norm of these variables:

$$\mathcal{C}_j := \| \sum_{i \in H_j} \epsilon_i Z_i^k \|.$$

Then, by substituting $r_k = \frac{1}{\mathcal{C}_j} \sum_{i \in H_j} \epsilon_i Z_i^k$ in Lemma 40, we get

$$\sum_{k=1}^{m_3} \Big( \text{trace}((W - W^{(0)})(\sum_{i \in H_j} \epsilon_i Z_i^k)) \Big)^2 \leq \mathcal{C}_j^2 (1 + m_3^2 O(\frac{n^2 d \log(m_1)}{\sqrt{m_1} \mathcal{C}_j^2})) \| W - W^{(0)} \|_F^2 \tag{84}$$

$$= (\mathcal{C}_j^2 + \frac{n^2 m_3^2 d \log(m_1)}{\sqrt{m_1}}) \| W - W^{(0)} \|_F^2. \tag{85}$$

Now recall from Equation (79), we have

$$\| \phi'(x_i) \| \leq \gamma_1. \tag{86}$$

Hence, we can apply Corollary 5.1 with $\phi^{(2)}(x_i)$ and $C_1$ substituted by $\phi'(x_i)$ and $\gamma_1$ respectively, to argue with high probability over the initialization, there exists a set $\tilde{P}_i$ such that for every $i \in [n]$ and $j \notin \tilde{P}_i$, sign of $V_j^T x_i'$ is the same as $V_j^{(0)T} \phi^{(0)}(x_i)$, and moreover,

$$|\tilde{P}_i| \lesssim \left(\frac{C_1^2}{(m_3 \kappa_1^2)}\right)^{1/3} m_2.$$

Now let

$$\bar{H}_j = \{i \in [n]: \ V_j^{(0)} \phi^{(0)}(x_i) \geq 0\}.$$

Note that for $j \notin \tilde{P} = \bigcup_i \tilde{P}_i$, we have $H_j = \bar{H}_j$. Now note that the norm of each $\sum_{i \in H_j} \epsilon_i Z_i^k$ is at most one. for each index $1 \leq \ell \leq m_1 d$, as the random variables $\sum_{i \in \bar{H}_j} \epsilon_i (Z_i^k)_\ell$ are $\sum_{i \in \bar{H}_j} (Z_i^k)_\ell^2 \leq \sum_{i \in [n]} (Z_i^k)_\ell^2$ subgaussian, we have with probability at least $1 - \frac{1}{n}$ over the randomness of $\epsilon_i$'s, for every $1 \leq \ell \leq m_1 d$ and every $1 \leq j \leq m_2$:

$$\Big| \sum_{i \in \bar{H}_j} \epsilon_i (Z_i^k)_\ell \Big| \leq \sqrt{\sum_{i \in [n]} (Z_i^k)_\ell^2 \log(m_1 d m_2 n)},$$

which implies for every $j \in [m_2]$:

$$\| \sum_{i \in \bar{H}_j} \epsilon_i Z_i^k \|^2 \leq \sum_\ell \sum_{i \in [n]} (Z_i^k)_\ell^2 \log(m_1 d) \leq n \log(m_1 d m_2 n).$$

Name this event $\mathcal{B}$, so

$$\mathbb{P}(\mathcal{B}) \leq \frac{1}{n}.$$

Note that although $H_j$ might depend on the randomness of $\epsilon_i$'s, $\bar{H}_j$ does not, and if $j \notin \tilde{P}$, we obtain

$$\mathcal{C}_j = \| \sum_{i \in \bar{H}_j} \epsilon_i Z_i^k \| \leq \sqrt{n \log(m_1 d m_2 n)}.$$

Moreover, note that we also have the following worse case bound:

$$\mathcal{C}_j = \| \sum_{i \in H_j} \epsilon_i Z_i^k \| \leq \sum_{i \in \bar{H}_j} \| Z_i^k \| \leq n.$$

Applying the last two inequalities into Equations (297) and (85):

$$\mathbb{E}_\epsilon \frac{1}{m_2} \sum_{j=1}^{m_2} \sup_{W \in S} \| \sum_{i \in H_j} \epsilon_i \phi'(x_i) \|^2$$

$$\leq \frac{C_1^3}{\kappa_1} \left(\frac{n^3 m_3^3}{m_1 \lambda_0}\right)^{1/2} + \frac{1}{m_2} \sum_{j=1}^{m_2} \mathbb{E}_\epsilon \sup_{W \in S} \sum_{k=1}^{m_3} \left( \text{trace}((W - W^{(0)})(\sum_{i \in H_j} \epsilon_i Z_i^k)) \right)^2$$

$$\leq \frac{C_1^3}{\kappa_1} \left(\frac{n^3 m_3^3}{m_1 \lambda_0}\right)^{1/2} + \frac{1}{m_2} \mathbb{E}_\epsilon \mathbb{1}\{\mathcal{B}\} \sum_{j \in \tilde{P}} (C_j^2 + \frac{n^2 m_3^2 d \log(m_1)}{\sqrt{m_1}}) \|W - W^{(0)}\|_F^2$$

$$+ \frac{1}{m_2} \mathbb{E}_\epsilon \mathbb{1}\{\mathcal{B}\} \sum_{j \notin \tilde{P}} (C_j^2 + \frac{n^2 m_3^2 d \log(m_1)}{\sqrt{m_1}}) \|W - W^{(0)}\|_F^2$$

$$+ \frac{1}{m_2} \mathbb{E}_\epsilon \mathbb{1}\{\mathcal{B}^c\} \sum_{j=1}^{m_2} (C_j^2 + \frac{n^2 m_3^2 d \log(m_1)}{\sqrt{m_1}}) \|W - W^{(0)}\|_F^2$$

$$\leq \frac{C_1^3}{\kappa_1} \left(\frac{n^3 m_3^3}{m_1 \lambda_0}\right)^{1/2} + \|W - W^{(0)}\|_F^2 \left[\frac{|\tilde{P}|}{m_2}\left(n^2 + \frac{n^2 m_3^2 d \log(m_1)}{\sqrt{m_1}}\right) + 2\left(n + \frac{n^2 m_3^2 d \log(m_1)}{\sqrt{m_1}}\right)\right]$$

$$\leq \frac{C_1^3}{\kappa_1} \left(\frac{n^3 m_3^3}{m_1 \lambda_0}\right)^{1/2} + \gamma_1^2 \left[n^3 \left(\frac{C_1^2}{(m_3 \kappa_1^2)}\right)^{1/3} + \left(\frac{C_1^2}{(m_3 \kappa_1^2)}\right)^{1/3} \frac{n^3 m_3^2 d \log(m_1)}{\sqrt{m_1}} + 2\frac{n^2 m_3^2 d \log(m_1)}{\sqrt{m_1}} + 2n\right].$$

$$(87)$$

Next, we analyze the term $\frac{1}{n}\mathbb{E}_\epsilon \sup_{W\in S} \text{trace}(V^{(0)}(\sum_{i=1}^n \epsilon_i Z_i'))$. Noting that $\|\phi^{(0)}(x_i)\| \lesssim \kappa_1\sqrt{m_3}$ with high probability and using Equation (86):

$$\|\sum_{i=1}^n \epsilon_i Z_i'\|_F \leq \sum_{i=1}^n \|Z_i'\|_F \leq \sum_{i=1}^n \|x_i'\| \leq \sum_i (\|\phi'(x_i)\| + \|\phi^{(0)}(x_i)\|) \lesssim n(\sqrt{m_3}\kappa_1 + \gamma_1). \quad (88)$$

Hence

$$\frac{1}{n}\mathbb{E}_\epsilon \sup_{W\in S} \text{trace}(V^{(0)}(\sum_{i=1}^n \epsilon_i Z_i')) = \frac{1}{n}\mathbb{E}_\epsilon \sup_{W\in S} \sum_{i=1}^n \epsilon_i \text{trace}(V^{(0)} Z_i')$$

$$= \frac{1}{n}\mathbb{E}_\epsilon \sup_{W\in S} \sum_{i=1}^n \epsilon_i \Big(\frac{1}{\sqrt{m_2}} \sum_{j=1}^{m_2} a_j V_{j,}^{(0)} x_i' \mathbb{1}\{V_{j,}^{(0)} x_i' \geq 0\}\Big).$$

$$= \frac{1}{n}\mathbb{E}_\epsilon \sup_{W\in S} \sum_{i=1}^n \epsilon_i \frac{1}{\sqrt{m_2}} a^T \sigma(V^{(0)} x_i') \leq \sup_{W\in S} \frac{1}{\sqrt{m_2}} a^T \sigma(V^{(0)} x_i').$$

But using Lemma 30:

$$LHS \lesssim \kappa_2\sqrt{m_3}\|x_i'\|.$$

Applying a similar bound as we did in Equation (88) on $\|x_i'\|$:

$$\|x_i'\| \leq \|\phi^{(0)}(x_i)\| + \|\phi'(x_i)\| \lesssim \kappa_1\sqrt{m_3} + \gamma_1.$$

Substituting above, we get

$$\frac{1}{n}\mathbb{E}_\epsilon \sup_{W\in S} \text{trace}(V^{(0)}(\sum_{i=1}^n \epsilon_i Z_i')) \leq \kappa_2\sqrt{m_3}(\kappa_1\sqrt{m_3} + \gamma_1). \quad (89)$$

Finally, Substituting Equations (87) into (81), then combining it with (89) into (80), we obtain a bound on Rademacher complexity which holds w.h.p over both the randomness of the initialization and the dataset:

$$\mathcal{R}(\mathcal{G}_{\gamma_1,\gamma_2})|_{x,y} \lesssim \sqrt{\frac{C_1^3}{\kappa_1}\Big(\frac{n^3 m_3^3}{m_1 \lambda_0}\Big)^{1/2}} \quad (90)$$

$$+ \frac{\gamma_1\gamma_2}{n}\sqrt{n^3\Big(\frac{C_1^2}{(m_3\kappa_1^2)}\Big)^{1/3} + \Big(\frac{C_1^2}{(m_3\kappa_1^2)}\Big)^{1/3}\frac{n^3 m_3^2 d\log(m_1)}{\sqrt{m_1}} + 2\frac{n^2 m_3^2 d\log(m_1)}{\sqrt{m_1}} + 2n} \quad (91)$$

$$+ \kappa_2\sqrt{m_3}(\kappa_1\sqrt{m_3} + \gamma_1) + \frac{\gamma_2^{4/3}\gamma_1}{(\kappa_2\sqrt{m_2})^{1/3}}.$$

Having enough overparameterization, we have for every dataset $(x, y)$ (i.e. worst-case Rademacher complexity):

$$\mathcal{R}(\mathcal{G}_{\gamma_1,\gamma_2})|_{x,y} \leq 2\gamma_1\gamma_2/\sqrt{n}. \quad (92)$$

Note that for the bound (92) to hold, the overparameterization should be picked poly large in $\gamma_1, \gamma_2$, as well as in other basic parameters. However, noting Equations (49) and (56) in the proof of Theorem 3, we set $\gamma_1 = 1, \gamma_2 \geq \Omega(B, n, 1/\gamma_0)$ in Theorem 3, so $\gamma_1\gamma_2$ is at most poly in the basic parameters. Therefore, again the overparameterization can be picked polynomially large in the basic only parameters (i.e. **independent** of $\gamma_1, \gamma_2$ or $\zeta$).

## A.12   Constructing $W^*, V^*$

This section consists of two subsections; First, we prove a structural result for the first layer weights $(W', V')$ that the algorithm visits, then construct a weight matrix $W^*$ for the first layer with some good properties. Second, we do the same thing for the second layer (however, the structure of the first and second layers are completely different). Through out this section, we assume we have the norm bounds $\|W'\| \leq C_1, \|V'\| \leq C_2$.

Notably, we rely on a number of basic Lemmas more related to the representation power of the network, which we defer their proof into a later Appendix B.2 and refer to them here as needed.

### A.12.1   First Layer, Construction of $W^*$

**Lemma 1** *Suppose $m_1 \geq 16n^2 m_3^2/\lambda_0^2$. Let $P_i = \{j \in [m_1] | |W_j^{(0)} x_i| \leq c_2/\sqrt{m_1}\}$ and $P = \cup P_i$. During SGD iterations, suppose we have $\|W'\|_F \leq C_1$. Then, for a value $c_2$ satisfying*

$$2C_1\sqrt{nm_3}/\sqrt{\lambda_0} \leq c_2 \leq \kappa_1 \lambda_0 \sqrt{m_1}/(2n^2),$$

*with high probability $\forall i$:*

$$|P_i| \lesssim c_2\sqrt{m_1}/\kappa_1,$$

*and for $j \notin P$, during the whole algorithm we have*

$$\|W_j'\| \leq \frac{\sqrt{nm_3}C_1}{\sqrt{m_1\lambda_0}} + c_2/(4\sqrt{m_1}) \leq c_2/(2\sqrt{m_1}),$$

$$c_2/\sqrt{m_2} \leq |W_j^{(0)} x_i|.$$

*So the signs of neurons outside $P$ never changes. In particular, we can set $c_2$ as small as $c_2 = C_1\sqrt{nm_3}/\sqrt{\lambda_0}$. In the rest of the proof (i.e. other sections), we set $c_2$ to this value.*

**Proof of Lemma 1**

Define the matrix

$$\tilde{Z}_k^i = \frac{1}{\sqrt{m_1}}(W_{k,j}^s x_i \mathbb{1}\{\forall i : W_j^{(0)T} x_i \geq c_2/\sqrt{m_1}\})_{j=1}^n.$$

Let $P_i$ be the set of indices $j$ such that $\mathbb{1}\{W_j^{(0)T} x_i \geq c_2/\sqrt{m_1}\}$ is zero. First of all, note that by Bernstein inequality:

$$|P_i| \leq c_2\sqrt{m_1}/\kappa_1 + O(\sqrt{c_2\sqrt{m_1}/\kappa_1} + 1) \lesssim c_2\sqrt{m_1}/\kappa_1.$$

Now suppose that until the current iteration of the algorithm the assumption has been true, i.e. the signs of the neurons outside of $P$ have never changed. As a result, due to the specific update of the SGD for both of the terms $\mathbb{E}_Z \ell(f_{V',W'}(x), y)$ and $\|W'\|_F^2$, if we define $W'|_P$ to be the restriction of $W'$ to indices that are not in $P$ (i.e. the columns in $P$ are equal to zero), then we can write

$$W'|_P^T = \sum_{k=1}^{m_3} \sum_{i=1}^{n} \alpha_{k,i} \tilde{Z}_k^i. \tag{93}$$

An issue here is that we also have some injected noise by PSGD into $W'$ which violates Equation (93). To handle the injected noise as well, we define the subspace $\Phi'$ of $\mathbb{R}^{m_1 \times d}$ matrices to be the set of vectors with arbitrary rows for $j \in [m_1]$ with $j \in P$, while restricted to the other rows $j \notin P$ in should be in the span of $(\tilde{Z}_k^i)_{i,k}$. Then, we decompose $W'$ into subspaces $\Phi'$ and $\Phi'^\perp$ respectively as $W' = W'^{(1)} + W'^{(2)}$, where $W'^{(1)} \in \Phi', W'^{(2)} \in \Phi'^\perp$. Here, we want to prove $\|W_j'^{(1)}\| \leq c_2/(4\sqrt{m_1})$. We handle the $\|W_j'^{(2)}\|$ part in Appendix 45. So instead of $W'|_P$ in Equation (93) we consider $W'^{(1)}|_P$:

$$W'^{(1)}|_P^T = \sum_{k=1}^{m_3} \sum_{i=1}^{n} \alpha_{k,i} \tilde{Z}_k^i. \tag{94}$$

We handle the other part $W'^{(2)}$ in Appendix 45. Now exactly similar to the drivation in Lemma 38, we can state with high probability

$$C_1^2 \geq \|W'\|^2 \geq \|W'^{(1)}\|^2$$
$$\geq \|W'^{(1)}|_P\|^2 \geq \sum_{k=1}^{m_3} \|\sum_{i=1}^{n} \alpha_{k,i}\tilde{Z}_k^i\|^2 - O(nm_3/\sqrt{m_1})\sum_k \|\alpha_k\|^2. \tag{95}$$

Note that we are exploiting the fact that the norm $\|W'\|$ remains bounded by $C_1$. Now using a Hoeffding bound for matrix $H^{\infty'}$ defined below, we write:

$$H^{\infty'}_{i_1,i_2} := \mathbb{E}_{w:\mathcal{N}(0,\mathbb{R}^d)}\mathbb{1}\{\forall i : |w^Tx_i| \geq c_2/\sqrt{m_1}\}x_{i_1}^Tx_{i_2}(\mathbb{1}\{w^Tx_{i_1} \geq 0\}\mathbb{1}\{w^Tx_{i_2} \geq 0\})$$
$$= \mathbb{E}_{w:\mathcal{N}(0,\mathbb{R}^d)}(\mathbb{1}\{w^Tx_{i_1} \geq 0\}\mathbb{1}\{w^Tx_{i_2} \geq 0\})x_{i_1}^Tx_{i_2}$$
$$\pm O(\mathbb{E}\mathbb{1}\{\exists i : |w^Tx_i| \leq c_2/\sqrt{m_1}\}(\mathbb{1}\{w^Tx_{i_1} \geq 0\}\mathbb{1}\{w^Tx_{i_2} \geq 0\}))x_{i_1}^Tx_{i_2}$$
$$= H^{\infty}_{i_1,i_2} \pm O(nc_2/(\sqrt{m_1}\kappa_1)\|x_{i_1}\|\|x_{i_2}\|)$$
$$= H^{\infty}_{i_1,i_2} \pm O(nc_2/(\sqrt{m_1}\kappa_1)). \tag{96}$$

Now opening Equation (95) and using the property $c_2 \leq k_1\lambda_0\sqrt{m_1}/(2n^2)$, we get

$$LHS = \sum_k \sum_{i_1,i_2} \alpha_{k,i_1}\alpha_{k,i_2}\langle \tilde{Z}_k^{i_1}, \tilde{Z}_k^{i_2}\rangle - O(nm_3/\sqrt{m_1}\sum_k \|\alpha_k\|^2)$$
$$= \sum_k \sum_{i_1,i_2} \alpha_{k,i_1}\alpha_{k,i_2}(H^{\infty'}_{i_1,i_2} \pm O(1/\sqrt{m_1})) - O(nm_3/\sqrt{m_1}\sum_k \|\alpha_k\|^2)$$
$$\geq \sum_k \sum_{i_1,i_2} \alpha_{k,i_1}\alpha_{k,i_2}H^{\infty}_{i_1,i_2} \pm \|\alpha_k\|_1^2 O(nc_2/\sqrt{m_1}\kappa_1) - O(nm_3/\sqrt{m_1}\sum_k \|\alpha_k\|^2)$$
$$\geq \sum_k \alpha_k^T H^{\infty}\alpha_k - O(nc_2/\sqrt{m_1}\kappa_1)\sum_k \|\alpha_k\|_1^2 - O(nm_3/\sqrt{m_1}\sum_k \|\alpha_k\|^2)$$
$$\geq \sum_k \alpha_k^T H^{\infty}\alpha_k - O(c_2n^2/\sqrt{m_1}\kappa_1)\sum_k \|\alpha_k\|_2^2 - O(nm_3/\sqrt{m_1}\sum_k \|\alpha_k\|^2)$$
$$= (\lambda_0 - O(nm_3/\sqrt{m_1}) - O(c_2n^2/\sqrt{m_1}\kappa_1))\sum_k \|\alpha_k\|^2$$
$$\geq \lambda_0/2\sum_k \|\alpha_k\|^2.$$

For the last line to hold, we need enough overparameterization. This implies

$$\sum_k \|\alpha_k\|^2 \lesssim C_1^2/\lambda_0.$$

Now again, exactly similar to the derivation in Lemma 38, for $j \notin P$ we have

$$\|W'^{(1)}_j\| \leq \sqrt{nm_3}/\sqrt{m_1}\sqrt{\sum_k \|\alpha_k\|^2} \lesssim \sqrt{m_3n}C_1/\sqrt{m_1\lambda_0},$$

which completes most of the proof. For the rest, we are left to show that for the other part $W'^{(2)}$, we have $\|W'^{(2)}_j\| \leq c_2/(4\sqrt{m_1})$, which we do in Appendix 45.

**Lemma 2** *Under condition $m_3n/\sqrt{m_1} \leq \lambda_0/4$, there exist matrices $\{W_k^*\}_{k=1}^{m_3} \in \mathbb{R}^{m_1 \times d}$ s.t. for every $k \neq k' \in [m_3]$ and $i \in [n]$:*

$$\langle W_k^*, Z_{k'}^i\rangle = 0,$$
$$\|W_k^* - W_k^+\| \lesssim \frac{n\sqrt{m_3}}{\lambda_0\sqrt{m_1}}\|\mathcal{V}_k\|_{H^{\infty}},$$
$$|\langle W_k^*, Z_k^i\rangle - \langle W_k^+, Z_k^i\rangle| \lesssim \frac{n\sqrt{m_3}}{\lambda_0\sqrt{m_1}}\|\mathcal{V}_k\|_{H^{\infty}}.$$

*Furthermore, for $k_1 \neq k_2$:*

$$|\langle W_{k_1}^*, W_{k_2}^* \rangle| \leq \frac{n}{\lambda_0^2} \frac{\sqrt{m_3}}{\sqrt{m_1}} (1 + \frac{\sqrt{m_3}}{\sqrt{m_1}}) \|\mathcal{V}_{k_1}\|_{H^\infty} \|\mathcal{V}_{k_2}\|_{H^\infty}. \tag{97}$$

**Proof of Lemma 2**

Let

$$W_k^+ = \sum_i \mathcal{V}_{k,i} Z_k^i.$$

we want to compute the norm of the projection $P(W_k^+)$ of $W_k^+$ onto the subspace spanned by all $Z_{k'}^i$ for $k' \neq k$ and $i \in [n]$:

$$\|P(W_k^+)\|^2 = (\langle W_k^+, Z_{k'}^i \rangle)_{k' \neq k, i \in [n]}^T \left( \langle Z_{k_1}^{i_1}, Z_{k_2}^{i_2} \rangle \right)_{(k_1,i_1),(k_2,i_2) \in [m_3]-\{k\} \times [n]}^{-1} (\langle W_k^+, Z_{k'}^i \rangle)_{k' \neq k, i \in [n]}, \tag{98}$$

where the first and third terms are vectors and the middle term is a matrix. Now note that for each $k', k_1, k_2 \neq k$, by Hoeffding inequality:

$$\left( \langle Z_{k'}^{i_1}, Z_{k'}^{i_2} \rangle \right)_{i_1, i_2 \in [n]} = H^\infty + (\pm 1/\sqrt{m_1})_{i_1, i_2 \in [n]}, \tag{99}$$

$$\langle W_k^+, Z_{k'}^i \rangle = \langle \sum_i \mathcal{V}_{k,i} Z_k^i, Z_{k'}^i \rangle \lesssim \frac{1}{\sqrt{m_1}} \sum_i |\mathcal{V}_{k,i}|$$

$$\leq \frac{\sqrt{n}}{\sqrt{m_1}} \|\mathcal{V}_k\|$$

$$\leq \frac{\sqrt{n}}{\sqrt{m_1} \lambda_0} \|\mathcal{V}_k\|_{H^\infty}. \tag{100}$$

Therefore,

$$\|(\langle W_k^+, Z_{k'}^i \rangle)_{k' \neq k, i \in [n]}^T\| \leq n \sqrt{\frac{m_3}{m_1 \lambda_0}} \|\mathcal{V}_k\|_{H^\infty}. \tag{101}$$

Now Equation (99) implies for small enough $m_1$

$$\lambda_{min}\left( \left( \langle Z_{k'}^{i_1}, Z_{k'}^{i_2} \rangle \right)_{i_1, i_2 \in [n]} \right) \geq \lambda_0/2, \tag{102}$$

as long as $\lambda_0 \geq 2n/m_1$. Moreover, define $\tilde{A}$ to be the block version of

$$A' = \left( \langle Z_{k_1}^{i_1}, Z_{k_2}^{i_2} \rangle \right)_{(k_1,i_1),(k_2,i_2) \in [m_3]-\{k\} \times [n]}^{-1},$$

i.e. for $k_1 = k_2$ they are the same but for $k_1 \neq k_2$ $\tilde{A}$ is zero. Then

$$\lambda_{min}(\tilde{A}) \geq \lambda_0/2,$$

because the eigenvalues of each block is at least $\lambda_0/2$ using Equation (102). But note that

$$\|A' - \tilde{A}\|_2 \leq \|A' - \tilde{A}\|_F \leq m_3 n/\sqrt{m_1}.$$

So as long as $m_3 n/\sqrt{m_1} \leq \lambda_0/4$, we have $\lambda_{min}(A) \geq \lambda_0/4$. Combining this fact with Equation (101) and plugging it into Equation (98), we obtain

$$\|P(W_k^+)\|^2 \lesssim \frac{n^2 m_3}{m_1 \lambda_0^2} \|\mathcal{V}_k\|_{H^\infty}^2.$$

Now define $W_k^* = W_k^+ - P(W_k^+)$. Then

$$\|W_k^* - W_k^+\| = \|P(W_k^+)\| \lesssim \frac{n\sqrt{m_3}}{\lambda_0\sqrt{m_1}}\|\mathcal{V}_k\|_{H^\infty},$$

$$|\langle W_k^* - W_k^+, Z_k^i\rangle| \le \|P(W_k^+)\|\|Z_k^i\| \lesssim \frac{n\sqrt{m_3}}{\lambda_0\sqrt{m_1}}\|\mathcal{V}_k\|_{H^\infty}.$$

Furthermore, note that $W_{k_2}^*$ is orthogonal to $W_{k_1}^+$ for $k_1 \neq k_2$, so

$$\begin{aligned}
|\langle W_{k_1}^*, W_{k_2}^*\rangle| &= |\langle W_{k_1}^+ - P(W_{k_1}^+), W_{k_2}^*\rangle| \\
&= |\langle P(W_{k_1}^+), W_{k_2}^+ - P(W_{k_2}^+)\rangle| \\
&\le \|P(W_{k_1}^+)\|\|W_{k_2}^+ - P(W_{k_2}^+)\| \\
&\le \|P(W_{k_1}^+)\|(\|W_{k_2}^+\| + \|P(W_{k_2}^+)\|).
\end{aligned} \tag{103}$$

But note that

$$\|W_k^+\| = \|\sum_i \mathcal{V}_{k,i}Z_k^i\| \le \sum_i |\mathcal{V}_{k,i}|\|Z_k^i\| \le \sum_i |\mathcal{V}_{k,i}| \le \sqrt{n}\|\mathcal{V}_k\|_2. \le \frac{\sqrt{n}}{\lambda_0}\|\mathcal{V}_k\|_{H^\infty}.$$

Therefore, we can bound Equation (103) as:

$$|\langle W_{k_1}^*, W_{k_2}^*\rangle| \le \frac{n}{\lambda_0^2}\frac{\sqrt{m_3}}{\sqrt{m_1}}(1 + \frac{\sqrt{m_3}}{\sqrt{m_1}})\|\mathcal{V}_{k_1}\|_{H^\infty}\|\mathcal{V}_{k_2}\|_{H^\infty}.$$

**Lemma 3** *There exists a matrix $W_k^{+2}$ such that for every $j \in P$, $W_{k\ j}^{+2} = 0$, and*

$$|trace(W_k^{+2}Z_i^k) - \bar{x}_{i,k}| \le C_1\sqrt{m_3}n^2/(\lambda_0\kappa_1\sqrt{m_1})\|\mathcal{V}_k\|_{H^\infty}.$$

**Proof of Lemma 3**

Define $W_k^{+2}$ to be equal to $W_k^+$ for $j \notin P$ and equal to zero vector otherwise. Then, by Lemma 38: (note that $|P_i| \le C_1\sqrt{nm_3}\sqrt{m_1}/(\sqrt{\lambda_0}\kappa_1)$)

$$\begin{aligned}
|\text{trace}(W_k^+Z_i^k) - \text{trace}(W_k^{+2}Z_i^k)| &\le 1/\sqrt{m_1}\sum_{j\in P}|W_{k\ j}^+x_i| \\
&\le \frac{|P|}{\sqrt{m_1}}\|W_k^+\| \\
&\le \sqrt{nm_3}/(m_1\sqrt{\lambda_0})\,|P|\|\mathcal{V}_k\|_{H^\infty} \\
&\le C_1m_3n^2/((\lambda_0\kappa_1\sqrt{m_1})\,\|\mathcal{V}_k\|_{H^\infty}.
\end{aligned}$$

Combining this with Lemma 37, the desired result follows.

**Lemma 4** *Under condition $m_3n/\sqrt{m_1} \le \lambda_0/4$, there exist matrix $W_k^*$'s exactly satisfying the same conditions in Lemma 2 but with respect to $W_k^{+2}$ instead of $W_k^+$, and moreover, for $j \in P$ we have $W_{k\ j}^* = 0$.*

**Proof of Lemma 4**

We can repeat the exact same procedure of Lemma 2 for $W_k^{+2}$. Using the bound in Equation (96), we have

$$\begin{aligned}
\left(\langle\tilde{Z}_{k'}^{i_1}, \tilde{Z}_{k'}^{i_2}\rangle\right)_{i_1,i_2\in[n]} &= H'^\infty + O(\pm 1/\sqrt{m_1})_{i_1,i_2\in[n]} \\
&= H^\infty + (\pm nc_2/\sqrt{m_1}\kappa_1)_{i_1,i_2\in[n]} + O(\pm 1/\sqrt{m_1})_{i_1,i_2\in[n]} \\
&= H^\infty + (\pm nc_2/\sqrt{m_1}\kappa_1)_{i_1,i_2\in[n]},
\end{aligned}$$

so as long as

$$n^2 c_2/\sqrt{m_1}\kappa_1 = n^2 C_1\sqrt{nm_3}/(\kappa_1\sqrt{m_1\lambda_0}) \leq \lambda_0/2,$$

with similar argument as in Lemma 2, we get

$$\lambda_{min}\left(\left(\langle \tilde{Z}_{k'}^{i_1}, \tilde{Z}_{k'}^{i_2}\rangle\right)_{i_1,i_2\in[n]}\right) \geq \lambda_0/2.$$

Moreover,

$$\langle W_k^{+2}, \tilde{Z}_{k'}^i\rangle = \langle \sum_i \mathcal{V}_{k,i}\tilde{Z}_k^i, \tilde{Z}_{k'}^i\rangle \lesssim \frac{1}{\sqrt{m_1}}\sum_i |\mathcal{V}_{k,i}|$$

$$\leq \frac{\sqrt{n}}{\sqrt{m_1}}\|\mathcal{V}_k\|$$

$$\leq \frac{\sqrt{n}}{\sqrt{m_1\lambda_0}}\|\mathcal{V}_k\|_{H^\infty}.$$

Thus, using the same argument as before the proof is complete.

**Lemma 5** *Suppose*

$$m_1 \geq n^7 m_3/\lambda_0.$$

*During SGD, suppose we are currently at $(V', W')$ with $W' \leq C_1$. For any matrix $W_1$, we denote the signs of the first layer imposed by $W_1$ by $D_{W_1,x_i}$. Then with high probability, there exists $W^* = \sum_{k\in[m_3]} W_k^*$ such that $W_k^*$'s is orthogonal to all other $Z_{k'}^i$'s for $k' \neq k$, and for every $i \in [n]$, we have:*

$$\|\frac{1}{\sqrt{m_1}}W^s D_{W^{(0)}+W',x_i}W^* x_i - \bar{x}_i\|_\infty \lesssim \frac{nm_3}{\sqrt{m_1}\lambda_0}\left[1 + \frac{nC_1}{\kappa_1}\right]\|\mathcal{V}_k\|_{H^\infty} := \Re\|\mathcal{V}_k\|_{H^\infty}.$$

*Moreover, we have*

$$\|W_j^*\| \leq \sqrt{nm_3}/(\sqrt{m_1\lambda_0})\sqrt{\sum_k \|\mathcal{V}_k\|_{H^\infty}^2} + \frac{n\sqrt{m_3}}{\lambda_0\sqrt{m_1}}(\sum_k \|\mathcal{V}_k\|_{H^\infty}) := \varrho\sqrt{\frac{m_3}{m_1}}, \qquad (104)$$

*Particularly, for any diagonal sign matrix $\Sigma \in \mathbb{R}^{m_3\times m_3}$, we have*

$$\|W_\Sigma^*\|_F^2 \leq (\frac{n\sqrt{n}}{\lambda_0^2}\frac{\sqrt{m_3}}{\sqrt{m_1}}(1 + \frac{\sqrt{m_3}}{\sqrt{m_1}}) + (1 + O(n/(\lambda_0\sqrt{m_1}) + \frac{n^2 m_3}{\lambda_0^2 m_1})))\sum_k \|\mathcal{V}_k\|_{H^\infty}^2. \qquad (105)$$

*which, by having enough overparameterization, implies*

$$\|W_\Sigma^*\|_F \leq \sqrt{2\sum_k \|\mathcal{V}_k\|_{H^\infty}^2} = \sqrt{2\zeta_1}, \qquad (106)$$

*where*

$$W_\Sigma^* := \sum_{k=1}^{m_3} \Sigma_k W_k^*. \qquad (107)$$

*Moreover, we have*

$$\frac{1}{\sqrt{m_1}}W^s D_{W^{(0)}+W',x_i}W_\Sigma^* x_i = \Sigma\frac{1}{\sqrt{m_1}}W^s D_{W^{(0)}+W',x_i}W^* x_i. \qquad (108)$$

**Proof of Lemma 5**

From Lemma 3, we have

$$|\bar{x}_{i,k} - \text{trace}(W_k^{+2}Z_k^i)| \leq C_1 m_3 n^2/(\lambda_0\kappa_1\sqrt{m_1})\|\mathcal{V}_k\|_{H^\infty}.$$

Combining this with the result of Lemma 4, we get:

$$|\bar{x}_{i,k} - \text{trace}(W_k^* Z_k^i)| \lesssim \left[\frac{n\sqrt{m_3}}{\lambda_0\sqrt{m_1}} + \frac{C_1 m_3 n^2}{\lambda_0 \kappa_1 \sqrt{m_1}}\right]\|\mathcal{V}_k\|_{H^\infty}$$

$$= \frac{nm_3}{\sqrt{m_1}\lambda_0}\left[1 + \frac{nC_1}{\kappa_1}\right]\|\mathcal{V}_k\|_{H^\infty}. \tag{109}$$

On the other hand, based on the property that $W_{k\,j}^* = 0$ for $j \in P$ and its orthogonal property from Lemma 4, for $j \in P$ we get

$$\frac{1}{\sqrt{m_1}}W_k^s D_{W^{(0)}+W',x_i}W^* x_i = \frac{1}{\sqrt{m_1}}W_k^s D_{W^{(0)},x_i}W^* x_i$$

$$= \text{trace}(W^* Z_k^i) = \text{trace}(W_k^* Z_k^i)$$

$$= \frac{1}{\sqrt{m_1}}W_k^s D_{W^{(0)},x_i}W_k^* x_i,$$

which combined with Equation (109) completes the proof. From the above, Equation (108) is also clear. Finally, note that by Lemma 38 we have

$$\|W^{+2}_{\,j}\| \le \sqrt{nm_3}/(\sqrt{m_1}\lambda_0)\sqrt{\sum_k \|\mathcal{V}_k\|_{H^\infty}^2}.$$

which Combined with Lemma 4 implies

$$\|W_j^*\| \le \sqrt{nm_3}/(\sqrt{m_1}\lambda_0)\sqrt{\sum_k \|\mathcal{V}_k\|_{H^\infty}^2} + \frac{n\sqrt{m_3}}{\lambda_0\sqrt{m_1}}(\sum_k \|\mathcal{V}_k\|_{H^\infty}) \coloneqq \varrho\sqrt{\frac{m_3}{m_1}},$$

while the other claims follows from Lemma 39 and Lemma 4, combined with Equation (97):

$$\|W_\Sigma^*\|_F^2 \le \sum_k \|W_k^*\|^2 + \sum_{k_1 \ne k_2} |\langle W_{k_1}^*, W_{k_2}^*\rangle|$$

$$\le \sum_k \|W_k^*\|^2 + \frac{n}{\lambda_0^2}\frac{\sqrt{m_3}}{\sqrt{m_1}}(1 + \frac{\sqrt{m_3}}{\sqrt{m_1}})(\sum_k \|\mathcal{V}_k\|_{H^\infty})^2$$

$$\le \sum_k \|W_k^*\|^2 + \frac{n}{\lambda_0^2}\frac{\sqrt{m_3}}{\sqrt{m_1}}(1 + \frac{\sqrt{m_3}}{\sqrt{m_1}})\sqrt{n}(\sum_k \|\mathcal{V}_k\|_{H^\infty}^2)$$

$$\le \sum_k \|W_k^*\|^2 + \frac{n\sqrt{n}}{\lambda_0^2}\frac{\sqrt{m_3}}{\sqrt{m_1}}(1 + \frac{\sqrt{m_3}}{\sqrt{m_1}})\zeta_1$$

$$\le \frac{n\sqrt{n}}{\lambda_0^2}\frac{\sqrt{m_3}}{\sqrt{m_1}}(1 + \frac{\sqrt{m_3}}{\sqrt{m_1}})\zeta_1 + \sum_k \|W_k^{+2}\|^2 + \frac{n^2 m_3}{\lambda_0^2 m_1}\sum_k \|\mathcal{V}_k\|_{H^\infty}^2$$

$$\le \frac{n\sqrt{n}}{\lambda_0^2}\frac{\sqrt{m_3}}{\sqrt{m_1}}(1 + \frac{\sqrt{m_3}}{\sqrt{m_1}})\zeta_1 + \sum_k \|W_k^+\|^2 + \frac{n^2 m_3}{\lambda_0^2 m_1}\sum_k \|\mathcal{V}_k\|_{H^\infty}^2$$

$$\lesssim \frac{n\sqrt{n}}{\lambda_0^2}\frac{\sqrt{m_3}}{\sqrt{m_1}}(1 + \frac{\sqrt{m_3}}{\sqrt{m_1}})\zeta_1 + (1 + O(n/(\lambda_0\sqrt{m_1}) + \frac{n^2 m_3}{\lambda_0^2 m_1}))\zeta_1.$$

Next, we move on to construct $V^*$ for the second layer.

### A.12.2 SECOND LAYER, CONSTRUCTION OF $V^*$

In this section, we present a couple of lemmas that step by step lead to the construction of $V^*$. we remind the reader that $\phi^{(0)}(x_i)$ is the output of the first layer at initialization weights, $\phi'(x_i)$ and $\phi^{(2)}(x_i)$ are the changes in the output of the first layer when $W'$ and $W' + W^\rho$ are added,

respectively, and finally $\phi^*(x_i)$ is the optimal features that are generated by the matrix $W^*$ but with the sign pattern of $W^{(0)} + W'$, i.e.

$$\phi^*(x_i) = \frac{1}{\sqrt{m_1}} W^s D_{W^{(0)}+W',x_i} W^* x_i.$$

We also define $x_i'$ as

$$x_i' = \phi^{(0)}(x_i) + \phi^{(2)}(x_i) = \frac{1}{\sqrt{m_1}} W^s \sigma((W^{(0)} + W' + W^\rho)x_i).$$

To begin, we state a lemma to bound the magnitude of $\|\phi'(x_i)\|$, given that the norm of $W'$ is bounded by $C_1$ and the sign pattern $\mathrm{Sgn}\big((W^{(0)} + W')x_i\big)$ satisfies condition stated for the set of indices $P$ in Lemma 1. Later on, we exploit this Lemma in Lemma 33 to state bounds for $\|\phi^{(2)}(x_i)\|$.

**Lemma 6** *Let the matrix $W'$ with norm bound $\|W'\| \leq C_1$, such that the signs of $(W_j^{(0)} + W_j')x_i$ and $W_j^{(0)}x_i$ can be different only for $j \in P$, for $P$ defined in Lemma 1. (Note that for $W'$ at every step of the algorithm, this is automatically satisfied by Lemma 1) Then*

$$\|\phi'(x_i)\| \leq \frac{2C_1^{3/2}}{\sqrt{\kappa_1}} (\frac{n^3 m_3^3}{m_1 \lambda_0})^{1/4} + (1 + O(m_3^2/\sqrt{m_1}))C_1.$$

*Particularly for large enough $m_1$ compared to $n, m_3, \lambda_0, \kappa_1, C_1$, we have*

$$\|\phi'(x_i)\| \lesssim C_1.$$

**Proof of Lemma 6**

We write

$$|\phi_k'(x_i) - \langle W', Z_k^i \rangle| \leq 2/\sqrt{m_1} \sum_{j \in P} |W_j' x_i| \leq 2/\sqrt{m_1} \sum_{j \in P} \|W_j'\|$$

$$\leq 2\sqrt{|P|}/\sqrt{m_1}\|W'\|_F \leq \frac{2C_1^{3/2}}{\sqrt{\kappa_1}} (\frac{n^3 m_3}{m_1 \lambda_0})^{1/4}, \tag{110}$$

where the last line follows from the bound on $|P|$ from Lemma 1.

On the other hand, because by Hoeffding we know that $\langle Z_k^i, Z_{k'}^i \rangle \lesssim 1/\sqrt{m_1}$ by Lemma 40, we get

$$\sum_{k=1}^{m_3} \langle W', Z_k^i \rangle^2 \leq (1 + O(m_3^2/\sqrt{m_1}))\|W'\|_F^2 \leq (1 + O(m_3^2/\sqrt{m_1}))C_1^2.$$

Combining this with Equation (110), we get

$$\|\phi'(x_i)\| \leq \sqrt{\sum_k |\phi_k^{(2)}(x_i) - \langle W', Z_k^i \rangle|^2} + \sqrt{\sum_k \langle W', Z_k^i \rangle^2}$$

$$\leq \frac{2C_1^{3/2}}{\sqrt{\kappa_1}} (\frac{n^3 m_3^3}{m_1 \lambda_0})^{1/4} + (1 + O(m_3^2/\sqrt{m_1}))C_1. \tag{111}$$

Next, we prove a structural lemma regarding the sign pattern in the second layer when we feed in $x_i'$ to it, with the important message that the dominance of sign patterns are specified by $\phi^{(0)}(x_i)$.

**Lemma 7** *Suppose we have $m_3 \kappa_1^2 \gtrsim C_1^2$, $\kappa_2 \sqrt{m_2} \geq C_2$, and $m_1$ satisfies the condition on Lemma 6. If we have the condition $\|\phi^{(2)}(x_i)\| \lesssim C_1$, which happens under the high probability event $E^c$ defined in Lemma 33, then for every $i \in [n]$, there exist a subset $\tilde{P}_i$ which might depend on $W^{(0)}, V^{(0)}, W', V$, such that*

$$|\tilde{P}_i| \lesssim \Big(\big(\frac{C_1^2}{(m_3 \kappa_1^2)}\big)^{1/3} + (c_3 + \frac{C_1^2}{c_3^2 m_3 \kappa_1^2})(\frac{C_2^2}{\kappa_2^2 m_2})^{1/3}\Big)m_2.$$

*Moreover, for every $i \in [n]$, for $j \notin \tilde{P}_i$:*

$$\frac{2}{3}|V_j^{(0)}\phi^{(0)}(x_i)| \geq |V_j^{(0)}\phi^{(2)}(x_i)| + |V_j'(\phi^{(0)}(x_i) + \phi^{(2)}(x_i))|,$$

$$|V_j^{(0)}\phi^{(0)}(x_i)| \gtrsim (\frac{\kappa_2}{m_2})^{1/3}C_2^{2/3}c_3\|\phi^{(0)}(x_i)\|,$$

$$|V_j^{(0)}\phi^{(0)}(x_i)| \gtrsim (\frac{\kappa_2}{m_2})^{1/3}C_2^{2/3}c_3\|x_i'\|.$$

**Proof of Lemma 7**

By assumption, we know that during the algorithm, we have $\|V'\| \leq C_2$. Also, we know by Lemma 33 that under $E^c$:

$$\|\phi^{(2)}(x_i)\| \leq 2C_1.$$

Define the set

$$P_i' = \{j|\ |V_j^{(0)}\phi^{(0)}(x_i)| \leq c_3(\frac{\kappa_2}{m_2})^{1/3}C_2^{2/3}\|\phi^{(0)}(x_i)\|\} \tag{112}$$

and $P' = \cup P_i'$. We have

$$\mathbb{P}(|V_j^{(0)}\phi^{(0)}(x_i)| \leq c_3(\frac{\kappa_2}{m_2})^{1/3}C_2^{2/3}\|\phi^{(0)}(x_i)\|) \leq c_3(\frac{\kappa_2}{m_2})^{1/3}C_2^{2/3}/\kappa_2,$$

so by Bernstein, with high probability:

$$|P_i'| \leq m_2 C_2^{2/3}c_3(\frac{\kappa_2}{m_2})^{1/3}/\kappa_2 + \sqrt{m_2 C_2^{2/3}c_3(\frac{\kappa_2}{m_2})^{1/3}/\kappa_2} + 1 \lesssim c_3 m_2 C_2^{2/3}(\frac{\kappa_2}{m_2})^{1/3}/\kappa_2,$$

so with high prob.

$$|P_i'| \lesssim c_3 C_2^{2/3}(\frac{m_2}{\kappa_2})^{2/3}. \tag{113}$$

On the other hand, Note that

$$\phi_k^{(0)}(x_i) = \sum_{j=1}^{m_1} 1/\sqrt{m_1} W_{k,j}^s \sigma(W_j^{(0)}x_i) \tag{114}$$

is subGaussian with parameter $\sigma^2 = O(1/m_1 \sum_j \sigma(W_j^{(0)}x_i)^2)$. Furthermore, note that if we compute the variance of $\phi_k^{(0)}(x_i)$ with respect to the randomness of $W^s$:

$$\mathbb{E}\phi_k^{(0)}(x_i)^2 = 1/m_1 \sum_{j=1}^{m_1} \sigma(W_j^{(0)}x_i)^2 := \aleph$$

which itself concentrates around $1/2\kappa_1^2\|x_i\|^2 = 1/2\kappa_1^2$ by another Bernstein, i.e. $\aleph = 1/2\kappa_1^2(1 \pm O(1/\sqrt{m_1}))$. Therefore, by concentration of subexponential variables (Bernstein), it is not hard to see that the squared norm of the vector $\phi^{(0)}(x_i)$ is $(m_3\kappa_1^4, \kappa_2)$-subexponential and concentrates around $m_3\aleph$, i.e.

$$\|\phi^{(0)}(x_i)\|^2 = m_3\aleph \pm O(\kappa_1^2\sqrt{m_3}) = m_3\kappa_1^2/2 \pm O(m_3\kappa_1^2/\sqrt{m_1}) \pm O(\kappa_1^2\sqrt{m_3}), \tag{115}$$

with high probability. Combining this with the fact that $\|\phi^{(2)}(x_i)\| \lesssim C_1$ implies with high probability:

$$\frac{\|\phi^{(0)}(x_i)\|}{\|\phi^{(2)}(x_i)\|} \gtrsim \frac{\sqrt{m_3}\kappa_1}{C_1}. \tag{116}$$

Now define $P_i'' = \{j|\ |V_j'x_i'| \geq |V_j^{(0)}\phi^{(0)}(x_i)|/3\}$. If $j \in P_i'' - P_i'$, then by Equation (116), with high probability

$$\|V_j'\|\|\phi^{(2)}(x_i)\| \geq |V_j'\phi^{(2)}(x_i)| = |V_j'(\phi^{(0)}(x_i) + \phi^{(2)}(x_i))| = |V_j'x_i'|$$

$$\geq |V_j^{(0)}\phi^{(0)}(x_i)|/3 \gtrsim c_3(\frac{\kappa_2}{m_2})^{1/3}C_2^{2/3}\|\phi^{(0)}(x_i)\|,$$

or

$$\|V_j'\|^2 \gtrsim c_3^2(\frac{\kappa_2}{m_2})^{2/3}C_2^{4/3}\frac{m_3\kappa_1^2}{C_1^2}.$$

But note that $\|V'\|_F^2 \leq C_2^2$ by our assumption, which implies

$$|P_i'' - P_i'| \lesssim C_2^2/[c_3^2(\frac{\kappa_2}{m_2})^{2/3}C_2^{4/3}\frac{m_3\kappa_1^2}{C_1^2}] = \frac{C_2^{2/3}C_1^2}{c_3^2 m_3\kappa_1^2}(\frac{m_2}{\kappa_2})^{2/3}. \tag{117}$$

Now combining Equations (113) and (117), we finally obtain

$$|P_i''| = |P_i'' - P_i'| + |P_i'| \lesssim (c_3 + \frac{C_1^2}{c_3^2 m_3\kappa_1^2})C_2^{2/3}(\frac{m_2}{\kappa_2})^{2/3}.$$

Now define the set

$$P_i''' = \{j| \; |V_j^{(0)}\phi^{(2)}(x_i)| \geq |V_j^{(0)}\phi^{(0)}(x_i)|/3\}. \tag{118}$$

Note that for every $j \in [m_2]$, $V_j^{(0)}\phi^{(0)}(x_i)$ is gaussian with variance $\|\phi^{(0)}(x_i)\|$ over the randomness of $V_j^{(0)}$, so

$$\mathbb{P}(V_j^{(0)}\phi^{(0)}(x_i) \leq \alpha\kappa_2\|\phi^{(0)}(x_i)\|) \lesssim \alpha.$$

Therefore, if we define the set

$$Q_i = \{j \in [m_2]| \; |V_j^{(0)}\phi^{(0)}(x_i)| \leq \alpha\kappa_2\|\phi^{(0)}(x_i)\|\},$$

then for large enough $m_2$, by Bernstein with high prob.:

$$|Q_i| \lesssim \alpha m_2. \tag{119}$$

Now note that $\phi^{(0)}(x_i)$ is fixed during the algorithm. On the other hand, by random matrix theory, we know that with high probability, the eigenvalues of the matrix $V^{(0)}$ are in $(\kappa_2(\sqrt{m_2} - \sqrt{m_3}), \kappa_2(\sqrt{m_2} + \sqrt{m_3}))$. Therefore, even if the vector $\phi^{(2)}(x_i)$ is picked adversarialy (because it keeps changing during the algorithm), we get that with high probability over the randomness of $V^{(0)}$:

$$\|V^{(0)}\phi^{(2)}(x_i)\|^2 \leq \kappa_2^2(\sqrt{m_2} + \sqrt{m_3})^2\|\phi^{(2)}(x_i)\|^2 \lesssim \kappa_2^2 m_2\|\phi^{(2)}(x_i)\|^2. \tag{120}$$

Moreover, because $\|\phi^{(2)}(x_i)\| \lesssim C_1$ and from Equation (115), with high probability over the randomness of $W^{(0)}$:

$$\frac{\|\phi^{(0)}(x_i)\|}{\|\phi^{(2)}(x_i)\|} \gtrsim \frac{\sqrt{m_3}\kappa_1}{C_1}.$$

This means that for $j \in P_i''' - Q_i$, combining these inequalities we conclude with high probability

$$|V_j^{(0)}\phi^{(2)}(x_i)| \geq |V_j^{(0)}\phi^{(0)}(x_i)|/3 \gtrsim \alpha\kappa_2\|\phi^{(0)}(x_i)\| \geq \alpha\kappa_2\frac{\sqrt{m_3}\kappa_1}{C_1}\|\phi^{(2)}(x_i)\|,$$

which combined with (120) implies

$$\|P_i'''\| \lesssim \frac{m_2 C_1^2}{m_3\kappa_1^2\alpha^2}.$$

Balancing this term with the one in Equation (119), we set

$$\alpha := \frac{C_1^{2/3}}{m_3^{1/3}\kappa_1^{2/3}},$$

which implies

$$|P_i'''| \lesssim |P_i''' - Q_i| + |Q_i| \leq (\frac{C_1^2}{(m_3\kappa_1^2)})^{1/3}m_2.$$

Defining $\tilde{P}_i = P_i'' \cup P_i'''$, we finally get

$$|\tilde{P}_i| \lesssim \Big((\frac{C_1^2}{(m_3\kappa_1^2)})^{1/3} + (c_3 + \frac{C_1^2}{c_3^2 m_3\kappa_1^2})C_2^{2/3}(\frac{1}{\kappa_2^2 m_2})^{1/3}\Big)m_2.$$

Clearly by the definition of $P_i''$ and $P_i'''$ the proof is complete.

**Corollary 5.1** *Under the condition $\|\phi^{(2)}(x_i)\| \lesssim C_1$ (which happens under the event $E^c$ defined in Lemma 33), setting $c_3 := \frac{C_1^{2/3}}{m_3^{1/3}\kappa_1^{2/3}}(\frac{\kappa_2^2 m_2}{C_2^2})^{1/3}$ in the previous Lemma, we obtain $\forall i \in [n]$:*

$$|\tilde{P}_i| \lesssim \big(\frac{C_1^2}{(m_3\kappa_1^2)}\big)^{1/3} m_2.$$

*Also for $j \notin \tilde{P}_i$, the conditions in (112) and (118) becomes the same as*

$$|W_j^{(0)}\phi^{(0)}(x_i)| \leq \kappa_2 \frac{C_1^{2/3}}{m_3^{1/3}\kappa_1^{2/3}}\|\phi^{(0)}(x_i)\|.$$

*Hence, for every $i \in [n]$ and for $j \notin \tilde{P}_i$, with high probability:*

$$\frac{2}{3}|W_j^{(0)}\phi^{(0)}(x_i)| \geq |W_j^{(0)}\phi^{(2)}(x_i)| + |W_j'(\phi^{(0)}(x_i) + \phi^{(2)}(x_i))|, \tag{121}$$

$$|W_j^{(0)}\phi^{(0)}(x_i)| \gtrsim \kappa_2 \frac{C_1^{2/3}}{m_3^{1/3}\kappa_1^{2/3}}\|\phi^{(0)}(x_i)\| \gtrsim \kappa_2(\sqrt{m_3}\kappa_1 C_1^2)^{1/3}, \tag{122}$$

$$|W_j^{(0)}\phi^{(0)}(x_i)| \gtrsim \frac{C_1^{2/3}}{m_3^{1/3}\kappa_2^{2/3}}\|x_i'\|. \tag{123}$$

Next, we state concentration result for the gram matrix of $\phi^{(0)}(x_i)$'s.

**Lemma 8** *For every $i_1, i_2 \in [n]$, with high probability over the randomness of $W^{(0)}$ and $V^{(0)}$ we have*

$$\langle\phi^{(0)}(x_{i_1}), \phi^{(0)}(x_{i_2})\rangle = m_3\mathbb{E}\sigma(W_j^{(0)}x_{i_1})\sigma(W_j^{(0)}x_{i_2}) \pm O(m_3\kappa_1^2/\sqrt{m_1} + \sqrt{m_3}\kappa_1^2).$$

**Proof of Lemma 8**

First, we compute the expectation:

$$\mathbb{E}\langle\phi^{(0)}(x_{i_1}), \phi^{(0)}(x_{i_2})\rangle = 1/m_1 \sum_{j_1,j_2\in[m_1]}\sum_{k\in[m_3]}\mathbb{E}W_{k,j_1}^s W_{k,j_2}^s \sigma(W_{j_1}^{(0)}x_{i_1})\sigma(W_{j_2}^{(0)}x_{i_2})$$

$$= 1/m_1 \sum_{j_1\neq j_2}\mathbb{E}\sum_{k\in[m_3]}W_{k,j_1}^s W_{k,j_2}^s \sigma(W_{j_1}^{(0)}x_{i_1})\sigma(W_{j_2}^{(0)}x_{i_2}) + m_3/m_1 \sum_{j\in[m_1]}\sigma(W_j^{(0)}x_{i_1})\sigma(W_j^{(0)}x_{i_2}).$$

But $\sigma(W_{j_1}^{(0)}x_{i_1})\sigma(W_{j_2}^{(0)}x_{i_2})$ is $(m_1\kappa_1^4, \kappa_1^2)$-sub-exponential, so

$$\sum_{j\in[m_1]}\sigma(W_j^{(0)}x_{i_1})\sigma(W_j^{(0)}x_{i_2}) = m_1\mathbb{E}\sigma(W_j^{(0)}x_{i_1})\sigma(W_j^{(0)}x_{i_2}) \pm O(\sqrt{m_1}\kappa_1^2),$$

which means with high probability:

$$\mathbb{E}\langle\phi^{(0)}(x_{i_1}), \phi^{(0)}(x_{i_2})\rangle = m_3\mathbb{E}\sigma(W_j^{(0)}x_{i_1})\sigma(W_j^{(0)}x_{i_2}) \pm O(m_3\kappa_1^2/\sqrt{m_1}).$$

On the other side, we know that $\phi_k^{(0)}(x_{i_1})$ is subgaussian with parameters $\sigma^2 = 1/m_1 \sum_j (W_j^{(0)}x_{i_1})^2 := \aleph_1$ and $\sigma^2 = 1/m_1 \sum_j (W_j^{(0)}x_{i_2})^2 := \aleph_2$ respectively. On the other hand, we know that by Bernstein w.h.p

$$\aleph_1 = 1/2\kappa_1^2(1 \pm O(1/\sqrt{m_1})),$$
$$\aleph_2 = 1/2\kappa_1^2(1 \pm O(1/\sqrt{m_1})).$$

Hence, $\phi_k^{(0)}(x_{i_1})\phi_k^{(2)}(x_{i_2})$ is $(\aleph_1\aleph_2, \sqrt{\aleph_1\aleph_2})$-subexponential, and so $\langle\phi^{(0)}(x_{i_1}), \phi^{(0)}(x_{i_2})\rangle$ is $(m_3\aleph_1\aleph_2, \sqrt{\aleph_1\aleph_2})$-subexponential. Therefore, applying another Bernstein on the top, we get

$$\langle\phi^{(0)}(x_{i_1}), \phi^{(0)}(x_{i_2})\rangle = \mathbb{E}\langle\phi^{(0)}(x_{i_1}), \phi^{(0)}(x_{i_2})\rangle \pm O(\sqrt{m_3}\sqrt{\aleph_1\aleph_2})$$

$$= m_3\mathbb{E}\sigma(W_j^{(0)}x_{i_1})\sigma(W_j^{(0)}x_{i_2}) \pm O(m_3\kappa_1^2/\sqrt{m_1}) \pm \frac{\sqrt{m_3}\kappa_1^2}{2}(1 \pm O(1/\sqrt{m_1}))$$

$$= m_3\mathbb{E}\sigma(W_j^{(0)}x_{i_1})\sigma(W_j^{(0)}x_{i_2}) \pm O(m_3\kappa_1^2/\sqrt{m_1} + \sqrt{m_3}\kappa_1^2).$$

Now we define the matrix $L_i \in \mathbb{R}^{m_3 \times m_2}$, with its $j$th column $L_{i,j}$ equal to

$$\frac{a_j}{\sqrt{m_2}} \mathbb{1}\{V_j^{(0)}\phi^{(0)}(x_i) \geq 0\}\phi^*(x_i).$$

First, we state the following lemma which characterize a concentration result for the gram matrix of $(L_i)_{i=1}^n$.

**Lemma 9** *With high probability, we have the following approximation:*

$$\langle L_{i_1}, L_{i_2} \rangle = \langle \phi^*(x_{i_1}), \phi^*(x_{i_2}) \rangle \Big[ F_2\Big(2F_3(\langle x_{i_1}, x_{i_2}\rangle)\Big) \pm O(m_1^{-1/4} + m_2^{-1/4} + m_3^{-1/4})\Big].$$

**Proof of Lemma 9**

By Hoeffding:

$$\langle L_{i_1}, L_{i_2}\rangle = 1/m_2 \sum_{j \in m_2} \phi^*(x_{i_1})^T \phi^*(x_{i_2}) \mathbb{1}\{V_j^{(0)}\phi^{(1)}(x_{i_1}) \geq 0\}\mathbb{1}\{V_j^{(0)}\phi^{(1)}(x_{i_2}) \geq 0\}$$

$$= \phi^*(x_{i_1})^T \phi^*(x_{i_2})\Big(\mathbb{E}\mathbb{1}\{V_j^{(0)}\phi^{(1)}(x_{i_1}) \geq 0\}\mathbb{1}\{V_j^{(0)}\phi^{(1)}(x_{i_2}) \geq 0\} \pm O(1/\sqrt{m_2})\Big)$$

$$= \phi^*(x_{i_1})^T \phi^*(x_{i_2})\Big(F_2\Big(\langle \phi^{(1)}(x_{i_1}), \phi^{(1)}(x_{i_2})\rangle/(\|\phi^{(1)}(x_{i_1})\|\|\phi^{(1)}(x_{i_2})\|)\Big) \pm O(1/\sqrt{m_2})\Big),$$

where recall

$$F_2(x) = 1/4 + \arcsin(x)/2\pi,$$

measures the angle between two unit vectors based on their dot product. Now notice that according to Lemma 8, with high probability:

$$\langle L_{i_1}, L_{i_2}\rangle/\langle \phi^*(x_{i_1}), \phi^*(x_{i_2})\rangle$$

$$= F_2\Big(\frac{m_3\mathbb{E}\sigma(W_j^{(0)}x_{i_1})\sigma(W_j^{(0)}x_{i_2}) \pm O((m_3/\sqrt{m_1} + \sqrt{m_3})\kappa_1^2)}{\sqrt{(m_3\mathbb{E}\sigma(W_j^{(0)}x_{i_1})^2 \pm O((m_3/\sqrt{m_1} + \sqrt{m_3})\kappa_1^2))(m_3\mathbb{E}\sigma(W_j^{(0)}x_{i_2})^2 \pm O(...)}} \pm O(1/\sqrt{m_2})\Big)$$

$$= F_2\Big(\frac{F_3(\langle x_{i_1}, x_{i_2}\rangle) \pm O(1/\sqrt{m_1} + 1/\sqrt{m_3})}{1/2 \pm O(1/\sqrt{m_1} + 1/\sqrt{m_3})} \pm O(1/\sqrt{m_2})\Big),$$

where recall $F_3 : [-1, +1] \to [-1/2, 1/2]$ is defined as:

$$F_3(x) := \frac{\sqrt{1-x^2}}{2\pi} + \frac{x}{4} + \frac{x\arcsin x}{2\pi}.$$

It is easy to see $F_3$ has the property that for unit vectors $x_1, x_2$ and $w$ sampled as standard normal:

$$F_3(\langle x_1, x_2\rangle) = \mathbb{E}\sigma(w^T x_1)\sigma(w^T x_2).$$

But because $|F_3(.)| = O(1)$, we have

$$\langle L_{i_1}, L_{i_2}\rangle/\langle \phi^*(x_{i_1}), \phi^*(x_{i_2})\rangle = F_2\Big(2F_3(\langle x_{i_1}, x_{i_2}\rangle) \pm O(1/\sqrt{m_1} + 1/\sqrt{m_2} + 1/\sqrt{m_3})\Big).$$

Now notice that the derivative of $F_2$, i.e. $1/2\pi\sqrt{1-x^2}$ is increasing in the interval $(0, 1)$, so for a fixed $\delta$, the maximum of $|F_2(x) - F_2(x - \delta)|$ happens at $x = 1$. On the other hand, by writing the first order approximation of $\arcsin(1 - t^2)$ around $t = 0$ and upper bounding its derivative in the interval $[0, 1]$, we get that for $0 \leq \delta \leq 1$:

$$\arcsin(1 - \delta) \geq \arcsin(1) - 2\sqrt{\delta}.$$

Therefore, $F_2(x \pm \delta) = F_2(x) \pm O(\sqrt{\delta})$. Hence:

$$\langle L_{i_1}, L_{i_2}\rangle/\langle \phi^*(x_{i_1}), \phi^*(x_{i_2})\rangle = F_2\Big(2F_3(\langle x_{i_1}, x_{i_2}\rangle)\Big) \pm O(\sqrt{1/\sqrt{m_1} + 1/\sqrt{m_2} + 1/\sqrt{m_3}})$$

$$= F_2\Big(2F_3(\langle x_{i_1}, x_{i_2}\rangle)\Big) \pm O(m_1^{-1/4} + m_2^{-1/4} + m_3^{-1/4}),$$

which completes the proof.

Finally, we are ready to construct the weights $V^*$ for the second layer.

### A.12.3 Construction of $V^*$

**Lemma 10** *Let*

$$\Re = \frac{n\sqrt{m_3}}{\sqrt{m_1}\lambda_0}\Big[1 + \frac{nC_1}{\kappa_1}\Big].$$

*Suppose we have the condition that for every $k \in [m_3]$:*

$$\max_k \|\mathcal{V}_k\| \leq \xi, \tag{124}$$

*where recall the definition of $\mathcal{V}_k$ in Equation (58). We assume enough overparameterization to make sure $\Re < 1$. Recall for the matrix $A$ defined by*

$$A = \Big(\langle \bar{x}_{i_1}, \bar{x}_{i_2}\rangle F_2(2F_3(\langle x_{i_1}, x_{i_2}\rangle))\Big)_{1 \leq i_1, i_2 \leq n}, \tag{125}$$

*we have*

$$(f^*(x_i))_{i=1}^{n}{}^T A^{-1}(f^*(x_i))_{i=1}^n \leq \zeta_2.$$

*Then, there exists weight matrix $V^*$ which only depends on the random initializations $W^{(0)}, V^{(0)}$ (e.g. not on $V'$ and $W'$) for the second layer, such that having enough overparameterization*

$$\|V^*\|_F^2 \leq 2\zeta_2, \tag{126}$$

*and for every $j \in [m_2]$:*

$$\|V_j^*\|_\infty \leq \frac{(1+\Re)n\sqrt{n\zeta_2}}{\sqrt{m_2}}\xi := \varrho_3 \xi/\sqrt{m_2}, \tag{127}$$

$$\|V_j^*\|_2 \leq \frac{1}{\sqrt{m_2}}n(1+\Re)\sqrt{\frac{\zeta_2 \sum_k \|\mathcal{V}_k\|_{H^\infty}^2}{\lambda_0}} := \varrho_2/\sqrt{m_2}, \tag{128}$$

*and further under the high probability event $E^c$ defined in Lemma 33:*

$$\Big|\frac{1}{\sqrt{m_2}}a^T D_{V^{(0)}+V', x_i} V^* \phi^*(x_i) - f^*(x_i)\Big| \lesssim \Big(\frac{C_1}{\sqrt{m_3}\kappa_1}\Big)^{1/3}(1+\Re)\sqrt{\zeta_2 \sum_k \|\mathcal{V}_k\|_{H^\infty}^2} := \Re_3. \tag{129}$$

**Proof of Lemma 10**

Let

$$V^* = \sum_{i=1}^{n} \mathcal{V}_i^* L_i,$$

be the minimum norm vector which maps $L_i$'s to $f^*(x_i)$'s. As a result, for the matrix

$$L = \Big(\langle L_{i_1}, L_{i_2}\rangle\Big)_{i_1, i_2}$$

it is easy to see

$$\|V^*\|_F^2 = (f^*(x_i))_{i=1}^{n}{}^T L^{-1}(f^*(x_i))_{i=1}^n.$$

Now combining Lemmas 5 and 41, we get

$$\|\phi^*(x_i)\|_\infty \leq (1+\Re)\xi, \tag{130}$$

$$\|\phi^*(x_i)\| \leq (1+\Re)\sqrt{\sum_k \|\mathcal{V}_k\|_{H^\infty}^2}, \tag{131}$$

and

$$\langle|\phi^*(x_{i_1}), \phi^*(x_{i_2})\rangle - \langle \bar{x}_{i_1}, \bar{x}_{i_2}\rangle| \leq (2\Re + \Re^2)\sum_k \|\mathcal{V}_k\|_{H^\infty}^2.$$

Now by Lemma 9:

$$|\langle L_{i_1}, L_{i_2}\rangle - A_{i_1,i_2}| \lesssim (2\Re + \Re^2)(\sum_k \|\mathcal{V}_k\|^2_{H^\infty})\Big|F_2\Big(2F_3(\langle x_{i_1}, x_{i_2}\rangle)\Big)\Big| + \langle \bar{x}_{i_1}, \bar{x}_{i_2}\rangle(m_1^{-1/4} + m_2^{-1/4} + m_3^{-1/4})$$
$$+ \text{ other cross term.}$$

By applying $\langle \bar{x}_{i_1}, \bar{x}_{i_2}\rangle \le \|\bar{x}_{i_1}\|\|\bar{x}_{i_2}\|$ we get

$$LHS \le (\sum_k \|\mathcal{V}_k\|^2_{H^\infty})\Big((2\Re + \Re^2)\Big|F_2\Big(2F_3(\langle x_{i_1}, x_{i_2}\rangle)\Big)\Big| + (m_1^{-1/4} + m_2^{-1/4} + m_3^{-1/4})\Big)$$
$$\lesssim (\sum_k \|\mathcal{V}_k\|^2_{H^\infty})\Big(\Re + m_1^{-1/4} + m_2^{-1/4} + m_3^{-1/4}\Big).$$

Therefore,

$$\|A - \Big(\langle L_{i_1}, L_{i_2}\rangle\Big)_{i_1,i_2}\|_2 \le \|A - \Big(\langle L_{i_1}, L_{i_2}\rangle\Big)_{i_1,i_2}\|_F$$
$$\le n(\sum_k \|\mathcal{V}_k\|^2_{H^\infty})\Big(\Re + m_1^{-1/4} + m_2^{-1/4} + m_3^{-1/4}\Big) := \Re_2.$$

Note that $\Re_2$ naturally goes to zero (with poly dependence) as $\Re \to 0$ and $m_1, m_2, m_3$ are large enough. Now if all of the eigenvalues of the matrix $A$ are $\Omega(1/n^2)$, then if we overparameterize enough such that $\Re_2 = O(1/n^2)$ with small enough constant so that $\Re_2$ is less than half of the smallest eigenvalue of $A$, then for the $i$th eigenvalue $\lambda_i$ of $A$ and $L$ we can write

$$\lambda_i(L) \ge \lambda_i(A) - \Re_2 \ge \lambda_i(A)/2,$$

so

$$\lambda_i(L^{-1}) \le 2\lambda_i(A^{-1}),$$

which implies the property

$$\|V^*\|^2_F \le 2\zeta_2. \tag{132}$$

However, $A$ might have very small eigenvalues. To remedie this, we use Lemma 42; we can substitute $f^*$ with some $\bar{f}^*$ such that

$$R_n(\bar{f}^*) \le 2R_n(f^*) + \frac{B^2}{n}, \tag{133}$$

$$\bar{f}^{*T}A^{-1}\bar{f}^* \le f^{*T}A^{-1}f^*, \tag{134}$$

where $\bar{f}^*$ is on the subspace of eigenvectors of $A$ whose eigenvalues are larger than $\Omega(1/n^2)$. But it is easy to check that in the context of Theorem 3, such substitution results in a $\bar{f}^{*T}A^{-1}\bar{f}^* \le f^{*T}A^{-1}f^* \le \zeta$ and $\bar{v}(\bar{f}^*)$ parameter (as defined in (41)) with respect to $\bar{f}^*$ which satisfies $\bar{\nu}/2 \le \nu$. Note that the algorithm is with respect to the setting $\nu$, however we want to exploit generalization bound with respect to $\bar{f}^*$ whose parameter is $\bar{\nu}$ as it enables us to use our analysis in this Lemma. Furthermore, note that using Equation (133) we can further upper bound the empirical risk of $\bar{f}^*$ with that of $f^*$, which makes it straightforward to derive a similar generalization bound as in (45) with respect to $f^*$, of course with a change of constants. Note that $\bar{f}^*$ is just the sum of $A$-eigenbasis directions in $f^*$ whose eigenvalues are larger than $\Omega(1/n^2)$. Hence, given a pair $(f^*, G)$, as we also point out in remark 1, we can construct the suitable pair $(\bar{f}^*, G)$ algorithmically and then use that pair to initialize the parameters of the algorithm (namely $\zeta$ and $\nu$). Otherwise, if we are not explicitly given a pair $(f^*, G)$ and instead want to run the doubling trick described in Theorem 1, we do not even have any additional computation; since using Theorem 1, within the framework of the doubling trick, the risk of the final network is competitive with respect to any choice of $(\bar{f}^*, G)$. Note that as we mentioned in Lemma 42, the constant 2 is arbitrary and can be reduced to any number less than 2, and it is easy to see that one can pick choice of constants along the way such that we end up with a factor two behind the risk (first) term in the definition of our complexity measure.

Therefore, without loss of generality we can use substitute $f^*$ by $\bar{f}^*$ and still obtain Equation (132).

On the other hand, the definition of $V^*$ implies

$$\frac{1}{\sqrt{m_2}} a^T D_{V^{(0)},x_i} V^* \phi^*(x_i) = f^*(x_i).$$

But note that by Corollary 5.1, under the high probability event $E^c$ defined in Lemma 33, $D_{V^{(0)},x_i}$ and $D_{V^{(0)}+V',x_i}$ can only be different in the index set $\tilde{P}_i$ and

$$|\tilde{P}_i| \lesssim \Big(\frac{C_1^2}{(m_3 \kappa_1^2)}\Big)^{1/3} m_2,$$

Therefore, for all $i \in [n]$:

$$|\frac{1}{\sqrt{m_2}} a^T D_{V^{(0)}+V',x_i} V^* \phi^*(x_i) - \frac{1}{\sqrt{m_2}} a^T D_{V^{(0)},x_i} V^* \phi^*(x_i)|$$

$$\leq 1/\sqrt{m_2} \sum_{j \in \tilde{P}} |V_j^* \phi^*(x_i)|$$

$$\leq 1/\sqrt{m_2} \sum_{j \in \tilde{P}} \|V_j^*\| \|\phi^*(x_i)\|$$

$$\leq \frac{\sqrt{|\tilde{P}|}}{\sqrt{m_2}} \|V^*\| (1 + \Re) \sqrt{\sum_k \|\mathcal{V}_k\|_{H^\infty}^2}$$

$$\lesssim \Big(\frac{C_1}{\sqrt{m_3}\kappa_1}\Big)^{1/3} \sqrt{\zeta_2}(1 + \Re) \sqrt{\sum_k \|\mathcal{V}_k\|_{H^\infty}^2},$$

which proves the first claim. On the other hand, we get:

$$2\zeta_2 \geq \|V^*\|_F^2 \geq \mathcal{V}^T L \mathcal{V}.$$

But because $\lambda_{min}(L) \gtrsim 1/n^2$, we get

$$\mathcal{V}^T L \mathcal{V} \gtrsim \|\mathcal{V}\|_2^2/n^2,$$

which implies

$$\|\mathcal{V}\|_2 \lesssim n\sqrt{\zeta_2}.$$

But now using Equation (130), we can write

$$|V_{j,k}^*| \leq \sum_{i=1}^n |\mathcal{V}_i| |L_{ij,k}| \leq \frac{1}{\sqrt{m_2}} \|\phi^*(x_i)\|_\infty \sum_i |\mathcal{V}_i|$$

$$\lesssim \frac{(1+\Re)\xi}{\sqrt{m_2}} \sum_i |\mathcal{V}_i| \leq (1+\Re)\xi\sqrt{n}\|\mathcal{V}\|/\sqrt{m_2}$$

$$\lesssim \frac{(1+\Re)n\sqrt{n\zeta_2}}{\sqrt{m_2}}\xi,$$

which proves the other part. Moreover,

$$\|V_j^*\|^2 \leq \|\mathcal{V}\|^2 \frac{1}{\sqrt{m_2}} \Big(\sum_i \|\phi^*(x_i)\|_2^2\Big) \leq \frac{1}{\sqrt{m_2}} n(1+\Re) \sqrt{\zeta_2 \sum_k \|\mathcal{V}_k\|_{H^\infty}^2/\lambda_0}. \tag{135}$$

A.13    EXISTENCE OF A GOOD DIRECTION

Our aim in this section is to show that if the objective value is above certain threshold, there exists a good random direction which reduces the objective in expectation. Particularly our aim is to prove the following theorem (informal):

**Theorem 6** *For a given pair $(f^*, G)$ with*

$$\langle H^\infty, G \rangle \leq \zeta_1,$$
$$f^{*T}(K^\infty \odot G)^{-1} f^* \leq \zeta_2,$$
$$R_n(f^*) \leq \Delta,$$

*recall the ideal random matrices $(W_\Sigma^*, V_\Sigma^*)$ constructed in Appendix A.12, where $\Sigma$ is a random diagonal sign matrix. Specifically, $W_\Sigma^*$ is defined in Equation (107), and $V_\Sigma^*$ is the projection of the rows of matrix $V^*$ onto the orthogonal subspace spanned by $(\phi^{(0)}(x_i))_{i=1}^n$.*

*Using the parameter setting for $i = 1, 2$*

$$\psi_i = \frac{\nu}{4\zeta_i}, \tag{136}$$

*with respect to an arbitrary parameter $\nu > 0$, then for every pair $(W', V')$ such that $\|W'\| \leq C_1, \|V'\| \leq C_2$ and*

$$L(W', V') \geq \Delta + \nu, \tag{137}$$

*for parameters $m_1, m_2, m_3, 1/\kappa_1, 1/\kappa_2$ polynomially large enough in $B, 1/\lambda_0, n, C_1, C_2$ and small enough step size $\eta$, we have*

$$\mathbb{E}_\Sigma L(W' - \eta/2W' + \sqrt{\eta}W_\Sigma^*, V' - \eta/2V' + \sqrt{\eta}V_\Sigma^*) \leq L(W', V') - \eta\nu/4. \tag{138}$$

In order to prove the above theorem, we first state and prove the following lemma which is the core of Theorem 6.

**Lemma 11** *For matrices $(W^*, V^*)$ constructed in Appendix A.12, specifically for their random coupling $(W_\Sigma^*, V_\Sigma^*)$ as denoted above, we have:*

$$\mathbb{E}_\Sigma \ell(f'_{(1-\eta/2)W' + \sqrt{\eta}W_\Sigma^*, (1-\eta/2)V' + \sqrt{\eta}V_\Sigma^*}(x_i), y_i) \leq (1 - \eta)\ell(f'_{W', V'}(x_i), y_i) + \eta\ell(f^*(x_i), y_i) \pm \eta\wp,$$

*where $\wp$ goes to zero with polynomially large overparameterization (the exact dependence is revealed via the proof).*

**Proof of Lemma 11**

For brevity, we use the notation $D'_{,\rho}$ here to refer to the diagonal binary sign matrix when the input is multiplied by the sum of weight and smoothing matrices. It will be clear in the context of the equation that what the "input" and the "weight" matrices are. This notation is also defined and used in Lemma 28). Here, we bound multiple cross terms that are created as a result of moving in the random direction. To simplify the presentation and avoid confusing recursions in the proof, we have made a sublemma for each of these cross terms and has deferred its proof to Appendix A.14. We use difference sub-indices of the symbol $\Re$ to illustrate terms that go to zero by growing the overparameterization in our architecture.

We start by using Lemma 28,

$$
\mathbb{E}_\Sigma \ell(f'_{(1-\eta/2)W' + \sqrt{\eta}W^*_\Sigma, (1-\eta/2)V' + \sqrt{\eta}V^*_\Sigma}(x_i), y_i)
$$
$$
= \mathbb{E}_\Sigma \ell(\mathbb{E}_{W^\rho, V^\rho} f_{(1-\eta/2)W' + \sqrt{\eta}W^*_\Sigma + W^\rho, (1-\eta/2)V' + \sqrt{\eta}V^*_\Sigma + V^\rho}(x_i), y_i)
$$
$$
= \mathbb{E}_\Sigma \ell\Big( \mathbb{E}_{W^\rho, V^\rho} a^T D_{',\rho}(V^{(0)} + (1-\eta/2)V' + V^\rho + \sqrt{\eta}V^*_\Sigma)W^s D_{',\rho}(W^{(0)} + (1-\eta/2)W' + W^\rho + \sqrt{\eta}W^*_\Sigma)x_i
$$
$$
+ \Re_8\eta, y_i\Big).
$$
$$
= \mathbb{E}_\Sigma \ell\Big( \mathbb{E}_{W^\rho, V^\rho} \Big[ a^T D_{',\rho}(V^{(0)} + (1-\eta/2)V' + V^\rho)W^s D_{',\rho}(W^{(0)} + (1-\eta/2)W' + W^\rho)x_i
$$
$$
+ \eta a^T D_{',\rho}V^*_\Sigma W^s D_{',\rho}W^*_\Sigma x_i \Big] + \sqrt{\eta}\mathbb{E}_{W^\rho, V^\rho}\Big[ a^T D_{',\rho}(V^{(0)} + (1-\eta/2)V' + V^\rho)W^s D_{',\rho}W^*_\Sigma x_i
$$
$$
+ a^T D_{',\rho}V^*_\Sigma W^s D_{',\rho}(W^{(0)} + (1-\eta/2)W' + W^\rho)x_i \Big]
$$
$$
+ \Re_8\eta, y_i\Big).
$$

Now using the notation introduced in Lemma 18, we have

$$
W^s D_{',\rho}(W^{(0)} + (1-\eta/2)W' + W^\rho)x_i = \phi^{(0)}(x_i) + (1-\eta/2)\phi^{(2)}(x_i) + \frac{\eta}{2}\phi^{(2)'}(x_i).
$$

By Lemma 18, we have the following bound for $\phi^{(2)'}(x_i)$:

$$
\mathbb{E}_{W^\rho, V^\rho}\Big| \frac{1}{\sqrt{m_2}}a^T D_{V^{(0)} + V^\rho + V', x_i}(V^{(0)} + V^\rho + (1-\eta/2)V')\phi^{(2)'}(x_i) \Big|
$$
$$
\lesssim (\kappa_2\sqrt{m_2 m_3} + \sqrt{m_3}\beta_2 + C_2)\Re_5.
$$

Therefore, Combining this with Lemma 20, we get

$$
= \mathbb{E}_\Sigma \ell\Big( \mathbb{E}_{W^\rho, V^\rho}\Big[ a^T D_{',\rho}(V^{(0)} + (1-\eta/2)V' + V^\rho)W^s(\phi^{(0)}(x_i) + (1-\eta/2)\phi^{(2)}(x_i))
$$
$$
+ \eta a^T D_{',\rho}V^*_\Sigma W^s D_{',\rho}W^*_\Sigma x_i \Big] + \sqrt{\eta}\mathbb{E}_{W^\rho, V^\rho}\Big[ a^T D_{',\rho}(V^{(0)} + (1-\eta/2)V' + V^\rho)W^s D_{',\rho}W^*_\Sigma x_i
$$
$$
+ a^T D_{',\rho}V^*_\Sigma W^s D_{',\rho}(W^{(0)} + (1-\eta/2)W' + W^\rho)x_i \Big]
$$
$$
\pm O((\kappa_2\sqrt{m_2 m_3} + \sqrt{m_3}\beta_2 + C_2)\Re_5\eta) \pm O(\Re_8\eta), y_i\Big)
$$
$$
= \mathbb{E}_\Sigma \ell\Big( \mathbb{E}_{W^\rho, V^\rho}\Big[ (1-\eta)a^T D_{',\rho}(V^{(0)} + V' + V^\rho)W^s(\phi^{(0)}(x_i) + \phi^{(2)}(x_i))
$$
$$
+ \eta a^T D_{',\rho}V^*_\Sigma W^s D_{',\rho}W^*_\Sigma x_i \Big] + \sqrt{\eta}\mathbb{E}_{W^\rho, V^\rho}\Big[ a^T D_{',\rho}(V^{(0)} + (1-\eta/2)V' + V^\rho)W^s D_{',\rho}W^*_\Sigma x_i
$$
$$
+ a^T D_{',\rho}V^*_\Sigma W^s D_{',\rho}(W^{(0)} + (1-\eta/2)W' + W^\rho)x_i \Big]
$$
$$
\pm O(\eta(\Re'_6 + \Re_4 + (\sqrt{m_3}\kappa_2 + \beta_2)(C_1 + \sqrt{m_3}\beta_1))) \pm O((\kappa_2\sqrt{m_2 m_3} + \sqrt{m_3}\beta_2 + C_2)\Re_5\eta) \pm O(\Re_8\eta), y_i\Big).
$$
$$
= \mathbb{E}_\Sigma \ell\Big( (1-\eta)f'_{W', V'}(x_i) + \eta\mathbb{E}_{W^\rho, V^\rho}a^T D_{',\rho}V^*_\Sigma W^s D_{',\rho}W^*_\Sigma x_i
$$
$$
+ \sqrt{\eta}\mathbb{E}_{W^\rho, V^\rho}\Big[ a^T D_{',\rho}(V^{(0)} + (1-\eta/2)V' + V^\rho)W^s D_{',\rho}W^*_\Sigma x_i
$$
$$
+ a^T D_{',\rho}V^*_\Sigma W^s D_{',\rho}(W^{(0)} + (1-\eta/2)W' + W^\rho)x_i \Big]
$$
$$
\pm O(\eta(\Re'_6 + \Re_4 + (\sqrt{m_3}\kappa_2 + \beta_2)(C_1 + \sqrt{m_3}\beta_1))) \pm O((\kappa_2\sqrt{m_2 m_3} + \sqrt{m_3}\beta_2 + C_2)\Re_5\eta) \pm O(\Re_8\eta), y_i\Big).
$$

Moreover, using the notation $\phi^{*\prime}(x_i)$ introduced in Lemma 15 and the bound in Lemma 17, we can rewrite the second term as:

$$
\begin{aligned}
LHS = {} & \mathbb{E}_\Sigma \ell \Big( (1-\eta) f'_{W',V'}(x_i) + \eta \mathbb{E}_{W^\rho,V^\rho} a^T D_{\cdot,\rho} V^*_\Sigma (\phi^*(x_i) + \phi^{*\prime}(x_i)) \\
& + \sqrt{\eta}\, \mathbb{E}_{W^\rho,V^\rho} \Big[ a^T D_{\cdot,\rho}(V^{(0)} + (1-\eta/2)V' + V^\rho) W^s D_{\cdot,\rho} W^*_\Sigma x_i \\
& + a^T D_{\cdot,\rho} V^*_\Sigma W^s D_{\cdot,\rho}(W^{(0)} + (1-\eta/2)W' + W^\rho) x_i \Big] \\
& \pm O(\eta(\Re'_6 + \Re_4 + (\sqrt{m_3}\kappa_2 + \beta_2)(C_1 + \sqrt{m_3}\beta_1))) \pm O((\kappa_2\sqrt{m_2 m_3} + \sqrt{m_3}\beta_2 + C_2)\Re_5\eta) \pm O(\Re_8\eta), y_i \Big) \\
= {} & \mathbb{E}_\Sigma \ell \Big( (1-\eta) f'_{W',V'}(x_i) + \eta \mathbb{E}_{W^\rho,V^\rho} a^T D_{\cdot,\rho} V^*_\Sigma \phi^*(x_i) \\
& + \sqrt{\eta}\, \mathbb{E}_{W^\rho,V^\rho} \Big[ a^T D_{\cdot,\rho}(V^{(0)} + (1-\eta/2)V' + V^\rho) W^s D_{\cdot,\rho} W^*_\Sigma x_i \\
& + a^T D_{\cdot,\rho} V^*_\Sigma W^s D_{\cdot,\rho}(W^{(0)} + (1-\eta/2)W' + W^\rho) x_i \Big] \\
& \pm O(\eta\Re_{10}) \pm O(\eta(\Re'_6 + \Re_4 + (\sqrt{m_3}\kappa_2 + \beta_2)(C_1 + \sqrt{m_3}\beta_1))) \pm O((\kappa_2\sqrt{m_2 m_3} + \sqrt{m_3}\beta_2 + C_2)\Re_5\eta) \pm \\
& O(\Re_8\eta), y_i \Big).
\end{aligned}
\tag{139}
$$

Now we write the gradient-lipshitz inequality for $\ell$ at point

$$
p^{(1)}_\Sigma := (1-\eta) f'_{W',V'}(x_i) + \eta \mathbb{E}_{W^\rho,V^\rho} a^T D_{\cdot,\rho} V^*_\Sigma \phi^*(x_i) \pm \eta\wp_1,
$$

and regarding the following vector, where $\wp_1$ is the sum of all the noise terms above and goes to zero by over parameterization:

$$
\begin{aligned}
p^{(2)}_\Sigma := {} & \sqrt{\eta}\, \mathbb{E}_{W^\rho,V^\rho} \Big[ a^T D_{\cdot,\rho}(V^{(0)} + (1-\eta/2)V' + V^\rho) W^s D_{\cdot,\rho} W^*_\Sigma x_i \\
& + a^T D_{\cdot,\rho} V^*_\Sigma W^s D_{\cdot,\rho}(W^{(0)} + (1-\eta/2)W' + W^\rho) x_i \Big].
\end{aligned}
$$

Hence, using the 1 smoothness of $\ell(., y_i)$:

$$
LHS \le \mathbb{E}_\Sigma \ell\Big(p^{(1)}_\Sigma\Big) + \mathbb{E}_\Sigma \dot{\ell}(p)\sqrt{\eta} p^{(2)}_\Sigma + \frac{1}{2}\eta\mathbb{E}_\Sigma (p^{(2)}_\Sigma)^2.
\tag{140}
$$

But note that

$$
\mathbb{E}_\Sigma \dot{\ell}(p)\sqrt{\eta} p^{(2)}_\Sigma = \dot{\ell}(p)\sqrt{\eta}\mathbb{E}_\Sigma p^{(2)}_\Sigma = 0.
\tag{141}
$$

On the other hand, using the notation of Lemma 15 and the result of Lemma 16:

$$
\mathbb{E}_\Sigma \Big( \mathbb{E}_{W^\rho,V^\rho} a^T D_{\cdot,\rho}(V^{(0)} + (1-\eta/2)V' + V^\rho) W^s D_{\cdot,\rho} W^*_\Sigma x_i \Big)^2
\tag{142}
$$

$$
= \mathbb{E}_\Sigma \Big( \mathbb{E}_{W^\rho,V^\rho} a^T D_{\cdot,\rho}(V^{(0)} + (1-\eta/2)V' + V^\rho)(\Sigma\phi^*(x_i) + \phi^{*\prime}_\Sigma(x_i)) \Big)^2
\tag{143}
$$

$$
\le 4\mathbb{E}_\Sigma \Big( \mathbb{E}_{W^\rho,V^\rho} a^T D_{\cdot,\rho}(V^{(0)} + (1-\eta/2)V' + V^\rho)\Sigma\phi^*(x_i) \Big)^2
\tag{144}
$$

$$
+ 4\mathbb{E}_\Sigma \Big( \mathbb{E}_{W^\rho,V^\rho} a^T D_{\cdot,\rho}(V^{(0)} + (1-\eta/2)V' + V^\rho)\phi^{*\prime}_\Sigma(x_i) \Big)^2
\tag{145}
$$

$$
\lesssim \Re^2_{12} + \Re^2_{11}.
\tag{146}
$$

Moreover, using again the result on $\phi^{(2)\prime}(x_i)$ from Lemma 18 and the fact that $\phi^{(0)}(x_i)$ is orthogonal to the rows of $V^*_\Sigma$:

$$
\begin{aligned}
& \mathbb{E}_{W^\rho,V^\rho} a^T D_{\cdot,\rho} V^*_\Sigma W^s D_{\cdot,\rho}(W^{(0)} + (1-\eta/2)W' + W^\rho) x_i \\
& = a^T D_{\cdot,\rho} V^*_\Sigma (\phi^{(0)}(x_i) + (1-\eta/2)\phi^{(2)}(x_i)) \\
& \quad + \frac{\eta}{2} a^T D_{\cdot,\rho} V^*_\Sigma \phi^{(2)\prime}(x_i) \\
& = (1-\frac{\eta}{2}) a^T D_{\cdot,\rho} V^*_\Sigma \phi^{(2)}(x_i) \\
& \quad + \frac{\eta}{2} a^T D_{\cdot,\rho} V^*_\Sigma W^s D_{\cdot,\rho} \phi^{(2)\prime}(x_i) \\
& \lesssim (1-\frac{\eta}{2}) a^T D_{\cdot,\rho} V^*_\Sigma \phi^{(2)}(x_i) \pm \Re_5.
\end{aligned}
$$

Combining the last Equation with Lemma 14:

$$
\begin{aligned}
&\mathbb{E}_\Sigma \Big( \mathbb{E}_{W^\rho, V^\rho} a^T D_{\prime,\rho} V_\Sigma^* W^s D_{\prime,\rho} (W^{(0)} + (1 - \eta/2)W' + W^\rho) x_i \Big)^2 \\
&\lesssim (1 - \frac{\eta}{2})^2 a^T D_{\prime,\rho} V_\Sigma^* W^s D_{\prime,\rho} \phi^{(2)}(x_i) + \Re_5^2 \\
&\lesssim \Re_0^2 + \Re_5^2.
\end{aligned}
\tag{147}
$$

Combining Equations (146) and (147):

$$
\mathbb{E}_\Sigma (p_\Sigma^{(2)})^2 \lesssim \Re_{11}^2 + \Re_{12}^2 + \Re_0^2 + \Re_5^2 := \wp_2.
\tag{148}
$$

Combining Equations (141) and (148), plugging into (140), and reopening the definition of $p_\Sigma^{(1)}$:

$$
LHS \lesssim \mathbb{E}_\Sigma \ell \Big( (1 - \eta) f'_{W',V'}(x_i) + \eta \mathbb{E}_{W^\rho, V^\rho} a^T D_{\prime,\rho} V_\Sigma^* \phi^*(x_i) \pm \eta \wp_1, y_i \Big) + \eta \mu_2 \wp_2.
\tag{149}
$$

Now note that we can easily bound the magnitude of the term $\eta \mathbb{E}_{W^\rho, V^\rho} a^T D_{\prime,\rho} V_\Sigma^* \phi^*(x_i)$ as:

$$
\begin{aligned}
|\mathbb{E}_{W^\rho, V^\rho} a^T D_{\prime,\rho} V_\Sigma^* \phi^*(x_i)| &\le \mathbb{E}_{W^\rho, V^\rho} |a^T D_{\prime,\rho} V_\Sigma^* \phi^*(x_i)| \\
&\le \|V_\Sigma^*\|_F \|\phi^*(x_i)\| \le \|V^*\| \|\phi^*(x_i)\| \le \sqrt{2\zeta_2}(1 + \Re) \sqrt{\sum_k \|\mathcal{V}_k\|_{H^\infty}^2},
\end{aligned}
$$

while using Lemma 34:

$$
|f'_{W',V'}(x_i)| \le (\kappa_2 \sqrt{m_3} + \beta_2)\Big( \sqrt{m_3}\kappa_1 + C_1 + \sqrt{m_3}\beta_1 \Big) + C_2(C_1 + \sqrt{m_3}\beta_1),
$$

which is $O(C_1 C_2)$ for enough overparameterization and smoothing parameters $\beta_1, \beta_2$ as defined in A.20.1. Furthermore, from Equations (131) and (126), we easily see that

$$
\mathbb{E}_{W^\rho, V^\rho} a^T D_{\prime,\rho} V_\Sigma^* \phi^*(x_i) \le \sqrt{2\zeta_2}(1 + \Re) \sqrt{\sum_k \|\mathcal{V}_k\|_{H^\infty}^2}.
$$

Now taking $\eta$ small enough so that the bound $\eta \sqrt{2\zeta_2}(1 + \Re) \sqrt{\sum_k \|\mathcal{V}_k\|_{H^\infty}^2}$ and $\eta \wp_1$ both also be bounded of order $O(C_1 C_2)$, we observe that the term inside the argument of $\ell(., y_i)$ Equation in (149) is $O(C_1 C_2)$. Hence, we can use the Lipschitz parameter of $\ell$ in the interval $[-O(C_1 C_2), O(C_1 C_2)]$, given by Lemma 9 to take out the noise term:

$$
LHS \lesssim \mathbb{E}_\Sigma \ell \Big( (1 - \eta) f'_{W',V'}(x_i) + \eta \mathbb{E}_{W^\rho, V^\rho} a^T D_{\prime,\rho} V_\Sigma^* \phi^*(x_i), y_i \Big) \pm \eta \wp_1 + \eta O(C_1 C_2 + B) \wp_2.
\tag{150}
$$

Now by applying Lemma 19 and writing the Lipchitz property of $\ell$ at point $(1 - \eta) f'_{W',V'}(x_i) = O(C_1 C_2)$:

$$
\begin{aligned}
LHS &\lesssim \mathbb{E}_\Sigma \ell \Big( (1 - \eta) f'_{W',V'}(x_i) + \eta f^*(x_i) \pm \eta \Re_9, y_i \Big) \pm \eta \wp_1 + \eta O(C_1 C_2 + B) \wp_2 \\
&:= \mathbb{E}_\Sigma \ell \Big( (1 - \eta) f'_{W',V'}(x_i) + \eta f^*(x_i), y_i \Big) \pm \eta \Re_9 \pm \eta \wp_1 + \eta O(C_1 C_2 + B) \wp_2 \\
&= \ell \Big( (1 - \eta) f'_{W',V'}(x_i) + \eta f^*(x_i), y_i \Big) \pm \eta \wp,
\end{aligned}
$$

where the last line is just definition. Now Convexity of $\ell$ finishes the proof.

Next, using Lemma 11 we prove Theorem 6.

**Restating Theorem 6** In the same setting as Theorem 6 and having enough overparameterization such that $\wp \le \frac{\nu}{8}$ ($\wp$ defined in Lemma 11) and polynomially small enough step size $\eta$, we have

$$
\mathbb{E}_\Sigma L(W' - \eta W' + \sqrt{\eta} W_\Sigma^*, V' - \eta V' + \sqrt{\eta} V_\Sigma^*) \le L(W', V') - \eta \nu/4.
$$

**Proof of Theorem 6**

First, note that taking expectation w.r.t $\Sigma$:

$$\mathbb{E}_\Sigma \|(1-\eta/2)W' + \sqrt{\eta}W_\Sigma^*\|^2 = \mathbb{E}_\Sigma(1-\eta/2)^2\|W'\|^2 + 2(1-\eta/2)\sqrt{\eta}\langle W', \sum_{k=1}^{m_3}\Sigma_k W_k^*\rangle + \eta\|\sum_{k=1}^{m_3}\Sigma_k W_k^*\|^2$$

$$= (1-\eta/2)^2\|W'\|^2 + \eta\sum_k\|W_k^*\|^2,$$

which by orthogonality of $W_k^*$'s:

$$LHS = (1-\eta/2)^2\|W'\|^2 + \eta\|W^*\|^2 = (1-\eta)\|W'\|^2 + \eta\|W^*\|^2 + \eta^2\|W'\|^2.$$

Similarly for $V'$:

$$\mathbb{E}_\Sigma\|(1-\eta/2)V' + \sqrt{\eta}V_\Sigma^*\| = (1-\eta/2)^2\|V'\|^2 + \eta\mathbb{E}_\Sigma\|V^*\Sigma\|^2 = (1-\eta)\|V'\|^2 + \eta\|V^*\|^2 + \eta^2\|V'\|^2.$$

Now using Lemma 11:

$$\mathbb{E}_\Sigma L(W' - \eta/2W' + \sqrt{\eta}W_\Sigma^*, V' - \eta/2V' + \sqrt{\eta}V_\Sigma^*)$$
$$\leq (1-\eta)\mathbb{E}_{\mathcal{Z}}\ell(f'_{W',V'}(x), y) + \eta\mathbb{E}_{\mathcal{Z}}\ell(f^*(x), y)$$
$$+ (1-\eta)\Big(\psi_1\|W'\|^2 + \psi_2\|V'\|^2\Big) + \eta\Big(\psi_1\|W^*\|^2 + \psi_2\|V^*\|^2\Big) + \eta\Big(\wp + \eta(\|W'\|^2 + \|V'\|^2)\Big)$$
$$\leq L(W', V') - \eta\Big(L(W', V') - \Delta - \psi_1\zeta_1 - \psi_2\zeta_2\Big) + \eta\Big(\wp + \eta(\|W'\|^2 + \|V'\|^2)\Big),$$

which by the choice of $\psi_i$'s is equal to

$$LHS \leq L(W', V') - \eta\Big(L(W', V') - \Delta - \nu/2\Big) + \eta\Big(\wp + \eta(\|W'\|^2 + \|V'\|^2)\Big)$$
$$LHS \leq L(W', V') - \eta\nu/2 + \eta\Big(\wp + \eta(\|W'\|^2 + \|V'\|^2)\Big).$$

Moreover, using the condition

$$\wp \leq \nu/8,$$

and picking $\eta$ as small as

$$\eta(\|W'\|^2 + \|V'\|^2) \leq \eta(C_1^2 + C_2^2) \leq \nu/8,$$

we finally get

$$LHS \leq L(W', V') - \eta\nu/4.$$

## A.14 Existence of a good direction Helper Lemmas

In this section, we state and prove the core lemmas that are used in the proof of Lemma 11. Notably, through all of this section, we assume the norm bounds $\|W'\| \leq C_1, \|V'\| \leq C_2$ and that as our usual assumption, the rows of $V'$ are orthogonal to $\phi^{(0)}(x_i)$'s for all $i \in [n]$. A notation that we use throughout the proofs is $V_\Sigma^*$ which refers to the projectiono of $V^*\Sigma$ onto the orthogonal subspace to $(\phi^{(0)}(x_i))_{i=1}^n$.

**Lemma 12** *Let $P(.)$ be the projection operator onto the subspace spanned by $(\phi^{(0)}(x_i))_{i=1}^n$. Also, we denote the projection of rows of $V^*\Sigma$ onto the orthogonal subspace to $(\phi^{(0)}(x_i))_{i=1}^n$ by $V_{\Sigma j}^*$. Then*

$$\mathbb{E}_\Sigma \|V_{\Sigma j}^* - V_j^*\Sigma)\|^2 \leq \varrho_3^2 \xi^2 n/m_2,$$

*with high probability*

$$\|V_{\Sigma j}^* - V_j^*\Sigma)\| \lesssim \frac{\varrho_3 \xi \sqrt{n}}{\sqrt{m_2}},$$

**Proof of Lemma 12**

By Equation (128), we have $\|V_j^*\|_\infty \leq \varrho_3 \xi/\sqrt{m_2}$. Now suppose that $u_1, ..., u_n$ are an orthonormal basis for the subspace $span(\phi^{(0)}(x_i))_{i=1}^n$. Then

$$\mathbb{E}_\Sigma \|V_{\Sigma j}^* - V_j^*\Sigma)\|^2 = \mathbb{E}_\Sigma \|P(V_j^*\Sigma)\|^2 = \sum_i \sum_k V_{j\,k}^{*\,2} u_{ik}^2 \leq \|V_j^*\|_\infty^2 n \leq \varrho_3^2 \xi^2 n/m_2.$$

Also, by Hoeffding, with high probability:

$$\|P(V_j^*\Sigma)\|^2 = \sum_i (\sum_{k=1}^{m_3} V_{j\,k}^* u_{ik}\Sigma_k)^2 \lesssim n\|V_j^*\|_\infty^2,$$

which implies the second part.

**Lemma 13** *The first cross term goes away because of the definition of $V_\Sigma^*$. (inside the expectations is zero almost surely)*

$$\mathbb{E}_\Sigma \left(\mathbb{E}_{V^\rho,W^\rho}[\frac{1}{\sqrt{m_2}} a^T D_{V^{(0)}+V^\rho+V',x_i} V_\Sigma^* \phi^{(0)}(x_i)]\right)^2 = 0.$$

**Lemma 14** *Second cross term:*

$$\mathbb{E}_\Sigma \left(\mathbb{E}_{V^\rho,W^\rho}[\frac{1}{\sqrt{m_2}} a^T D_{V^{(0)}+V^\rho+V',x_i} V_\Sigma^* \phi^{(2)}(x_i)]\right)^2 \tag{151}$$

$$\lesssim \xi^2((1+\Re)^2 n\zeta_2 + \varrho_3^2 n)(C_1^2 + m_3\beta_1^2) = \Re_0^2. \tag{152}$$

**Proof of Lemma 14**

This time we use Equation (128) in Lemma 10 and Lemma 12:

$$\mathbb{E}_{\Sigma}\Big(\mathbb{E}_{V^{\rho},W^{\rho}}\big[\frac{1}{\sqrt{m_2}}a^T D_{V^{(0)}+V^{\rho}+V',x_i}V_{\Sigma}^*\phi^{(2)}(x_i)\big]\Big)^2$$

$$\leq \mathbb{E}_{\Sigma}\Big(\mathbb{E}_{V^{\rho},W^{\rho}}1/\sqrt{m_2}\sum_j |V_{\Sigma j}^*\phi^{(2)}(x_i)|\Big)^2$$

$$\leq \frac{1}{m_2}\mathbb{E}_{\Sigma,V^{\rho},W^{\rho}}\Big(\sum_j |V_{\Sigma j}^*\phi^{(2)}(x_i)|\Big)^2$$

$$\leq \mathbb{E}_{V^{\rho},W^{\rho}}E_{\Sigma}\sum_j |V_{\Sigma j}^*\phi^{(2)}(x_i)|^2$$

$$\lesssim E_{V^{\rho},W^{\rho}}E_{\Sigma}\sum_j |(V_{\Sigma j}^* - V_j^*\Sigma)\phi^{(2)}(x_i)|^2 + \sum_j |V_j^*\Sigma\phi^{(2)}(x_i)|^2$$

$$\lesssim \mathbb{E}_{V^{\rho},W^{\rho}}E_{\Sigma}\sum_j \|V_{\Sigma j}^* - V_j^*\Sigma)\|^2\|\phi^{(2)}(x_i)\|^2 + \sum_j \|V_j^*\|_{\infty}^2\|\phi^{(2)}(x_i)\|_2^2$$

$$\lesssim ((1+\Re)^2 n\zeta_2\xi^2 + \varrho_3^2\xi^2 n)\mathbb{E}_{V^{\rho},W^{\rho}}\|\phi^{(2)}(x_i)\|_2^2.$$

Now according to Lemma 33, we have

$$\mathbb{E}_{V^{\rho},W^{\rho}}\|\phi^{(2)}(x_i)\|^2 \lesssim C_1{}^2 + m_3\beta_1{}^2,$$

which completes the proof.

**Lemma 15** *We get an additional term $\phi^{*\prime}(x_i)$ as a result of smoothing which we define as*

$$\phi^{*\prime}(x_i) = \frac{1}{\sqrt{m_1}}W^s D_{W^{(0)}+W'+W^{\rho},x_i}W_{\Sigma}^* x_i - \phi^*{}_{\Sigma}(x_i). \tag{153}$$

*Then*

$$\mathbb{P}(\phi^{*\prime}(x_i) \neq 0) \leq m_1 \exp\{-c_2^2/(8\beta_1^2)\}.$$

*Moreover, we have the following inequality almost surely (over the randomness of $W^{\rho}$):*

$$\|\phi^{*\prime}(x_i)\|_{\infty} \lesssim \sqrt{\sum_k \|\mathcal{V}_k\|_{H^{\infty}}^2}.$$

**Proof of Lemma 15**

According to Lemma 1, for $j \notin P$, for every $i \in [n]$ we have

$$|(W_j^{(0)} + W_j')x_i| \geq c_2/2\sqrt{m_1}.$$

Now note that as long as the sign patterns for $j \notin P$ does not change, $\phi^{*\prime}(x_i)$ will be zero. Therefore by union bound

$$\mathbb{P}(\phi^{*\prime}(x_i) \neq 0) \leq \sum_{j=1}^{m_1}\mathbb{P}(\text{sign change in } j) \leq m_1\mathbb{P}(|(W_j^{(0)} + W_j')x_i| \leq |W_j^{\rho}x_i|)$$

$$\leq m_1\mathbb{P}(|W_j^{\rho}x_i| \geq c_2/(2\sqrt{m_1})).$$

But $(W_j^{\rho})x_i$ is Gaussian with variance $\beta_1^2/m_1$. Hence

$$LHS \lesssim m_1 \exp\{-c_2^2/(8\beta_1^2)\},$$

which proves the first part. For the second part, according to Equation (105) in Lemma 5, for every $k \in [m_3]$:

$$|\phi^{*\prime}{}_k(x_i)| \leq |\frac{1}{\sqrt{m_1}}W_k^s D_{W^{(0)}+W'+W^{\rho},x_i}W^* x_i| + |\phi^*{}_k(x_i)| \tag{154}$$

$$\leq 2/\sqrt{m_1}\sum_j \|W_j^*\| \leq 2\|W^*\|_F \lesssim \sqrt{\sum_k \|\mathcal{V}_k\|_{H^{\infty}}^2}, \tag{155}$$

which implies the second part.

**Lemma 16** *Fourth Extra term:*

$$\mathbb{E}_{W^\rho, V^\rho}[\frac{1}{\sqrt{m_2}} a^T D_{V^{(0)}+V^\rho+V', x_i}(V^{(0)} + V^\rho + (1-\eta/2)V')\phi^{*'}(x_i)]$$

$$\lesssim (\kappa_2\sqrt{m_2 m_3} + C_2 + \sqrt{m_3}\beta_2)\, m_1 \exp\{-c_2^2/(8\beta_1^2)\} \sqrt{\sum_k \|\mathcal{V}_k\|_{H^\infty}^2} := \Re_{11}.$$

**Proof of Lemma 16**

Note that with high probability over the randomness of $V^{(0)}$, we have $\|V^{(0)}\|_F \lesssim \sqrt{m_2 m_3}\kappa_2$. Now according to Lemma 15 and using the fact that $\|V'\|_F \le C_2$:

$$\le \mathbb{E}_{W^\rho, V^\rho} \frac{1}{\sqrt{m_2}} \|a\| \|V^{(0)} + V^\rho + (1-\eta)V'\|_2 \|\phi^{*'}(x_i)\|$$

$$= \mathbb{E}_{W^\rho, V^\rho} \|V^{(0)} + V^\rho + (1-\eta/2)V'\|_F \|\phi^{*'}(x_i)\|$$

$$\le \sqrt{\mathbb{E}_{V^\rho}(\|V^{(0)} + (1-\eta/2)V'\|_F^2 + \|V^\rho\|_F^2)}\, m_1 \exp\{-c_2^2/(8\beta_1^2)\} \sqrt{\sum_k \|\mathcal{V}_k\|_{H^\infty}^2}$$

$$\lesssim \sqrt{\|V^{(0)}\|_F^2 + \|V'\|_F^2 + m_3\beta_2^2}\, m_1 \exp\{-c_2^2/(8\beta_1^2)\} \sqrt{\sum_k \|\mathcal{V}_k\|_{H^\infty}^2}$$

$$\lesssim (\kappa_2\sqrt{m_2 m_3} + C_2 + \sqrt{m_3}\beta_2)\, m_1 \exp\{-c_2^2/(8\beta_1^2)\} \sqrt{\sum_k \|\mathcal{V}_k\|_{H^\infty}^2}.$$

**Lemma 17** *Fifth extra term:*

$$\left|\mathbb{E}_{W^\rho, V^\rho}[\frac{1}{\sqrt{m_2}} a^T D_{V^{(0)}+V^\rho+V', x_i} V_\Sigma^* \phi^{*'}(x_i)]\right| \lesssim \sqrt{\zeta_2} m_1 \exp\{-c_2^2/(8\beta_1^2)\} \sqrt{\sum_k \|\mathcal{V}_k\|_{H^\infty}^2} := \Re_{10}.$$

**Proof of Lemma 17**

Similar to the previous Lemma, the inner expectation can be bounded as:

$$\le \mathbb{E} \frac{1}{\sqrt{m_2}} \|a\| \|V_\Sigma^*\|_F \|\phi^{*'}(x_i)\| \le \mathbb{E}\|V^*\|_F \|\phi^{*'}(x_i)\| \lesssim \sqrt{\zeta_2} m_1 \exp\{-c_2^2/(8\beta_1^2)\} \sqrt{\sum_k \|\mathcal{V}_k\|_{H^\infty}^2}.$$

**Lemma 18** *We have another extra term as a product of the movement $-\eta/2W'$ in the first layer:*

$$\phi^{(2)'}(x_i) = \frac{2}{\eta\sqrt{m_1}} \left[W^s D_{W^{(0)}+W^\rho+W'}(W^{(0)}+W^\rho+(1-\eta/2)W')x_i - \phi^{(0)}(x_i) - (1-\eta/2)\phi^{(2)}(x_i)\right].$$

*Then*

$$\left|\mathbb{E}_{W^\rho, V^\rho} \frac{1}{\sqrt{m_2}} a^T D_{V^{(0)}+V^\rho+V', x_i} V_\Sigma^* \phi^{(2)'}(x_i)\right|$$

$$\lesssim \sqrt{\zeta_2 m_3 \left(\frac{\beta_1^2}{m_1} + \frac{c_2 C_1^2}{\sqrt{m_1}\kappa_1} + m_1 \exp\{-c_2^2/(8\beta_1^2)\}C_1^2\right)} := \Re_5. \tag{156}$$

$$\mathbb{E}_{W^\rho, V^\rho} \left|\frac{1}{\sqrt{m_2}} a^T D_{V^{(0)}+V^\rho+V', x_i}(V^{(0)} + V^\rho + (1-\eta/2)V')\phi^{(2)'}(x_i)\right|$$

$$\lesssim (\kappa_2\sqrt{m_2 m_3} + \sqrt{m_3}\beta_2 + C_2)\Re_5. \tag{157}$$

**Proof of Lemma 18**

First we prove the following approximation argument (for all $k \in [m_3]$):

$$\mathbb{E}_{W^\rho}|\phi^{(2)'}(x_i)_k|^2 \le \frac{\beta_1^2}{m_1} + \frac{c_2 C_1^2}{\sqrt{m_1}\kappa_1} + m_1 \exp\{-c_2^2/(8\beta_1^2)\}C_1^2. \tag{158}$$

We have

$$LHS = \mathbb{E}_{W^\rho} \Big| \frac{1}{\sqrt{m_1}} W_k^s D_{W^{(0)}+W^\rho+W',x_i}(W^{(0)}+W^\rho)x_i - \frac{1}{\sqrt{m_1}} W_k^s D_{W^{(0)},x_i} W^{(0)} x_i \Big|^2$$

$$\lesssim \mathbb{E}_{W^\rho} \Big| \frac{1}{\sqrt{m_1}} W_k^s D_{W^{(0)}+W^\rho,x_i}(W^{(0)}+W^\rho)x_i - \frac{1}{\sqrt{m_1}} W_k^s D_{W^{(0)},x_i} W^{(0)} x_i \Big|^2$$

$$+\mathbb{E}_{W^\rho} \Big| \frac{1}{\sqrt{m_1}} W_k^s D_{W^{(0)}+W^\rho,x_i}(W^{(0)}+W^\rho)x_i - \frac{1}{\sqrt{m_1}} W_k^s D_{W^{(0)}+W^\rho+W',x_i}(W^{(0)}+W^\rho)x_i \Big|^2$$

By the independence of $W_j^\rho$'s, the first term can be upper bounded as

$$= \mathbb{E}_{W^\rho} \frac{1}{m_1} \sum_{j=1}^{m_1} \Big( (W_j^{(0)}+W_j^\rho)x_i \mathbb{1}\{(W_j^{(0)}+W_j^\rho)x_i \geq 0\} - W_j^{(0)} x_i \mathbb{1}\{W_j^{(0)} x_i \geq 0\} \Big)^2 =$$

$$\leq \frac{1}{m_1} \sum_{j=1}^{m_1} \mathbb{E}_{W^\rho} |W_j^\rho x_i|^2 = \frac{1}{m_1} \sum \frac{\beta_1^2}{m_1} = \frac{\beta_1^2}{m_1}.$$

For the second term, note that for every $j \notin P$, the $j$th entries of $D_{W^{(0)}+W^\rho,x_i}$ and $D_{W^{(0)}+W^\rho+W',x_i}$ are different only if $W_j^\rho$ can make a sign change in the $j$th row, i.e. $|(W_j^{(0)}+W_j')x_i| \leq |W_j^\rho x_i|$ should happen. We denote this event for every $j \notin P$ by $\tilde{E}_j$. Furthermore, if this happens for some $j$, then the value of $(W^{(0)}+W^\rho)_j x_i$ is upper bounded by $|W_j' x_i|$. Now similar to our discussion in Lemma 15 and using the result of Lemma 1:

$$\mathbb{P}(\cup_{j\notin P}\tilde{E}_j) = \mathbb{P}(\text{sign change in some } j \notin P) \leq \sum_{j\notin P}^{m_1} \mathbb{P}(\text{sign change in } j)$$

$$\leq m_1 \mathbb{P}(|(W_j^{(0)}+W_j')x_i| \leq |W_j^\rho x_i|) \leq m_1 \mathbb{P}(|W_j^\rho x_i| \geq c_2/(2\sqrt{m_1})).$$

But note that $(W_j^\rho)x_i$ is Gaussian with variance $\beta_1^2/m_1$. Hence

$$LHS \lesssim m_1 \exp\{-c_2^2/(8\beta_1^2)\},$$

So finally we can write

$$\lesssim \frac{\beta_1^2}{m_1} + \mathbb{E}_{W^\rho} \frac{1}{m_1}\Big(\sum_{j\in P}|W_j'x_i|\Big)^2 + E_{W^\rho}\frac{1}{m_1}\Big(\mathbb{1}\{\cup_{j\notin P}\tilde{E}_j\}\sum_{j\notin P}|W_j'x_i|\Big)^2$$

$$\lesssim \frac{\beta_1^2}{m_1} + \frac{|P|}{m_1}\|W'\|^2 + \mathbb{P}(\cup_{j\notin P}\tilde{E}_j)\|W'\|^2$$

$$\leq \frac{\beta_1^2}{m_1} + \frac{c_2 C_1^2}{\sqrt{m_1}\kappa_1} + m_1 \exp\{-c_2^2/(8\beta_1^2)\}C_1^2.$$

which completes the proof for Equation (158). This immediately implies

$$\mathbb{E}_{W^\rho}\|\phi^{(2)\prime}(x_i)\| \leq \sqrt{\mathbb{E}_{W^\rho}\|\phi^{(2)\prime}(x_i)\|^2} \leq \sqrt{m_3\Big(\frac{\beta_1^2}{m_1} + \frac{c_2 C_1^2}{\sqrt{m_1}\kappa_1} + m_1 \exp\{-c_2^2/(8\beta_1^2)\}C_1^2\Big)}.$$

Now we first prove Equation (156):

$$\Big|\mathbb{E}_{W^\rho,V^\rho}\Big[\frac{1}{\sqrt{m_2}}a^T D_{V^{(0)}+V^\rho+V',x_i} V_\Sigma^* \phi^{(2)\prime}(x_i)\Big]\Big| \leq \mathbb{E}_{W^\rho}\frac{1}{\sqrt{m_2}}\|a\|\|D_{V^{(0)}+V^\rho+V',x_i}V_\Sigma^*\|_F \|\phi^{(2)\prime}(x_i)\|\|$$

$$\leq \|V_\Sigma^*\|_F \mathbb{E}_{W^\rho}\|\phi^{(2)\prime}(x_i)\|\|\| \leq \|V^*\|_F \mathbb{E}_{W^\rho}\|\phi^{(2)\prime}(x_i)\|\|\|$$

$$\lesssim \sqrt{\zeta_2 m_3\Big(\frac{\beta_1^2}{m_1} + \frac{c_2 C_1^2}{\sqrt{m_1}\kappa_1} + m_1 \exp\{-c_2^2/(8\beta_1^2)\}C_1^2\Big)}.$$

To prove Equation (157):

$$\mathbb{E}_{W^\rho, V^\rho} \Big| \frac{1}{\sqrt{m_2}} a^T D_{V^{(0)}+V^\rho+V',x_i}(V^{(0)} + V^\rho + (1-\eta/2)V')\phi^{(2)\prime}(x_i) \Big|$$

$$\lesssim \mathbb{E}_{W^\rho, V^\rho} \frac{1}{\sqrt{m_2}} \|a\| \|D_{V^{(0)}+V^\rho+(1-\eta/2)V',x_i}(V^{(0)} + V^\rho + (1-\eta/2)V')\|_F \|\phi^{(2)\prime}(x_i)\|$$

$$\lesssim \mathbb{E}_{W^\rho, V^\rho} \frac{1}{\sqrt{m_2}} \|a\| \|V^{(0)} + V^\rho + (1-\eta/2)V'\|_F \|\phi^{(2)\prime}(x_i)\|$$

$$\lesssim \frac{1}{\sqrt{m_2}} \|a\| \sqrt{\mathbb{E}_{V^\rho}(\|V^{(0)}\|^2 + \|V^\rho\|_F^2 + \|(1-\eta/2)V'\|_F^2)} \mathbb{E}_{W^\rho} \|\phi^{(2)\prime}(x_i)\|$$

$$\lesssim \sqrt{(\kappa_2^2 m_2 m_3 + m_3 \beta_2^2 + C_2^2)} \Re_5 \lesssim (\kappa_2 \sqrt{m_2 m_3} + \sqrt{m_3}\beta_2 + C_2)\Re_5.$$

**Lemma 19** *Closeness condition:*

$$\mathbb{E}_\Sigma \Big| \mathbb{E}_{W^\rho, V^\rho} \big[ \frac{1}{\sqrt{m_2}} a^T D_{V^{(0)}+V^\rho+V',x_i} V_\Sigma^* \phi^*{}_\Sigma(x_i) \big] - f^*(x_i) \Big| \lesssim \Re_9,$$

*where*

$$\Re_9 := \varrho_3 \xi \sqrt{n}(1+\Re)\sqrt{\sum_k \|\mathcal{V}_k\|_{H^\infty}^2} + \Re_3 \tag{159}$$

$$+ m_2 \Big( \exp\{-(m_2\kappa_2^2 C_2^4)^{1/3}/(2\beta_2^2)\} + m_1 \exp\{-C_1^2/(8m_3\beta_1^2)\} \Big) \sqrt{\zeta_2}(1+\Re)\sqrt{\sum_k \|\mathcal{V}_k\|_{H^\infty}^2}. \tag{160}$$

**Proof of Lemma 19**

Note that by Corollary 5.1 and according to the proof of Equation 129 in Lemma 10, if for every $j \notin \tilde{P}$ we don't have a sign change in $D_{V^{(0)}+V^\rho+V',x_i} V^* \phi^*(x_i)$, then get

$$|\frac{1}{\sqrt{m_2}} a^T D_{V^{(0)}+V^\rho+V',x_i} V^* \phi^*(x_i) - f^*(x_i)| \le \Re_3.$$

Also, note that we need the event $E^c$ (defined in Lemma 33) to happen in order to be able to use Corrolary 5.1. Hence, given a $W^\rho$ for which $E^c$ happens, we upper bound the probability of sign change with respect to the randomness of $V^\rho$. We define the following event with respect to the randomness of $V^\rho$ when conditioned on a $W^\rho$ for which $E^c$ happens ($P_i$'s are defined in Lemma 7):

$$SC := \{\exists j \notin P_i \text{ s.t.} |V_j^\rho x_i'| \gtrsim (\frac{\kappa_2}{m_2})^{1/3} C_2^{2/3} \|x_i'\|\}.$$

Now from the result in Corollary 5.1 we have $\le \mathbb{1}\{\text{sign change in } j \notin P_i\} \le \mathbb{1}\{SC\}$. Therefore,

$$E\mathbb{1}\{\text{sign change}\} \le \mathbb{1}\{SC\} \le \sum_{j \notin P_i} \mathbb{P}(|V_j^\rho x_i'| \gtrsim (\frac{\kappa_2}{m_2})^{1/3} C_2^{2/3} \|x_i'\|)$$

$$\le m_2 \mathbb{P}(|V_j^\rho x_i'| \gtrsim (\frac{\kappa_2}{m_2})^{1/3} C_2^{2/3} \|x_i'\|).$$

But note that $(V_j^\rho)x_i'$ is Gaussian with variance $\beta_2^2 \|x_i'\|^2/m_2$. Hence

$$LHS \lesssim m_2 \exp\{-(m_2\kappa_2^2 C_2^4)^{1/3}/(2\beta_2^2)\}. \tag{161}$$

Now let $D$ be a sign matrix random variable such that if $E^c$ and $SC^c$ both happens, then it is equal to the valid sign matrix $D_{V^{(0)}+V^\rho+V',x_i}$, and otherwise it is equal to an arbitrary valid sign matrix in the case when both $E^c$ and $SC^c$ happen. Now using Equation (116) we have with high probability

over the initialization:

$$\mathbb{E}_\Sigma \left| \mathbb{E}_{W^\rho, V^\rho} [\frac{1}{\sqrt{m_2}} a^T D_{V^{(0)}+V^\rho+V', x_i} V_\Sigma^* \phi^*_\Sigma(x_i)] - f^*(x_i) \right|$$

$$\leq \mathbb{E}_\Sigma \left| \mathbb{E}_{W^\rho, V^\rho} [\frac{1}{\sqrt{m_2}} a^T D_{V^{(0)}+V^\rho+V', x_i} (V_\Sigma^* - V^*\Sigma) \phi^*_\Sigma(x_i)] \right.$$

$$+ \mathbb{E}_\Sigma \left| \mathbb{E}_{W^\rho, V^\rho} [\frac{1}{\sqrt{m_2}} a^T D_{V^{(0)}+V^\rho+V', x_i} V^*\Sigma \phi^*_\Sigma(x_i)] - f^*(x_i) \right.$$

$$\leq \mathbb{E}_{W^\rho, V^\rho} \mathbb{E}_\Sigma \left| \frac{1}{\sqrt{m_2}} a^T D_{V^{(0)}+V^\rho+V', x_i} (V_\Sigma^* - V^*\Sigma) \phi^*_\Sigma(x_i) \right|$$

$$+ \mathbb{E}_\Sigma \left| \mathbb{E}_{W^\rho, V^\rho} [\frac{1}{\sqrt{m_2}} a^T D_{V^{(0)}+V^\rho+V', x_i} V^* \phi^*(x_i)] - f^*(x_i) \right|$$

$$\leq \mathbb{E}_{W^\rho, V^\rho} \mathbb{E}_\Sigma \frac{1}{m_2} \sum_j \|V_{\Sigma j}^* - V_j^*\Sigma\| \|\phi^*_\Sigma(x_i)\|$$

$$+ \mathbb{E}_\Sigma \left| \mathbb{E}_{W^\rho, V^\rho} [\left( \frac{1}{\sqrt{m_2}} a^T D V^* \phi^*(x_i) - f^*(x_i) \right) \right.$$

$$+ \mathbb{1}\{SC \cup E\} \left( \frac{1}{\sqrt{m_2}} a^T D_{V^{(0)}+V', x_i} V^* \phi^*(x_i) - D \right)] \Big|$$

$$\leq \mathbb{E}_{W^\rho, V^\rho} \mathbb{E}_\Sigma \frac{1}{m_2} \sum_j \|V_{\Sigma j}^* - V_j^*\Sigma\| \|\phi^*_\Sigma(x_i)\|$$

$$+ \mathbb{E}_\Sigma \left| \mathbb{E}_{W^\rho, V^\rho} \left[ \left( \frac{1}{\sqrt{m_2}} a^T D V^* \phi^*(x_i) - f^*(x_i) \right) \right| \right.$$

$$+ \mathbb{E}_\Sigma \mathbb{E}_{W^\rho, V^\rho} \left| \mathbb{1}\{SC \cup E\} \left( \frac{1}{\sqrt{m_2}} a^T D_{V^{(0)}+V', x_i} V^* \phi^*(x_i) - \frac{1}{\sqrt{m_2}} a^T D V^* \phi^*(x_i) \right) \right] \Big|$$

$$\leq \mathbb{E}_{W^\rho, V^\rho} \frac{1}{\sqrt{m_2}} \sum_j \sqrt{\mathbb{E}_\Sigma \|V_{\Sigma j}^* - V_j^*\Sigma\|^2} \|\phi^*(x_i)\|$$

$$+ \Re_3 + 2\mathbb{P}(SC \cup E) \max_{D'} \left| \frac{1}{\sqrt{m_2}} a^T D' V^* \phi^*(x_i) \right|.$$

Now note that for any sign matrix $D'$, we have the following bound:

$$\left| \frac{1}{\sqrt{m_2}} a^T D' V^* \phi^*(x_i) \right| \leq \frac{1}{\sqrt{m_2}} \|a\| \|V^*\|_F \|\phi^*(x_i)\| \lesssim \sqrt{\zeta_2}(1 + \Re) \sqrt{\sum_k \|\mathcal{V}_k\|_{H^\infty}^2}.$$

Also, applying a union bound and using Lemmas 33

$$\mathbb{P}(SC \cup E) \leq \mathbb{P}(SC) + \mathbb{P}(E)$$
$$\lesssim \exp\{-(m_2 \kappa_2^2 C_2^4)^{1/3}/(2\beta_2^2)\} + m_1 \exp\{-C_1^2/(8m_3\beta_1^2)\}.$$

Hence, also applying Lemma 40, we further write

$$LHS \lesssim \varrho_3 \xi \sqrt{n}(1 + \Re) \sqrt{\sum_k \|\mathcal{V}_k\|_{H^\infty}^2} + \Re_3$$

$$+ m_2 \left( \exp\{-(m_2 \kappa_2^2 C_2^4)^{1/3}/(2\beta_2^2)\} + m_1 \exp\{-C_1^2/(8m_3\beta_1^2)\} \right) \sqrt{\zeta_2}(1 + \Re) \sqrt{\sum_k \|\mathcal{V}_k\|_{H^\infty}^2}.$$

**Lemma 20** *Suppose we have* $m_3 \kappa_1^2 \geq C_1^2$. *Then, for the following basic term we have:*

$$\mathbb{E}_{W^\rho, V^\rho} [\frac{1}{\sqrt{m_2}} a^T D_{V^{(0)}+V^\rho+V', x_i} (V^{(0)} + V^\rho + (1 - \eta/2)V')(\phi^{(0)}(x_i) + (1 - \eta/2)\phi^{(2)}(x_i))$$

$$\lesssim (1 - \eta) \mathbb{E}_{W^\rho, V^\rho} [\frac{1}{\sqrt{m_2}} a^T D_{V^{(0)}+V^\rho+V', x_i} (V^{(0)} + V^\rho + V')(\phi^{(0)}(x_i) + \phi^{(2)}(x_i))$$

$$\pm \eta \Big( \Re_6' + \Re_4 + (\sqrt{m_3}\kappa_2 + \beta_2)(C_1 + \sqrt{m_3}\beta_1) \Big),$$

*where*

$$\Re_4 := C_2(C_1 + \sqrt{m_3}\beta_1)m_2 \exp\{-C_2^{4/3}(\sqrt{m_2}\kappa_2)^{2/3}/8\beta_2^2\} + \frac{C_2^{1/3}}{(\sqrt{m_2}\kappa_2)^{1/3}}C_2(C_1 + \sqrt{m_3}\beta_1),$$

$$\Re_6' := m_1 \exp\{-C_1^2/(8m_3\beta_1^2)\}\sqrt{m_3}\kappa_1(\sqrt{m_2} + \beta_2) + \Re_6,$$

*and $\Re_6$ is defined in Lemma 22.*

**Proof of Lemma 20**

First, note that by orthogonality of $\phi^{(0)}(x_i)$ to the rows of $V'$:

$$LHS - \eta/2\mathbb{E}_{W^\rho, V^\rho}[\frac{1}{\sqrt{m_2}}a^T D_{V^{(0)}+V^\rho+V', x_i}(V^{(0)} + V^\rho)\phi^{(0)}(x_i)$$

$$= LHS - \eta/2\mathbb{E}_{W^\rho, V^\rho}[\frac{1}{\sqrt{m_2}}a^T D_{V^{(0)}+V^\rho+V', x_i}(V^{(0)} + V^\rho + (1-\eta/2)V')\phi^{(0)}(x_i)$$

$$= (1-\eta/2)\mathbb{E}_{W^\rho, V^\rho}[\frac{1}{\sqrt{m_2}}a^T D_{V^{(0)}+V^\rho+V', x_i}(V^{(0)} + V^\rho + (1-\eta/2)V')(\phi^{(0)}(x_i) + \phi^{(2)}(x_i))]$$

$$= (1-\eta/2)^2\mathbb{E}_{V^\rho}[\frac{1}{\sqrt{m_2}}a^T D_{V^{(0)}+V^\rho+V', x_i}(V^{(0)} + V^\rho + V')(\phi^{(0)}(x_i) + \phi^{(2)}(x_i))]$$

$$+ (1-\eta/2)(\eta/2)\mathbb{E}_{V^\rho}[\frac{1}{\sqrt{m_2}}a^T D_{V^{(0)}+V^\rho+V', x_i}(V^{(0)} + V^\rho)(\phi^{(0)}(x_i) + \phi^{(2)}(x_i))] \tag{162}$$

But note that for the second term:

$$\mathbb{E}_{W^\rho, V^\rho}[\frac{1}{\sqrt{m_2}}a^T D_{V^{(0)}+V^\rho+V', x_i}(V^{(0)} + V^\rho)(\phi^{(0)}(x_i) + \phi^{(2)}(x_i))]$$

$$\lesssim \mathbb{E}_{W^\rho, V^\rho}[\frac{1}{\sqrt{m_2}}a^T D_{V^{(0)}+V^\rho, x_i}(V^{(0)} + V^\rho)(\phi^{(0)}(x_i) + \phi^{(2)}(x_i))]$$

$$\pm \frac{1}{\sqrt{m_2}} \sum_{j:\text{ sign change}} |V_j'(\phi^{(0)}(x_i) + \phi^{(2)}(x_i))|$$

$$= \mathbb{E}_{V^\rho}[\frac{1}{\sqrt{m_2}}a^T D_{V^{(0)}+V^\rho, x_i}(V^{(0)} + V^\rho)(\phi^{(0)}(x_i) + \phi^{(2)}(x_i))]$$

$$\pm \frac{1}{\sqrt{m_2}} \sum_{j:\text{ sign change}} |V_j'\phi^{(2)}(x_i)|. \tag{163}$$

Now conditioned on $x_i'$, by the result of Lemma 31 we know there exists a set of indices $O \subseteq [m_2]$, s.t. $|O| \leq \frac{C_2^{2/3}}{(\sqrt{m_2}\kappa_2)^{2/3}}m_2$ and for $j \notin O$ we have

$$|V_j^{(0)}x_i'| \geq \frac{C_2^{2/3}(\sqrt{m_2}\kappa_2)^{1/3}}{\sqrt{m_2}}\|x_i'\|$$

and

$$|V_j'x_i'| \leq \frac{C_2^{2/3}(\sqrt{m_2}\kappa_2)^{1/3}}{2\sqrt{m_2}}\|x_i'\|.$$

Now for $j \in [m_2]$, define the event

$$R_j = \{|W_j^\rho x_i'| \geq \frac{C_2^{2/3}(\sqrt{m_2}\kappa_2)^{1/3}}{2\sqrt{m_2}}\|x_i'\|\},$$

and $R = \cup_j R_j$. First, note that using Gaussian tail bound, $R$ is a rare event:

$$\mathbb{P}(R) \leq \sum_j \mathbb{P}(R_j) \leq m_2 \exp\{-C_2^{4/3}(\sqrt{m_2}\kappa_2)^{2/3}/8\beta_2^2\}.$$

Now for $j \notin O$ and under $R^c$, clearly we have that the signs of $(V_j^{(0)}+V_j^\rho)x_i'$ and $(V_j^{(0)}+V_j^\rho+V_j')x_i'$ are the same. Therefore, applying Lemma 33, we can argue under $R^c$:

$$\mathbb{E}_{W^\rho, V^\rho} \frac{1}{\sqrt{m_2}} \sum_{j:\ \text{sign change}} |V_j'\phi^{(2)}(x_i)| \leq \mathbb{E}_{W^\rho} \frac{1}{\sqrt{m_2}} \sum_{j \in O} |V_j'\phi^{(2)}(x_i)|$$

$$\leq \sqrt{\frac{|O|}{m_2}} \|V'\| E_{W^\rho} \|\phi^{(2)}(x_i)\| \leq \frac{C_2^{1/3}}{(\sqrt{m_2}\kappa_2)^{1/3}} C_2(C_1 + \sqrt{m_3}\beta_1).$$

Hence, overall, using Cauchy-Shwartz

$$\mathbb{E}_{W^\rho, V^\rho} \frac{1}{\sqrt{m_2}} \sum_{j:\ \text{sign change}} |V_j'\phi^{(2)}(x_i)| \leq \|V'\| \mathbb{E}_{W^\rho} \|\phi^{(2)}(x_i)\| \mathbb{P}(R) + \frac{C_2^{1/3}}{(\sqrt{m_2}\kappa_2)^{1/3}} C_2(C_1 + \sqrt{m_3}\beta_1)$$

$$\leq C_2(C_1 + \sqrt{m_3}\beta_1) m_2 \exp\{-C_2^{4/3}(\sqrt{m_2}\kappa_2)^{2/3}/8\beta_2^2\} + \frac{C_2^{1/3}}{(\sqrt{m_2}\kappa_2)^{1/3}} C_2(C_1 + \sqrt{m_3}\beta_1) := \Re_4. \tag{164}$$

On the other hand, using Lemma 30, we have with high probability over the randomness of initialization

$$\frac{1}{\sqrt{m_2}} a^T D_{V^{(0)},x_i} V^{(0)}\phi^{(2)}(x_i) \leq \sqrt{m_3}\kappa_2 \|\phi^{(2)}(x_i)\|.$$

Hence:

$$\mathbb{E}_{V^\rho}\left[\frac{1}{\sqrt{m_2}} a^T D_{V^{(0)}+V^\rho,x_i}(V^{(0)}+V^\rho)\phi^{(2)}(x_i)\right]$$

$$\leq \mathbb{E}_{W^\rho,V^\rho}\left[\frac{1}{\sqrt{m_2}} a^T D_{V^{(0)},x_i} V^{(0)}\phi^{(2)}(x_i) + \frac{1}{\sqrt{m_2}} \sum_j |V_j^\rho \phi^{(2)}(x_i)|\right]$$

$$\leq \mathbb{E}_{W^\rho} \frac{1}{\sqrt{m_2}} a^T D_{V^{(0)},x_i} V^{(0)}\phi^{(2)}(x_i) + \beta_2 \mathbb{E}_{W^\rho} \|\phi^{(2)}(x_i)\|$$

$$\lesssim (\sqrt{m_3}\kappa_2 + \beta_2) \mathbb{E}_{W^\rho} \|\phi^{(2)}(x_i)\|$$

$$\lesssim (\sqrt{m_3}\kappa_2 + \beta_2)(C_1 + \sqrt{m_3}\beta_1). \tag{165}$$

Combining Equations (164) and (165) into Equation (163):

$$\left|\mathbb{E}_{V^\rho}\left[\frac{1}{\sqrt{m_2}} a^T D_{V^{(0)}+V^\rho+V',x_i}(V^{(0)}+V^\rho)\phi^{(2)}(x_i)\right]\right| \lesssim \Re_4 + (\sqrt{m_3}\kappa_2 + \beta_2)(C_1 + \sqrt{m_3}\beta_1). \tag{166}$$

Moreover, for the first term in (162), using Equation (165) and Lemmas 33 and Lemma 30 we have

$$|\mathbb{E}_{V^\rho}\left[\frac{1}{\sqrt{m_2}} a^T D_{V^{(0)}+V^\rho+V',x_i}(V^{(0)}+V^\rho+V')\phi^{(2)}(x_i)\right]|$$

$$\lesssim |\mathbb{E}_{W^\rho,V^\rho} \frac{1}{\sqrt{m_2}} a^T D_{V^{(0)}} V^{(0)}\phi^{(2)}(x_i)| + \mathbb{E}_{W^\rho,V^\rho} \frac{1}{\sqrt{m_2}} \sum_j |V_j^\rho\phi^{(2)}(x_i)| + \frac{1}{\sqrt{m_2}} \sum_j |V_j'\phi^{(2)}(x_i)|$$

$$\lesssim \kappa_2\sqrt{m_3}(C_1 + \beta_1\sqrt{m_3}) + (C_2 + \beta_2)\mathbb{E}_{W^\rho}\|\phi^{(2)}(x_i)\|$$

$$\lesssim \kappa_2\sqrt{m_3}(C_1 + \beta_1\sqrt{m_3}) + (C_2 + \beta_2)(C_1 + \sqrt{m_3}\beta_1). \tag{167}$$

Substituting Equations (166) and (167) into Equation (162), we finally get

$$
\begin{aligned}
LHS &- \eta/2\mathbb{E}_{W^\rho,V^\rho}[\frac{1}{\sqrt{m_2}}a^T D_{V^{(0)}+V^\rho+V',x_i}(V^{(0)}+V^\rho)\phi^{(0)}(x_i) \\
&\lesssim (1-\eta/2)^2\mathbb{E}_{V^\rho}[\frac{1}{\sqrt{m_2}}a^T D_{V^{(0)}+V^\rho+V',x_i}(V^{(0)}+V^\rho+V')\phi^{(2)}(x_i)] \\
&\pm \frac{\eta}{2}\Big(\Re_4 + (\sqrt{m_3}\kappa_2 + \beta_2)(C_1 + \sqrt{m_3}\beta_1)\Big) \\
&\lesssim (1-\eta)\mathbb{E}_{V^\rho}[\frac{1}{\sqrt{m_2}}a^T D_{V^{(0)}+V^\rho+V',x_i}(V^{(0)}+V^\rho+V')\phi^{(2)}(x_i)] \\
&\pm \frac{\eta^2}{4}\Big|\mathbb{E}_{V^\rho}[\frac{1}{\sqrt{m_2}}a^T D_{V^{(0)}+V^\rho+V',x_i}(V^{(0)}+V^\rho+V')\phi^{(2)}(x_i)]\Big| \\
&\pm \frac{\eta}{2}\Big(\Re_4 + (\sqrt{m_3}\kappa_2 + \beta_2)(C_1 + \sqrt{m_3}\beta_1)\Big) \\
&\lesssim (1-\eta)\mathbb{E}_{V^\rho}[\frac{1}{\sqrt{m_2}}a^T D_{V^{(0)}+V^\rho+V',x_i}(V^{(0)}+V^\rho+V')\phi^{(2)}(x_i)] \\
&\pm \eta^2(\kappa_2\sqrt{m_3}(C_1 + \beta_1\sqrt{m_3}) + (C_2 + \beta_2)(C_1 + \sqrt{m_3}\beta_1)) \\
&\pm \eta\Big(\Re_4 + (\sqrt{m_3}\kappa_2 + \beta_2)(C_1 + \sqrt{m_3}\beta_1)\Big).
\end{aligned}
\tag{168}
$$

Now by picking $\eta$ small enough so that the second term is dominated by the third term we get:

$$
LHS - \eta/2\mathbb{E}_{W^\rho,V^\rho}[\frac{1}{\sqrt{m_2}}a^T D_{V^{(0)}+V^\rho+V',x_i}(V^{(0)}+V^\rho)\phi^{(0)}(x_i)
\tag{169}
$$

$$
\lesssim (1-\eta)\mathbb{E}_{V^\rho}[\frac{1}{\sqrt{m_2}}a^T D_{V^{(0)}+V^\rho+V',x_i}(V^{(0)}+V^\rho+V')\phi^{(2)}(x_i)]
\tag{170}
$$

$$
\pm \eta\Big(\Re_4 + (\sqrt{m_3}\kappa_2 + \beta_2)(C_1 + \sqrt{m_3}\beta_1)\Big).
\tag{171}
$$

Now we aim to bound the term $\mathbb{E}_{W^\rho,V^\rho}[\frac{1}{\sqrt{m_2}}a^T D_{V^{(0)}+V^\rho+V',x_i}(V^{(0)}+V^\rho)\phi^{(0)}(x_i)]$. First assume that we are in the event $E^c$ defined in Lemma 33, i.e. we have $\|\phi^{(2)}(x_i)\| \lesssim C_1$. Conditioned on such $W^\rho$, we now work with the randomness of the initialization and $V^\rho$. Note that the random matrix $V^{(0)} + V^\rho$ jointly over the randomness of $V^\rho$ and the initialization is also Gaussian, and its variance is

$$
\kappa_2^2 \le \kappa_2^2 + \frac{\beta_2^2}{m_2} \le 2\kappa_2^2,
\tag{172}
$$

where the inequality follows from the fact that $\kappa_2 \ge \frac{1}{\sqrt{m_2}}$ and $\beta_2 \le 1$. Therefore, applying Lemma 22 for the random matrix $V^{(0)}$ in the Lemma as $V^{(0)} + V^\rho$ here, the bound does not change up to constants because of the inequality (172). Hence, with high probability, lets say with prob. $1 - \delta_1$ this time over both the randomness of initialization and $V^\rho$:

$$
\mathcal{L} = \Big|\frac{1}{\sqrt{m_2}}a^T D_{V^{(0)}+V^\rho+V',x_i}(V^{(0)}+V^\rho)\phi^{(0)}(x_i)\Big| \le \Re_6
\tag{173}
$$

This means that with probability at least $1 - \sqrt{\delta_1}$ over the random initialization, then we have (173) with prob. at least $1 - \sqrt{\delta_1}$ over the randomenss of $V^\rho$. We name the latter high probability statement as $(\star)$. Moreover, note that by Lemma 32 and assuming $m_3 \log(m_2) < m_2$, we have the following

almost surely bound (also note that $V_j^\rho \phi^{(0)}(x_i)$ is Gaussian with std $\frac{\beta_2}{\sqrt{m_2}}\|\phi^{(0)}(x_i)\|$):

$$\mathbb{E}_{V^\rho}\left|\frac{1}{\sqrt{m_2}}a^T D_{V^{(0)}+V^\rho+V',x_i}(V^{(0)}+V^\rho)\phi^{(0)}(x_i)\right| \tag{174}$$

$$= \mathbb{E}_{V^\rho}\left|\frac{1}{\sqrt{m_2}}a^T D_{V^{(0)}+V^\rho+V',x_i}(V^{(0)}+V^\rho)\phi^{(0)}(x_i)\right| \tag{175}$$

$$\leq \frac{1}{\sqrt{m_2}}\sum_{j=1}^{m_2}|V_j^{(0)}\phi^{(0)}(x_i)| + \mathbb{E}_{V^\rho}\frac{1}{\sqrt{m_2}}\sum_{j=1}^{m_2}|V_j^\rho\phi^{(0)}(x_i)| \tag{176}$$

$$\lesssim \|\phi^{(0)}(x_i)\|\sup_{\|x'\|=1}\frac{1}{\sqrt{m_2}}\sum_{j=1}^{m_2}|V_j^{(0)}x'| + \frac{1}{\sqrt{m_2}}\sum_{j=1}^{m_2}\frac{\beta_2}{\sqrt{m_2}}\|\phi^{(0)}(x_i)\| \tag{177}$$

$$\lesssim \|\phi^{(0)}(x_i)\|(\sqrt{m_2}+\beta_2). \tag{178}$$

Furthermore, note because each variable $|V_j^\rho\phi^{(0)}(x_i)|$ is $\frac{\beta_2}{\sqrt{m_2}}\|\phi^{(0)}(x_i)\|$ subGaussian. Therefore, $\mathcal{L}$ is subGaussian with parameter $\|\phi^{(0)}(x_i)\|\beta_2$ with respect to the randomness of $V^\rho$. Now the point is that the high probability argument in $(\star)$ is much stronger than what one can get from the subGaussian ineqaulity with parameter $\|\phi^{(0)}(x_i)\|\beta_2$ (with the corresponding expectation term $\|\phi^{(0)}(x_i)\|(\sqrt{m_2}+\beta_2)$). However, the disadvantage of $(\star)$ is that it only works for a fixed $\delta_1$. In other words, at least it is not obvious from this argument that why for a fixed $W^{(0)}$ in a high probaiblity region of the random initialization, whether we can send $\delta_1$ to zero by growing the constant behind $\Re_6$ with logarithmic rate $\log(1/\delta)$. This makes our job hard for bounding the expectation with respect to $V^\rho$ if we only wish to rely on $(\star)$. Therefore, we combine it with the inequality that we get from the subGaussian parameter that we introudced above. More rigorously, we define the thresholding parameter

$$\mho := \|\phi^{(0)}(x_i)\|(\sqrt{m_2}+\beta_2) + \|\phi^{(0)}(x_i)\|\beta_2\log(\|\phi^{(0)}(x_i)\|(\sqrt{m_2}+\beta_2)/\Re_6)$$

$$= \Theta\Big(\|\phi^{(0)}(x_i)\|(\sqrt{m_2}+\beta_2\log(\|\phi^{(0)}(x_i)\|(\sqrt{m_2}+\beta_2)/\Re_6))\Big),$$

for which we have

$$\mathbb{E}\Big[\mathcal{L}\big|\ \mho\leq\mathcal{L}\Big]\mathbb{P}(\mho\leq\mathcal{L})\lesssim\Re_6.$$

we divide the range of values for $\mathcal{L}$ into three parts:

$$\mathbb{E}[\mathcal{L}] = \mathbb{E}\Big[\mathcal{L}\big|\ \mathcal{L}\leq\Re\Big]\mathbb{P}(\mathcal{L}\leq\Re_6)$$

$$+ \mathbb{E}\Big[\mathcal{L}\big|\ \Re_6\leq\mathcal{L}\leq\mho\Big]\mathbb{P}\Big(\Re_6\leq\mathcal{L}\leq\mho\Big)$$

$$+ \mathbb{E}\Big[\mathbb{L}\big|\ \mho\leq\mathcal{L}\Big]\mathbb{P}(\mho\leq\mathcal{L})$$

$$\leq E\Big[\mathcal{L}\big|\ \mathcal{L}\leq\Re_6\Big] + \mathbb{P}(\Re_6\leq\mathcal{L}\leq\mho) + \Re_6$$

$$\lesssim \Re_6 + \sqrt{\delta_1}\mho.$$

Now by choosing $\delta_1\lesssim 1/\mho$, we conclude with high probability over initialization and conditioned on $W^\rho$'s such that $E^c$ happens we have

$$\mathbb{E}_{V^\rho}\left|\frac{1}{\sqrt{m_2}}a^T D_{V^{(0)}+V^\rho+V',x_i}(V^{(0)}+V^\rho)\phi^{(0)}(x_i)\right| = \mathbb{E}[\mathcal{L}]\lesssim\Re_6.$$

Finally, we integrate also with respect to $W^\rho$. To control the random variable when $E$ happens, we use the bound in (178) and the fact that $E$ is a rare event due to Lemma 33:

$$\mathbb{E}_{W^\rho,V^\rho}\left|\frac{1}{\sqrt{m_2}}a^T D_{V^{(0)}+V^\rho+V',x_i}(V^{(0)}+V^\rho)\phi^{(0)}(x_i)\right| \lesssim \mathbb{P}(E)\|\phi^{(0)}(x_i)\|(\sqrt{m_2}+\beta_2) + \mathbb{P}(E^c)\Re_6$$

$$\leq m_1\exp\{-C_1^2/(8m_3\beta_1^2)\}\sqrt{m_3}\kappa_1(\sqrt{m_2}+\beta_2) + \Re_6 := \Re_6'.$$

Substituting this into (171) the proof is finally complete.

**Lemma 21** *Third cross term: with high probability over initialization, we have*

$$\mathbb{E}_{\Sigma}\Big(\mathbb{E}_{W^{\rho},V^{\rho}}[\frac{1}{\sqrt{m_2}}a^T D_{V^{(0)}+V^{\rho}+V',x_i}(V^{(0)}+V^{\rho}+(1-\eta)V')\Sigma\phi^*(x_i)]\Big)^2$$

$$\lesssim \xi^2(m_1\exp\{-C_1^2/(8m_3\beta_1^2)\}(\kappa_2^2 m_2 m_3 + \beta_1^2 m_3) + \Re_7^2 C_2^2) \coloneqq \Re_{12}^2.$$

**Proof of Lemma 21**

Note that the way we defined the matrix $W^*$ and as a result $\phi^*(x_i)$ only depends on the randomness of $W^{(0)}$, not on $W'$ or the randomness of $V^{(0)}$. Now using Equation (130) and Jensen inequality we can write (for vector $v$, the notation $v^{2\odot}$ is another vector with each entry as the second power of the corresponding entry in $v$):

$$= \mathbb{E}_{\Sigma}\Big(\mathbb{E}_{W^{\rho},V^{\rho}}[\frac{1}{\sqrt{m_2}}a^T D_{V^{(0)}+V^{\rho}+V',x_i}(V^{(0)}+V^{\rho}+(1-\eta)V')\Sigma\phi^*(x_i)]\Big)^2$$

$$\leq \mathbb{E}_{\Sigma}\mathbb{E}_{W^{\rho},V^{\rho}}\Big(\frac{1}{\sqrt{m_2}}a^T D_{V^{(0)}+V^{\rho}+V',x_i}(V^{(0)}+V^{\rho}+(1-\eta)V')\Sigma\phi^*(x_i)\Big)^2$$

$$= \mathbb{E}_{W^{\rho},V^{\rho}}\mathbb{E}_{\Sigma}\Big(\langle \frac{1}{\sqrt{m_2}}a^T D_{V^{(0)}+V^{\rho}+V',x_i}(V^{(0)}+V^{\rho}+(1-\eta)V') \,,\, \Sigma\phi^*(x_i)\rangle\Big)^2$$

$$= \mathbb{E}_{W^{\rho},V^{\rho}}\Big\langle \Big(\frac{1}{\sqrt{m_2}}a^T D_{V^{(0)}+V^{\rho}+V',x_i}(V^{(0)}+V^{\rho}+(1-\eta)V')\Big)^{2\odot} \,,\, \phi^*(x_i)^{2\odot}\Big\rangle$$

$$\leq \mathbb{E}_{W^{\rho},V^{\rho}}\Big\|\frac{1}{\sqrt{m_2}}a^T D_{V^{(0)}+V^{\rho}+V',x_i}(V^{(0)}+V^{\rho}+(1-\eta)V')\Big\|_2^2 \Big\|\phi^*(x_i)\Big\|_\infty^2$$

$$\leq \xi^2 \mathbb{E}_{W^{\rho},V^{\rho}}\Big\|\frac{1}{\sqrt{m_2}}a^T D_{V^{(0)}+V^{\rho}+V',x_i}(V^{(0)}+V^{\rho}+(1-\eta)V')\Big\|_2^2$$

$$\leq 2\xi^2 \mathbb{E}_{W^{\rho},V^{\rho}}\Big\|\frac{1}{\sqrt{m_2}}a^T D_{V^{(0)}+V^{\rho}+V',x_i}(V^{(0)}+V^{\rho})\Big\|_2^2 + 2\xi^2 \mathbb{E}_{W^{\rho},V^{\rho}}\Big\|\frac{1}{\sqrt{m_2}}a^T D_{V^{(0)}+V^{\rho}+V',x_i}(1-\eta)V'\Big\|_2^2$$

$$\leq \xi^2 \mathbb{E}_{W^{\rho},V^{\rho}}\Big\|\frac{1}{\sqrt{m_2}}a^T D_{V^{(0)}+V^{\rho}+V',x_i}(V^{(0)}+V^{\rho})\Big\|_2^2 + \xi^2(1-\eta)^2\|V'\|_F^2$$

$$\leq \xi^2 \mathbb{E}_{W^{\rho},V^{\rho}}\Big\|\frac{1}{\sqrt{m_2}}a^T D_{V^{(0)}+V^{\rho}+V',x_i}(V^{(0)}+V^{\rho})\Big\|_2^2 + \xi^2(1-\eta)^2 C_2^2.$$

Now under the event $E^c$ defined in Lemma 33 we get that $\|\phi^{(2)}(x_i)\| \lesssim C_1$, so we can bound the above as

$$\leq \xi^2 \mathbb{E}_{V^{\rho}} \sup_{\|V'\|\leq C_2, V'\perp\phi^{(0)}(x_i),\|x'\|\leq C_1} \frac{1}{m_2}\Big\|\sum_j a_j \mathbb{1}\{(V_j^{(0)}+V_j^{\rho}+V_j')(\phi^{(0)}(x_i)+x')\geq 0\}(V_j^{(0)}+V_j^{\rho})\Big\|^2 \tag{179}$$

$$+ \xi^2(1-\eta)^2 C_2^2. \tag{180}$$

Now defining

$$\mathcal{L}_2 \coloneqq \frac{1}{m_2}\Big\|\sum_j a_j \mathbb{1}\{(V_j^{(0)}+V_j^{\rho}+V_j')(\phi^{(0)}(x_i)+x')\geq 0\}(V_j^{(0)}+V_j^{\rho})\Big\|^2,$$

to bound the first term, we want to apply Lemma 23 using the same trick that we did in the proof of Lemma 20. Note that $\mathcal{L}_2$ is the same term as $\Gamma_{x',V'}^2$ in Lemma 23 except that it is defined with respect to $V^{(0)}+\mathcal{V}^{\rho}$ instead of $V^{(0)}$. On the other hand, note that $V^{(0)}+V^{\rho}$ has Gaussian entries with variance $\kappa_2^2 + \frac{\beta_2^2}{m_2}$ and we know $\kappa_2^2 \leq \kappa_2^2 + \frac{\beta_2^2}{m_2} \leq 2\kappa_2^2$, which means the argument of Lemma 23 holds true here up to constants:

$$\sup_{\|V'\|\leq C_2, V'\perp\phi^{(0)}(x_i),\|x'\|\leq C_1} \mathcal{L}_2 \lesssim \Re_7^2.$$

This holds with probability say $1-\delta_2$ over the randomness of both $V^{(0)}$ and $V^{\rho}$. Therefore, with probability $1-\sqrt{\delta_2}$ over the initialization, then with probability at least $1-\sqrt{\delta_2}$ over the randomness

of $V^\rho$ we have the above. Moreover, with a simple Cauchy-Swuartz we get the following almost surely bound:

$$\mathcal{L}_2 \lesssim \|V^{(0)}\|_F^2 + \|V^\rho\|_F^2. \tag{181}$$

Now the variable $\|V^\rho\|^2$ is subexponential with parameter $(\beta_1^4 m_3^2, \beta_1^2 m_3)$. Furthermore, with high probability we have $\|V^{(0)}\|_F^2 \lesssim m_2 m_3 \kappa_2^2$. Therefore, taking

$$\mho_2 := \Theta\Big(\kappa_2^2 m_2 m_3 + \beta_1^2 m_3 \log\big((\kappa_2^2 m_2 m_3 + \beta_1^2 m_3)/\Re_7\big)\Big),$$

then one can easily see by the subexponential tail:

$$\mathbb{E}[\mathcal{L}_2|\,\mathcal{L}_2 \geq \mho_2] = \Theta\Big(\mho_2\Big),$$
$$\mathbb{P}(\mathcal{L}_2 \geq \mho_2) \leq \Re_7^2/\mho_2.$$

Hence, we can apply the same trick as Lemma 20 as

$$\begin{aligned}
\mathbb{E}[\mathcal{L}_2] &= \mathbb{E}\Big[\mathcal{L}_2|\,\mathcal{L}_2 \leq \Re_7^2\Big]\mathbb{P}(\mathcal{L}_2 \leq \Re_7^2) \\
&\quad + \mathbb{E}\Big[\mathcal{L}_2|\,\Re_7^2 \leq \mathcal{L}_2 \leq \mho_2\Big]\mathbb{P}\Big(\Re_7^2 \leq \mathcal{L}_2 \leq \mho_2\Big) \\
&\quad + \mathbb{E}\Big[\mathcal{L}_2|\,\mho_2 \leq \mathcal{L}_2\Big]\mathbb{P}(\mho_2 \leq \mathcal{L}_2) \\
&\lesssim E\Big[\mathcal{L}_2|\,\mathcal{L}_2 \leq \Re_7^2\Big] + \mathbb{P}(\Re_7^2 \leq \mathcal{L}_2 \leq \mho_2) + \Re_7^2 \\
&\lesssim \Re_7^2 + \sqrt{\delta_2}\mho_2.
\end{aligned}$$

Now taking $\delta_2 \lesssim \Re_7^4/\mho_2^2$, we finally get that conditioned on $W^\rho$'s where $E$ happens, then

$$\mathbb{E}_{V^\rho}\mathcal{L}_2 \leq \Re_7^2.$$

On the other hand, to handle the case when $E$ happens, we can use the bound in (181) as it does not depend on the occurrence of $E$ as well:

$$\begin{aligned}
\mathbb{E}_{W^\rho,V^\rho}\mathcal{L}_2 &\leq \mathbb{P}(E)\mathbb{E}_{V^\rho}(\|V^{(0)}\|^2 + \|V^\rho\|^2) + \mathbb{P}(E^c)\Re_7^2 \\
&\lesssim m_1 \exp\{-C_1^2/(8m_3\beta_1^2)\}(\kappa_2^2 m_2 m_3 + \beta_1^2 m_3) + \Re_7^2.
\end{aligned}$$

Plugging this back into (180) we finally get

$$LHS \lesssim \xi^2(m_1 \exp\{-C_1^2/(8m_3\beta_1^2)\}(\kappa_2^2 m_2 m_3 + \beta_1^2 m_3) + \Re_7^2) + \xi^2 C_2^2.$$

### A.15 BOUNDING THE WORST-CASE SENARIO

**Lemma 22** *Suppose $m_3 \geq \log(m_2)$ and $\sqrt{m_3}\kappa_1 \gtrsim C_1$. We define the sign matrices $D_{V^{(0)}+V',x_i}^{x'}$ and $D_{V^{(0)},x_i}^{x'}$ with respect to the multiplications*

$$(V^{(0)} + V')(\phi^{(0)}(x_i) + x'),$$

*and*

$$V^{(0)}(\phi^{(0)}(x_i) + x').$$

*Then, with high probability:*

$$\sup_{\|x'\| \lesssim C_1, \|V'\|_F \leq C_2, V' \perp \phi^{(0)}} \frac{1}{\sqrt{m_2}} a^T D_{V^{(0)}+V',x_i}^{x'} V^{(0)} \phi^{(0)}(x_i)$$

$$\lesssim \left( \frac{(C_1 C_2)^{4/3}}{(\sqrt{m_2}\kappa_2)^{1/3}(\sqrt{m_3}\kappa_1)^{1/3}} + \frac{(C_1 C_2)^{2/3} m_3^{2/3}(\kappa_1 \kappa_2)^{1/3}\sqrt{\log(m_2)}}{m_2^{1/3}} \right) \left( 1 + \log(m_2)\frac{C_1^{1/3}(\kappa_2\sqrt{m_2})^{2/3}}{C_2^{2/3}(\kappa_1\sqrt{m_3})^{1/3}} \right)$$

$$+ \frac{m_3^{3/2}\kappa_1\kappa_2}{\sqrt{m_2}}\sqrt{\log(m_2)}(\log(m_3) + \log(\log(m_2))) + \kappa_1\kappa_2\sqrt{m_3\log(m_2)} := \Re_6.$$

**Proof of Lemma 22**

Consider a cover for the euclidean ball of radius $C_1$ in $\mathbb{R}^{m_3}$ with precision $\epsilon$, i.e. $B_{C_1}(\epsilon)$. So for every $x' \in \mathbb{R}^{m_3}$, there exists an $x \in B_{C_1}(\epsilon)$ such that $\|x - x'\| \leq \epsilon$, and $|B_{C_1}(\epsilon)| \lesssim (\frac{1}{\epsilon})^{m_3}$. Now fix $x'$ and $x$. We have

$$\Gamma_{x',V'} := \frac{1}{\sqrt{m_2}} a^T D_{V^{(0)}+V',x_i}^{x'} V^{(0)} \phi^{(0)}(x_i) = \frac{1}{\sqrt{m_2}} \sum_{j=1}^{m_2} a_j \mathbb{1}\{(V_j^{(0)}+V_j')(\phi^{(0)}(x_i)+x') \geq 0\}V_j^{(0)}\phi^{(0)}(x_i).$$

Now by a union bound, because each variable $V_j^{(0)}\phi^{(0)}(x_i)$ is Gaussian with parameter $\kappa_2\|\phi^{(0)}(x_i)\|$ and using Equation (115), with high probability we have for every $j \in [m_2]$:

$$V_j^{(0)}\phi^{(0)}(x_i) \lesssim \kappa_2\|\phi^{(0)}(x_i)\|\sqrt{\log(m_2)} \lesssim \kappa_1\kappa_2\sqrt{m_3\log(m_2)}. \tag{182}$$

Therefore, by Hoeffding over the randomness of the Bernoulli variables $a_j$, for a fixed $x'$ with high probability:

$$\Gamma_{x'} := \frac{1}{\sqrt{m_2}} \sum_{j=1}^{m_2} a_j \mathbb{1}\{V_j^{(0)}(\phi^{(0)}(x_i) + x') \geq 0\}V_j^{(0)}\phi^{(0)}(x_i) \lesssim \kappa_1\kappa_2\sqrt{m_3\log(m_2)}.$$

On the other hand, We know that the VC-dimension of the class of binary functions with respect to halfspaces in $\mathbb{R}^{m_3}$ is $m_3 + 1$. Therefore, the set of different sign patterns in matrices $D_{V^{(0)},x_i}^{x'}$ is bounded by $m_2^{m_3+1}$, i.e. for

$$\mathcal{D} = \{D_{V^{(0)},x_i}^{x'} \mid x' \in \mathbb{R}^{m_3}\},$$

we have

$$|\mathcal{D}| \lesssim m_2^{m_3+1}.$$

Therefore, by taking a union bound over all sign matrices in $\mathcal{D}$, we get with high probability

$$\sup_{x'} \Gamma_{x'} \lesssim \kappa_1\kappa_2\sqrt{m_3\log(m_2)}\sqrt{\log(m_2^{m_3+1})} = \kappa_1\kappa_2 m_3\log(m_2). \tag{183}$$

Now for a threshold $r$ which satisfies

$$r \geq 2\sqrt{m_3}\kappa_2\epsilon, \tag{184}$$

we define

$$\mathcal{J}_{x,r} = \{j \in [m_2]\mid |V_j^{(0)}(\phi^{(0)}(x_i) + x)| \leq r\}.$$

Now by Equation (115)and the assumption of the Lemma $\sqrt{m_3}\kappa_1 \gtrsim C_1$, we have

$$\|\phi^{(0)}(x_i) + x\| \leq \|\phi^{(0)}(x_i)\| + \|x\| \lesssim \sqrt{m_3}\kappa_1 + C_1. \tag{185}$$

$$\|\phi^{(0)}(x_i) + x\| \geq \|\phi^{(0)}(x_i)\| - \|x\| \gtrsim \sqrt{m_3}\kappa_1 - C_1 \gtrsim \sqrt{m_3}\kappa_1. \tag{186}$$

Hence, $V_j^{(0)}(\phi^{(0)}(x_i) + x)$ is Gaussian with standard deviation at least $\Omega(\kappa_2\sqrt{m_3}\kappa_1)$. Therefore,

$$\mathbb{P}(|V_j^{(0)}(\phi^{(0)}(x_i) + x)| \leq r) \lesssim \frac{r}{\sqrt{m_3}\kappa_1\kappa_2}.$$

This implies

$$\mathbb{E}[|\mathcal{J}_{x,r}|] \lesssim \frac{r}{\sqrt{m_3}\kappa_1\kappa_2} m_2. \tag{187}$$

On the other hand, note that $|\mathcal{J}_{x,r}|$ is the sum of $m_2$ Bernoulli random variables, so it is subGaussian with parameter $m_2$. Therefore, with high probability

$$|\mathcal{J}_{x,r}| \lesssim \frac{r}{\sqrt{m_3}\kappa_1\kappa_2} m_2 + \sqrt{m_2}.$$

Now taking maximum over all $x \in B_{C_1}(\epsilon)$ and exploiting the subGaussian tail of the random variables, we get with high probability

$$\max_{x \in B_{C_1}(\epsilon)} |\mathcal{J}_{x,r}| \lesssim \frac{r}{\sqrt{m_3}\kappa_1\kappa_2} m_2 + \sqrt{m_2 \log(|B_{C_1}(\epsilon)|)} \lesssim \frac{r}{\sqrt{m_3}\kappa_1\kappa_2} m_2 + \sqrt{m_2 m_3 \log(1/\epsilon)}. \tag{188}$$

Moreover, consider a threshold $1 < \theta$, such that $e^{-\theta^2/8} \leq m_2/m_3$, and define the following set of indices

$$\mathcal{J}_{x',\theta}^{(2)} := \{j \in [m_2] | \, |V_j^{(0)}x'| \geq \theta\kappa_2 C_1\}.$$

Then, using Lemma 29 and noting the fact that the standard deviation of Gaussians in $V^{(0)}$ is $\kappa_2$ and that $\|\phi^{(2)}(x_i)\| \leq C_1$, with high probability:

$$\sup_{x': \|x'\|=1} |\mathcal{J}_{x',\theta}^{(2)}| \leq m_3(\log(m_3) + \log(\log(m_2))). \tag{189}$$

Now note that for each $j \in [m_2]$, $\|V_j^{(0)}\|^2$ is subexponential with parameters $(m_3\kappa_2^4, \kappa_2^2)$, which means that with high probability:

$$\max_j \|V_j^{(0)}\|^2 \lesssim m_3\kappa_2^2 + \sqrt{m_3}\kappa_2^2\sqrt{\log(m_2)} + \kappa_2^2 \log(m_2).$$

But with condition $m_3 \geq \log(m_2)$, we can further upper bound it as

$$\max_j \|V_j^{(0)}\|^2 \lesssim m_3\kappa_2^2.$$

Now for fixed $x, x'$, for $j \in \mathcal{J}_{x,r}$ we have

$$\begin{aligned} |V_j^{(0)}(\phi^{(0)}(x_i) + x')| &\leq |V_j^{(0)}(\phi^{(0)}(x_i) + x)| + |V_j^{(0)}(x' - x)| \\ &\leq |V_j^{(0)}(\phi^{(0)}(x_i) + x)| + \|V_j^{(0)}\|\|x' - x\| \\ &\lesssim r + \sqrt{m_3}\kappa_2\epsilon. \end{aligned}$$

On the other hand, for $j \notin \mathcal{J}_{x',\theta}^{(2)}$:

$$|V_j^{(0)}x'| \leq \theta\kappa_2 C_1. \tag{190}$$

Therefore, for $j \in \mathcal{J}_{x,r} - \mathcal{J}_{x',\theta}^{(2)}$:

$$|V_j^{(0)}\phi^{(0)}(x_i)| \leq |V_j^{(0)}(\phi^{(0)}(x_i) + x')| + |V_j^{(0)}x'| \lesssim r + \sqrt{m_3}\kappa_2\epsilon + \theta\kappa_2 C_1. \tag{191}$$

In a similar fashion, if $j \notin \mathcal{J}_{x,r}$, then using assumption (184):

$$|V_j^{(0)}(\phi^{(0)}(x_i) + x')| \geq |V_j^{(0)}(\phi^{(0)}(x_i) + x)| - |V_j^{(0)}(x - x')| \gtrsim r - \sqrt{m_3}\kappa_2\epsilon \geq r/2. \tag{192}$$

Hence, using the fact that $\phi^{(0)}(x_i)$ is orthogonal to $V_j'$:

$$
\begin{aligned}
&\left| \mathbb{1}\{(V_j^{(0)} + V_j')(\phi^{(0)}(x_i) + x') \geq 0\} - \mathbb{1}\{V_j^{(0)}(\phi^{(0)}(x_i) + x') \geq 0\} \right| \\
&\leq \mathbb{1}\{|V_j'(\phi^{(0)}(x_i) + x')| \gtrsim |V_j^{(0)}(\phi^{(0)}(x_i) + x')|\} \\
&\leq \mathbb{1}\{|V_j'x'| \gtrsim |V_j^{(0)}(\phi^{(0)}(x_i) + x')|\} \\
&\leq \mathbb{1}\{\|V_j'\|\|x'\| \gtrsim |V_j^{(0)}(\phi^{(0)}(x_i) + x')|\} \\
&\leq \mathbb{1}\{\|V_j'\|C_1 \gtrsim |V_j^{(0)}(\phi^{(0)}(x_i) + x')|\} \\
&\leq \mathbb{1}\{\|V_j'\| \gtrsim \frac{|V_j^{(0)}(\phi^{(0)}(x_i) + x')|}{2C_1} + \frac{r}{4C_1}\}.
\end{aligned}
\tag{193}
$$

Now by triangle inequality and Equations (193), (191), (190) and the fact that $\|V'\|_F \leq C_2$, we can write:

$$
\begin{aligned}
&|\Gamma_{x'} - \Gamma_{x',V'}| \\
&\leq \frac{1}{\sqrt{m_2}} \sum_{j \in \mathcal{J}_{x,r} - \mathcal{J}_{x',\theta}^{(2)}} \left| \mathbb{1}\{(V_j^{(0)} + V_j')(\phi^{(0)}(x_i) + x') \geq 0\} - \mathbb{1}\{V_j^{(0)}(\phi^{(0)}(x_i) + x') \geq 0\} \right| |V_j^{(0)}\phi^{(0)}(x_i)| \\
&+ \frac{1}{\sqrt{m_2}} \sum_{j \notin (\mathcal{J}_{x,r} \cup \mathcal{J}_{x',\theta}^{(2)})} \left| \mathbb{1}\{(V_j^{(0)} + V_j')(\phi^{(0)}(x_i) + x') \geq 0\} - \mathbb{1}\{V_j^{(0)}(\phi^{(0)}(x_i) + x') \geq 0\} \right| |V_j^{(0)}\phi^{(0)}(x_i)| \\
&+ \frac{1}{\sqrt{m_2}} \sum_{j \in \mathcal{J}_{x',\theta}^{(2)}} \left| \mathbb{1}\{(V_j^{(0)} + V_j')(\phi^{(0)}(x_i) + x') \geq 0\} - \mathbb{1}\{V_j^{(0)}(\phi^{(0)}(x_i) + x') \geq 0\} \right| |V_j^{(0)}\phi^{(0)}(x_i)| \\
&\leq \frac{1}{\sqrt{m_2}} |\mathcal{J}_{x,r} - \mathcal{J}_{x',\theta}^{(2)}| \max_{j \in \mathcal{J}_{x,r} - \mathcal{J}_{x',\theta}^{(2)}} |V_j^{(0)}\phi^{(0)}(x_i)| \\
&+ \frac{1}{\sqrt{m_2}} \sum_{j \notin (\mathcal{J}_{x,r} \cup \mathcal{J}_{x',\theta}^{(2)})} \mathbb{1}\{\|V_j'\| \gtrsim \frac{|V_j^{(0)}(\phi^{(0)}(x_i) + x')|}{2C_1} + \frac{r}{4C_1}\} |V_j^{(0)}\phi^{(0)}(x_i)| \\
&+ \frac{1}{\sqrt{m_2}} |J_{x',\theta}^{(2)}| \max_{j \in m_2} |V_j^{(0)}\phi^{(0)}(x_i)| \\
&\leq \frac{1}{\sqrt{m_2}} |\mathcal{J}_{x,r} - \mathcal{J}_{x',\theta}^{(2)}| \max_{j \in \mathcal{J}_{x,r} - \mathcal{J}_{x',\theta}^{(2)}} |V_j^{(0)}\phi^{(0)}(x_i)| \\
&+ \frac{1}{\sqrt{m_2}} \sum_{j \notin (\mathcal{J}_{x,r} \cup \mathcal{J}_{x',\theta}^{(2)})} \mathbb{1}\{\|V_j'\| \gtrsim \frac{|V_j^{(0)}(\phi^{(0)}(x_i) + x')|}{2C_1} + \frac{r}{4C_1}\} (|V_j^{(0)}(\phi^{(0)}(x_i) + x')| + \theta\kappa_2 C_1) \\
&+ \frac{1}{\sqrt{m_2}} |J_{x',\theta}^{(2)}| \max_{j \in m_2} |V_j^{(0)}\phi^{(0)}(x_i)| \\
&\lesssim \frac{1}{\sqrt{m_2}} |\mathcal{J}_{x,r} - \mathcal{J}_{x',\theta}^{(2)}| \max_{j \in \mathcal{J}_{x,r} - \mathcal{J}_{x',\theta}^{(2)}} |V_j^{(0)}\phi^{(0)}(x_i)| \\
&+ \frac{1}{\sqrt{m_2}} \sum_{j \notin (\mathcal{J}_{x,r} \cup \mathcal{J}_{x',\theta}^{(2)})} \mathbb{1}\{\|V_j'\| \gtrsim \frac{r}{C_1}\} (C_1\|V_j'\| + \theta\kappa_2 C_1) \\
&+ \frac{1}{\sqrt{m_2}} |J_{x',\theta}^{(2)}| \max_{j \in m_2} |V_j^{(0)}\phi^{(0)}(x_i)| \\
&\lesssim \frac{1}{\sqrt{m_2}} |\mathcal{J}_{x,r} - \mathcal{J}_{x',\theta}^{(2)}|(r + \sqrt{m_3}\kappa_2\epsilon + \theta\kappa_2 C_1) + \frac{C_1}{\sqrt{m_2}} \sqrt{\#\left(j : \|V_j'\| \gtrsim \frac{r}{C_1}\right)\|V'\|_F^2}
\end{aligned}
$$

$$+ \frac{1}{\sqrt{m_2}} \# \left( j : \|V_j'\| \gtrsim \frac{r}{C_1} \right) \theta \kappa_2 C_1 + \frac{1}{\sqrt{m_2}} |\mathcal{J}_{x',\theta}^{(2)}| \max_{j \in m_2} |V_j^{(0)} \phi^{(0)}(x_i)|$$

$$\lesssim \frac{1}{\sqrt{m_2}} |\mathcal{J}_{x,r} - \mathcal{J}_{x',\theta}^{(2)}| (r + \sqrt{m_3} \kappa_2 \epsilon + \theta \kappa_2 C_1) + \frac{C_1^2 C_2^2}{\sqrt{m_2} r}$$

$$+ \frac{C_1^3 C_2^2}{\sqrt{m_2} r^2} \theta \kappa_2 + \frac{1}{\sqrt{m_2}} |\mathcal{J}_{x',\theta}^{(2)}| \max_{j \in m_2} |V_j^{(0)} \phi^{(0)}(x_i)|.$$

Now using Equations (207), (189), (182), and (184), and the bound on $|\mathcal{J}_{x',\theta}^{(2)}|$ from Lemma 29, we write

$$\lesssim \frac{1}{\sqrt{m_2}} |\mathcal{J}_{x,r}| (r + \theta \kappa_2 C_1) + \frac{1}{\sqrt{m_2}} |\mathcal{J}_{x',\theta}^{(2)}| \kappa_1 \kappa_2 \sqrt{m_3 \log(m_2)} + \frac{C_1^2 C_2^2}{\sqrt{m_2} r} + \frac{C_1^3 C_2^2}{\sqrt{m_2} r^2} \theta \kappa_2$$

$$\leq \frac{1}{\sqrt{m_2}} \left( \frac{r}{\sqrt{m_3} \kappa_1 \kappa_2} m_2 + \sqrt{m_2 m_3 \log(1/\epsilon)} \right) (r + \theta \kappa_2 C_1)$$

$$+ \frac{1}{\sqrt{m_2}} \left( m_3 (\log(m_3) + \log(\log(m_2))) \right) \kappa_1 \kappa_2 \sqrt{m_3 \log(m_2)} + \frac{C_1^2 C_2^2}{r \sqrt{m_2}} + \frac{C_1^3 C_2^2}{\sqrt{m_2} r^2} \theta \kappa_2.$$

$$\leq \frac{1}{\sqrt{m_2}} \left( \frac{r}{\sqrt{m_3} \kappa_1 \kappa_2} m_2 + \sqrt{m_2 m_3 \log(1/\epsilon)} \right) (r + \theta \kappa_2 C_1)$$

$$+ \frac{m_3^{3/2} \kappa_1 \kappa_2}{\sqrt{m_2}} \sqrt{\log(m_2)} (\log(m_3) + \log(\log(m_2))) + \frac{C_1^2 C_2^2}{r \sqrt{m_2}} + \frac{C_1^3 C_2^2}{\sqrt{m_2} r^2} \theta \kappa_2. \tag{194}$$

Now setting

$$r^* := (C_1 C_2)^{2/3} \frac{m_3^{1/6} (\kappa_1 \kappa_2)^{1/3}}{m_2^{1/3}}.$$

By this choice, from (194) we obtain

$$|\Gamma_{x'} - \Gamma_{x',V'}| \leq \left( \frac{(C_1 C_2)^{4/3}}{(\sqrt{m_2} \kappa_2)^{1/3} (\sqrt{m_3} \kappa_1)^{1/3}} + \frac{(C_1 C_2)^{2/3} m_3^{2/3} (\kappa_1 \kappa_2)^{1/3} \sqrt{\log(1/\epsilon)}}{m_2^{1/3}} \right) \left( 1 + \theta \frac{C_1^{1/3} (\kappa_2 \sqrt{m_2})^{2/3}}{C_2^{2/3} (\kappa_1 \sqrt{m_3})^{1/3}} \right)$$

$$+ \frac{m_3^{3/2} \kappa_1 \kappa_2}{\sqrt{m_2}} \sqrt{\log(m_2)} (\log(m_3) + \log(\log(m_2))).$$

Now we set

$$\theta^* := 3 \log(m_2),$$

which also satisfies the condition of Lemma 29 and combining with Equation (183), we get that with high probability

$$|\Gamma_{x',V'}| \leq |\Gamma_{x',V'} - \Gamma_{x'}| + |\Gamma_{x'}|$$

$$\lesssim \left( \frac{(C_1 C_2)^{4/3}}{(\sqrt{m_2} \kappa_2)^{1/3} (\sqrt{m_3} \kappa_1)^{1/3}} + \frac{(C_1 C_2)^{2/3} m_3^{2/3} (\kappa_1 \kappa_2)^{1/3} \sqrt{\log(1/\epsilon)}}{m_2^{1/3}} \right) \left( 1 + \log(m_2) \frac{C_1^{1/3} (\kappa_2 \sqrt{m_2})^{2/3}}{C_2^{2/3} (\kappa_1 \sqrt{m_3})^{1/3}} \right)$$

$$+ \frac{m_3^{3/2} \kappa_1 \kappa_2}{\sqrt{m_2}} \sqrt{\log(m_2)} (\log(m_3) + \log(\log(m_2))) + \kappa_1 \kappa_2 \sqrt{m_3 \log(m_2)},$$

where $\|x'\| \lesssim C_2$ and $\|V'\|_F \leq C_2$, $\forall j : V_j' \phi^{(0)}(x_i) = 0$. We also need to satisfy condition (184), which regarding this choice for $\theta = \theta^*$ becomes

$$r^* = (C_1 C_2)^{2/3} \frac{m_3^{1/6} (\kappa_1 \kappa_2)^{1/3}}{m_2^{1/3}} \geq 2 \sqrt{m_3} \kappa_2 \epsilon, \tag{195}$$

for which it suffices to set

$$\epsilon^* := (C_1 C_2)^{2/3} \frac{\kappa_1^{1/3}}{2 (m_2 m_3)^{1/3} \kappa_2^{2/3}}, \tag{196}$$

Substituting this choice of $\epsilon$ above and picking the overparameterization large enough to dominate the magnitude of $C_1, C_2$ so that $\log(1/\epsilon^*) \lesssim \log(m_2)$, the proof is complete.

**Lemma 23** *Under the following condition*

$$(\sqrt{m_2}\kappa_2)^{1/3}(\sqrt{m_3}\kappa_1)^{2/3} \geq \log^{-7/6}(m_2)(C_1 C_2)^{1/3},$$

*with high probability we have*

$$\sup_{\|x'\|\lesssim C_1, \|V'\|_F \leq C_2, V' \perp \phi^{(0)}} \frac{1}{\sqrt{m_2}} \| \sum_j a_j \mathbb{1}\{(V_j^{(0)} + V_j')(\phi^{(0)}(x_i) + x') \geq 0\} V_j^{(0)} \|$$

$$\lesssim \sqrt{m_3}\kappa_2 \log(m_2) + \frac{(\sqrt{m_2}\kappa_2)^{1/3}}{(\sqrt{m_3}\kappa_1)^{2/3}}(C_1 C_2)^{2/3} \log^{1/6}(m_2) := \Re_7.$$

**Proof of Lemma 23**

Similar to Lemma 22, define the helper functions $\Gamma_{x'}$ and $\Gamma_{x',V'}$ as

$$\Gamma_{x',V'} = \frac{1}{\sqrt{m_2}} \| \sum_j a_j \mathbb{1}\{(V_j^{(0)} + V_j')(\phi^{(0)}(x_i) + x') \geq 0\} V_j^{(0)} \|, \tag{197}$$

$$\Gamma_{x'} = \frac{1}{\sqrt{m_2}} \| \sum_j a_j \mathbb{1}\{V_j^{(0)}(\phi^{(0)}(x_i) + x') \geq 0\} V_j^{(0)} \|. \tag{198}$$

First we bound $\sup_{x'} \Gamma_{x'}$. To this end, note that because $V_j^{(0)} \in \mathbb{R}^{m_3}$ and the VC-dimension of half-planes is $m_3 + 1$, then by Sauer's Lemma, the set

$$\mathcal{D} = \{D_{V^{(0)},x_i}^{x'} \mid x' \in \mathbb{R}^{m_3}, \|x'\| \lesssim C_1\}$$

of all sign pattern matrices has cardinality at most

$$|\mathcal{D}| \leq m_2^{m_3+1}.$$

Now note that with high probability, the entries of the matrix $V^{(0)}$ are all less than $O(\kappa_2 \sqrt{\log(m_2 m_3)})$. On the other hand, for each fixed sign pattern $D_{V^{(0)},x_i}^{x'}$, we have for the sum with respect to this sign pattern:

$$\| \frac{1}{\sqrt{m_2}} \sum_j a_j \mathbb{1}\{V_j^{(0)}(\phi^{(0)}(x_i) + x') \geq 0\} V_j^{(0)} \|^2 \tag{199}$$

is $(m_3 \kappa_2^4 \log^2(m_2 m_3), \kappa_2^2 \log(m_2 m_3))$ sub-exponential with respect to the randomness of $a$, because each entry of the vector $\frac{1}{\sqrt{m_2}} \sum_j a_j \mathbb{1}\{V_j^{(0)}(\phi^{(0)}(x_i) + x') \geq 0\} V_j^{(0)}$ is $(\kappa_2 \sqrt{\log(m_2 m_3)})$-subGaussian. Therefore, with high probability we have

$$\| \frac{1}{\sqrt{m_2}} \sum_j a_j \mathbb{1}\{V_j^{(0)}(\phi^{(0)}(x_i) + x') \geq 0\} V_j^{(0)} \|^2 \tag{200}$$

$$\leq \mathbb{E}_a[\| \frac{1}{\sqrt{m_2}} \sum_j a_j \mathbb{1}\{V_j^{(0)}(\phi^{(0)}(x_i) + x') \geq 0\} V_j^{(0)} \|^2] + \text{deviation} \tag{201}$$

$$\lesssim m_3 \kappa_2^2 \log(m_2 m_3) + \sqrt{m_3}\kappa_2^2 \log(m_2 m_3) + \kappa_2^2 \log(m_2 m_3). \tag{202}$$

Similarly, if we take a union bound over all sign matrices in $\mathcal{D}$ and using the fact that $m_2 > m_3$:

$$\sup_{x'} \Gamma_{x'}^2 = \sup_{x' \in \mathbb{R}^{m_3}} \| \frac{1}{\sqrt{m_2}} \sum_j a_j \mathbb{1}\{V_j^{(0)}(\phi^{(0)}(x_i) + x') \geq 0\} V_j^{(0)} \|^2 \tag{203}$$

$$\lesssim m_3 \kappa_2^2 \log(m_2 m_3) + \sqrt{m_3}\kappa_2^2 \log(m_2 m_3)\sqrt{\log(m_2^{m_3+1})} + \kappa_2^2 \log(m_2 m_3) \log(m_2^{m_3+1}) \tag{204}$$

$$\lesssim m_3 \kappa_2^2 \log^2(m_2), \tag{205}$$

which implies

$$\sup_{x'} \Gamma_{x'} \lesssim \sqrt{m_3}\kappa_2 \log(m_2). \tag{206}$$

Moreover, defining $\mathcal{J}_{v,x}$ similar to Lemma 22 and using the similar approach we get with high probability

$$\max_{x \in B_{C_1}(\epsilon)} |\mathcal{J}_{x,r}| \lesssim \frac{r}{\sqrt{m_3}\kappa_1\kappa_2}m_2 + \sqrt{m_2 \log(|B_{C_1}(\epsilon)|)} \lesssim \frac{r}{\sqrt{m_3}\kappa_1\kappa_2}m_2 + \sqrt{m_2 m_3 \log(1/\epsilon)}.$$
(207)

Now for simplifying the analysis, we assume that for indices $j \in \mathcal{J}_{x,r}$ we can change the sign pattern with no cost on $V'$, i.e. we can pick any subset of them. Therefore, we first compute a high probability upper bound on the following quantity:

$$\frac{1}{\sqrt{m_2}} \sup_{S \subset \mathcal{J}_{x,r}, \pm\text{signs}} \| \sum_{j \in S} \pm V_j^{(0)} \|.$$
(208)

If we form the matrix $V^{(0)}(\mathcal{J}_{x,r})$ be the matrix which only keeps the rows with indices in $\mathcal{J}_{x,r}$, then the above quantity can be computed as

$$\frac{1}{\sqrt{m_2}} \sup_{S \subset \mathcal{J}_{x,r}, \pm\text{signs}} \| \sum_{j \in S} \pm V_j^{(0)} \| = \frac{1}{\sqrt{m_2}} \sup_{v \in \{0,1,-1\}^{|\mathcal{J}_{x,r}|}} \| v^T V^{(0)}(\mathcal{J}_{x,r}) \|$$
(209)

$$\leq \frac{1}{\sqrt{m_2}} \lambda_{\max}(V^{(0)}(\mathcal{J}_{x,r})) \sup \|v\| \leq \frac{1}{\sqrt{m_2}} \lambda_{\max}(V^{(0)}(\mathcal{J}_{x,r}))|\mathcal{J}_{x,r}|,$$
(210)

where $\lambda_{max}$ is the maximum singular value of the matrix. Now by random matrix theory, we know for a fixed $x$ and arbitrary $t \geq 0$, the following argument holds:

$$\mathbb{P}(\lambda_{\max}(V^{(0)}(\mathcal{J}_{x,r}))/\kappa_2 \gtrsim \sqrt{m_3} + \sqrt{|\mathcal{J}_{x,r}|} + t) \leq 2e^{-ct^2}.$$
(211)

Therefore, as $|\mathcal{D}| \leq m_2^{m_3+1}$, we get with high probability

$$\max_{x \in B_{C_1}(\epsilon)} \lambda_{\max}(V^{(0)}(\mathcal{J}_{x,r})) \lesssim \kappa_2(\sqrt{m_3} + \sqrt{|\mathcal{J}_{x,r}|} + \sqrt{\log(m_2^{m_3+1})})$$
(212)

$$\leq \kappa_2\sqrt{\log(m_2)m_3} + \kappa_2\sqrt{|\mathcal{J}_{x,r}|}.$$
(213)

Therefore, with high probability

$$\sup_{x \in B_{C_1}(\epsilon)} \frac{1}{\sqrt{m_2}} \sup_{S \subset \mathcal{J}_{x,r}} \| \sum_{j \in S} V_j^{(0)} \| \leq \frac{\kappa_2}{\sqrt{m_2}}(\sqrt{\log(m_2)m_3|\mathcal{J}_{x,r}|} + |\mathcal{J}_{x,r}|).$$
(214)

On the other hand, as in Equation (192) in the proof of Lemma 22, for $j \notin \mathcal{J}_{x,r}$ we have:

$$|V_j^{(0)}(\phi^{(0)}(x_i) + x')| \geq |V_j^{(0)}(\phi^{(0)}(x_i) + x)| - |V_j^{(0)}(x - x')| \gtrsim r - \sqrt{m_3}\kappa_2\epsilon.$$
(215)

Picking

$$\epsilon^* := \frac{r}{2\sqrt{m_3}\kappa_2},$$

we get for $j \notin \mathcal{J}_{x,r}$

$$|V_j^{(0)}(\phi^{(0)}(x_i) + x')| \gtrsim r.$$

Now similar to the derivation in (193) we have

$$\left| \mathbb{1}\{(V_j^{(0)} + V_j')(\phi^{(0)}(x_i) + x') \geq 0\} - \mathbb{1}\{V_j^{(0)}(\phi^{(0)}(x_i) + x') \geq 0\} \right|$$
(216)

$$\leq \mathbb{1}\{\|V_j'\|C_1 \gtrsim |V_j^{(0)}(\phi^{(0)}(x_i) + x')|\}$$
(217)

$$\leq \mathbb{1}\{\|V_j'\| \gtrsim \frac{r}{C_1}\}.$$
(218)

Hence, because $\|V'\|_F \leq C_2$, the number of indices for which $\mathbb{1}\{(V_j^{(0)} + V_j')(\phi^{(0)}(x_i) + x') \geq 0\} \neq \mathbb{1}\{V_j^{(0)}(\phi^{(0)}(x_i) + x') \geq 0\}$ is at most $l = \frac{(C_1 C_2)^2}{r^2}$. Therefore, we bound the following quantity to use in the analysis:

$$\sup_{S \subset [m_2] \ \& \ |S| \leq l, \pm\text{signs}} \| \sum_{j \in S} \pm V_j^{(0)} \|.$$
(219)

But if we define for $m_2 < j \leq 2m_2$,

$$V_j^{(0)} = -V_{j-m_2}^{(0)},$$

then

$$\sup_{S \subset [m_2] \ \& \ |S| \leq l, \pm \text{signs}} \|\sum_{j \in S} \pm V_j^{(0)}\| \leq \sup_{S \subset [2m_2] \ \& \ |S| \leq l} \|\sum_{j \in S} V_j^{(0)}\|.$$

Now note that each entry of $\sum_{j \in S} V_j^{(0)}$ is $\sqrt{l}\kappa_2$ subGaussian. Hence, the quantity $\|\sum_{j \in S} V_j^{(0)}\|^2$ is $(m_3 l^2 \kappa_2^4, l\kappa_2^2)$ subexponential. Therefore, we have with high probability

$$\sup_{|S| \leq l} \|\sum_{j \in S} V_j^{(0)}\|^2 \lesssim \mathbb{E}\|\sum_{j \in S} V_j^{(0)}\|^2 + \sqrt{m_3} l\kappa_2^2 \sqrt{\log \binom{2m_2}{l}} + l\kappa_2^2 \log \binom{2m_2}{l}$$

$$\lesssim m_3 l\kappa_2^2 + \sqrt{m_3} l\kappa_2^2 \sqrt{\log \binom{2m_2}{l}} + l\kappa_2^2 \log \binom{2m_2}{l}$$

$$\lesssim m_3 l\kappa_2^2 + \sqrt{m_3} l^{3/2}\kappa_2^2 \sqrt{\log(m_2)} + l^2 \kappa_2^2 \log(m_2).$$

Hence

$$\sup_{|S| \leq l} \|\sum_{j \in S} V_j^{(0)}\| \lesssim \sqrt{m_3}\sqrt{l}\kappa_2 + l\kappa_2 \sqrt{\log(m_2)}.$$

Now using Equation , we can write

$$|\Gamma_{x'} - \Gamma_{x',V'}|$$

$$\leq \frac{1}{\sqrt{m_2}} \| \sum_{j \in \mathcal{J}_{x,r}} (\mathbb{1}\{(V_j^{(0)} + V_j')(\phi^{(0)}(x_i) + x') \geq 0\} - \mathbb{1}\{V_j^{(0)}(\phi^{(0)}(x_i) + x') \geq 0\})V_j^{(0)}\|$$

$$+ \frac{1}{\sqrt{m_2}} \| \sum_{j \notin \mathcal{J}_{x,r}} (\mathbb{1}\{(V_j^{(0)} + V_j')(\phi^{(0)}(x_i) + x') \geq 0\} - \mathbb{1}\{V_j^{(0)}(\phi^{(0)}(x_i) + x') \geq 0\})V_j^{(0)}\|$$

$$\leq \frac{1}{\sqrt{m_2}} \sup_{S \subset \mathcal{J}_{x,r}, \pm\text{signs}} \|\sum_{j \in S} \pm V_j^{(0)}\|$$

$$+ \frac{1}{\sqrt{m_2}} \sup_{S \subset [m_2] \ \& \ |S| \leq (\frac{C_1 C_2}{r})^2, \pm\text{signs}} \|\sum_{j \in S} \pm V_j^{(0)}\|$$

$$\leq \frac{\kappa_2}{\sqrt{m_2}}(\sqrt{\log(m_2)m_3|\mathcal{J}_{x,r}|} + |\mathcal{J}_{x,r}|) + \frac{1}{\sqrt{m_2}}\left(\sqrt{m_3}\sqrt{l}\kappa_2 + l\kappa_2 \sqrt{\log(m_2)}\right)$$

$$\leq \frac{\kappa_2}{\sqrt{m_2}}\left(\frac{rm_2}{\sqrt{m_3}\kappa_1 \kappa_2} + \sqrt{m_2 m_3 \log(\frac{\sqrt{m_3}\kappa_2}{r})}\right)^{1/2}(\sqrt{\log(m_2)m_3} + \sqrt{|\mathcal{J}_{x,r}|})$$

$$+ \frac{1}{\sqrt{m_2}}\left(\sqrt{m_3}(C_1 C_2/r)\kappa_2 + (C_1 C_2/r)^2 \kappa_2 \sqrt{\log(m_2)}\right)$$

$$\lesssim \frac{\kappa_2}{\sqrt{m_2}}\left(\frac{rm_2}{\sqrt{m_3}\kappa_1 \kappa_2} + \sqrt{m_2 m_3 \log(\frac{\sqrt{m_3}\kappa_2}{r})}\right)$$

$$+ \frac{1}{\sqrt{m_2}}\left(\sqrt{m_3}(C_1 C_2/r)\kappa_2 + (C_1 C_2/r)^2 \kappa_2 \sqrt{\log(m_2)}\right).$$

Combining this with (206):

$$|\Gamma_{x',V'}| \lesssim \sqrt{m_3}\kappa_2 \log(m_2) \tag{220}$$

$$+ \frac{r\sqrt{m_2}}{\sqrt{m_3}\kappa_1} + \kappa_2 \sqrt{m_3 \log(\frac{\sqrt{m_3}\kappa_2}{r})} + \frac{1}{\sqrt{m_2}}\left(\sqrt{m_3}(C_1 C_2/r)\kappa_2 + (C_1 C_2/r)^2 \kappa_2 \sqrt{\log(m_2)}\right). \tag{221}$$

Now setting

$$r^* := \frac{m_3^{1/6}(\kappa_1 \kappa_2)^{1/3}}{m_2^{1/3}}(C_1 C_2)^{2/3} \log^{1/6}(m_2),$$

we get

$$LHS \lesssim \sqrt{m_3}\kappa_2 \log(m_2) + \frac{(\sqrt{m_2}\kappa_2)^{1/3}}{(\sqrt{m_3}\kappa_1)^{2/3}}(C_1 C_2)^{2/3}\log^{1/6}(m_2) \tag{222}$$

$$+ \kappa_2 m_3^{1/2}\log^{1/2}(m_2) + \frac{m_3^{1/3}\kappa_2^{2/3}(C_1 C_2)^{1/3}}{m_2^{1/6}\kappa_1^{1/3}\log^{1/6}(m_2)} \tag{223}$$

$$\lesssim \sqrt{m_3}\kappa_2 \log(m_2) + \frac{(\sqrt{m_2}\kappa_2)^{1/3}}{(\sqrt{m_3}\kappa_1)^{2/3}}(C_1 C_2)^{2/3}\log^{1/6}(m_2) \tag{224}$$

$$+ \frac{m_3^{1/2}\kappa_2(C_1 C_2)^{1/3}}{(\sqrt{m_2}\kappa_2)^{1/3}(\sqrt{m_3}\kappa_1)^{1/3}\log^{1/6}(m_2)}. \tag{225}$$

Now under the condition

$$(\sqrt{m_2}\kappa_2)^{1/3}(\sqrt{m_3}\kappa_1)^{2/3} \geq \log^{-7/6}(m_2)(C_1 C_2)^{1/3},$$

The final term is dominated by the first term, which finally completes the proof.

## A.16 Convergence

The goal of this section is to prove Theorem 7.

**Theorem 7** *Letting $\aleph = 4B^2$, by Corollary 8.1, we have $L^\Pi(0) \leq \aleph$. We define the domain $\mathcal{D}_l := \{\|w'\| \leq C_1 := \frac{\aleph + 4l}{\psi_1}, \|v'\| \leq C_2 := \frac{\aleph + 4l}{\psi_2}\}$. For a large enough constant $l = O(1)$ and function $L^\Pi(w := (w', v')) : \mathbb{R}^N \to \mathbb{R}$. Moreover, suppose $L^\Pi$ is $\rho_1$ Lipschitz, $\rho_2$ gradient Lipschitz, and $\rho_3$ hessian Lipschitz in the domain $\mathcal{D}_l$ ($\rho_1, \rho_2, \rho_3 \geq 1$), in the sense that their first, second, and third directional derivatives in an arbitrary unit direction is bounded by the corresponding parameters. Suppose we have access to the gradient of $L^\Pi$ at each point in $\mathcal{D}_l$ plus a zero mean noise vector $\pounds$ such that $\sigma_1^2 I \leq \mathbb{E}\pounds\pounds^T \leq \sigma_2^2 I$ and $\|\pounds\| \leq Q$ almost surely. Also, suppose for a threshold $\aleph_\ell \leq \aleph$, if $L^\Pi(w) \geq \aleph_\ell$ and $w \in \mathcal{D}_l$, then we have at least one of the following conditions holds:*

$$(1) \ \|\nabla L^\Pi(w)\| \geq \frac{\nu}{16\sqrt{C_1^2 + C_2^2}}, \tag{226}$$

$$(2) \ \lambda_{min}(\nabla^2 L^\Pi(w)) \leq -\gamma. \tag{227}$$

*Then starting from $w_0 = 0$, with probability at least $0.999$ after at most $poly(\rho_1, \rho_2, \rho_3, Q, \aleph, C_1, C_2, 1/\gamma, \log(\sigma_1/\sigma_2))$ number of iterations, we reach a point $w_t$ such that $L^\Pi(w_t) \leq \aleph_\ell$.*

**Proof** Our proof here is a refined version of that in Ge et al. (2015a). As we mentioned in section 5, the key fact that we are using in the other parts of our proof is a uniform upper bound $\|w'\| \leq C_1$, $\|v'\| \leq C_2$ which is unjustified by only naively using Ge et al. (2015a). Here, first we restate a refined version of Lemmas 14 and 16 in Ge et al. (2015a) in Lemmas 24 and 26 respectively, and then use them to also bound the upward deviations of $L^\Pi$. Moreover, to avoid writing repeated proofs and overwhelm the reader, we mostly treat the arguments in Lemma 16 of Ge et al. (2015a) as blackbox and use them for our purpose here. A point to mention before we start, unlike Lemma 14 of Ge et al. (2015a) where the dependency on other parameters than the step size $\eta$ is more explicit, Lemma 16 hides the dependencies on all the other parameters (which is polynomial). Here, we follow the same style.

We refer to the trajectory of the steps of algorithm by $(w_t)_{t \geq 0}$. In Lemmas of this section, To avoid introducing new notation and complicating things, we refer to the current point of the algorithm by $w_0$, while for the next point of the algorithm we use $w_1$ (in Lemma 24), and $w_T$ (in Lemma 26) respectively. Also, similar to Ge et al. (2015a), $\tilde{O}$ and $\tilde{\Omega}$ below means we are looking at the dependency on $\eta$.

**Lemma 24** *Suppose $L^\Pi(w_0) \leq \aleph + 2l$, and consider a parameter $\chi > 1$ which can be set arbitrarily. For every point $w_0$ such that $L^\Pi(w_0) \leq \aleph + 2l$, $\|\nabla L^\Pi(w_0)\| \geq 2\sqrt{\eta(Q^2 + \sigma_2^2 N)\rho_2\rho_1^2(2\chi + \frac{1}{2})}$, then for $w_1 := w_0 - \eta(\nabla L^\Pi(w_0) + \pounds)$ and random variable $\mathfrak{R}_1$ (depending on $w_0$) defined as*

$$\mathbb{E}L^\Pi(w_1) - L^\Pi(w_0) = -\eta^2\mathfrak{R}_1^2, \tag{228}$$

*we have $\mathfrak{R}_1 = \tilde{\Omega}(1)$ a.s., and almost surely:*

$$\left| L^\Pi(w_1) - L^\Pi(w_0) \right| \leq \eta\mathfrak{R}_1/\sqrt{\rho_2\chi}.$$

*(the expectation is over the randomness of $\pounds$).*

**Proof** This lemma is a tuned version of Lemma 14 in Ge et al. (2015a). First, note that the condition $L^\Pi(w_0) \leq \aleph + 2l$ assures the smoothness coefficients $\rho_1, \rho_2$ and $\rho_3$ for $L^\Pi$ by Corollary **??**. We follow similar to Ge et al. (2015a) (picking $\eta < 1/(2\rho_2)$):

$$\mathbb{E}L^\Pi(w_1) - L^\Pi(w_0) \leq -\frac{\eta}{2}\|\nabla L^\Pi(w_0)\|^2 + \frac{\eta^2\sigma_2^2\rho_2 N}{2}$$

$$\leq -\frac{\eta}{4}\|\nabla L^\Pi(w_0)\|^2 - \eta^2(\sigma_2^2 N + Q^2)\rho_2\rho_1^2(2\chi + \frac{1}{2}) + \frac{\eta^2\sigma_2^2\rho_2 N}{2}$$

$$\leq -\frac{\eta}{4}\|\nabla L^\Pi(w_0)\|^2 - 2\eta^2 Q^2\rho_2\rho_1^2\chi. \tag{229}$$

where we used the fact that $\rho_1 \geq 1$. On ther other hand, $L^\Pi$ is $\rho_1$ Lipcshitz, so we have almost surely

$$|L^\Pi(w_1) - L^\Pi(w_0)| \leq \rho_1\eta(\|\nabla L^\Pi(w_0) + \pounds\|) \leq \rho_1\eta(\|\nabla L^\Pi(w_0)\| + \|\pounds\|)$$
$$\leq \rho_1\eta(\|\nabla L^\Pi(w_0)\| + Q). \tag{230}$$

(To be completely precise, we should justify that we can write the Lipschitz inequality at point $w_0$, we also need to make sure that $w_1$ remains in the domain that we have the Lipschitz parameter in, i.e. $\mathcal{D}_l$. To see why this is true, see the next Corollary).

Therefore

$$(\rho_2\chi)\Big|L^\Pi(w_1) - L^\Pi(w_0)\Big|^2 \leq 2\eta^2\rho_1^2\rho_2\chi\|\nabla L^\Pi(w_0)\|^2 + 2\rho_2\chi\eta^2\rho_1^2 Q^2. \tag{231}$$

Taking

$$\eta \leq (8\rho_1^2\rho_2\chi)^{-1}, \tag{232}$$

we get from Equation (229):

$$\mathbb{E}L^\Pi(w_1) - L^\Pi(w_0) \leq -2\eta^2\rho_1^2\rho_2\chi\|\nabla L^\Pi(w_0)\|^2 - 2\rho_2\chi\eta^2.\rho_1^2 Q^2. \tag{233}$$

Combining Equations (231) and (233), we see that

$$(\rho_2\chi)\Big|L^\Pi(w_1) - L^\Pi(w_0)\Big|^2 \leq -(\mathbb{E}L^\Pi(w_1) - L^\Pi(w_0)).$$

Hence, if we define

$$\mathbb{E}L^\Pi(w_1) - L^\Pi(w_0) \coloneqq -\eta^2\mathfrak{R}_1^2,$$

we get

$$\Big|L^\Pi(w_1) - L^\Pi(w_0)\Big| \leq \eta\mathfrak{R}_1/\sqrt{\rho_2\chi}, \tag{234}$$

and from Equation (233), that

$$\mathfrak{R}_1^2 \geq 2\rho_1^2\rho_2\chi\|\nabla L^\Pi(w_0)\|^2 + 2\rho_2\chi\rho_1^2 Q^2 \geq 2\rho_2\chi\rho_1^2 Q^2 = \tilde{\Omega}(1).$$

Moreover, because the function is $\rho_1$-Lipshitz at the domain point $w_0$, we get from Equation (230):

$$-\eta^2\mathfrak{R}_1^2 = \mathbb{E}L^\Pi(w_1) - L^\Pi(w_0) \geq -\eta\rho_1(\|\nabla L^\Pi(w_0)\| + Q) \geq -\eta\rho_1(\rho_1 + Q) \geq -\frac{1}{\rho_2\chi},$$

by taking

$$\eta \leq (\rho_1(\rho_1 + Q)\rho_2\chi)^{-1},$$

which implies

$$\eta\mathfrak{R}_1 \leq \frac{1}{\sqrt{\rho_2\chi}}.$$

This, combined with Equation (234) and triangle inequality implies:

$$\Big|L^\Pi(w_1) - \mathbb{E}L^\Pi(w_1)\Big| \leq \eta\mathfrak{R}_1/\sqrt{\rho_2\chi} + \eta^2\mathfrak{R}_1^2 \leq 2\eta\mathfrak{R}_1/\sqrt{\rho_2\chi}. \tag{235}$$

**Lemma 25** *As long as the value of the function at some $w$ is bounded by $\aleph + 2l$ ($L^\Pi(w) \leq \aleph + 2l$), then $\eta$ can be picked small enough (polynomially in other parameters), namely $\eta \leq l/(\|\nabla L^\Pi(w_0)\| + Q)$, so that the change of the function by a step is bounded by $l$.*

**Proof** Let $\psi = \min\{\psi_1, \psi_2\}$. First, note that as the function is bounded by $\aleph + 2l$, we have the Lipschitz parameter $\rho_1$, hence $\|\nabla L^\Pi(w)\| \leq \rho_1$. Therefore, the change in $w$ in a step is bounded as

$$\|\nabla L^\Pi(w) + \pounds\| \leq Q + \rho_1.$$

So by picking

$$\eta \leq \Big(\sqrt{\frac{\aleph + 3l}{\psi}} - \sqrt{\frac{\aleph + 2l}{\psi}}\Big)/(Q + \rho_1),$$

we guarantee that the value of $w$ after a step remains in the ball of radius $\sqrt{\frac{\aleph+3l}{\psi}}$, hence we still have the smoothing parameters even after one step. Therefore, now we can use the Lipcshitz parameter $\rho_1$ to bound the value of the function after one step as it is written in Equation Equation (230). Using this Equation, it is enough to pick $\eta$ as small as:

$$\eta \le l/(\|\nabla L^\Pi(w_0)\| + Q), \tag{236}$$

so that the change in the function would be at most $l$ as desired.

**Lemma 26** *For a fixed point $w_0$ s.t. $L^\Pi(w_0) \le \aleph + 2l$, suppose we pick $\eta$ small enough such that*

$$\S(\eta) := 2\sqrt{\eta(Q^2 + \sigma_2^2 N)\rho_2\rho_1^2(2\chi + \frac{1}{2})} < \frac{\nu}{16\sqrt{C_1^2 + C_2^2}}.$$

*Then, note that for $\|\nabla L^\Pi(w_0)\| \le \S(\eta)$, condition 227 implies:*

$$\lambda_{min}\left(\nabla^2 L^\Pi(w_0)\right) \le \gamma.$$

*Then, using the notation $\mathfrak{E}_T$ for the high probability event corresponding to Equations (36) and (44) in Ge et al. (2015a), for small enough $\eta$ (polynomially small w.r.t other parameters), for $\mathfrak{R}_2$ defined as*

$$\mathbb{E}[L^\Pi(w_T) - L^\Pi(w_0)]\mathbb{1}\{\mathfrak{E}_T\} = -\mathfrak{R}_2^2\eta^2, \tag{237}$$

*we have almost surely*

$$\left|[L^\Pi(w_T) - L^\Pi(w_0)]\mathbb{1}\{\mathfrak{E}_T\}\right| \le \mathfrak{R}_2\eta/\sqrt{\rho_2\chi}. \tag{238}$$

*Note that in the expectations above $w_0$ is assumed fixed. Furthermore, we can assume $\mathbb{P}(\mathfrak{E}_T) \ge 1 - \tilde{O}(\eta^5)$.*

**Proof** This Lemma is a tuned version of Lemma 16 in Ge et al. (2015a). We change a couple of things here. First, we consider an implicit coupling that if $w_t$ exits $\mathcal{D}_l$ we do not move it anymore, i.e. $w_{t'} = w_t, \forall t' \ge t$, which means the noise vectors also becomes zero, i.e. $\mathcal{L}_{t'} = 0, \forall t' \ge t$. This way, the sequence of noise vectors remain bounded by $Q$, because if $w_t$ is inside $\mathcal{D}_l$, then by assumption $\|\mathcal{L}_t\| \le Q$, while otherwise $\mathcal{L}_t = 0$. We denote the event that the sequence $w_0, \dots w_T$ remain in $D_l$ by $\mathcal{E}_T$, where $T$ is defined in Lemma 16 of Ge et al. (2015a).

Note that we also have the smoothing parameters $\rho_1, \rho_2, \rho_3$ for all $(w_t)$ because of this coupling. In fact, we will use a more strict coupling; we consider the event $\mathfrak{E}_T$ to be the high probability event corresponding to the bounds in Equations (44) and (36) of Ge et al. (2015a) holding for all $t \le T$; We will see that $\mathfrak{E}_T \subseteq \mathcal{E}_T$ at the end of this proof, but for now we assume it is true. An important point to note here is that in Ge et al. (2015a), $\mathbb{P}(\mathfrak{E}_T)$ is bounded by $O(\eta^2)$. However, the exponent dependency of $\eta$ in this bound comes from Azuma-Hoeffding type inequalities, particularly used in Equations (60) and (42) in Ge et al. (2015a), in which by considering larger constants one can easily get higher exponents. For our analysis, a bit stronger dependence of $\eta^5$ is required.

Also, because the distribution of our noise depends on the point $w$, our sequence of noise vectors $(\mathcal{L}_t)$ is a martingale instead of being i.i.d, so we apply Azuma-Hoeffding inequality instead of the simple Hoeffdings in Lemma 16 of Ge et al. (2015a). (because we are also sampling a random $(x_i, y_i)$ to compute the estimate of the gradient, this could be simplified to the case where we compute the actual gradient and then inject an i.i.d noise vector in each step, but it is an overhead to compute the actual gradient, so here we choose to analyze the more complicated case.)

Next, notice the definition of $\Lambda$ and $\tilde{\Lambda}$ right after Equation (66) in Ge et al. (2015a), which in our notation translates to

$$\tilde{\Lambda} := \nabla L^\Pi(w_0)^T\tilde{\delta} + \frac{1}{2}\tilde{\delta}^T\mathcal{H}\tilde{\delta}, \ \Lambda = \nabla L^\Pi(w_0)^T\delta + \frac{1}{2}\delta^T\mathcal{H}\delta + \tilde{\delta}^T\mathcal{H}\delta + \frac{\rho_3}{6}\|\tilde{\delta} + \delta\|^3. \tag{239}$$

where

$$\tilde{\delta} = \tilde{w}_T - w_0, \ \delta = w_T - \tilde{w}_T,$$

for $(\tilde{w}_t)$ which is a coupled sequence with $(w_t)$ as defined in Ge et al. (2015a). Note that we apply the coupling for the sequence $\tilde{w}_t$ as well, i.e. if $w_{t+1} = w_t$, we also set $\tilde{w}_{t+1} = \tilde{w}_t$.

To show Equation (237), we want to use Equation (67) in Ge et al. (2015a), though we only use the expansion for the first term which is under $\mathbb{1}\{\mathcal{E}_T\}$, i.e.

$$\mathbb{E}[L^\Pi(w_T) - L^\Pi(w_0)]\mathbb{1}_{\mathfrak{E}_T} = \mathbb{E}\tilde{\Lambda}\mathbb{1}_{\mathfrak{E}_T} + \mathbb{E}\Lambda\mathbb{1}_{\mathfrak{E}_T}. \tag{240}$$

First of all, as it is mentioned in Lemma 16 of (Ge et al., 2015a), in the case where the noise vector $\sigma_1{}^2 I \leq \mathbb{E}\pounds\pounds^T \leq \sigma_2{}^2 I$ instead of having $\mathbb{E}\pounds\pounds^T = \sigma^2 I$ for a fixed $\sigma$, in order to still get a negative term of order $\eta$ in Equation (68) of (Ge et al., 2015a), we just need the size of $T_{\max}$ to be as large as $O(\frac{1}{\gamma\eta}(\log d + \log \frac{\sigma_2}{\sigma_1}))$, and it does not change the order of $\eta$ in any other part of Lemma 16. Now similar to Equation (68) of (Ge et al., 2015a), if w.l.o.g we assume the smallest eigenvalue $\gamma_0$ corresponds to $i = 1$:

$$\mathbb{E}\tilde{\Lambda}\mathbb{1}_{\mathfrak{E}_T} \leq \frac{1}{2}\sum_{i=1}^{N}\lambda_i \sum_{\tau=0}^{T-1}\mathbb{1}_{\{\lambda_i<0\}}(1-\eta\lambda_i)^{2\tau}\eta^2\sigma_1^2\mathbb{P}(\mathfrak{E}_T)+ \tag{241}$$

$$\frac{1}{2}\sum_{i=1}^{N}\lambda_i \sum_{\tau=0}^{T-1}\mathbb{1}_{\{\lambda_i\geq0\}}(1-\eta\lambda_i)^{2\tau}\eta^2\sigma_2^2 \tag{242}$$

$$\leq \frac{\eta^2}{2}\left[\sigma_2^2\frac{N-1}{\eta} - \gamma_0\sigma_1^2\mathbb{P}(\mathfrak{E}_T)\sum_{\tau=0}^{T-1}(1+\eta\gamma_0)^{2\tau}\right] \leq -\frac{\eta\sigma_1^2}{2}. \tag{243}$$

where in the last line we use the fact that $\mathbb{P}(\mathfrak{E}_T) \leq 1/2$ plus the additional $\log(\sigma_2/\sigma_1)$ factor. Second, note that our threshold $\S(\eta)$ for the size of gradient in Lemmas 24 and 26 has the same order of $\eta$ compared to that of Lemmas 14 and 16 in Ge et al. (2015a). Therefore, the arguments in Lemma 16 that considers the order of $\eta$ and treat the other parameters as constants is true here as well. Hence, we still have Equation (69) of Ge et al. (2015a) which is under the event $\mathcal{E}_T$. Applying it to Equation (240),

Hence, finally by a similar derivation of Equation (67) in Ge et al. (2015a):

$$\mathbb{E}[L^\Pi(w_T) - L^\Pi(w_0)]\mathbb{1}\{\mathfrak{E}_T\} \leq -\tilde{\Omega}(\eta). \tag{244}$$

Next, we turn to prove the second bound (238). Combining Equations (36) and (44) in (Ge et al., 2015a), we get with high probability (we use the final high probability parameter of Lemma 16 which is the result of a union bound over all the high probability arguments which is equivalent to the occurrence of $\mathfrak{E}_T$), i.e. when $\mathfrak{E}_T$ happens,

$$\|w_T - w_0\| \leq \tilde{O}(\eta^{\frac{1}{2}}\log\frac{1}{\eta}). \tag{245}$$

Picking $\eta$ small enough such that for the bound above we have

$$O(\eta^{\frac{1}{2}}\log\frac{1}{\eta}) \leq \sqrt{\frac{\aleph+3l}{\psi}} - \sqrt{\frac{\aleph+2l}{\psi}},$$

we get for every $\bar{w}$ in the line connecting $w_0$ to $w_T$ :

$$\|\bar{w}\| \leq \sqrt{\frac{\aleph+3l}{\psi}},$$

which implies that $L^\Pi$ has the smoothing parameters $\rho_1, \rho_2, \rho_3$ along $w_0$ to $w_T$. Therefore, by the $\rho_2$-gradient smoothness property of $L^\Pi$:

$$\|\nabla L^\Pi(\bar{w}) - \nabla L^\Pi(w_0)\| \leq \rho_2\|w_0 - \bar{w}\| \leq O(\rho_2\eta^{\frac{1}{2}}\log\frac{1}{\eta}).$$

Combining the assumption of the Lemma $\|\nabla L^\Pi(w_0)\| \leq \tilde{O}(\eta^{\frac{1}{2}})$, we get

$$\|\nabla L^\Pi(\bar{w})\| = \tilde{O}(\eta^{1/2}\log(1/\eta)).$$

(The last $\tilde{O}$ also hides the dependency on $\rho_2$). Now integrating over the derivative along the direction from $w_0$ to $w_T$:

$$L^\Pi(w_T) = L^\Pi(w_0) + \int_{t=0}^1 \nabla L^\Pi(tw_0 + (1-t)w_T)^T(w_T - w_0)dt.$$

Therefore, using (245) one more time, under the event $\mathcal{E}_T$:

$$|L^\Pi(w_T) - L^\Pi(w_0)| \leq \int \left| \nabla L^\Pi(tw_0 + (1-t)w_T)^T(w_T - w_0) \right| dt$$

$$\leq \int_0^1 \|\nabla L^\Pi(tw_0 + (1-t)w_T)\| \|w_T - w_0\| dt$$

$$\leq \tilde{O}(\eta^{1/2}\log 1/\eta)\|w_T - w_0\| = \tilde{O}(\eta \log^2 1/\eta).$$

Hence

$$\left| [L^\Pi(w_T) - L^\Pi(w_0)]\mathbb{1}\{\mathfrak{E}_T\} \right| \leq \tilde{O}(\eta \log^2 1/\eta), \tag{246}$$

which

Now comparing Equations (244) and (246), it is clear that one can pick $\eta$ small enough (again polynomially small in the other parameters) such that for some random variable $\mathfrak{R}_2$, which also depends on $\eta$, so that equations (237) and (238) hold.

It remains to show $\mathcal{E}_T \subseteq \mathfrak{E}_T$. This is desirable as up until now we have only proved (237) and (238) for the coupled sequence (which does not move outside the ball $D_l$), but we know that under the event $\mathcal{E}_T$, the coupled sequence and the original sequence are the same, which automatically implies the conclusion for the original sequence. Notice that the bound in (246) is an a.s. upper bound on the change of the function value under the event $\mathcal{E}_T$ for every $1 \leq t \leq T$. Therefore, by picking $\eta$ small enough (polynomially) s.t. the quantity $O(\eta \log(1/\eta)^2)$ in Equation (246) is bounded by $l$, we again make sure that the value of function during these steps changes by at most $l$ compared to $w_0$, i.e. for every $1 \leq t \leq T$:

$$\left| [L^\Pi(w_t) - L^\Pi(w_0)]\mathbb{1}\{\mathfrak{E}_t\} \right| \leq l, \tag{247}$$

hence, remains bounded by $\aleph + 3l$. This implies $\mathfrak{E}_T \subseteq \mathcal{E}_T$ as promised.

### A.17 PROCESS FROM A HIGHER VIEW: DEFINITION OF THE $(X)$ SEQUENCE

The goal here is to find a $w^*$ with $L^\Pi(w^*) \leq \aleph_\ell$ using Lemmas 24 and 26 (recall the definition of $\aleph_\ell$ from Theorem 7). The main result of this section is Lemma 27. For this purpose we define a useful coupling: to begin, as done in Ge et al. (2015a), define a sequence of times $\tau_i$ inductively in the following way: To define $\tau_{i+1}$ based on $\tau_i$, if the condition

$$\aleph_\ell \leq L^\Pi(w_{\tau_{i+1}}) \leq \aleph + 2l \tag{248}$$

does not hold, then just set $\tau_{i+1} = \tau_i \star (1)$. Otherwise, using the conditions (227), we are either in the situation of Lemma 24 or Lemma 26 by setting the value of $w_0$ in these Lemmas as $w_0 = w_{\tau_i}$. In the first case, define $\tau_{i+1} = \tau_i + 1 \star (2)$. In the latter case, Let $\mathfrak{E}_T$ be the same high probability event that we consider in Lemma (26), which happens when the aggregate behavior of the noise vectors is normal, as a result of which $w$ remains close to the starting point $w_0$. Note that from Lemma 26, we know $\mathbb{P}(\mathfrak{E}_T) \geq 1 - O(\eta^5)$. Now if the event $\mathfrak{E}_T$ happens, define $\tau_{i+1} := \tau_i + T$ $\star(3)$, for $T$ also from Lemma 26 and defined originally in Lemma 16 of Ge et al. (2015a), while otherwise, define $\tau_{i+1} = \tau_i \star(4)$. Moreover, if $\mathfrak{E}_T$ does not happen, define the rest of $\tau_{i'}$'s equal to $\tau_i$: $\tau_{i'} = \tau_i$ for every $i' \geq i$. At the same time, we define the monotone increasing events $\{\mathcal{G}_i\}$, where $\mathcal{G}_i$ happens in the case $\star(4)$, and $\mathcal{G}_{i+1}$ happens in case $\star(4)$. Also, $\mathcal{G}_i$ happens if any of the previous $\mathcal{G}_{i'}$'s happen for $i' < i$; in other words, $\mathcal{G}_i$ is included in $\mathcal{G}_{i+1}$. We use these events to bound the probability that the process remains above $\aleph_\ell$. Moreover, define the sequence of random variables $(X_i)$ as $X_i := L^\Pi(w_{\tau_i})$. Note that by Lemma 25 and Equation (247) in Lemma 26, we have $\aleph_\ell - l \leq X_i \leq \aleph + 3l$. The key idea behind defining $X_i$'s is that we want to bound the MGF of $L^\Pi(w_t)$, without worrying about falling out of the assumptions of Lemmas 24 and 26. With the definition of $(X_i)$ and $\mathcal{G}_i$, we are ready to state the theorem which roughly says the sequence $\tau_i$ will most likely stop after a number of steps.

**Lemma 27** *Let $\mathcal{Q}_R := \bigcup_{i=1}^{\infty} \left( \{ \tau_i \leq R \} \cap \bar{\mathcal{G}}_i \right)$. Then, for some*

$$R = \frac{O(\log(1/\delta_1))(\aleph + 3l)}{\theta \eta^2}, \tag{249}$$

*we have $\mathbb{P}(\mathcal{Q}_R) \leq \delta_1$. In other words, after $R$ iterations of PSGD, the defined sequence $(X_i)$ above has either been in situation $\star(1)$ or $\star(4)$. Here, $\theta$ depends polynomially in the other parameters.*

**Proof** By Equations (228) and (244) in Lemmas 24 and 26, there exist a constant $\theta$ depending polynomially on all parameters except $\eta$ such that

$$\mathbb{E}[X_{i+1} - X_i | \bar{\mathcal{G}}_i] \leq -\theta(\tau_{i+1} - \tau_i)\eta^2. \tag{250}$$

Now for some constant $C$ that we specify later, define the random time $\imath$ as the largest $i$ where $\tau_i \leq C/\eta^2$. Using the fact that $\mathcal{G}_{i-1} \subseteq \mathcal{G}_i$, for every $i$ we have a.s.:

$$X_{i+1}\mathbb{1}\{\bar{\mathcal{G}}_i\} - X_i\mathbb{1}\{\bar{\mathcal{G}}_{i-1}\} = \mathbb{1}\{\mathcal{G}_i - \mathcal{G}_{i-1}\}(-X_i) + (X_{i+1} - X_i)\mathbb{1}\{\bar{\mathcal{G}}_i\}.$$

Now summing this for $i = 1$ to $\imath$, taking expectation from both sides and using (250):

$$\mathbb{E}X_{\imath+1}\mathbb{1}\{\bar{\mathcal{G}}_\imath\} - X_0 = \sum_{i=1}^{\infty} \mathbb{E}\mathbb{1}\{\mathcal{G}_i - \mathcal{G}_{i-1}\}(-X_i)\mathbb{1}\{\imath \geq i\}$$

$$+ \sum_{i=1}^{\infty} (X_{i+1} - X_i)\mathbb{1}\{\bar{\mathcal{G}}_i \cap \{\imath \geq i\}\}$$

$$\leq \sum_{i=1}^{\infty} \mathbb{E}(\mathbb{1}\{\mathcal{G}_i\} - \mathbb{1}\{\mathcal{G}_{i-1}\})(-X_i)$$

$$+ \sum_{i=1}^{\infty} \mathbb{E}(X_{i+1} - X_i \mid \bar{\mathcal{G}}_i \cap \{\imath \geq i\})\mathbb{P}(\bar{\mathcal{G}}_i \cap \{\imath \geq i\})$$

$$\leq \sup_i \sup |X_i|$$

$$+ \theta \sum_{i=1}^{\infty} \mathbb{E}(-(\tau_{i+1} - \tau_i)\eta^2 \mid \bar{\mathcal{G}}_i \cap \{\imath \geq i\})\mathbb{P}(\bar{\mathcal{G}}_i \cap \{\imath \geq i\})$$

$$= \sup_i \sup |X_i| - \eta^2\theta \sum_{i=1}^{\infty} \mathbb{E}(\tau_{i+1} - \tau_i)\mathbb{1}\{\bar{\mathcal{G}}_i \cap \{\imath \geq i\}\}.$$

Now using Lemma 25, we know that in except when $\mathfrak{E}_T$ happens (in which we stop the time sequence $\tau_i$), the increments of $X_i$ are at most $l$. Therefore, the value of $X_i$'s always remain bounded by $\aleph + 3l$, hence:

$$LHS \leq \aleph + 3l - \theta\eta^2 \sum_{i=1}^{\infty} \mathbb{E}(\tau_{i+1} - \tau_i)\mathbb{1}\{\bar{\mathcal{G}}_i \cap \{\imath \geq i\}\}.$$

Also, by restricting the integration of the second term to the part $\bigcup_{i=1}^{\infty} \left( \bar{\mathcal{G}}_i \cap \{\tau_i \geq 2C/\eta^2\} \right)$ of the sample space, we know that under the event $\{\imath \geq i\}$, $\bar{\mathcal{G}}_i$ automatically happens when $\tau_{i+1} \neq \tau_i$ (it is easy to check). Therefore:

$$LHS \leq \aleph + 3l - \theta\eta^2\mathbb{E}\mathbb{1}\{\bigcup_{i=1}^{\infty} \left( \bar{\mathcal{G}}_i \cap \{\tau_i \geq C/\eta^2\} \right)\} \sum_{i=1}^{\infty} (\tau_{i+1} - \tau_i)\mathbb{1}\{\bar{\mathcal{G}}_i \cap \{\imath \geq i\}\}$$

$$= \aleph + 3l - \theta\eta^2\mathbb{E}\mathbb{1}\{\bigcup_{i=1}^{\infty} \left( \bar{\mathcal{G}}_i \cap \{\tau_i \geq C/\eta^2\} \right)\} \sum_{i=1}^{\infty} (\tau_{i+1} - \tau_i)\mathbb{1}\{\imath \geq i\}$$

$$= \aleph + 3l - \theta\eta^2\mathbb{E}\mathbb{1}\{\bigcup_{i=1}^{\infty} \left( \bar{\mathcal{G}}_i \cap \{\tau_i \geq C/\eta^2\} \right)\} \sum_{i=1}^{\imath} (\tau_{i+1} - \tau_i)$$

$$= \aleph + 3l - \theta\eta^2\mathbb{E}\mathbb{1}\{\bigcup_{i=1}^{\infty} \left( \bar{\mathcal{G}}_i \cap \{\tau_i \geq C/\eta^2\} \right)\}\tau_{\imath+1}.$$

Now by the definition of $\imath$, $\tau_{i+1} \geq C/\eta^2$. Hence, we can write

$$LHS \leq \aleph + 3l - \theta\eta^2 \mathbb{E}\mathbb{1}\{\bigcup_{i=1}^{\infty}\Big(\bar{\mathcal{G}}_i \cap \{\tau_i \geq C/\eta^2\}\Big)\}(C/\eta^2)$$

$$\aleph + 3l - C\theta\mathbb{P}\Big(\bigcup_{i=1}^{\infty}\Big(\bar{\mathcal{G}}_i \cap \{\tau_i \geq C/\eta^2\}\Big)\Big).$$

But note that $X_i$'s are a.s. bounded between 0 and $\aleph + 3l$, which implies the LHS above is at least $-(\aleph + 3l)$. Therefore, we finally get:

$$\mathbb{P}(\bigcup_{i=1}^{\infty}\Big(\bar{\mathcal{G}}_i \cap \{\tau_i \geq C/\eta^2\}\Big)) \leq \frac{2\aleph + 6l}{C\theta}.$$

Picking $C = C^* := 2(2\aleph + 6l)/\theta$:

$$\mathbb{P}(\bigcup_{i=1}^{\infty}\Big(\bar{\mathcal{G}}_i \cap \{\tau_i \geq C^*/\eta^2\}\Big)) \leq \frac{1}{2}. \tag{251}$$

Note that the differences between $\tau_i$'s is at most $T_{\max} = \tilde{O}(1/\eta)$ Ge et al. (2015a). Hence, again for $\eta$ polynomially small in other parameters, (251) implies that for $R = 2C^*/\eta^2$, there exists $\tilde{R} = poly(.)$ such that after $\tilde{R}$ iterations on the main sequence $(w_t)$, the corresponding sequence $(\tau_i)$ has either been in $\star(1)$ or $\star(4)$ with chance at least $1/2$. Repeating this argument $\log(1/\delta_1)$ times (using the markov property of the process) we conclude the proof.

## A.18 BOUNDING THE MGF OF $X_i$'S

Next, we want to exploit $X_i$'s to bound the upward deviation of $L^{\Pi}(w_t)$. For a fix $\theta$ the goal here is to bound $\mathbb{E}[\exp\{\theta X_i\}]$ (this is a different $\theta$!). More precisely, let $F_t$ be the sub-sigma field generated by variables $w_t$ from time zero to $t$, and $\mathcal{F}_i := F_{\tau_i}$ be the sigma field of the stop time $\tau_i$. Then, obviously, $X_i$ is measurable w.r.t $\mathcal{F}_i$. We prove the following theorem:

**Theorem 8** *For any $\theta > 0$, the sequence $(\mathbb{E}e^{\theta(X_i - X_0)})_{i=1}^{\infty}$ is a supermartingale with respect to the filteration $(\mathcal{F}_i)$,*

**Proof** We proceed inductively by jointly conditioning on the previous $X_i$ and whether $\mathcal{G}_i$ has happened or not, and whether we are in situation $\star(2)$ or $\star(3)$. We have

$$\mathbb{E}[\exp\{\theta(X_{i+1} - X_0)\}|\ \mathcal{F}_i]$$
$$= \mathbb{E}[\exp\{\theta(X_{i+1} - X_i + X_i - X_0)\}\mathbb{1}\{\mathcal{G}_i\}|\mathcal{F}_i]$$
$$+ \mathbb{E}[\exp\{\theta(X_{i+1} - X_i + X_i - X_0)\}\mathbb{1}\{\bar{\mathcal{G}}_i \cap \star(2)\}|\mathcal{F}_i]$$
$$+ \mathbb{E}[\exp\{\theta(X_{i+1} - X_i + X_i - X_0)\}\mathbb{1}\{\bar{\mathcal{G}}_i \cap \star(3)\}|\mathcal{F}_i].$$

Now by the a.s. bounds of Lemmas 24 and 26:

$$\mathbb{E}[X_{i+1} - X_i|\ \bar{\mathcal{G}}_i,\ w_{\tau_i}s.t. \star(2)] = -\mathfrak{R}_1^2\eta^2,$$
$$\mathbb{E}[X_{i+1} - X_i|\ \bar{\mathcal{G}}_i,\ w_{\tau_i}s.t. \star(3)] = -\mathfrak{R}_2^2\eta^2,$$
$$(X_{i+1} - X_i)\mathbb{1}\{\mathcal{G}_i\} = 0. \text{ (a.s.)}$$

Where $\mathfrak{R}_1$ and $\mathfrak{R}_2$ are r.v. defined in Lemmas 24 and 26 and are clearly $\mathcal{F}_i$ measurable. This implies

$$\mathbb{E}[(X_{i+1} - X_i)\mathbb{1}\{\bar{\mathcal{G}}_i,\ w_{\tau_i}s.t. \star(2)\}|\ \mathcal{F}_i] = -\mathfrak{R}_1^2\eta^2\mathbb{1}\{\bar{\mathcal{G}}_i,\ w_{\tau_i}s.t. \star(2)\},$$
$$\mathbb{E}[(X_{i+1} - X_i)\mathbb{1}\{\bar{\mathcal{G}}_i,\ w_{\tau_i}s.t. \star(3)\}|\ \mathcal{F}_i] = -\mathfrak{R}_2^2\eta^2\mathbb{1}\{\bar{\mathcal{G}}_i,\ w_{\tau_i}s.t. \star(3)\}.$$

Now we mention the following fact:

**Fact** For a $\sigma$ subGaussian random variable $X$ we have $\mathbb{E}[\exp\{\theta X\}] \leq \exp\{\theta^2\sigma^2\}$.

Using the a.s. bounds of Lemmas 24 and 26, we get that conditioned on $\{\bar{\mathcal{G}}_i,\ w_{\tau_i}\ s.t. \star(2)\}$,

$X_{i+1} - \mathbb{E}X_{i+1}$ is a.s. bounded by $2\eta\theta\mathfrak{R}_1/(\rho_2\chi)$, and conditioned on $\{\bar{\mathcal{G}}_i, \ w_{\tau_i} \ s.t. \ \star (3)\}$, $X_{i+1} - \mathbb{E}X_{i+1}$ is bounded by $2\eta\theta\mathfrak{R}_2/(\rho_2\chi)$. Therefore, using the above fact

$$\mathbb{E}\left[\exp\{\theta(X_{i+1} - \mathbb{E}(X_{i+1}|\ \bar{\mathcal{G}}_i, \ w_{\tau_i} \ s.t. \ \star (2))))\}\Big|\ \bar{\mathcal{G}}_i, \ w_{\tau_i} \ s.t. \ \star (2)\right] \leq \exp\{4\eta^2\theta^2\mathfrak{R}_1^2/(\rho_2\chi)\}, \tag{252}$$

$$\mathbb{E}\left[\exp\{\theta(X_{i+1} - \mathbb{E}(X_{i+1}|\ \bar{\mathcal{G}}_i, \ w_{\tau_i} \ s.t. \ \star (3)))\}\Big|\ \bar{\mathcal{G}}_i, \ w_{\tau_i} \ s.t. \ \star (3)\right] \leq \exp\{4\eta^2\theta^2\mathfrak{R}_2^2/(\rho_2\chi)\}, \tag{253}$$

which implies in the notation of conditional expectation on sigma field:

$$\mathbb{E}\left[\exp\{\theta(X_{i+1} - \mathbb{E}(X_{i+1}|\mathcal{F}_i))\}\mathbb{1}\{\bar{\mathcal{G}}_i, \star(2)\}\Big|\mathcal{F}_i\right] \leq \exp\{4\eta^2\theta^2\mathfrak{R}_1^2/(\rho_2\chi)\}\mathbb{1}\{\bar{\mathcal{G}}_i, \star(2)\}, \tag{254}$$

$$\mathbb{E}\left[\exp\{\theta(X_{i+1} - \mathbb{E}(X_{i+1}|\mathcal{F}_i))\}\mathbb{1}\{\bar{\mathcal{G}}_i, \star(3)\}\Big|\mathcal{F}_i\right] \leq \exp\{4\eta^2\theta^2\mathfrak{R}_2^2/(\rho_2\chi)\}\mathbb{1}\{\bar{\mathcal{G}}_i, \star(3)\}. \tag{255}$$

Now we write:

$$LHS \leq \mathbb{E}[\exp\{\theta(X_{i+1} - X_0)\}\mathbb{1}\{\mathcal{G}_i\}|\ \mathcal{F}_i]$$
$$+\mathbb{E}[\exp\{\theta(X_{i+1} - \mathbb{E}[X_{i+1}|\ \mathcal{F}_i])\}\exp\{\theta(\mathbb{E}[X_{i+1}|\ \mathcal{F}_i] - X_i)\}\exp\{\theta(X_i - X_0)\}\mathbb{1}\{\bar{\mathcal{G}}_i \cap \star(2)\}|\ \mathcal{F}_i]$$
$$+\mathbb{E}[\exp\{\theta(X_{i+1} - \mathbb{E}[X_{i+1}|\ \mathcal{F}_i])\}\exp\{\theta(\mathbb{E}[X_{i+1}|\ \mathcal{F}_i] - X_i)\}\exp\{\theta(X_i - X_0)\}\mathbb{1}\{\bar{\mathcal{G}}_i \cap \star(3)\}|\mathcal{F}_i]$$
$$\leq \exp\{\theta(X_i - X_0)\}\mathbb{1}\{\mathcal{G}_i\}$$
$$+ \exp\{\theta(X_i - X_0)\}\mathbb{E}[\exp\{\theta^2\mathfrak{R}_1^2\eta^2/(\rho_2\chi)\}\exp\{-\theta(\mathfrak{R}_1^2\eta^2)\}\mathbb{1}\{\bar{\mathcal{G}}_i \cap \star(2)\}|\ \mathcal{F}_i]$$
$$+ \exp\{\theta(X_i - X_0)\}\mathbb{E}[\exp\{\theta^2\mathfrak{R}_2^2\eta^2/(\rho_2\chi)\}\exp\{-\theta(\mathfrak{R}_2^2\eta^2)\}\mathbb{1}\{\bar{\mathcal{G}}_i \cap \star(3)\}|\ \mathcal{F}_i]$$
$$\leq \exp\{\theta(X_i - X_0)\}\mathbb{E}\Big[\Big(\mathbb{1}\{\mathcal{G}_i\}$$
$$+ \exp\{\theta^2\mathfrak{R}_1^2\eta^2/(\rho_2\chi) - \theta(\mathfrak{R}_1^2\eta^2)\}\mathbb{1}\{\bar{\mathcal{G}}_i \cap \star(2)\}$$
$$+ \exp\{\theta^2\mathfrak{R}_2^2\eta^2/(\rho_2\chi) - \theta(\mathfrak{R}_2^2\eta^2)\}\mathbb{1}\{\bar{\mathcal{G}}_i \cap \star(3)\}\Big)|\ \mathcal{F}_i\Big].$$

Now setting $\theta := 1$ and picking $\chi \geq 1/\rho_2$:

$$LHS \leq \exp\{(X_i - X_0)\}\mathbb{E}\Big[\mathbb{1}\{\mathcal{G}_i\} + \mathbb{1}\{\bar{\mathcal{G}}_i \cap \star(2)\} + \mathbb{1}\{\bar{\mathcal{G}}_i \cap \star(3)\}|\ \mathcal{F}_i\Big].$$
$$= \exp\{X_i - X_0\}.$$

Now by hypothesis of Induction we have

$$\mathbb{E}[\exp\{X_{i+1} - X_0\}] = \mathbb{E}[\mathbb{E}[\exp\{X_{i+1} - X_0\}|\ \mathcal{F}_i]] \leq \mathbb{E}[\exp\{X_i - X_0\}] \leq 1,$$

which finishes the proof of step of induction.

Now using Doob's Maximal inequality for positive supermartingales and $R$ defined in (249):

$$\mathbb{P}(\sup_{1 \leq i \leq R} (X_i - X_0) \geq z)$$
$$= \mathbb{P}(\sup_{1 \leq i \leq R} \exp\{X_i - X_0\} \geq \exp\{z\}) \leq \mathbb{E}[\exp\{X_R - X_0\}]/\exp\{z\} \leq e^{-z}. \tag{256}$$

## A.19 PROOF OF THEOREM 7

Finally with the developed tools, we are ready to prove Theorem 7.

**Proof** *of Theorem 7* Starting from $w_0 = 0$ with $L^\Pi(w_0) \leq \aleph$, we use Equation (256) to get $\mathbb{P}(\sup_{1 \leq i \leq R} \exp\{X_i - X_0\} \geq \Omega(\log(1/\delta_1))) \leq \delta_1$. Therefore, setting $l = \Theta(\log(1/\delta_1))$ and a union bound implies with probability at least $1 - 2\delta_1$ we should have gotten into situation $\star(1)$ or $\star(4)$ without the value of $X_i$ exceeding $\aleph + 2l$. On the other hand, using Lemma 26 we know that $\mathfrak{E}_T$ happens with probability at least $1 - \tilde{O}(\eta^5)$ for every $1 \leq t \leq R$ which is equal to $\tau_i$ for some $i$

and when we are in the situation of Lemma 26. As a result, the chance that even one of $\mathfrak{E}_T$'s happen along $R$ iterations is at most

$$R\tilde{O}(\eta^5) = \eta^3 \frac{O(\log(1/\delta_1))(\aleph + 3l)}{\theta}.$$

But picking $\eta$ small enough with respect to $\log(\delta_1)$ and other parameters, we conclude that with probability at least $1 - 3\delta$, after $R$ rounds, we should have gotten into situation $\star(1)$ and not $\star(4)$ and not exceeding $\aleph + 2l$, which means that $X_i = L^{\Pi}(w_{\tau_i})$ has gotten under the threshold $\aleph_\ell$. Note that as soon as that happens, we terminate the algorithm. We elaborate on this more in Appendix A.10.

### A.20 GAUSSIAN SMOOTHING

In this section, we describe our smoothing scheme and the approximation that it provides which enables us to keep the signs from the case $\eta = 0$. Recall that we use Gaussian smoothing matrices $V_{j,k}^\rho \sim \mathcal{N}(0, \beta_1^2/m_1)$ and $W_{j,k}^\rho \sim \mathcal{N}(0, \beta_2^2/m_2)$. Here, we will particularly specify lower bounds for $\beta_1$ and $\beta_2$ in order for our sign approximation to be precise. On the other hand, we normally prefer the smoothing noise to be as low as possible so the primary and smoothed functions are close, so we set $\beta_1, \beta_2$ equal to their lower bounds, and use this setting in the other parts.

To begin fix one of the inputs $x_i$. In order to reduce and simplify the amount of notations, we refer to the sign pattern matrix (diagonal sign matrix) of both the first and second layers by $D$ with the appropriate indices. More specifically, for the first layer, we refer to $\text{Sgn}(W^{(0)} + W' + W^\rho)x_i$ by $D',_\rho$ and $\text{Sgn}(W^{(0)} + (1 - \eta/2)W' + W^\rho + \sqrt{\eta}W^*)x_i$ by $D',_\rho$. Similarly, for the second layer, of course depending on the input vector, we refer to the sign matrix with respect to the matrices $V^{(0)} + V' + V^\rho$ and $V^{(0)} + (1 - \eta/2)V' + V^\rho + \sqrt{\eta}V^*$ by by $D',_\rho$ and $D',_{\rho,\eta}$, respectively. We introduce two new notations as well for the output of the first layer with respect to different matrix and sign patterns:

$$x'^{(1)} := W^s D',_\rho (W^{(0)} + (1 - \eta)W' + W^\rho + \sqrt{\eta}V^*)x_i, \tag{257}$$

$$x'^{(2)} := W^s D',_{\rho,\eta}(W^{(0)} + (1 - \eta)W' + W^\rho + \sqrt{\eta}V^*)x_i. \tag{258}$$

For further brevity, we sometimes refer to $x'^{(2)}$ by $x'$.

Now we are ready to mention our approximation theorem regarding the smoothing and the sign changes.

**Lemma 28** *Under the conditions $\kappa_1\sqrt{m_3} \gtrsim C_1 + \beta_1\sqrt{m_3}$ and $m_2 \geq m_3 \log(m_2)$, then for every $i \in [n]$:*

$$\Big| \mathbb{E}_{W^\rho, V^\rho} \, a^T D',_{\rho,\eta}(V^{(0)} + (1 - \eta)V' + V^\rho + \sqrt{\eta}V^*)W^s D',_{\rho,\eta}(W^{(0)} + (1 - \eta)W' + W^\rho + \sqrt{\eta}V^*)x_i$$

$$- a^T D',_\rho(V^{(0)} + (1 - \eta)V' + V^\rho + \sqrt{\eta}V^*)W^s D',_\rho(W^{(0)} + (1 - \eta)W' + W^\rho + \sqrt{\eta}V^*)x_i \Big|$$

$$\leq \eta\varrho_2^2\beta_2^{-1}\Big[(C_1 + \sqrt{m_3}\beta_1)^2/(\kappa_1\sqrt{m_3}) + \big[\sqrt{m_3}m_1\beta_1 + C_1\big]\exp\{-C_1^2/(8m_3\beta_1^2)\}\Big]$$

$$\times \Big[\exp\{-C_2^{4/3}(\sqrt{m_2}\kappa_2)^{2/3}/(8\beta_2^2)\} + \frac{C_2^{2/3}}{(\sqrt{m_2}\kappa_2)^{2/3}}\Big]$$

$$+ \eta\Big(\kappa_2\sqrt{m_2} + C_1\Big)\Big(\exp\{-c_2^2/(32\beta_1^2)\} + \frac{c_2}{\kappa_1\sqrt{m_1}}\Big)\frac{\varrho^2 m_3\sqrt{m_3}}{\beta_1} := \eta\Re_8. \tag{259}$$

**Proof of Lemma 28**

We can bound the Left hand side above as

$LHS \leq$

$\left| \mathbb{E}_{W^\rho, V^\rho} a^T D_{\prime,\rho,\eta} (V^{(0)} + (1-\eta)V' + V^\rho + \sqrt{\eta}V^*) W^s D_{\prime,\rho,\eta} (W^{(0)} + (1-\eta)W' + W^\rho + \sqrt{\eta}W^*) x_i \right.$

$\left. - a^T D_{\prime,\rho} (V^{(0)} + (1-\eta)V' + V^\rho + \sqrt{\eta}V^*) W^s D_{\prime,\rho,\eta} (W^{(0)} + (1-\eta)W' + W^\rho + \sqrt{\eta}W^*) x_i \right|$

$+ \left| \mathbb{E}_{W^\rho, V^\rho} a^T D_{\prime,\rho} (V^{(0)} + (1-\eta)V' + V^\rho + \sqrt{\eta}V^*) W^s D_{\prime,\rho,\eta} (W^{(0)} + (1-\eta)W' + W^\rho + \sqrt{\eta}W^*) x_i \right.$

$\left. - a^T D_{\prime,\rho} (V^{(0)} + (1-\eta)V' + V^\rho + \sqrt{\eta}V^*) W^s D_{\prime,\rho} (W^{(0)} + (1-\eta)W' + W^\rho + \sqrt{\eta}W^*) x_i \right|.$

$:= A_1 + A_2.$ \hfill (260)

We bound $A_1$ and $A_2$ separately. First, we start with $A_1$.

Let $\hat{P}_i$ be the set of indices $j$ for which $\mathbb{1}\{|(V_j^{(0)} + V_j')x'| \leq R^*\kappa_2 \|x'\|\}$ happens. Then, from Lemma 31, we have $|\hat{P}_i| \lesssim R^* m_2$. Now for $j \in [m_2]$, we write

$$\mathbb{1}\{\text{sign change in the } j\text{th neuron}\} \times |\text{amount of change}| \hfill (261)$$

$$\leq \mathbb{1}\{V_j^\rho x' \in (-V_j^{(0)} x' - V_j' x' + \eta V_j' x' - \sqrt{\eta}V_j^* x', -V_j^{(0)} x' - V_j' x')\} \times \frac{1}{\sqrt{m_2}} (|\eta V_j' x'| + |\sqrt{\eta}V_j^* x'|)$$

$$(262)$$

Moreover, note that

$$|V_j' x'| = |V_j'(x' - \phi^{(0)}(x_i))|,$$
$$|V_j^* x'| = |V_j^*(x' - \phi^{(0)}(x_i))|.$$

Also, because $\|V_j'\| \leq \|V'\| \leq 2C_2$ plus using Equation (128), we can further upper bound the above indicator as:

$$\leq \mathbb{1}\{V_j^\rho x' \in (-V_j^{(0)} x' - V_j' x' - (\eta\|V_j'\| + \sqrt{\eta}\|V_j^*\|) \min\{\|x'\|, \|x' - \phi^{(0)}(x_i)\|\}, -V_j^{(0)} x' - V_j' x')\}$$

$$\times \frac{1}{\sqrt{m_2}} (\eta\|V_j'\| + \sqrt{\eta}\|V_j^*\|) \|x' - \phi^{(0)}(x_i)\|$$

$$\leq \mathbb{1}\{V_j^\rho x' \in (-V_j^{(0)} x' - V_j' x' - (2\eta C_2 + \sqrt{\eta}\varrho_2/\sqrt{m_2}) \min\{\|x'\|, \|x' - \phi^{(0)}(x_i)\|\}, -V_j^{(0)} x' - V_j' x')\}$$

$$\times \frac{1}{\sqrt{m_2}} (2\eta C_2 + \sqrt{\eta}\varrho_2/\sqrt{m_2}) \|x' - \phi^{(0)}(x_i)\|.$$

Taking $\sqrt{\eta} \leq \varrho/(2C_2\sqrt{m_2})$, we can further upper bound as

$$\lesssim \mathbb{1}\{V_j^\rho x' \in (-V_j^{(0)} x' - V_j' x' - 2\sqrt{\eta}\varrho_2 \min\{\|x' - \phi^{(0)}(x_i)\|, \|x'\|\}/\sqrt{m_2}, -V_j^{(0)} x' - V_j' x')\}$$

$$\times (\sqrt{\eta}\varrho_2/m_2) \|x' - \phi^{(0)}(x_i)\|.$$

Therefore, conditioned on $x'$:

$$\mathbb{E}_{V^\rho}[\mathbb{1}\{\text{sign change in the } j\text{th neuron}\} \times |\text{amount of change}| \mid x'] \leq$$

$$\mathbb{P}(V_j^\rho x' \in (-V_j^{(0)} x' - V_j' x' - 2\sqrt{\eta}\varrho_2 \min\{\|x' - \phi^{(0)}(x_i)\|, \|x'\|\}/\sqrt{m_2}, -V_j^{(0)} x' - V_j' x'))$$

$$\times (\sqrt{\eta}\varrho_2/m_2) \|x' - \phi^{(0)}(x_i)\|.$$

Now notice that for $j \notin \hat{P}_i$, we have

$$|-V_j^{(0)} x' - V_j' x'| \geq R^*\kappa_2 \|x'\|.$$

Also, note that the variable $V_j^\rho x'$ is gaussian with variance $\|x'\|\beta_2/\sqrt{m_2}$ Therefore, conditioned on $x'$, for $j \notin \hat{P}_i$, we have (note that $x'$ does not depend on the randomness of $V^\rho$):

$$\mathbb{P}(V_j^\rho x' \in (-V_j^{(0)} x' - V_j' x' - 2\sqrt{\eta}\varrho_2 \min\{\|x' - \phi^{(0)}(x_i)\|, \|x'\|\}/\sqrt{m_2}, -V_j^{(0)} x' - V_j' x'))$$

$$\lesssim \exp\{\min\{|-V_j^{(0)}x' - V_j'x' - 2\sqrt{\eta}\varrho_2\|x'\|/\sqrt{m_2}|, |-V_j^{(0)}x' - V_j'x'|\}/(\sqrt{2}\|x'\|\beta_2/\sqrt{m_2})\}^2$$
$$\times(\|x'\|\beta_2/\sqrt{m_2})^{-1} \times (\sqrt{\eta}\varrho_2\min\{\|x' - \phi^{(0)}(x_i)\|, \|x'\|\}/\sqrt{m_2}).$$

This equation follows from the fact that

$$\mathbb{P}(a \leq \mathcal{N} \leq b) \lesssim \frac{|a-b|}{\sigma}e^{\min^2\{a,b\}/\sigma^2}. \tag{263}$$

On the other hand, note that for $\sqrt{\eta} \lesssim \frac{C_2^{2/3}(\sqrt{m_2}\kappa_2)^{1/3}}{\varrho_2}$ we have:

$$R^*\kappa_2\|x'\|/2 = \frac{C_2^{2/3}(\sqrt{m_2}\kappa_2)^{1/3}}{2\sqrt{m_2}}\|x'\| \geq 2\sqrt{\eta}\varrho_2\|x'\|/\sqrt{m_2}$$

which implies

$$\lesssim \exp\{R^*\kappa_2\|x'\|/(2\sqrt{2}\|x'\|\beta_2/\sqrt{m_2})\}^2(\sqrt{\eta}\varrho_2/\beta_2)$$
$$\leq \exp\{-C_2^{4/3}(\sqrt{m_2}\kappa_2)^{2/3}/(8\beta_2^2)\}(\sqrt{\eta}\varrho_2/\beta_2).$$

On the other side, for $j \in \hat{P}_i$, we can write

$$\mathbb{P}(V_j^\rho x' \in (-V_j^{(0)}x' - V_j'x' - 2\sqrt{\eta}\varrho_2\min\{\|x' - \phi^{(0)}(x_i)\|, \|x'\|\}/\sqrt{m_2}, -V_j^{(0)}x' - V_j'x'))$$
$$\lesssim (\|x'\|\beta_2/\sqrt{m_2})^{-1}(\sqrt{\eta}\varrho_2\min\{\|x' - \phi^{(0)}(x_i)\|, \|x'\|\}/\sqrt{m_2}) = \min\{\|x' - \phi^{(0)}(x_i)\|/\|x'\|, 1\}\sqrt{\eta}\varrho_2/\beta_2.$$

Therefore, overall using the fact that $\|V_j^*\| \leq \varrho_2/\sqrt{m_2}$, we can write

$$A_1 \lesssim \sum_{j\notin\hat{P}} \exp\{-C_2^{4/3}(\sqrt{m_2}\kappa_2)^{2/3}/(8\beta_2^2)\}(\sqrt{\eta}\varrho_2/\beta_2)\min\{\|x' - \phi^{(0)}(x_i)\|^2/\|x'\|, \|x' - \phi^{(0)}(x_i)\|\}$$
$$+ \sum_{j\in\hat{P}}(\sqrt{\eta}\varrho_2/\beta_2)\min\{\|x' - \phi^{(0)}(x_i)\|/\|x'\|, 1\} \times (\sqrt{\eta}\varrho_2/m_2)\|x' - \phi^{(0)}(x_i)\|$$
$$\lesssim \Big[m_2 \times \exp\{-C_2^{4/3}(\sqrt{m_2}\kappa_2)^{2/3}/(8\beta_2^2)\}(\sqrt{\eta}\varrho_2/\beta_2) \times (\sqrt{\eta}\varrho_2/m_2)$$
$$+ \eta\varrho_2^2\beta_2^{-1}\frac{|\hat{P}_i|}{m_2}\Big]\min\{\|x' - \phi^{(0)}(x_i)\|^2/\|x'\|, \|x' - \phi^{(0)}(x_i)\|\}$$
$$\leq \eta\varrho_2^2\beta_2^{-1}\min\{\|x' - \phi^{(0)}(x_i)\|^2/\|x'\|, \|x' - \phi^{(0)}(x_i)\|\}\Big[\exp\{-C_2^{4/3}(\sqrt{m_2}\kappa_2)^{2/3}/(8\beta_2^2)\} + \frac{C_2^{2/3}}{(\sqrt{m_2}\kappa_2)^{2/3}}\Big]. \tag{264}$$

Next, we bound $A_2$. First we bound $\mathbb{E}_{W^\rho}\|x'^{(1)} - x'^{(2)}\|$. Recalling the setting $c_2 = 2\sqrt{nm_3}C_1/\sqrt{\lambda_0}$ and the definition of in $P$ from Lemma 1, we obtain that for $j \notin P$, we have for all $i \in [n]$:

$$|W_j^{(0)}x_i| \geq c_2/\sqrt{m_1},$$
$$|W_j'x_i| \leq c_2/(2\sqrt{m_1}),$$

which means for $j \notin P$:

$$|(W_j^{(0)} + W_j')x_i| \geq c_2/(2\sqrt{m_1}). \tag{265}$$

Also, we have

$$|P| \leq c_2\sqrt{m_1}/\kappa_1. \tag{266}$$

Now using Equation (104) in Lemma 5, we can write for every $i \in [n]$:

$$val_j := \mathbb{1}\{\text{sign change in the } j\text{th neuron}\} \times \Big|\text{amount of change}\Big|$$
$$\leq \mathbb{1}\{W_j^\rho x_i \in (-W_j^{(0)}x - W_j'x_i + \eta W_j'x_i - \sqrt{\eta}W_j^*x_i, -W_j^{(0)}x_i - W_j'x_i)\}$$
$$\times \frac{1}{\sqrt{m_1}}(|\sqrt{\eta}W_j^*x_i + \eta W_j'x_i|).$$

Using the fact that $\|W_j'\| \leq \|W'\|_F \leq C_1$, and Equation (104) ($\|W_j^*\| \leq \varrho\sqrt{\frac{m_3}{m_1}}$) and picking $\sqrt{\eta} \leq \frac{\varrho\sqrt{m_3}}{C_1\sqrt{m_1}}$, we obtain

$$
\begin{aligned}
&\leq \mathbb{1}\{W_j^\rho x_i \in (-W_j^{(0)}x_i - W_j'x_i - \eta\|W_j'\| - \sqrt{\eta}\|W_j^*\|, -W_j^{(0)}x_i - W_j'x_i)\} \\
&\quad \times \frac{1}{\sqrt{m_1}}(\sqrt{\eta}\|W_j^*\| + \eta\|W_j'\|) \\
&\leq \mathbb{1}\{W_j^\rho x_i \in (-W_j^{(0)}x_i - W_j'x_i - 2\sqrt{\eta}\varrho\frac{\sqrt{m_3}}{\sqrt{m_1}}, -W_j^{(0)}x_i - W_j'x_i)\} \\
&\quad \times \frac{2}{\sqrt{m_1}}(\sqrt{\eta}\varrho\sqrt{\frac{m_3}{m_1}}).
\end{aligned}
$$

Now for $j \notin P$, because $W_j^\rho x_i$ is Gaussian with std $\frac{\beta_1}{\sqrt{m_1}}$:

$$
\begin{aligned}
\mathbb{E}_{W^\rho}[val_j] &\leq \mathbb{P}(W_j^\rho x_i \in (-W_j^{(0)}x_i - W_j'x_i - 2\sqrt{\eta}\varrho\frac{\sqrt{m_3}}{\sqrt{m_1}}, -W_j^{(0)}x_i - W_j'x_i)) \times \frac{1}{\sqrt{m_1}}\sqrt{\eta}\varrho\frac{\sqrt{m_3}}{\sqrt{m_1}} \\
&\lesssim \exp-\{\min\{|-W_j^{(0)}x_i - W_j'x_i - 2\sqrt{\eta}\varrho\frac{\sqrt{m_3}}{\sqrt{m_1}}|, |-W_j^{(0)}x' - W_j'x'|\}/(\sqrt{2}\beta_1/\sqrt{m_1})\}^2 \\
&\quad \times (\beta_1/\sqrt{m_1})^{-1} \times (\sqrt{\eta}\varrho\frac{\sqrt{m_3}}{\sqrt{m_1}}) \times \frac{1}{\sqrt{m_1}}\sqrt{\eta}\varrho\frac{\sqrt{m_3}}{\sqrt{m_1}}.
\end{aligned}
$$

Now from Equation (265) and by picking $\sqrt{\eta} \lesssim \frac{c_2}{\varrho\sqrt{m_3}}$ so that

$$
2\sqrt{\eta}\varrho\frac{\sqrt{m_3}}{\sqrt{m_1}} \leq c_2/(4\sqrt{m_1}),
$$

then

$$
LHS \lesssim \exp\{-c_2^2/(32\beta_1^2)\}\eta\frac{\varrho^2 m_3}{\beta_1 m_1}. \tag{267}
$$

On the other hand, for $j \in P$ we have

$$
\begin{aligned}
\mathbb{E}val_j &\leq \mathbb{P}(W_j^\rho x_i \in (-W_j^{(0)}x_i - W_j'x_i - 2\sqrt{\eta}\varrho\frac{\sqrt{m_3}}{\sqrt{m_1}}, -W_j^{(0)}x_i - W_j'x_i)) \times \frac{1}{\sqrt{m_1}}\sqrt{\eta}\varrho\frac{\sqrt{m_3}}{\sqrt{m_1}} \\
&\lesssim (\beta_1/\sqrt{m_1})^{-1}\sqrt{\eta}\varrho\frac{\sqrt{m_3}}{\sqrt{m_1}} \times \sqrt{\eta}\varrho\frac{\sqrt{m_3}}{m_1} = \eta\frac{\varrho^2 m_3}{\beta_1 m_1}. \tag{268}
\end{aligned}
$$

Now define the following random variable with respect to the randomness of $W^\rho$:

$$
Val := \sum_{j=1}^{m_1} \mathbb{1}\{\text{sign change in the}j\text{th neuron}\} \times |\text{amount of change}|
$$

then for every $k \in [m_3]$, we have

$$
|x_k'^{(1)} - x_k'^{(2)}| \leq Val,
$$

which implies

$$
\|x'^{(1)} - x'^{(2)}\| \leq \sqrt{m_3}Val.
$$

But Combining Equations (267) and (268):

$$
\begin{aligned}
\mathbb{E}Val &\leq \Big(\exp\{-c_2^2/(32\beta_1^2)\} + \frac{|P|}{m_1}\Big)\eta\frac{\varrho^2 m_3}{\beta_1} \\
&\leq \Big(\exp\{-c_2^2/(32\beta_1^2)\} + \frac{c_2}{\kappa_1\sqrt{m_1}}\Big)\eta\frac{\varrho^2 m_3}{\beta_1},
\end{aligned}
$$

which implies

$$
\mathbb{E}_{W^\rho}\|x'^{(1)} - x'^{(2)}\| \leq \Big(\exp\{-c_2^2/(32\beta_1^2)\} + \frac{c_2}{\kappa_1\sqrt{m_1}}\Big)\eta\frac{\varrho^2 m_3\sqrt{m_3}}{\beta_1}. \tag{269}
$$

Now we can write

$$\left| a^T D_{\prime,\rho}(V^{(0)} + (1-\eta)V' + V^\rho + \sqrt{\eta}V^*)x'^{(2)} - a^T D_{\prime,\rho}(V^{(0)} + (1-\eta)V' + V^\rho + \sqrt{\eta}V^*)x'^{(1)} \right|$$

$$\leq \frac{1}{\sqrt{m_2}} \sum_{j=1}^{m_2} |(V_j^{(0)} + (1-\eta)V_j' + V_j^\rho + \sqrt{\eta}V_j^*)(x'^{(2)} - x'^{(1)})|$$

$$\leq \frac{1}{\sqrt{m_2}} \sum_{j=1}^{m_2} |V_j^{(0)}(x'^{(2)} - x'^{(1)})| + (1-\eta)|V_j'(x'^{(2)} - x'^{(1)})| + |V_j^\rho(x'^{(2)} - x'^{(1)})| + \sqrt{\eta}|V_j^*(x'^{(2)} - x'^{(1)})|$$

$$= \frac{1}{\sqrt{m_2}} \sum_{j=1}^{m_2} |V_j^{(0)}(x'^{(2)} - x'^{(1)})| + |V_j^\rho(x'^{(2)} - x'^{(1)})| + \frac{1}{\sqrt{m_2}} \sum_{j=1}^{m_2} ((1-\eta)\|V_j'\| + \sqrt{\eta}\|V_j^*\|)\|x'^{(2)} - x'^{(1)}\|$$

$$\leq \frac{1}{\sqrt{m_2}} \sum_{j=1}^{m_2} |V_j^{(0)}(x'^{(2)} - x'^{(1)})| + |V_j^\rho(x'^{(2)} - x'^{(1)})| + \left((1-\eta)\|V'\|_F + \sqrt{\eta}\|V^*\|_F\right)\|x'^{(2)} - x'^{(1)}\|.$$

Now by Equation (126) in Lemma 10 (i.e. $\|V^*\|_F \lesssim \sqrt{\zeta_2}$) and the fact that $\|V'\|_F \leq C_2$, and by taking

$$\sqrt{\eta} \leq \frac{C_2}{\sqrt{\zeta_2}},$$

we have

$$\left((1-\eta)\|V'\|_F + \sqrt{\eta}\|V^*\|_F\right) \lesssim C_1, \tag{270}$$

so we can bound the above as

$$LHS \lesssim \frac{1}{\sqrt{m_2}} \sum_{j=1}^{m_2} \left(|V_j^{(0)}(x'^{(2)} - x'^{(1)})| + |V_j^\rho(x'^{(2)} - x'^{(1)})|\right) + C_1\|x'^{(2)} - x'^{(1)}\|.$$

Furthermore, using Lemma 32 and noting the fact that the entries of $V^{(0)}$ are normal with standard deviation $\kappa_2$, we get with high probability over the randomness of $V^{(0)}$:

$$\lesssim \frac{1}{\sqrt{m_2}} \sum_{j=1}^{m_2} |V_j^\rho(x'^{(2)} - x'^{(1)})| + \left(\kappa_2\sqrt{m_2} + \kappa_2\sqrt{m_3(\log(m_3) + \log(\log(m_2)))} + C_1\right)\|x'^{(2)} - x'^{(1)}\|.$$

Now note that $V_j^\rho(x'^{(2)} - x'^{(1)})$ is normal with standard deviation $\frac{\beta_2}{\sqrt{m_2}}\|x'^{(2)} - x'^{(1)}\|$. Hence, taking expectation with respect to $V^\rho$:

$$\mathbb{E}_{V^\rho}\left| a^T D_{\prime,\rho,\eta}(V^{(0)} + (1-\eta)V' + V^\rho + \sqrt{\eta}V^*)x'^{(2)} - a^T D_{\prime,\rho,\eta}(V^{(0)} + (1-\eta)V' + V^\rho + \sqrt{\eta}V^*)x'^{(1)} \right|$$

$$\lesssim \left(\kappa_2\sqrt{m_2} + \kappa_2\sqrt{m_3(\log(m_3) + \log(\log(m_2)))} + C_1\right)\|x'^{(2)} - x'^{(1)}\|. \tag{271}$$

Finally, Combining Equation (264) and (271) and applying it to Equation (260) implies with high probability over the random initialization:

$$\left| \mathbb{E}_{W^\rho,V^\rho} a^T D_{\prime,\rho,\eta}(V^{(0)} + (1-\eta)V' + V^\rho + \sqrt{\eta}V^*)W^s x'^{(2)} \right.$$

$$\left. - a^T D_{\prime,\rho}(V^{(0)} + (1-\eta)V' + V^\rho + \sqrt{\eta}V^*)W^s x'^{(1)} \right|$$

$$\leq A_1 + A_2$$

$$\lesssim \mathbb{E}_{W^\rho}\eta\varrho_2^2\beta_2^{-1}\min\{\|x' - \phi^{(0)}(x_i)\|^2/\|x'\|, \|x' - \phi^{(0)}(x_i)\|\}\left[\exp\{-C_2^{4/3}(\sqrt{m_2}\kappa_2)^{2/3}/(8\beta_2^2)\} + \frac{C_2^{2/3}}{(\sqrt{m_2}\kappa_2)^{2/3}}\right]$$

$$+ \mathbb{E}_{W^\rho}\left(\kappa_2\sqrt{m_2} + \kappa_2\sqrt{m_3(\log(m_3) + \log(\log(m_2)))} + C_1\right)\|x'^{(2)} - x'^{(1)}\|$$

Now notice that under $E^c$, using the assumption $\kappa_1\sqrt{m_3} \gtrsim C_1 + \sqrt{m_3}\beta_1$ and Lemma 33 we have

$$
\begin{aligned}
\|x'\| &\geq \|\phi^{(0)}(x_i)\| - \|x' - \phi^{(0)}(x_i)\| \\
&\geq \kappa_1\sqrt{m_3} - (C_1 + \sqrt{m_3}\beta_1) \\
&\gtrsim \kappa_1\sqrt{m_3},
\end{aligned}
$$

and

$$
\|x' - \phi^{(0)}(x_i)\|^2 \leq (C_1 + \sqrt{m_3}\beta_1)^2, \tag{272}
$$

which implies:

$$
\begin{aligned}
&\mathbb{E}_{W^\rho} \min\{\|x' - \phi^{(0)}(x_i)\|^2/\|x'\|, \|x' - \phi^{(0)}(x_i)\|\} \\
&\leq \mathbb{E}_{W^\rho} \mathbb{1}\{E^c\}\|x' - \phi^{(0)}(x_i)\|^2/\|x'\| + \mathbb{1}\{E\}\|x' - \phi^{(0)}(x_i)\| \\
&\lesssim (C_1 + \sqrt{m_3}\beta_1)^2/(\kappa_1\sqrt{m_3}) + \mathbb{E}_{W^\rho} \mathbb{1}\{E\}\|x' - \phi^{(0)}(x_i)\| \\
&\lesssim (C_1 + \sqrt{m_3}\beta_1)^2/(\kappa_1\sqrt{m_3}) + \left[\sqrt{m_3}m_1\beta_1 + C_1\right] \exp\{-C_1^2/(8m_3\beta_1^2)\}.
\end{aligned}
$$

Substituting this above and further applying the result of Lemma 33 and Equation (269) and the assumption that $m_2 \geq m_3\log(m_2)$:

$$
\begin{aligned}
A_1 + A_2 &\lesssim \eta\varrho_2^2\beta_2^{-1}\left[(C_1 + \sqrt{m_3}\beta_1)^2/(\kappa_1\sqrt{m_3}) + \left[\sqrt{m_3}m_1\beta_1 + C_1\right]\exp\{-C_1^2/(8m_3\beta_1^2)\}\right] \\
&\times \left[\exp\{-C_2^{4/3}(\sqrt{m_2}\kappa_2)^{2/3}/(8\beta_2^2)\} + \frac{C_2^{2/3}}{(\sqrt{m_2}\kappa_2)^{2/3}}\right] \\
&+ \eta\left(\kappa_2\sqrt{m_2} + C_1\right)\left(\exp\{-c_2^2/(32\beta_1^2)\} + \frac{c_2}{\kappa_1\sqrt{m_1}}\right)\frac{\varrho^2 m_3\sqrt{m_3}}{\beta_1},
\end{aligned}
$$

which completes the proof.

### A.20.1 SETTING $\beta_1$ AND $\beta_2$

As we mentioned, to minimize the amount of deviation of the smoothed function compared to the original one, we prefer to choose $\beta_1, \beta_2$ as small as possible. (The benefit of such choice, indeed, can be observed more explicitly in other parts of the proof, e.g. Appendix A.14.) Observing the bound in Equation (259) and noting that we can easily make the exponential terms orders of magnitude smaller than the poly terms, it is easy to find the following optimal setting for the smoothing parameters:

$$
\begin{aligned}
\beta_2 &:= \Theta_p\left((\kappa_1\sqrt{m_3})^{-1}(\sqrt{m_2}\kappa_2)^{-\frac{2}{3}}\right), \\
\beta_1 &:= \Theta_p\left(m_3\sqrt{m_3}/(\kappa_1\sqrt{m_1})\right).
\end{aligned}
$$

Using this setting, we still can make $\Re_8$ arbitrarily small. Here, we remind the reader that $O_p$ only cares about the non-logarithmic dependencies on the overparameterization, i.e. $m_1, m_2, m_3, \kappa_1, \kappa_2$.

### A.21 BASIC TOOLS

In this section, we introduce and prove some lemmas that we use in our analysis as basic tools.

**Lemma 29** *Suppose $V^{(0)} \in \mathbb{R}^{m_2 \times m_3}$ has standard normal entries and $a$ is a random sign vector. Suppose theta $> 1, R < 1$ are given thresholds, such that*

$$m_2 R \gtrsim m_3(\log(1/R) + \log(m_3) + \log(\log(m_2))),$$

$$e^{-\theta^2/8} \lesssim m_3/m_2.$$

*Then, for the following quantities:*

$$N_R^1(x) = \#\left(j \in [m] : |V_j^{(0)}x| \le R\right)$$

$$N_\theta^2(x) = \#\left(j \in [m] : |V_j^{(0)}x| \ge \theta\right),$$

*with high probability we have*

$$\sup_{\|x'\|=1} N_R^1(x') \lesssim m_2 R,$$

$$\sup_{\|x'\|=1} N_\theta^2(x') \lesssim m_3(\log(m_3) + \log(\log(m_2))).$$

**Proof of Lemma 29**

Suppose $B_1(\epsilon)$ is a cover for the Euclidean ball in $\mathbb{R}^{m_3}$ with precision $\epsilon$. We know

$$|B_1(\epsilon)| \lesssim (1/\epsilon)^{m_3}.$$

Now for a fixed $\|x\| = 1$, we have

$$\mathbb{P}(W_j^{(0)}x \le 2R) \lesssim R.$$

Therefore, using Bernstein, with high probability we have

$$\#\left(j \in [m_2] : |V_j^{(0)}x| \le 2R\right) \lesssim m_2 R + \sqrt{m_2 R} + 1.$$

Hence, using union bound, we have with high probability

$$\sup_{x \in B_1(\epsilon)} \#\left(j \in [m] : |V_j^{(0)}x| \le 2R\right) \lesssim m_2 R + \sqrt{\log |B_1(\epsilon)|}\sqrt{m_2 R} + \log |B_1(\epsilon)|$$

$$= m_2 R + \sqrt{m_2 R m_3 \log(1/\epsilon)} + m_3 \log(1/\epsilon).$$

By picking

$$\epsilon \lesssim R/(\sqrt{m_3 \log(m_2 m_3)}),$$

The assumption implies $m_2 R \ge m_3 \log(1/\epsilon)$, which implies

$$LHS \lesssim m_2 R.$$

On the other hand, note that with high probability we have

$$\sup_{j \in [m_2], k \in [m_3]} |V_{j,k}^{(0)}| \le \sqrt{\log(m_2 m_3)}. \tag{273}$$

Now for $\|x'\| = 1$ which is not in the cover, if $x$ is the closest point to it in the cover, i.e. $x \in B_1(\epsilon)$ and $\|x - x'\| \le \epsilon$, then for every $j \in [m_2]$ we have

$$||V_j^{(0)}x| - |V_j^{(0)}x'|| \le \|V_j^{(0)}\|\|x - x'\| \le \sqrt{m_3 \log(m_2 m_3)}\epsilon \le R,$$

by picking a small enough constant. Therefore, for a $j$ that $|V_j^{(0)}x| \ge 2R$, then

$$|V_j^{(0)}x'| \ge 2R - R = R.$$

Therefore, we get that with high probability, for every $\|x'\| = 1$:

$$\sup_{\|x'\|=1} \# \left( j \in [m_2] : |V_j^{(0)} x'| \leq R \right) \lesssim m_2 R.$$

For the second part, note that for $\|x\| = 1$, by the tail bound for normal vars:

$$\mathbb{P}(W_j^{(0)} x \geq \theta/2) \lesssim e^{-\theta^2/8}.$$

Hence, again using Bernstein, we have with high probability

$$\sup_{x \in B_1(\epsilon)} \# \left( j \in [m_2] : |V_j^{(0)} x| \geq \theta/2 \right) \lesssim m_2 e^{-\theta^2/8} + \sqrt{\log |B_1(\epsilon)|} \sqrt{m_2 e^{-\theta^2/8}} + \log |B_1(\epsilon)|$$

$$\lesssim m_2 e^{-\theta^2/8} + \sqrt{m_3 \log(1/\epsilon)} \sqrt{m_2 e^{-\theta^2/8}} + m_3 \log(1/\epsilon).$$

By picking

$$\epsilon \lesssim 1/(\sqrt{m_3 \log(m_2 m_3)}),$$

and using the assumption $m_2 e^{-\theta^2/8} \lesssim m_3$, all terms are dominated by the third term so we can bound the above as

$$\sup_{x \in B_1(\epsilon)} \# \left( j \in [m_2] : |V_j^{(0)} x| \geq \theta/2 \right) \lesssim m_3 (\log(m_3) + \log(\log(m_2))).$$

Now for $\|x'\| = 1$ not in the cover, for the new $\epsilon$ we can write

$$||V_j^{(0)} x| - |V_j^{(0)} x'|| \leq \|V_j^{(0)}\| \|x - x'\| \leq \sqrt{m_3 \log(m_2 m_3)} \epsilon \leq 1/2 \leq \theta/2.$$

Hence, with high prob.

$$\sup_{\|x'\|=1} \# \left( j \in [m_2] : |V_j^{(0)} x| \geq \theta \right) \lesssim m_3 (\log(m_3) + \log(\log(m_2))).$$

**Lemma 30** *For $x \in \mathbb{R}^d$ and $W^{(0)} \in \mathbb{R}^{m \times d}$ which has standard normal entries (and $a$ is a random sign vector), we have with high probability:*

$$\sup_{\|x\|=1} f(x) := \frac{1}{\sqrt{m}} a^T \sigma(W^{(0)} x) \leq \sqrt{d}.$$

**Proof of Lemma 30**

For the first part, we first compute an upper bound on

$$\mathbb{E} \sup_{\|x\|=1} \frac{1}{\sqrt{m}} a^T \sigma(W^{(0)} x).$$

To do so, we use Dudley's chaining. Note that the for $x_1, x_2 \in \mathbb{R}^d$, the variable $\sigma(W_j^{(0)} x_1) - \sigma(W_j^{(0)} x_2)$ is subGaussian with parameter $\|x_1 - x_2\|$, so the variable $f(x_1) - f(x_2)$ is also sub-Gaussian with parameter $\|x_1 - x_2\|$. Hence, by Dudley's integral:

$$\mathbb{E} \sup_{\|x\|=1} \frac{1}{\sqrt{m}} a^T \sigma(W^{(0)} x) \leq \int_0^1 \sqrt{\log(\mathcal{N}(\mathcal{B}_1^d, \epsilon))} \lesssim \sqrt{d}.$$

Now for a fixed $x$, note that

$$\frac{1}{\sqrt{m}} a^T \sigma(W_1 x) - \frac{1}{\sqrt{m}} a^T \sigma(W_2 x) \leq \frac{1}{\sqrt{m}} \sum_{j=1}^m \|W_{1j} - W_{2j}\| \leq \|W_1 - W_2\|_F.$$

Hence, the function $f(x)$ is 1-lipchitz with respect to $W$ and $l2$ norm, so is the function $\sup f(x)$. Hence, by Gaussian concentration, $\sup f(x)$ is 1-subGaussian around its mean, so we finally get with high probability

$$\sup f(x) \lesssim \sqrt{d} + 1 \lesssim \sqrt{d}.$$

**Lemma 31** *For*

$$R^* := \frac{C_2^{2/3}}{(\sqrt{m_2}\kappa_2)^{2/3}},$$

*we have with high probability over the randomness of $V^{(0)}$:*

$$\sup_{x', V':\ \|V'\| \le C_2} \#\Big(j \in [m_2] :\ |(V_j^{(0)} + V_j')x'| \le R^*\kappa_2\|x'\|\Big) \lesssim R^* m_2.$$

**Proof of Lemma 31**

Note that obviously the condition of Lemma 29 is satisfied with this choice of $R = R^*$. Therefore, with high probability we have for an arbitrary $x'$:

$$\#\Big(|V_j^{(0)}x'| \le 2R^*\kappa_2\|x'\|\Big) \le m_2 R^*.$$

On the other hand, note that for $j \in [m_2]$ such that $|V_j'x'| \ge R\kappa_2\|x'\|$, we have

$$\|V_j'\|\|x'\| \ge |V_j'x'| \ge R\kappa_2\|x'\|,$$

which implies

$$\|V_j'\| \ge R\kappa_2.$$

Therefore, there are at most $\frac{C_2^2}{R^2\kappa_2^2}$. Therefore, setting aside $m_2 R + \frac{C_2^2}{R^2\kappa_2^2}$ of $j$'s, for the rest we have

$$|(V_j^{(0)} + V_j')x'| \ge |V_j^{(0)}x'| - |V_j'x'| \ge 2R\kappa_2\|x'\| - R\kappa_2\|x'\| = R\kappa_2\|x'\|.$$

Setting $R^*$ as defined above balances the terms $m_2 R$ and $\frac{C_2^2}{R^2\kappa_2^2}$, which completes the proof.

**Lemma 32** *If $V^{(0)} \in \mathbb{R}^{m_2 \times m_3}$ is a matrix with standard normal entries, then with high probability*

$$\sup_{\|x'\|=1} \frac{1}{\sqrt{m_2}} \sum_{j=1}^{m_2} |V_j^{(0)}x'| \lesssim \sqrt{m_2} + \sqrt{m_3(\log(m_3) + \log(\log(m_2)))}.$$

**Proof of Lemma 32**

Let $B_1(\epsilon)$ be a cover for the unit Euclidean ball with precision $\epsilon$, for which we have $|B_1(\epsilon)| \lesssim (\frac{1}{\epsilon})^{m_3}$. Now for a fixed $x \in B_1(\epsilon)$, note that because $V_j^{(0)}x$ is a standard normal variable, the random variable $|V_j^{(0)}x'| - \mathbb{E}|V_j^{(0)}x'|$ is $O(1)$-subGaussian, which means $\frac{1}{\sqrt{m_2}}\sum_{j=1}^{m_2}(|V_j^{(0)}x'| - \mathbb{E}|V_j^{(0)}x'|)$ is also $O(1)-$subGaussian. Now from the tail of maximum of subGaussian variables:

$$\sup_{x \in B_1(\epsilon)} \frac{1}{\sqrt{m_2}} \sum_{j=1}^{m_2}(|V_j^{(0)}x| - \mathbb{E}|V_j^{(0)}x|) \lesssim \sqrt{\log(|B_1(\epsilon)|)} = \sqrt{m_3 \log(1/\epsilon)}.$$

On the other hand, note that $\mathbb{E}|V_j^{(0)}x'|) = O(1)$, which implies w.h.p:

$$\sup_{x \in B_1(\epsilon)} \frac{1}{\sqrt{m_2}} \sum_{j=1}^{m_2} |V_j^{(0)}x| \lesssim \sqrt{m_2} + \sqrt{m_3 \log(1/\epsilon)}.$$

Moreover, note that again by the tail of subGaussian variables, we have w.h.p:

$$\max_{j \in [m_2], k \in [m_3]} |V_{j,k}^{(0)}| \lesssim \sqrt{\log(m_2 m_3)},$$

which implies with high prob for every $j \in [m_2]$:

$$\|V_j^{(0)}\| \lesssim \sqrt{m_3 \log(m_2 m_3)}.$$

Now by picking

$$\epsilon := \left(\sqrt{m_3 \log(m_2 m_3)}\right)^{-1},$$

we get with high probability

$$\sup_{x \in B_1(\epsilon)} \frac{1}{\sqrt{m_2}} \sum_{j=1}^{m_2} |V_j^{(0)} x| \lesssim \sqrt{m_2} + \sqrt{m_3(\log(m_3) + \log(\log(m_2)))}. \tag{274}$$

On the other hand, for an arbitrary $x'$ with $\|x'\| = 1$, if $x \in B_1(\epsilon)$ is the representative of $x'$, we have by definition $\|x' - x\| \le \epsilon$, which combined with (A.21) implies

$$\left| |V_j^{(0)} x'| - |V_j^{(0)} x| \right| \le |V_j^{(0)}(x' - x)| \le \|V_j^{(0)}\| \|x' - x\|,$$

$$\lesssim \sqrt{m_3 \log(m_2 m_3)} \left(\sqrt{m_3 \log(m_2 m_3)}\right)^{-1} \le 1.$$

Therefore

$$\left| \frac{1}{\sqrt{m_2}} \sum_{j=1}^{m_2} |V_j^{(0)} x'| - \frac{1}{\sqrt{m_2}} \sum_{j=1}^{m_2} |V_j^{(0)} x| \right| \le \sqrt{m_2}. \tag{275}$$

Combining Equations (274) and (275), we conclude the result.

### A.21.1 Defining the rare events $E_j$

**Lemma 33** *For $x'^{(2)}$ defined in Equation (258) we have*

$$\mathbb{E}_{W^\rho}\|x'^{(2)} - \phi^{(0)}(x_i)\|, \mathbb{E}_{W^\rho}\|\phi^{(2)}(x_i)\| \leq C_1 + \sqrt{m_3}\beta_1,$$
$$\mathbb{E}_{W^\rho}\|\phi^{(2)}(x_i)\|^2 \lesssim C_1^2 + m_3\beta_1^2.$$

*Moreover, for the events*

$$E_j = \{|W_j^\rho x_i| \geq C_1/\sqrt{m_3 m_1}\}, \ E = \cup_j E_j,$$

*we have under $E^c$:*

$$\|x'^{(2)} - \phi^{(0)}(x_i)\|, \|\phi^{(2)}(x_i)\| \lesssim C_1.$$

*Furthermore, $E$ happens rarely:*

$$\mathbb{P}(E) \lesssim m_1 \exp\{-C_1^2/(8m_3\beta_1^2)\},$$
$$\mathbb{E}_{W^\rho}\mathbb{1}\{E\}\|\phi^{(2)}(x_i)\| \leq \left[\sqrt{m_3}m_1\beta_1 + C_1\right]\exp\{-C_1^2/(8m_3\beta_1^2)\}.$$
$$\mathbb{E}_{W^\rho}\mathbb{1}\{E\}\|x'^{(2)} - \phi^{(0)}(x_i)\| \leq \left[\sqrt{m_3}m_1\beta_1 + C_1\right]\exp\{-C_1^2/(8m_3\beta_1^2)\}.$$

*Finally, we have the following almost surely bound:*

$$\|\phi^{(2)}(x_i)\| \leq C_1 + \sqrt{m_3}\sum_{j=1}^{m_1}\frac{1}{\sqrt{m_1}}|W_j^\rho x_i|.$$

**Proof of Lemma 33**

We start by writing

$$|x_k'^{(2)} - \frac{1}{\sqrt{m_1}}W_k^s\sigma(W^{(0)} + (1-\eta)W' + \sqrt{\eta}W^*)x_i| \leq \sum_{j=1}^{m_1}\frac{1}{\sqrt{m_1}}|W_j^\rho x_i|. \tag{276}$$

Now notice that by Lemma 1, we know for every $j \notin \tilde{P}$:

$$|W_j^{(0)}x_i| \geq c_2/\sqrt{m_2}, \tag{277}$$
$$|(1-\eta)W_j'x_i| \leq c_2/(2\sqrt{m_1}). \tag{278}$$

In addition, by Equations in (104) from Lemma 5, for every $j \in [m_1]$:

$$\|W_j^*\| \leq \varrho_1\sqrt{\frac{m_3}{m_1}},$$

so by picking

$$\eta \leq c_2/(4\varrho\sqrt{m_3})$$

we obtain

$$\eta|W_j^*x_i| \leq \frac{c_2}{4\sqrt{m_1}}. \tag{279}$$

Combining this with Equations in (278), we see that the signs of $(W_j^{(0)} + (1-\eta)W_j' + \sqrt{\eta}W_j^*)x_i$ and $W_j^{(0)}x_i$ are the same for $j \notin \tilde{P}$.

Moreover, the matrix $(1-\eta)W' + \sqrt{\eta}W^*$ satisfies

$$\|(1-\eta)W' + \sqrt{\eta}W^*\| \leq (1-\eta)C_1 + \sqrt{\eta}\sqrt{2\zeta_2} \leq C_1,$$

by picking $\sqrt{\eta} \leq C_1/\sqrt{\zeta_2}$. Hence, the conditions of Lemma 6 are satisfied and we get:

$$\|\frac{1}{\sqrt{m_1}}W^s\sigma(W^{(0)} + (1-\eta)W' + \sqrt{\eta}W^*)x_i - \phi^{(0)}(x_i)\| \leq C_1. \tag{280}$$

Combining Equations (276) and (280):

$$\|x'^{(2)} - \phi^{(0)}(x_i)\| \leq C_1 + \sqrt{m_3} \sum_{j=1}^{m_1} \frac{1}{\sqrt{m_1}} |W_j^\rho x_i|. \tag{281}$$

In exactly similar fashion, one can derive

$$\|\phi^{(2)}(x_i)\| \leq C_1 + \sqrt{m_3} \sum_{j=1}^{m_1} \frac{1}{\sqrt{m_1}} |W_j^\rho x_i|. \tag{282}$$

Now first of all, note

$$\mathbb{E}_{W^\rho} \sum_{j=1}^{m_1} \frac{1}{\sqrt{m_1}} |W_j^\rho x_i| \leq \beta_1,$$

which proves the first part of the claims. For the second part, note that by the Gaussian tail bound

$$\mathbb{P}(|W_j^\rho x_i| \geq C_1/\sqrt{m_3 m_1}) \lesssim \exp\{-C_1^2/(8 m_3 \beta_1^2)\}.$$

Therefore,

$$\mathbb{P}(E) \leq \sum_j \mathbb{P}(E_j) \leq m_1 \exp\{-C_1^2/(8 m_3 \beta_1^2)\}.$$

Moreover

$$\mathbb{E}_{W^\rho} \mathbb{1}\{E\} \sum_{j=1}^{m_1} \frac{1}{\sqrt{m_1}} |W_j^\rho x_i| \leq \mathbb{E}_{W^\rho} \frac{1}{\sqrt{m_1}} \sum_{j_2} \sum_{j \neq j_2} \mathbb{1}\{E_{j_2}\} |W_j^\rho x_i| + \frac{1}{\sqrt{m_1}} \sum_j \mathbb{E}_{W^\rho} \mathbb{1}\{E_j\} |W_j^\rho x_i|,$$

$$= \left[ \frac{1}{\sqrt{m_1}} \sum_{j_2} \sum_{j \neq j_2} \mathbb{E}_{W^\rho} |W_j^\rho x_i| + \frac{1}{\sqrt{m_1}} \sum_j E_{W^\rho}[|W_j^\rho x_i| \mid E_j] \right] \mathbb{P}(E_j)$$

$$\lesssim \left[ m_1 \beta_1 + C_1/\sqrt{m_3} \right] \exp\{-C_1^2/(8 m_3 \beta_1^2)\}.$$

Plugging this into (281) finishes the proof. Also, under $E^c$ by Equation (281) we have

$$\|x'^{(2)} - \phi^{(0)}(x_i)\|, \|\phi^{(2)}(x_i)\| \lesssim C_1.$$

Finally, exploiting Equation (282):

$$\mathbb{E}_{W^\rho} \|\phi^{(2)}(x_i)\|^2$$
$$\lesssim C_1^2 + m_3 \frac{1}{m_1} \mathbb{E}_{W^\rho}(\sum_j |W_j^\rho x_i|)^2 \leq C_1^2 + m_3 \mathbb{E}_{W^\rho} \sum_j |W_j^\rho x_i|^2$$
$$\leq C_1^2 + m_3 \beta_1^2.$$

### A.21.2 Bounding the value of $f'$

The following Lemma provides a reasonable bound on the value of the smoothed function.

**Lemma 34** *We have the following general bound on the values of the smoothed function: With high probability over the initialization, for $\|W'\| \leq C_1, \|V'\| \leq C_2$ and $\forall i \in [n]$ (having small enough choices of $\beta_1, \beta_2$ described in Appendix A.20):*

$$|f'_{W',V'}(x_i)| \lesssim (\kappa_2\sqrt{m_3} + \beta_2)\left(\sqrt{m_3}\kappa_1 + C_1 + \sqrt{m_3}\beta_1\right) + C_2(C_1 + \sqrt{m_3}\beta_1),$$

*which is $O(C_1C_2)$ for large enough overparameterization as described in AppendixA.3. Moreover, we have the following almost surely bound (with respect to the randomness of $W^\rho$ and $V^\rho$):*

$$|f_{W'+W^\rho,V'+V^\rho}(x_i)|$$
$$\lesssim (\kappa_2\sqrt{m_3} + \|V^\rho\|_F)\left(\sqrt{m_3}\kappa_1 + C_1 + \sqrt{m_3}(\frac{1}{\sqrt{m_1}}\sum_j |W_j^\rho x_i|)\right) + C_2(C_1 + \sqrt{m_3}(\frac{1}{\sqrt{m_1}}\sum_j |W_j^\rho x_i|)).$$

*Notably, with slightly higher overparameterization, the high probability bound in (34) holds even if we take supremum over $x$.*

**Proof of Lemma 34**

Using Lemmas 30 and 33 and using the fact that $\phi^{(0)}(x_i)$ is orthogonal to the rows of $V'$ (recall $x_i' = \phi^{(0)}(x_i) + \phi^{(2)}(x_i)$):

$$|f_{W'+W^\rho,V'+V^\rho}(x_i)|$$
$$\leq \frac{1}{\sqrt{m_2}}a^T\sigma(V^{(0)}x_i') + \frac{1}{\sqrt{m_2}}\sum_j |(V_j^\rho + V_j')x_i'|$$
$$\leq (\kappa_2\sqrt{m_3})\|x_i'\| + \frac{1}{\sqrt{m_2}}\sum_j |V_j^\rho x_i'| + C_2\|\phi^{(2)}(x_i)\|$$
$$\leq (\kappa_2\sqrt{m_3} + \|V^\rho\|_F)\|x_i'\| + C_2\|\phi^{(2)}(x_i)\|$$
$$\lesssim (\kappa_2\sqrt{m_3} + \|V^\rho\|_F)\left(\sqrt{m_3}\kappa_1 + C_1 + \sqrt{m_3}(\frac{1}{\sqrt{m_1}}\sum_j |W_j^\rho x_i|)\right) + C_2(C_1 + \sqrt{m_3}(\frac{1}{\sqrt{m_1}}\sum_j |W_j^\rho x_i|)).$$
$$\tag{283}$$

Note that above, if we apply the stronger worst-case norm bound of the first layer's output presented in Lemma 46, we would get $\sup_{x,\|x\|=1} |f_{W'+W^\rho,V'+V^\rho}(x)|$ is bounded by the RHS, which in turn proves a stronger uniform bound on $f'$.

Similarly, this time by taking expectation with respect to $W^\rho$ and $V^\rho$:

$$|f'_{W',V'}(x_i)| = |\mathbb{E}_{W^\rho,V^\rho}f_{W',V'}(x_i)|$$
$$\leq \mathbb{E}_{W^\rho,V^\rho}|f_{W',V'}(x_i)|$$
$$= \mathbb{E}_{W^\rho}(\kappa_2\sqrt{m_3} + \beta_2)\|x_i'\| + C_2\|x_i' - \phi^{(0)}(x_i)\|$$
$$\lesssim (\kappa_2\sqrt{m_3} + \beta_2)\left(\sqrt{m_3}\kappa_1 + C_1 + \sqrt{m_3}\beta_1\right) + C_2(C_1 + \sqrt{m_3}\beta_1).$$

**Corollary 8.1** *If we set $C_1 = C_2 = 0$ above, we get*

$$|f'_{0,0}(x_i)| \leq (\kappa_2\sqrt{m_3} + \beta_2)\left(\sqrt{m_3}\kappa_1 + \sqrt{m_3}\beta_1\right),$$

*the point being these terms go to zero by an order of $O((\sqrt{m_2}\kappa_2)^{-\frac{2}{3}})$. Therefore, taking $(\sqrt{m_2}\kappa_2)^{-\frac{2}{3}} << B$, we make sure that $|f'_{0,0}| < B$, so by the 1 smoothness of $\ell$ and $B$ boundedness of the labels we get $\ell(f'_{0,0}(x_i), y_i) < 4B^2$.*

### A.21.3 Bounding the difference between Original and Smoothed Functions

The following Lemma bounds the difference between the smoothed function and original function of the network.

**Lemma 35** *Bound on the smoothing change under the assumption $m_2 \geq m_3 \log(m_2)$: with high probability over the initialization, for any $(W', V')$ with $\|W'\| \leq C_1, \|V'\| \leq C_2$:*

$$|f_{W',V'}(x_i) - f'_{W',V'}(x_i)|$$
$$\leq \beta_2(\kappa_1\sqrt{m_3} + C_1 + \sqrt{m_3}\beta_1) + \left(C_2 + \kappa_2\sqrt{m_2}\right)\sqrt{m_3}\beta_1.$$

**Proof of Lemma 35**

We write

$$|f_{W',V'}(x_i) - f'_{W',V'}(x_i)| = |f_{W',V'}(x_i) - \mathbb{E}_{W^\rho,V^\rho}f_{W'+W^\rho,V'+V^\rho}(x_i)|$$
$$= \left|\mathbb{E}_{W^\rho,V^\rho}\left(f_{W',V'}(x_i) - f_{W'+W^\rho,V'+V^\rho}(x_i)\right)\right|$$
$$\leq \mathbb{E}_{W^\rho,V^\rho}\left|f_{W',V'}(x_i) - f_{W'+W^\rho,V'+V^\rho}(x_i)\right|.$$

In the following, $\sigma$ means we apply Relu activation to the vector in front of it (entrywise):

$$LHS \leq \mathbb{E}_{W^\rho,V^\rho}\left|\frac{1}{\sqrt{m_2}}a^T\sigma(V^{(0)} + V' + V^\rho)\frac{1}{\sqrt{m_1}}W^s\sigma(W^{(0)} + W' + W^\rho)x_i\right.$$
$$\left. - \frac{1}{\sqrt{m_2}}a^T\sigma(V^{(0)} + V')\frac{1}{\sqrt{m_1}}W^s\sigma(W^{(0)} + W' + W^\rho)x_i\right|$$
$$+ \mathbb{E}_{W^\rho,V^\rho}\left|\frac{1}{\sqrt{m_2}}a^T\sigma(V^{(0)} + V')\frac{1}{\sqrt{m_1}}W^s\sigma(W^{(0)} + W' + W^\rho)x_i\right.$$
$$\left. - \frac{1}{\sqrt{m_2}}a^T\sigma(V^{(0)} + V')\frac{1}{\sqrt{m_1}}W^s\sigma(W^{(0)} + W')x_i\right|.$$

Now for the first term above, using the previous notation of $x_i'$ representing the output of the first layer and using Lemma 33:

$$\mathbb{E}_{W^\rho,V^\rho}\left|\frac{1}{\sqrt{m_2}}a^T\sigma(V^{(0)} + V' + V^\rho)\frac{1}{\sqrt{m_1}}W^s\sigma(W^{(0)} + W' + W^\rho)x_i\right.$$
$$\left. - \frac{1}{\sqrt{m_2}}a^T\sigma(V^{(0)} + V')\frac{1}{\sqrt{m_1}}W^s\sigma(W^{(0)} + W' + W^\rho)x_i\right|$$
$$\leq \mathbb{E}_{W^\rho,V^\rho}\frac{1}{\sqrt{m_2}}\sum_j|V_j^\rho x_i'|$$
$$\lesssim \beta_2\mathbb{E}_{W^\rho}\|x_i'\|$$
$$\leq \beta_2\mathbb{E}_{W^\rho,V^\rho}(\|\phi^{(0)}(x_i)\| + \|\phi^{(2)}(x_i)\|)$$
$$\lesssim \beta_2(\kappa_1\sqrt{m_3} + C_1 + \sqrt{m_3}\beta_1). \tag{284}$$

For the second term, by starting off with a simple triangle inequality:

$$\mathbb{E}_{W^\rho,V^\rho}\left|\frac{1}{\sqrt{m_2}}a^T\sigma(V^{(0)} + V')x_i' - \frac{1}{\sqrt{m_2}}a^T\sigma(V^{(0)} + V')(\phi^{(0)}(x_i) + \phi^{(2)\prime}(x_i))\right|$$
$$\leq \mathbb{E}_{W^\rho,V^\rho}\frac{1}{\sqrt{m_2}}\sum_{j=1}^{m_2}\left|(V_j^{(0)} + V_j')(x_i' - \phi^{(0)}(x_i) - \phi^{(2)\prime}(x_i))\right|$$
$$\leq \mathbb{E}_{W^\rho,V^\rho}\frac{1}{\sqrt{m_2}}\sum_{j=1}^{m_2}\left|V_j^{(0)}(x_i' - \phi^{(0)}(x_i) - \phi^{(2)\prime}(x_i))\right| + \left|V_j'(x_i' - \phi^{(0)}(x_i) - \phi^{(2)\prime}(x_i))\right|$$
$$\leq C_2E_{W^\rho}\|x_i' - \phi^{(0)}(x_i) - \phi^{(2)\prime}(x_i)\| + \mathbb{E}_{W^\rho}\frac{1}{\sqrt{m_2}}\sum_{j=1}^{m_2}\left|V_j^{(0)}(x_i' - \phi^{(0)}(x_i) - \phi^{(2)\prime}(x_i))\right|.$$

Now using Lemma 32:

$$
\begin{aligned}
&\lesssim \left(C_2 + \kappa_2\sqrt{m_2}\right)\mathbb{E}_{W^\rho}\|x_i' - \phi^{(0)}(x_i) - \phi^{(2)\prime}(x_i)\| \\
&\lesssim \left(C_2 + \kappa_2\sqrt{m_2}\right)\sqrt{m_3}E_{W^\rho}\sum_j \frac{1}{\sqrt{m_1}}|W_j^\rho x_i| \\
&\lesssim \left(C_2 + \kappa_2\sqrt{m_2}\right)\sqrt{m_3}\beta_1.
\end{aligned}
\tag{285}
$$

Combining Equations (284) and (285) we conclude the proof.

# B APPENDIX

## CONTENTS

## B.1 SMOOTHNESS COEFFICIENTS

Recall that for a function $f \in \mathcal{C}^3$ on $\mathbb{R}^d$, we say it is $\mu_1$ Lipschitz, $\mu_2$ gradient Lipschitz, and $\mu_3$ hessian Lipschitz at point $x$ if for every unit direction $v$, $|\frac{d}{d\lambda}f(x+\lambda v)| \leq \mu_1$, $|\frac{d^2}{d\lambda^2}f(x+\lambda v)| \leq \mu_2$, and $|\frac{d^2}{d\lambda^2}f(x+\lambda v)| \leq \mu_3$.

The aim of this section is to bound the Lipschitz coefficients of the loss $\ell(,y)$ and objective $L(W', V')$ in a bounded domain $\|W'\| \leq C_1, \|V'\| \leq C_2$. The following is our main Theorem in this regard:

**Theorem 9** *For given values $C_1, C_2 > 0$, in the domain $\|W'\| \leq C_1, \|V'\| \leq C_2$, for any label $|y| \leq B$, the loss function $\ell(.,y)$ is $O((C_1C_2 + B^2))$-Lipschitz (having enough overparameterization) and 1 gradient-Lipschitz $x = f'_{W',V'}$. Moreover, the loss function $L(W', V')$ is $(O(C_1C_2) + B)\Psi_1 + 2(C_1 + C_2)$ Lipschitz, $\Psi_1^2 + (O(C_1C_2) + B)\Psi_2 + 4$ gradient Lipschitz, and $3\Psi_2\Psi_1 + (O(C_1C_2) + B)\Psi_3$ hessian Lipschitz, where $\Psi_1, \Psi_2, \Psi_3$ are defined in Lemma 36.*

**Proof of Lemma 9**

As in the proof of Lemma 36, let $(\tilde{W}, \tilde{V})$ be a unit direction, i.e. $\|\tilde{W}\|^2 + \|\tilde{V}\|^2 = 1$. Then, using Lemma 34, we know that for every $i \in [n]$: $|f'_{W',V'}(x_i)| = O(C_1C_2)$, so by 1-smoothness of the loss and $B$-boundedness of the labels, we get that $\ell(.,y)$ is $(O(C_1C_2) + B)$ lipshcitz at point $f'_{W',V'}$. The gradient smoothness parameter of the square loss $\ell$ is bounded by 1 and its third derivative is zero. Now using these coefficients, we can easily compute the coefficients for $L$ as well by simple differentiation:

$$|\frac{d}{d\lambda}\ell(f'_{W'+\lambda\tilde{W},V'+\lambda\tilde{V}}(x_i), y_i)| = |\dot{\ell}(f', y_i)\frac{d}{d\lambda}f'| \leq (O(C_1C_2) + B)\Psi_1.$$

$$|\frac{d}{d\lambda^2}\ell(f'_{W'+\lambda\tilde{W},V'+\lambda\tilde{V}}(x_i), y_i)| = |\ddot{\ell}(f', y_i)(\frac{d}{d\lambda}f')^2 + \dot{\ell}(f', y_i)\frac{d^2}{d\lambda^2}f'| \leq \Psi_1^2 + (O(C_1C_2) + B)\Psi_2.$$

$$|\frac{d}{d\lambda^3}\ell(f'_{W'+\lambda\tilde{W},V'+\lambda\tilde{V}}(x_i), y_i)| = |\dddot{\ell}(f', y_i)(\frac{d}{d\lambda}f')^3 + 3\ddot{\ell}(f', y_i)\frac{d^2}{d\lambda^2}f'\frac{d}{d\lambda}f' + \dot{\ell}(f', y_i)\frac{d^3}{d\lambda^3}f'|$$
$$\leq \Psi_1^3 + 3\Psi_2\Psi_1 + (O(C_1C_2) + B)\Psi_3.$$

Moreover, note that

$$\frac{d}{d\lambda}\|W' + \lambda\tilde{W}\|^2 = 2\langle W' + \lambda\tilde{W}, \tilde{W}\rangle\Big|_{\lambda=0} = 2\langle W', \tilde{W}\rangle \leq 2\|W'\| = 2C_1,$$

$$\frac{d^2}{d\lambda^2}\|W' + \lambda\tilde{W}\|^2 = \langle\tilde{W}, \tilde{W}\rangle = 2,$$

$$\frac{d^3}{d\lambda^3}\|W' + \lambda\tilde{W}\|^2 = 0,$$

and similarly for $\|V' + \lambda \tilde{V}\|^2$. Combining these results finishes the proof.

Above, we used parameters $\Psi_1, \Psi_2, \Psi_3$, the Lipschitz coefficients of $f'$ in domain $\|W'\| \leq C_1, \|V'\| \leq C_2$, which we bound in Lemma 36 below.

### B.1.1 Computing the Lipschitz Coefficients of $f'_{W',V'}$

In this section, we bound the Lipschitz coefficients of $f'_{W',V'}$ in the domain $\|W'\| \leq C_1, \|V'\| \leq C_2$ by $poly(m_1, m_2, m_3, \beta_1, \beta_2)$ functions.

**Lemma 36** *For every point $(W', V')$ in the domain $\|W'\| \leq C_1, \|V'\| \leq C_2$, we have the following bounds on the Lipschitz coefficients of $f'_{W',V'}$ ($(\tilde{W}, \tilde{V})$ is a unit direction with $\|\tilde{W}\|^2 + \|\tilde{V}\|^2 = 1$):*

$$
\left| \frac{d}{d\lambda} f'_{W'+\lambda\tilde{W}, V'+\lambda\tilde{V}}(x_i)\Big|_{\lambda=0} \right|
$$
$$
\lesssim \frac{m_2}{\beta_2^2}\left( \frac{\beta_2}{\sqrt{m_2}}\sqrt{m_3}(\kappa_1 + C_1 + \beta_1)\left(\kappa_2\sqrt{m_3} + C_2\right) + \frac{\beta_2^2}{\sqrt{m_2}}\sqrt{m_3}\left(\kappa_1 + C_1 + \beta_1\right)\right)
$$
$$
+\frac{m_1}{\beta_1^2}\left(\sqrt{m_3}\left(\frac{\beta_1}{\sqrt{m_1}}\left(\kappa_1 + C_1\right) + \frac{\beta_1^2}{\sqrt{m_1}}\right)\left(\kappa_2\sqrt{m_3} + C_2\right) + \beta_2\sqrt{m_3}\left(\frac{\beta_1}{\sqrt{m_1}}\left(\kappa_1 + C_1\right) + \frac{\beta_1^2}{\sqrt{m_1}}\right)\right) := \Psi_1,
$$
$$
\frac{d^2}{d\lambda^2} f'_{W'+\lambda\tilde{W}, V'+\lambda\tilde{V}}(x_i)\Big|_{\lambda=0}
$$
$$
\lesssim \left(\frac{m_1}{\beta_1^2}\|\tilde{W}\|^2 + \frac{m_2}{\beta_2^2}\|\tilde{V}\|^2\right)\sqrt{m_3\left((\kappa_2\sqrt{m_3} + C_2)^2 + \beta_2^2\right)\left[(\kappa_1 + C_1)^2 + \beta_1^2\right]} := \Psi_2
$$
$$
\left| \frac{d^3}{d\lambda^3} f'_{W'+\lambda\tilde{W}, V'+\lambda\tilde{V}}(x_i)\Big|_{\lambda=0} \right|
$$
$$
\lesssim \left(\frac{m_1}{\beta_1^2}\|\tilde{W}\|^2 + \frac{m_2}{\beta_2^2}\|\tilde{V}\|^2\right)^{3/2}\sqrt{m_3\left((\kappa_2\sqrt{m_3} + C_2)^2 + \beta_2^2\right)\left[(\kappa_1 + C_1)^2 + \beta_1^2\right]} := \Psi_3
$$

$$(286)$$

**Proof of Lemma 36**

Let

$$
\rho(W^\rho, V^\rho) := \frac{1}{(\sqrt{2\pi})^{m_2 m_3 + m_1 d}(\beta_1/\sqrt{m_1})^{m_1 d}(\beta_2/\sqrt{m_2})^{m_2 m_3}} \exp\left\{-\frac{\|W^\rho\|^2}{2\beta_1^2/m_1} - \frac{\|V^\rho\|^2}{2\beta_2^2/m_2}\right\},
$$

be the density function of the law of $W^\rho$ and $V^\rho$ which is a joint Gaussian. Then to compute the derivative and second derivative of the function in the unit direction $(\tilde{W}, \tilde{V})$, s.t. $\|\tilde{W}\|_F^2 + \|\tilde{V}\|_F^2$, we can write the value of the smoothed function as an integration with density $\rho$, change variable, and then take derivatives:

$$
\frac{d}{d\lambda} f'_{W'+\lambda\tilde{W}, V'+\lambda\tilde{V}}(x_i)\Big|_{\lambda=0}
$$
$$
= \frac{d}{d\lambda}\mathbb{E}_{W^\rho, V^\rho} f_{W'+\lambda\tilde{W}+W^\rho, V'+\lambda\tilde{V}+V^\rho}(x_i)
$$
$$
= \frac{d}{d\lambda}\int f_{W'+\lambda\tilde{W}+W^\rho, V'+\lambda\tilde{V}+V^\rho}(x_i)\rho(W^\rho, V^\rho)d(W^\rho, V^\rho)
$$
$$
= \frac{d}{d\lambda}\int f_{W^+, V^+}(x_i)\rho(W^+ - (W' + \lambda\tilde{W}), V^+ - (V' + \lambda\tilde{V}))d(W^+, V^+).
$$

But one can easily see that for fixed $V'$ and $\tilde{V}$, the set of functions $f_{W^+, V^+}(x_i)\rho(W^+ - (W' + \lambda\tilde{W}), V^+ - (V' + \lambda\tilde{V}))$ for a small neighborhood of $\lambda$ can simultaneously be upper bounded by an integrable function. Hence, the Leibnitz rule holds here because of dominated convergence theorem,

and we can change the order of integration and derivation:

$$
= \int f_{W^+,V^+}(x_i)\frac{d}{d\lambda}\rho(W^+ - (W' + \lambda\tilde{W}), V^+ - (V' + \lambda\tilde{V}))d(W^+, V^+)
$$

$$
= -\int f_{W^+,V^+}(x_i)\Big\langle(\tilde{W},\tilde{V}),((\frac{\beta_1^2}{m_1})^{-1}\Big(W^+ - (W' + \lambda\tilde{W})\Big),(\frac{\beta_2^2}{m_2})^{-1}\Big(V^+ - (V' + \lambda\tilde{V})\Big))\Big\rangle
$$
$$
\rho(W^+ - (W' + \lambda\tilde{W}), V^+ - (V' + \lambda\tilde{V}))d(W^+, V^+)
$$

$$
= \int f_{W^+,V^+}(x_i)\Big\langle(\tilde{W},\tilde{V}),(\frac{m_1}{\beta_1^2}W^\rho, \frac{m_2}{\beta_2^2}V^\rho)\Big\rangle\rho(W^\rho, V^\rho)d(W^\rho, V^\rho)
$$

$$
= \mathbb{E}_{W^\rho,V^\rho}\Big(\frac{m_1}{\beta_1^2}\Big\langle\tilde{W}, W^\rho\Big\rangle + \frac{m_2}{\beta_2^2}\Big\langle\tilde{V}, V^\rho\Big\rangle\Big)f_{W'+\lambda\tilde{W}+W^\rho,V'+\lambda\tilde{V}+V^\rho(x_i)}(x_i)\Big|_{\lambda=0}
$$

$$
= \mathbb{E}_{W^\rho,V^\rho}\Big(\frac{m_1}{\beta_1^2}\Big\langle\tilde{W}, W^\rho\Big\rangle + \frac{m_2}{\beta_2^2}\Big\langle\tilde{V}, V^\rho\Big\rangle\Big)f_{W'+W^\rho,V'+V^\rho(x_i)}(x_i). \tag{287}
$$

Similarly we can compute the second derivative:

$$
\frac{d^2}{d\lambda^2}f'_{W'+\lambda\tilde{W},V'+\lambda\tilde{V}}(x_i)\Big|_{\lambda=0}
$$
$$
= \frac{d}{d\lambda}\int f_{W^+,V^+}(x_i)\Big\langle(\tilde{W},\tilde{V}),((\frac{\beta_1^2}{m_1})^{-1}\Big(W^+ - (W' + \lambda\tilde{W})\Big),(\frac{\beta_2^2}{m_2})^{-1}\Big(V^+ - (V' + \lambda\tilde{V})\Big))\Big\rangle
$$
$$
\rho(W^+ - (W' + \lambda\tilde{W}), V^+ - (V' + \lambda\tilde{V}))d(W^+, V^+)
$$
$$
= \int f_{W^+,V^+}(x_i)\Big[\Big\langle(\tilde{W},\tilde{V}),((\frac{\beta_1^2}{m_1})^{-1}\Big(W^+ - (W' + \lambda\tilde{W})\Big),(\frac{\beta_2^2}{m_2})^{-1}\Big(V^+ - (V' + \lambda\tilde{V})\Big))\Big\rangle^2 -
$$
$$
\Big\langle(\tilde{W},\tilde{V}),((\frac{\beta_1^2}{m_1})^{-1}\tilde{W},(\frac{\beta_2^2}{m_2})^{-1}\tilde{V})\Big\rangle\Big]\rho(W^+ - (W' + \lambda\tilde{W}), V^+ - (V' + \lambda\tilde{V}))d(W^+, V^+)
$$
$$
= \int f_{W^+,V^+}(x_i)\Big[\Big\langle(\tilde{W},\tilde{V}),(\frac{m_1}{\beta_1^2}W^\rho, \frac{m_2}{\beta_2^2}V^\rho)\Big\rangle^2 + \Big\langle(\tilde{W},\tilde{V}),(\frac{m_1}{\beta_1^2}\tilde{W}, \frac{m_2}{\beta_2^2}\tilde{V})\Big\rangle\Big]\rho(W^\rho, V^\rho)d(W^\rho, V^\rho)
$$
$$
= \mathbb{E}_{W^\rho,V^\rho}\Big[\Big(\frac{m_1}{\beta_1^2}\langle\tilde{W}, W^\rho\rangle + \frac{m_2}{\beta_2^2}\langle\tilde{V}, V^\rho\rangle\Big)^2 - \Big(\frac{m_1}{\beta_1^2}\|\tilde{W}\|^2 + \frac{m_2}{\beta_2^2}\|\tilde{V}\|^2\Big)\Big]f_{W'+\lambda\tilde{W}+W^\rho,V'+\lambda\tilde{V}+V^\rho(x_i)}(x_i)\Big|_{\lambda=0}
$$
$$
= \mathbb{E}_{W^\rho,V^\rho}\Big[\Big(\frac{m_1}{\beta_1^2}\langle\tilde{W}, W^\rho\rangle + \frac{m_2}{\beta_2^2}\langle\tilde{V}, V^\rho\rangle\Big)^2 - \Big(\frac{m_1}{\beta_1^2}\|\tilde{W}\|^2 + \frac{m_2}{\beta_2^2}\|\tilde{V}\|^2\Big)\Big]f_{W'+W^\rho,V'+V^\rho(x_i)}(x_i). \tag{288}
$$

Similarly for the third derivative:

$$
\frac{d^3}{d\lambda^3}f'_{W'+\lambda\tilde{W},V'+\lambda\tilde{V}}(x_i)\Big|_{\lambda=0}
$$
$$
\frac{d}{d\lambda}\int f_{W^+,V^+}(x_i)\Big[\Big\langle(\tilde{W},\tilde{V}),((\frac{\beta_1^2}{m_1})^{-1}\Big(W^+ - (W' + \lambda\tilde{W})\Big),(\frac{\beta_2^2}{m_2})^{-1}\Big(V^+ - (V' + \lambda\tilde{V})\Big))\Big\rangle^2 -
$$
$$
\Big\langle(\tilde{W},\tilde{V}),((\frac{\beta_1^2}{m_1})^{-1}\tilde{W},(\frac{\beta_2^2}{m_2})^{-1}\tilde{V})\Big\rangle\Big]\rho(W^+ - (W' + \lambda\tilde{W}), V^+ - (V' + \lambda\tilde{V}))d(W^+, V^+)
$$
$$
= \int f_{W^+,V^+}(x_i)\Big[\Big\langle(\tilde{W},\tilde{V}),((\frac{\beta_1^2}{m_1})^{-1}\Big(W^+ - (W' + \lambda\tilde{W})\Big),(\frac{\beta_2^2}{m_2})^{-1}\Big(V^+ - (V' + \lambda\tilde{V})\Big))\Big\rangle^3
$$
$$
- 3\Big\langle(\tilde{W},\tilde{V}),((\frac{\beta_1^2}{m_1})^{-1}\Big(W^+ - (W' + \lambda\tilde{W})\Big),(\frac{\beta_2^2}{m_2})^{-1}\Big(V^+ - (V' + \lambda\tilde{V})\Big))\Big\rangle\Big\langle(\tilde{W},\tilde{V}),\Big((\frac{\beta_1^2}{m_1})^{-1}\tilde{W},(\frac{\beta_2^2}{m_2})^{-1}\tilde{V}\Big)\Big\rangle\Big]
$$
$$
\rho(W^+ - (W' + \lambda\tilde{W}), V^+ - (V' + \lambda\tilde{V}))d(W^+, V^+)
$$
$$
= \mathbb{E}_{W^\rho,V^\rho}\Big[\Big(\frac{m_1}{\beta_1^2}\langle\tilde{W}, W^\rho\rangle + \frac{m_2}{\beta_2^2}\langle\tilde{V}, V^\rho\rangle\Big)^3 - 3\Big(\frac{m_1}{\beta_1^2}\langle\tilde{W}, W^\rho\rangle + \frac{m_2}{\beta_2^2}\langle\tilde{V}, V^\rho\rangle\Big)\Big(\frac{m_1}{\beta_1^2}\|\tilde{W}\|^2 + \frac{m_2}{\beta_2^2}\|\tilde{V}\|^2\Big)
$$
$$
f_{W'+W^\rho,V'+V^\rho(x_i)}(x_i)\Big].
$$

Now for first derivative, exactly similar to the derivation in (283), we can write

$$\left| \mathbb{E}_{W^\rho, V^\rho} \left( \frac{m_1}{\beta_1^2} \left\langle \tilde{W}, W^\rho \right\rangle + \frac{m_2}{\beta_2^2} \left\langle \tilde{V}, V^\rho \right\rangle \right) f_{W'+W^\rho, V'+V^\rho(x_i)}(x_i) \right|$$

$$\leq \mathbb{E}_{W^\rho, V^\rho} \left| \frac{m_1}{\beta_1^2} \left\langle \tilde{W}, W^\rho \right\rangle + \frac{m_2}{\beta_2^2} \left\langle \tilde{V}, V^\rho \right\rangle \right| \left| f_{W'+W^\rho, V'+V^\rho(x_i)}(x_i) \right|$$

$$\leq \mathbb{E}_{W^\rho, V^\rho} \left| \frac{m_1}{\beta_1^2} \left\langle \tilde{W}, W^\rho \right\rangle + \frac{m_2}{\beta_2^2} \left\langle \tilde{V}, V^\rho \right\rangle \right| \left( \kappa_2 \sqrt{m}_3 \|x_i'^{(2)}\| + \|V'\|_F \|x_i'^{(2)}\| + \mathbb{E}_{V^\rho} \frac{1}{\sqrt{m_2}} \sum_j |V_j^\rho x_i'^{(2)}| \right)$$

$$\leq \mathbb{E}_{W^\rho, V^\rho} \left( \left| \frac{m_1}{\beta_1^2} \left\langle \tilde{W}, W^\rho \right\rangle \right| + \left| \frac{m_2}{\beta_2^2} \left\langle \tilde{V}, V^\rho \right\rangle \right| \right) \left( \kappa_2 \sqrt{m}_3 \|x_i'^{(2)}\| + \|V'\|_F \|x_i'^{(2)}\| + \frac{1}{\sqrt{m_2}} \sum_j |V_j^\rho x_i'^{(2)}| \right)$$

$$= \frac{m_2}{\beta_2^2} \left( E_{V^\rho} \left| \left\langle \tilde{V}, V^\rho \right\rangle \right| \mathbb{E}_{W^\rho} \|x_i'^{(2)}\| \left( \kappa_2 \sqrt{m}_3 + \|V'\|_F \right) + \mathbb{E}_{W^\rho} \mathbb{E}_{V^\rho} \frac{1}{\sqrt{m_2}} \left| \left\langle \tilde{V}, V^\rho \right\rangle \right| \sum_j |V_j^\rho x_i'^{(2)}| \right)$$

$$+ \frac{m_1}{\beta_1^2} \left( \left( \mathbb{E}_{W^\rho} \|x_i'^{(2)}\| \left| \left\langle \tilde{W}, W^\rho \right\rangle \right| \right) \left( \kappa_2 \sqrt{m}_3 + \|V'\|_F \right) + \mathbb{E}_{W^\rho} \left| \left\langle \tilde{W}, W^\rho \right\rangle \right| \mathbb{E}_{V^\rho} \frac{1}{\sqrt{m_2}} \sum_j |V_j^\rho x_i'^{(2)}| \right)$$

$$\lesssim \frac{m_2}{\beta_2^2} \left( E_{V^\rho} \left| \left\langle \tilde{V}, V^\rho \right\rangle \right| \mathbb{E}_{W^\rho} \|x_i'^{(2)}\| \left( \kappa_2 \sqrt{m}_3 + C_2 \right) + \mathbb{E}_{W^\rho} \mathbb{E}_{V^\rho} \frac{1}{\sqrt{m_2}} \left| \left\langle \tilde{V}, V^\rho \right\rangle \right| \sum_j |V_j^\rho x_i'^{(2)}| \right)$$

$$+ \frac{m_1}{\beta_1^2} \left( \left( \mathbb{E}_{W^\rho} \|x_i'^{(2)}\| \left| \left\langle \tilde{W}, W^\rho \right\rangle \right| \right) \left( \kappa_2 \sqrt{m}_3 + C_2 \right) + \mathbb{E}_{W^\rho} \left| \left\langle \tilde{W}, W^\rho \right\rangle \right| \mathbb{E}_{V^\rho} \frac{1}{\sqrt{m_2}} \sum_j |V_j^\rho x_i'^{(2)}| \right),$$

where the last line follows because $\|V'\| \lesssim C_2$. But notice that because $\|\tilde{V}\|_F \leq 1$, $\|\tilde{W}\|_F \leq 1$, then $\left\langle \tilde{V}, V^\rho \right\rangle$ and $\left\langle \tilde{W}, W^\rho \right\rangle$ are Gaussian variables with variances at most $\beta_2^2/m_2$ and $\beta_1^2/m_1$. Hence

$$E_{V^\rho} \left| \left\langle \tilde{V}, V^\rho \right\rangle \right| \lesssim \beta_2/\sqrt{m}_2, \tag{289}$$

$$E_{W^\rho} \left| \left\langle \tilde{W}, W^\rho \right\rangle \right| \lesssim \beta_1/\sqrt{m}_1. \tag{290}$$

Similarly, using the same derivation as in (33), one can also get the following a.s. bound (over the randomness of $W^\rho$):

$$\|x_i'^{(2)}\| \lesssim \sqrt{m}_3 \left( \kappa_1 + C_1 + \frac{1}{\sqrt{m_1}} \sum_j |W_j^\rho x_i| \right), \tag{291}$$

therefore

$$E_{W^\rho} \|x_i'^{(2)}\| \lesssim \sqrt{m}_3 \left( \kappa_1 + C_1 + E_{W^\rho} \frac{1}{\sqrt{m_1}} \sum_j |W_j^\rho x_i| \right) \tag{292}$$

$$\lesssim \sqrt{m}_3 \left( \kappa_1 + C_1 + \beta_1 \right). \tag{293}$$

Moreover, for every $j \in [m_2]$:

$$E_{V^\rho} \left| \left\langle \tilde{V}, V^\rho \right\rangle \right| |V_j^\rho x_i'^{(2)}| \leq \sqrt{E_{V^\rho} \left| \left\langle \tilde{V}, V^\rho \right\rangle \right|^2} \sqrt{E_{V^\rho} |V_j^\rho x_i'^{(2)}|^2}$$

$$= \frac{\beta_2}{\sqrt{m}_2} \frac{\beta_2}{\sqrt{m}_2} \|x_i'^{(2)}\| = \frac{\beta_2^2}{m_2} \|x_i'^{(2)}\|,$$

$$E_{W^\rho} \left| \left\langle \tilde{W}, W^\rho \right\rangle \right| |W_j^\rho x_i| \leq \frac{\beta_1^2}{m_1}. \tag{294}$$

Similarly, using Equation (291) we bound

$$\mathbb{E}_{W^\rho} \|x_i'^{(2)}\| \left| \left\langle \tilde{W}, W^\rho \right\rangle \right| \lesssim \sqrt{m}_3 \left( \mathbb{E}_{W^\rho} \left| \left\langle \tilde{W}, W^\rho \right\rangle \right| \left( \kappa_1 + C_1 \right) + \mathbb{E}_{W^\rho} \frac{1}{\sqrt{m_1}} \sum_j \left| \left\langle \tilde{W}, W^\rho \right\rangle \right| |W_j^\rho x_i| \right)$$

$$\leq \sqrt{m}_3 \left( \frac{\beta_1}{\sqrt{m}_1} \left( \kappa_1 + C_1 \right) + \frac{\beta_1^2}{\sqrt{m}_1} \right). \tag{295}$$

Now applying these bounds (290), (293), (294), and (295) to (289) and using the fact that

$$\mathbb{E}_{V^\rho} \frac{1}{\sqrt{m_2}} \sum_j |V_j^\rho x_i'^{(2)}| \le \beta_2 \|x_i'^{(2)}\|,$$

we get

$$
\begin{aligned}
LHS \lesssim & \frac{m_2}{\beta_2^2} \Big( \frac{\beta_2}{\sqrt{m_2}} \sqrt{m_3}(\kappa_1 + C_1 + \beta_1) \Big( \kappa_2 \sqrt{m_3} + C_2 \Big) + \mathbb{E}_{W^\rho} \frac{\beta_2^2}{\sqrt{m_2}} \|x_i'^{(2)}\| \Big) \\
& + \frac{m_1}{\beta_1^2} \Big( \sqrt{m_3} \Big( \frac{\beta_1}{\sqrt{m_1}} \big( \kappa_1 + C_1 \big) + \frac{\beta_1^2}{\sqrt{m_1}} \Big) \Big( \kappa_2 \sqrt{m_3} + C_2 \Big) + \mathbb{E}_{W^\rho} \big| \langle \tilde{W}, W^\rho \rangle \big| \beta_2 \|x_i'^{(2)}\| \Big) \\
\lesssim & \frac{m_2}{\beta_2^2} \Big( \frac{\beta_2}{\sqrt{m_2}} \sqrt{m_3}(\kappa_1 + C_1 + \beta_1) \Big( \kappa_2 \sqrt{m_3} + C_2 \Big) + \frac{\beta_2^2}{\sqrt{m_2}} \sqrt{m_3} \big( \kappa_1 + C_1 + \beta_1 \big) \Big) \\
& + \frac{m_1}{\beta_1^2} \Big( \sqrt{m_3} \Big( \frac{\beta_1}{\sqrt{m_1}} \big( \kappa_1 + C_1 \big) + \frac{\beta_1^2}{\sqrt{m_1}} \Big) \Big( \kappa_2 \sqrt{m_3} + C_2 \Big) + \beta_2 \sqrt{m_3} \Big( \frac{\beta_1}{\sqrt{m_1}} \big( \kappa_1 + C_1 \big) + \frac{\beta_1^2}{\sqrt{m_1}} \Big) \Big).
\end{aligned}
$$

To make it easier for handling the second and third derivatives, we first bound the expectations of $f^2_{W'+W^\rho, V'+V^\rho(x_i)}(x_i)$ which enables us to use Cauchy-schwartz. Again using similar derivation as in (283) and Equation (291):

$$
\begin{aligned}
& \mathbb{E}_{W^\rho, V^\rho} f^2_{W'+W^\rho, V'+V^\rho(x_i)}(x_i) \\
& \le \mathbb{E}_{W^\rho, V^\rho} \Big( \kappa_2 \sqrt{m_3} \|x_i'^{(2)}\| + C_2 \|x_i'^{(2)}\| + \frac{1}{\sqrt{m_2}} \sum_j |V_j^\rho x_i'^{(2)}| \Big)^2 \\
& \lesssim (\kappa_2 \sqrt{m_3} + C_2)^2 \mathbb{E}_{W^\rho} \|x_i'^{(2)}\|^2 + E_{W^\rho} E_{V^\rho} \frac{1}{m_2} \Big( \sum_j |V_j^\rho x_i'^{(2)}| \Big)^2 \\
& \lesssim (\kappa_2 \sqrt{m_3} + C_2)^2 \mathbb{E}_{W^\rho} \|x_i'^{(2)}\|^2 + E_{W^\rho} \frac{1}{m_2} \sum_j E_{V_j^\rho} |V_j^\rho x_i'^{(2)}|^2 + E_{W^\rho} \frac{1}{m_2} \sum_{j_1 \ne j_2} E_{V_{j_1}^\rho} |V_{j_1}^\rho x_i'^{(2)}| E_{V_{j_2}^\rho} |V_{j_2}^\rho x_i'^{(2)}| \\
& \lesssim (\kappa_2 \sqrt{m_3} + C_2)^2 \mathbb{E}_{W^\rho} \|x_i'^{(2)}\|^2 + E_{W^\rho} \frac{\beta_2^2}{m_2} \|x_i'^{(2)}\|^2 + E_{W^\rho} \beta_2^2 \|x_i'^{(2)}\|^2 \frac{m(m-1)}{m^2} \\
& \lesssim \Big( (\kappa_2 \sqrt{m_3} + C_2)^2 + \beta_2^2 \Big) \mathbb{E}_{W^\rho} \|x_i'^{(2)}\|^2 \\
& \lesssim \Big( (\kappa_2 \sqrt{m_3} + C_2)^2 + \beta_2^2 \Big) E_{W^\rho} m_3 \Big( \kappa_1 + C_1 + \frac{1}{\sqrt{m_1}} \sum_j |W_j^\rho x_i| \Big)^2 \\
& \lesssim \Big( (\kappa_2 \sqrt{m_3} + C_2)^2 + \beta_2^2 \Big) m_3 \Big[ (\kappa_1 + C_1)^2 + \frac{1}{m_1} \sum_j \mathbb{E}_{W^\rho} |W_j^\rho x_i|^2 + \frac{1}{m_1} \sum_{j_1 \ne j_2} \mathbb{E}_{W_{j_1}^\rho} |W_{j_1}^\rho x_i| \mathbb{E}_{W_{j_1}^\rho} |W_{j_2}^\rho x_i| \Big] \\
& \lesssim \Big( (\kappa_2 \sqrt{m_3} + C_2)^2 + \beta_2^2 \Big) m_3 \Big[ (\kappa_1 + C_1)^2 + \frac{\beta_1^2}{m_1} + \beta_1^2 \frac{m(m-1)}{m^2} \Big] \\
& = m_3 \Big( (\kappa_2 \sqrt{m_3} + C_2)^2 + \beta_2^2 \Big) \Big[ (\kappa_1 + C_1)^2 + \beta_1^2 \Big]. \qquad (296)
\end{aligned}
$$

Now for the second derivative, we can proceed by applying Cauchy-Swartz:

$$\frac{d^2}{d\lambda^2} f'_{W'+\lambda\tilde{W}, V'+\lambda\tilde{V}}(x_i)\Big|_{\lambda=0}$$

$$\leq \mathbb{E}_{W^\rho, V^\rho} \left| \left( \frac{m_1}{\beta_1^2} \langle \tilde{W}, W^\rho \rangle + \frac{m_2}{\beta_2^2} \langle \tilde{V}, V^\rho \rangle \right)^2 - \left( \frac{m_1}{\beta_1^2} \|\tilde{W}\|^2 + \frac{m_2}{\beta_2^2} \|\tilde{V}\|^2 \right) \right| \left| f_{W'+W^\rho, V'+V^\rho(x_i)}(x_i) \right|$$

$$\leq \mathbb{E}_{W^\rho, V^\rho} \left| \frac{m_1^2}{\beta_1^4} \langle \tilde{W}, W^\rho \rangle^2 - \frac{m_1}{\beta_1^2} \|\tilde{W}\|^2 + \frac{m_2^2}{\beta_2^4} \langle \tilde{V}, V^\rho \rangle^2 - \frac{m_2}{\beta_2^2} \|\tilde{V}\|^2 + \frac{2m_1 m_2}{\beta_1^2 \beta_2^2} \langle \tilde{W}, W^\rho \rangle \langle \tilde{V}, V^\rho \rangle \right|$$

$$\left| f_{W'+W^\rho, V'+V^\rho(x_i)}(x_i) \right|$$

$$\leq \sqrt{ \mathbb{E}_{W^\rho, V^\rho} \left| \frac{m_1^2}{\beta_1^4} \langle \tilde{W}, W^\rho \rangle^2 - \frac{m_1}{\beta_1^2} \|\tilde{W}\|^2 + \frac{m_2^2}{\beta_2^4} \langle \tilde{V}, V^\rho \rangle^2 - \frac{m_2}{\beta_2^2} \|\tilde{V}\|^2 + \frac{2m_1 m_2}{\beta_1^2 \beta_2^2} \langle \tilde{W}, W^\rho \rangle \langle \tilde{V}, V^\rho \rangle \right|^2 }$$

$$\sqrt{ \mathbb{E}_{W^\rho, V^\rho} \left| f_{W'+W^\rho, V'+V^\rho(x_i)}(x_i) \right|^2 }.$$

Now note that the cross terms have expectation zero, so we get

$$= \sqrt{ \mathbb{E}_{W^\rho, V^\rho} \left( \frac{m_1^2}{\beta_1^4} \langle \tilde{W}, W^\rho \rangle^2 - \frac{m_1}{\beta_1^2} \|\tilde{W}\|^2 \right)^2 + \left( \frac{m_2^2}{\beta_2^4} \langle \tilde{V}, V^\rho \rangle^2 - \frac{m_2}{\beta_2^2} \|\tilde{V}\|^2 \right)^2 + \frac{4m_1^2 m_2^2}{\beta_1^4 \beta_2^4} \mathbb{E}_{W^\rho, V^\rho} \langle \tilde{W}, W^\rho \rangle^2 \langle \tilde{V}, V^\rho \rangle^2 }$$

$$\sqrt{ \mathbb{E}_{W^\rho, V^\rho} \left| f_{W'+W^\rho, V'+V^\rho(x_i)}(x_i) \right|^2 }$$

$$\lesssim \sqrt{ \frac{m_1^2}{\beta_1^4} \|\tilde{W}\|^4 + \frac{m_2^2}{\beta_2^4} \|\tilde{V}\|^4 + \frac{4m_1 m_2}{\beta_1^2 \beta_2^2} \|\tilde{W}\|^2 \|\tilde{V}\|^2 } \sqrt{ \mathbb{E}_{W^\rho, V^\rho} \left| f_{W'+W^\rho, V'+V^\rho(x_i)}(x_i) \right|^2 }$$

$$\lesssim \left( \frac{m_1}{\beta_1^2} \|\tilde{W}\|^2 + \frac{m_2}{\beta_2^2} \|\tilde{V}\|^2 \right) \sqrt{ \mathbb{E}_{W^\rho, V^\rho} \left| f_{W'+W^\rho, V'+V^\rho(x_i)}(x_i) \right|^2 }.$$

Now applying Cauchy-shwartz and Equation (296) to above and combining it with Equations (288):

$$\frac{d^2}{d\lambda^2} f'_{W'+\lambda\tilde{W}, V'+\lambda\tilde{V}}(x_i)\Big|_{\lambda=0}$$

$$\lesssim \left( \frac{m_1}{\beta_1^2} \|\tilde{W}\|^2 + \frac{m_2}{\beta_2^2} \|\tilde{V}\|^2 \right) \sqrt{ m_3 \left( (\kappa_2 \sqrt{m_3} + C_2)^2 + \beta_2^2 \right) \left[ (\kappa_1 + C_1)^2 + \beta_1^2 \right] }.$$

Similarly for the third derivative:

$$\left| \frac{d^3}{d\lambda^3} f'_{W'+\lambda\tilde{W}, V'+\lambda\tilde{V}}(x_i)\Big|_{\lambda=0} \right|$$

$$= \mathbb{E}_{W^\rho, V^\rho} \left| \left( \frac{m_1}{\beta_1^2} \langle \tilde{W}, W^\rho \rangle + \frac{m_2}{\beta_2^2} \langle \tilde{V}, V^\rho \rangle \right)^3 - 3 \left( \frac{m_1}{\beta_1^2} \langle \tilde{W}, W^\rho \rangle + \frac{m_2}{\beta_2^2} \langle \tilde{V}, V^\rho \rangle \right) \left( \frac{m_1}{\beta_1^2} \|\tilde{W}\|^2 + \frac{m_2}{\beta_2^2} \|\tilde{V}\|^2 \right) \right|$$

$$\left| f_{W'+W^\rho, V'+V^\rho}(x_i) \right|$$

$$\leq \sqrt{ \mathbb{E}_{W^\rho, V^\rho} \left[ \left( \frac{m_1}{\beta_1^2} \langle \tilde{W}, W^\rho \rangle + \frac{m_2}{\beta_2^2} \langle \tilde{V}, V^\rho \rangle \right)^3 - 3 \left( \frac{m_1}{\beta_1^2} \langle \tilde{W}, W^\rho \rangle + \frac{m_2}{\beta_2^2} \langle \tilde{V}, V^\rho \rangle \right) \left( \frac{m_1}{\beta_1^2} \|\tilde{W}\|^2 + \frac{m_2}{\beta_2^2} \|\tilde{V}\|^2 \right) \right]^2 }$$

$$\sqrt{ \mathbb{E}_{W^\rho, V^\rho} \left| f_{W'+W^\rho, V'+V^\rho}(x_i) \right|^2 }. \tag{297}$$

But note that

$$\mathbb{E}_{W^\rho, V^\rho} \left[ \left( \frac{m_1}{\beta_1^2} \langle \tilde{W}, W^\rho \rangle + \frac{m_2}{\beta_2^2} \langle \tilde{V}, V^\rho \rangle \right)^3 - 3 \left( \frac{m_1}{\beta_1^2} \langle \tilde{W}, W^\rho \rangle + \frac{m_2}{\beta_2^2} \langle \tilde{V}, V^\rho \rangle \right) \left( \frac{m_1}{\beta_1^2} \|\tilde{W}\|^2 + \frac{m_2}{\beta_2^2} \|\tilde{V}\|^2 \right) \right]^2$$

$$\leq 2 \mathbb{E}_{W^\rho, V^\rho} \left( \frac{m_1}{\beta_1^2} \langle \tilde{W}, W^\rho \rangle + \frac{m_2}{\beta_2^2} \langle \tilde{V}, V^\rho \rangle \right)^6$$

$$+ 18 \left( \frac{m_1}{\beta_1^2} \|\tilde{W}\|^2 + \frac{m_2}{\beta_2^2} \|\tilde{V}\|^2 \right)^2 \mathbb{E}_{W^\rho, V^\rho} \left( \frac{m_1}{\beta_1^2} \langle \tilde{W}, W^\rho \rangle + \frac{m_2}{\beta_2^2} \langle \tilde{V}, V^\rho \rangle \right)^2$$

Now note that $\frac{m_1}{\beta_1^2}\langle \tilde{W}, W^\rho\rangle + \frac{m_2}{\beta_2^2}\langle \tilde{V}, V^\rho\rangle$ is a normal variable with variance $\frac{m_1}{\beta_1^2}\|\tilde{W}\|^2 + \frac{m_2}{\beta_2^2}\|\tilde{V}\|^2$. Therefore, by the bound on the moments of normal random variables:

$$LHS \lesssim \left(\frac{m_1}{\beta_1^2}\|\tilde{W}\|^2 + \frac{m_2}{\beta_2^2}\|\tilde{V}\|^2\right)^3. \tag{298}$$

Plugging this into Equation (297) and also using Equation (296):

$$\left|\frac{d^3}{d\lambda^3}f'_{W'+\lambda\tilde{W},V'+\lambda\tilde{V}}(x_i)\Big|_{\lambda=0}\right|$$
$$\lesssim \left(\frac{m_1}{\beta_1^2}\|\tilde{W}\|^2 + \frac{m_2}{\beta_2^2}\|\tilde{V}\|^2\right)^{3/2}\sqrt{m_3\left((\kappa_2\sqrt{m_3}+C_2)^2+\beta_2^2\right)\left[(\kappa_1+C_1)^2+\beta_1^2\right]}.$$

## B.2 Representation Lemmas

In this section, we prove lemmas mostly related to the representation power of the network, which we mainly use in Appendix A.12.

### B.2.1 Representation Toolbox

**Lemma 37** *Recall the definitions of $W_k^+$, and $\bar{x}_{i,k}$ from Equations (64), and (60). For all $k, i \in [n]$:*

$$|\bar{x}_{i,k} - trace(W_k^+ Z_k^i)| \leq \sqrt{n/(m_1\lambda_0)}\|\mathcal{V}_k\|_{H^\infty}.$$

**Proof of Lemma 37**

$$\text{trace}(W_k^+ Z_k^i) = \text{trace}(Z_k^i \sum_{j=1}^{n} \mathcal{V}_{k,j} Z_k^j) = \sum_{i'=1}^{n} \mathcal{V}_{k,i'}\langle Z_k^i, Z_k^{i'}\rangle. \tag{299}$$

But note

$$\langle Z_k^i, Z_k^{i'}\rangle = 1/m_1 \sum_{j=1}^{m_1} W_{k,j}'^2 \mathbb{1}\{W_j^{(0)T}x_i\}\mathbb{1}\{W_j^{(0)T}x_{i'}\}\langle x_{i'}, x_i\rangle$$

$$= \frac{\langle x_{i'}, x_i\rangle}{m_1} \sum_{j=1}^{m_1} \mathbb{1}\{W_j^{(0)T}x_i\}\mathbb{1}\{W_j^{(0)T}x_{i'}\}.$$

Now note that $\langle x_{i'}, x_i\rangle \leq 1$, and $\mathbb{1}\{W_j^{(0)T}x_i\}\mathbb{1}\{W_j^{(0)T}x_{i'}\}$ is a Bernoulli with

$$\mathbb{E}\mathbb{1}\{W_j^{(0)T}x_i\}\mathbb{1}\{W_j^{(0)T}x_{i'}\} = 1/4 + \arcsin(\langle x, y\rangle)/2\pi.$$

Therefore, by Hoeffding inequality we get

$$|\langle Z_k^i, Z_k^{i'}\rangle - H_{i,i'}^\infty| = O(1/\sqrt{m_1}).$$

Hence, because obviously $\|H^\infty\|_2 \leq 1$, we get

$$\text{trace}(W_k^+ Z_k^i) = \sum_{i'=1}^{n} \mathcal{V}_{k,i'} H_{i,i'}^\infty + O(1/\sqrt{m_1}) \sum_{i'=1}^{n} \mathcal{V}_{k,i'} = \bar{x}_{i,k} + O(1/\sqrt{m_1}) \sum_{i'=1}^{n} \mathcal{V}_{k,i'},$$

which implies

$$|\text{trace}(W_k^+ Z_k^i) - \bar{x}_{i,k}| \leq O(1/\sqrt{m_1})\sqrt{n}\|\mathcal{V}_k\|_2$$
$$\lesssim O(\sqrt{n}/(\sqrt{m_1}\sqrt{\lambda_0}))\|\mathcal{V}_k\|_{H^\infty}$$
$$\leq \sqrt{n/(m_1\lambda_0)}\|\mathcal{V}_k\|_{H^\infty}.$$

**Lemma 38** *(Bounding the rows norm) For every $1 \leq j \leq m_1$, we have*

$$\|W_j^+\| \leq \sqrt{nm_3}/(\sqrt{m_1\lambda_0})\sqrt{\sum_k \|\mathcal{V}_k\|_{H^\infty}^2}.$$

*Furthermore, for every $k \in [m_3]$, we have*

$$\|W_j^{k+}\| \leq \sqrt{n}/\sqrt{\lambda_0 m_1}\|\mathcal{V}_k\|_{H^\infty}. \tag{300}$$

For the ease of notation, because here we want to work with row sub indices of the matrix $W_k^+$, we refer to it by $W^{k+}$. **Proof of Lemma 38**

For a fixed $1 \leq j \leq m_1$ we have with high probability over the randomness of the sign matrix $W^s$:

$$\|W_j^+\| = \|\sum_{k=1}^{m_3} W_j^{k+}\| = \|\sum_{k=1}^{m_3} W_{k,j}^s 1/\sqrt{m_1} \sum_{i=1}^n \mathcal{V}_{k,i} x_i \mathbb{1}\{W_j^{(0)} x_i \geq 0\}\|$$

$$\leq \sqrt{m_3}/\sqrt{m_1} \sqrt{\sum_{k=1}^{m_3} \|\sum_{i=1}^n \mathcal{V}_{k,i} x_i \mathbb{1}\{W_j^{(0)} x_i \geq 0\}\|^2}$$

$$\leq \sqrt{m_3}/\sqrt{m_1} \sqrt{\sum_k \left(\sum_i |\mathcal{V}_{k,i}|\right)^2}$$

$$\leq \sqrt{m_3}/\sqrt{m_1} \sqrt{n \sum_k \|\mathcal{V}_k\|^2}$$

$$\leq \sqrt{m_3 n}/(\sqrt{m_1}\lambda_0) \sqrt{\sum_k \|\mathcal{V}_k\|_{H^\infty}^2}.$$

Furthermore, for every $k \in [m_3]$, we have

$$\|W_j^{k+}\| \leq 1/\sqrt{m_1} \sum_i |\mathcal{V}_{k,i}| \leq (\sqrt{n}/\sqrt{m_1})\|\mathcal{V}_k\|_2 \leq (\sqrt{n}/\sqrt{\lambda_0 m_1})\|\mathcal{V}_k\|_{H^\infty}.$$

**Lemma 39** *With high probability over the initialization, we have*

$$\|W^{k+}\|_F^2 \leq (1 \pm O(n/(\lambda_0\sqrt{m_1})))\|\mathcal{V}_k\|_{H^\infty}^2.$$

**Proof of Lemma 39**

Recall from the definition of $W^{k+}$ in Equation (64):

$$\|W^{k+}\|_F^2 = \|\sum_{i=1}^n \mathcal{V}_{k,i} Z_k^i\|_F^2 = \sum_{i=1}^n \sum_{i'=1}^n \mathcal{V}_{k,i} \mathcal{V}_{k,i'} \langle Z_k^i, Z_k^{i'} \rangle$$

$$= \sum_{i=1}^n \sum_{i'=1}^n \mathcal{V}_{k,i} \mathcal{V}_{k,i'}(H_{i,i'}^\infty \pm O(1/\sqrt{m_1})) \lesssim \mathcal{V}_k^T H^\infty \mathcal{V}_k \pm O((\|\mathcal{V}_k\|_1)^2/\sqrt{m_1})$$

$$= \|\mathcal{V}_k\|_{H^\infty}^2 \pm \|\mathcal{V}_k\|_{H^\infty}^2 O(n/(\lambda_0\sqrt{m_1})) = (1 \pm O(n/(\lambda_0\sqrt{m_1})))\|\mathcal{V}_k\|_{H^\infty}^2$$

### B.2.2    SOME LINEAR ALGEBRA

**Lemma 40** *For $n \leq s$, let $r_1, \ldots, r_n$ be $s$-dimensional vectors that are approximately normalized and orthogonal to one another, i.e. given some $\delta > 0$, for every $1 \leq i \neq j \leq n$:*

$$-\delta \leq \langle r_i, r_j \rangle \leq \delta, \ \|r_i\|^2 \leq 1 + \delta.$$

*Then, for any vector $v$ we have*

$$\sum_{i=1}^n \langle v, r_i \rangle^2 \leq (1 + \delta + n(n-1)\delta(1+\delta)^2)\|v\|^2.$$

**Proof of Lemma 40**

Define

$$v_1 = \sum_{i=1}^n \langle v, r_i \rangle r_i, \ v_2 = v - v_1.$$

First, note that

$$\sum_{i=1}^{n}\langle v, r_i\rangle^2 \leq (1+\delta)\sum_{i=1}^{n}\langle v, r_i\rangle^2\|r_i\|^2$$

$$= (1+\delta)\|\sum_i\langle v, r_i\rangle r_i\|^2 - 2(1+\delta)\sum_{1\leq i\neq j\leq n}\langle v, r_i\rangle\langle v, r_i\rangle\langle r_i, r_j\rangle$$

$$= (1+\delta)\|v_1\|^2 - 2(1+\delta)\sum_{1\leq i\neq j\leq n}\langle v, r_i\rangle\langle v, r_i\rangle\langle r_i, r_j\rangle.$$

Next, we write

$$\langle v_1, v - v_1\rangle = \langle v, \sum_{i=1}^{n}\langle v, r_i\rangle r_i\rangle - \langle\sum_{i=1}^{n}\langle v, r_i\rangle r_i, \sum_{i=1}^{n}\langle v, r_i\rangle r_i\rangle$$

$$= \sum_i\langle v, r_i\rangle^2 - \sum_i\langle v, r_i\rangle^2 - 2\sum_{i\neq j}\langle v, r_i\rangle\langle v, r_j\rangle\langle r_i, r_j\rangle$$

$$= -2\sum_{i\neq j}\langle v, r_i\rangle\langle v, r_j\rangle\langle r_i, r_j\rangle.$$

Therefore

$$\sum_{i=1}^{n}\langle v, r_i\rangle^2 \leq (1+\delta)(\|v_1\|^2 + \|v - v_1\|^2) - 2(1+\delta)\sum_{1\leq i\neq j\leq n}\langle v, r_i\rangle\langle v, r_i\rangle\langle r_i, r_j\rangle.$$

$$\leq (1+\delta)(\|v_1\|^2 + \|v - v_1\|^2 + 2\langle v_1, v - v_1\rangle)$$

$$+ 2(1+\delta)\sum_{1\leq i\neq j\leq n}\langle v, r_i\rangle\langle v, r_i\rangle\langle r_i, r_j\rangle$$

$$\leq (1+\delta)\|v\|^2 + 2(1+\delta)\sum_{1\leq i\neq j\leq n}\|v\|^2\|r_i\|\|r_j\|\delta$$

$$\leq (1+\delta)\|v\|^2 + 2(1+\delta)\sum_{1\leq i\neq j\leq n}\|v\|^2(1+\delta)\delta$$

$$= (1+\delta)\|v\|^2 + n(n-1)\delta(1+\delta)^2\|v\|^2$$

$$= (1+\delta + n(n-1)\delta(1+\delta)^2)\|v\|^2,$$

which completes the proof.

In the following lemma, we state a trivial bound on the norm of $\bar{x}_i$ based on $\zeta_1$.

### B.2.3   Bound on the norm of $\bar{x}_i$'s

**Lemma 41** *For every $i \in [n]$, we have*

$$\|\bar{x}_i\| \leq \sqrt{\sum_k \|\mathcal{V}_k\|_{H^\infty}^2} = \sqrt{\zeta_1}.$$

**Proof of Lemma 41**

By definition:

$$\bar{x}_i^T = \left(H_{i,}^\infty\mathcal{V}_k\right)_{k=1}^{m_3}.$$

Now consider the Cholskey factorization $H^\infty = KK^T$. Because of the assumption $\|x_i\| = 1$, we know that the diagonal of $H^\infty$ is all $1/2$. Hence, for the $i$th row of $K$ we have $\|K_i\| = 1/2$. Now by Cauchy-Swartz, we have

$$x_{i_1}^2 = \langle\sum_i \mathcal{V}_{k,i}K_i, K_{i_1}\rangle^2 \leq \|\sum_i \mathcal{V}_{k,i}K_i\|^2\|K_{i_1}\|^2 = 1/2\|\mathcal{V}_k\|_{H^\infty}^2.$$

Summing over $i$ and noting Equation (59) completes the proof.

**Lemma 42** *In the context of Lemma 10, for $\zeta_2 \leq 2nB^2$, one can substitute $f^*$ by $\bar{f}^*$ such that*

$$R_n(\bar{f}^*) \leq 2R_n(f^*) + \frac{B^2}{n},$$

$$\bar{f}^{*T} A^{-1} \bar{f}^* \leq f^{*T} A^{-1} f^*,$$

*and furthermore, $\bar{f}^*$ is in the subspace of eigenvectors of $A$ with eigenvaue larger than $\Omega(\frac{1}{n^2})$. Moreover, the constant 2 is arbitrary and can be changed to any constant more than one, with the cost of an additional constant behind the second term.*

**Proof of Lemma 42**

For an arbitrary $i \in [n]$ and some given vector $\bar{f}^*$ (we will specify later), we define

$$\delta = |f_i^* - \bar{f}_i^*|,$$

and suppose the slope of $\ell(., y_i)$ at point $f_i^*$ is equal to $c$. Then, using the convexity, the fact that $\ell(y_i, y_i) = 0$, and the 1-smoothness of $\ell(., y_i)$, it is not hard to see the following poincare inequality between the value and derivative of $\ell(., y_i)$ at point $f_i^*$:

$$c \leq \sqrt{2\ell(f_i^*, y_i)} := 2\ell. \tag{301}$$

where from now on, for brevity, we refer to $\ell(f_i^*, y_i)$ by $\ell$. Also, from the definition of $\delta$ and again using 1 smoothness property, it is easy to see that

$$\ell(\bar{f}_i^*, y_i) \leq (c + \delta)\delta + \ell(f_i^*, y_i) = (c + \delta)\delta + \ell, \tag{302}$$

Plugging Equation (301) into (302) and using AM-GM inequality:

$$\begin{aligned} \ell(\bar{f}_i^*, y_i) &\leq \delta^2 + c\delta + \ell \leq \delta^2 + \sqrt{2\ell}\delta + \ell \\ &\leq \delta^2 + \ell + \delta^2/2 + \ell \\ &\leq 2\ell + 3\delta^2/2. \end{aligned}$$

Summing above for $i \in [n]$, we obtain

$$R_n(\bar{f}^*) \leq 2R_n(f^*) + 3\|f^* - \bar{f}^*\|_2^2/2. \tag{303}$$

Now we write an eigendecomposition for $A$ as $A = \sum_{i=1}^n \lambda_i u_i u_i^T$ for orthonormal basis $\{u_i\}$, and let $f^* = \sum_i \gamma_i u_i$ be the representation of $f^*$ in this basis. Then, from our assumption, for arbitrary $\omega > 0$

$$\sum_i \gamma_i^2 \lambda_i^{-1} = f^{*T} A^{-1} f^* \leq 4nB^2,$$

which implies

$$\omega^{-1} \sum_{i:\ \lambda_i \leq \omega} \gamma_i^2 \leq 4nB^2,$$

or equivalently

$$\sum_{i:\ \lambda_i \leq \omega} \gamma_i^2 \leq 4nB^2\omega, \tag{304}$$

where notice that $\sum_{i:\ \lambda_i \leq \omega} \gamma_i^2$ is the squared norm of the projection of $f^*$ onto the directions whose eigenvalue is at most $\omega$. Now taking $\omega = \frac{1}{12n^2}$ and defining $\bar{f}^*$ by keeping only the directions in the expansion of $f^*$ in the eigenbasis of $A$, for which $\lambda_i > \omega$, completes the proof.

### B.3 Coupling for $\hat{\nabla}_W, \hat{\nabla}_V$

In general, because the gaussian smoothing matrices $(W^\rho, V^\rho)$ can become unbounded, the gradient estimates $(\hat{\nabla}_W, \hat{\nabla}_V) = \nabla_{W,V}\ell(f_{W'+W^\rho, V'+V^\rho}(x_i), y_i)$ also become unbounded. However, in analyzing the stochastic behavior of SGD and showing that it can escape saddle points, it is convenient to assume the gradient's noise vector is almost surely bounded. The goal of this section is to introduce a coupling between $(W^\rho, V^\rho)$ and another random variable that is a.s. bounded polynomially in other parameters. As that the coupled random variables take different values is exponentially small while the number of iterations in our algorithm is only polynomially large, without any concern we instead work with this new random varaible, and with an overload of notation we also denote it by $(W^\rho, V^\rho)$.

**Lemma 43** *For an arbitrary parameter $\chi >> 1$, On any pair for $(W', V')$ with $\|W'\| \le C_1$, $\|V'\| \le C_2$, there exist a mean zero random vector $\bar{\Lambda}$ with respect to the randomness of the uniformly picked data point $(x_i, y_i)$ and the smoothing matrices $W^{\rho,1}$, $W^{\rho,2}$, $V^{\rho,1}$, and $V^{\rho,2}$ which define $\hat{\nabla}_{W',V'}$ (meaning it is a function of those variables), such that with probability at least*

$$1 - 2\exp\{-(\chi^2 - 1)dm_1/4\} - 2\exp\{-(\chi^2 - 1)m_3 m_2/4\} := 1 - \delta_1,$$

*we have*

$$\hat{\nabla}_{W',V'} = \nabla_{W',V'}L(W', V') + \bar{\Lambda}, \tag{305}$$

*and finally $\bar{\Lambda}$ is a.s. polynomially bounded, i.e. almost surely we have*

$$\|\bar{\Lambda}\| \le poly(m_1, m_2, m_3, C_1, C_2, B, \chi).$$

**Proof of Lemma 43**

Remember that $x_i'$ was the output of the first layer (by considering the smoothing matrix $W^\rho$). Now with high probability over the initialization,

$$\|\nabla_{W'} f_{W'+W^\rho, V'+V^\rho}(x_i)\|_F = \|\nabla_{x_i'} f_{W'+W^\rho, V'+V^\rho}(x_i)^T \frac{D(x_i')}{dW'}\|$$

$$\|\nabla_{x_i'} f_{W'+W^\rho, V'+V^\rho}(x_i)\| \|\frac{D(x_i')}{dW'}\|$$

$$\|\frac{1}{\sqrt{m_2}} a^T D_{V'+V^\rho}(V^{(0)} + V' + V^\rho)\| \|\frac{1}{\sqrt{m_1}} \sum_{k=1}^{m_3} \text{diag}(W_k^s) D_{W'+W^\rho, x_i} x_i^T\|_F$$

$$\le (\|V^{(0)}\|_F + \|V'\|_F + \|V^\rho\|_F)\Big(\frac{1}{\sqrt{m_1}} \sum_k \|\text{diag}(W_k^s) D_{W'+W^\rho, x_i} x_i^T\|_F\Big)$$

$$\le (\kappa_2 \sqrt{m_2 m_3} + C_2 + \|V^\rho\|_F)\Big(\frac{1}{\sqrt{m_1}} \sum_k \|\text{diag}(W_k^s) D_{W'+W^\rho, x_i} x_i^T\|_F\Big)$$

$$\le (\kappa_2 \sqrt{m_2 m_3} + C_2 + \|V^\rho\|_F). \tag{306}$$

On the other hand, using the final bound in Lemma 33:

$$\|\nabla_{V'} f_{W'+W^\rho, V'+V^\rho}(x_i)\|_F = \|\frac{1}{\sqrt{m_2}} \text{diag}(a) D_{V'+V^\rho} x_i'^T\|_F \le \|x_i'\|$$

$$\le \kappa_1 \sqrt{m_3} + C_1 + \sqrt{m_3} \sum_j \frac{1}{\sqrt{m_1}} |W_j^\rho x_i|$$

$$\le \kappa_1 \sqrt{m_3} + C_1 + \sqrt{m_3} \|W^\rho\|_F. \tag{307}$$

Denoting $\dot{\ell}(f_{W'+W^{\rho,1}, V'+V^{\rho,1}}, y_i)\nabla_{W',V'}\ell(f_{W'+W^{\rho,2}, V'+V^{\rho,2}}(x_i), y_i)$ by $\tilde{\nabla}_{W',V'}$, then combining Equations (306) and (307) and using the 1 gradient lipschitzness property of the square loss,

$$\|\tilde{\nabla}_{W',V'}\|_F$$

$$= |\frac{d(\ell(f, y_i))}{df}|\sqrt{\|\nabla_{W'} f_{W'+W^{\rho,2}, V'+V^{\rho,2}}(x_i)\|_F^2 + \|\nabla_{V'} f_{W'+W^{\rho,2}, V'+V^{\rho,2}}(x_i)\|_F^2}$$

$$\le \Big(|f_{W'+W^{\rho,1}, V'+V^{\rho,1}}(x_i)| + |B|\Big)\Big(\kappa_1 \sqrt{m_3} + C_1 + \sqrt{m_3}\|W^{\rho,2}\|_F + \kappa_2 \sqrt{m_2 m_3} + C_2 + \|V^{\rho,2}\|_F\Big).$$

$$\tag{308}$$

Finally, applying Cauchy-Swartz to the second a.s. bound in Lemma 34 we have:

$$\left| f_{W'+W^\rho, V'+V^\rho}(x_i) \right|$$
$$\leq (\kappa_2\sqrt{m_3} + \|V^\rho\|_F)\left(\sqrt{m_3}\kappa_1 + C_1 + \sqrt{m_3}\|W^\rho\|\right) + C_2(C_1 + \sqrt{m_3}\|W^\rho\|).$$

Combining this with (308):

$$\|\tilde{\nabla}_{W',V'}\|_F$$
$$\leq \left[ B + (\kappa_2\sqrt{m_3} + \|V^{\rho,1}\|_F)\left(\sqrt{m_3}\kappa_1 + C_1 + \sqrt{m_3}\|W^{\rho,1}\|\right) + C_2(C_1 + \sqrt{m_3}\|W^{\rho,1}\|) \right]$$
$$\times \left( \kappa_1\sqrt{m_3} + C_1 + \sqrt{m_3}\|W^{\rho,2}\|_F + \kappa_2\sqrt{m_2 m_3} + C_2 + \|V^{\rho,2}\|_F \right).$$

Therefore, using the Lipschitz bound in Theorem 9:

$$\|\tilde{\nabla}_{W',V'} - \nabla_{W',V'}\mathbb{E}_{(x_i,y_i)\sim\mathcal{Z}}\ell(f'_{W',V'}(x_i), y_i)\|_F$$
$$\leq \|\hat{\nabla}_{W',V'}\|_F + \|\mathbb{E}_{(x_i,y_i)\sim\mathcal{Z}}\nabla_{W',V'}\ell(f_{W'+W^\rho,V'+V^\rho}(x_i), y_i)\|_F$$
$$\leq \left[ B + (\kappa_2\sqrt{m_3} + \|V^{\rho,1}\|_F)\left(\sqrt{m_3}\kappa_1 + C_1 + \sqrt{m_3}\|W^{\rho,1}\|\right) + C_2(C_1 + \sqrt{m_3}\|W^{\rho,1}\|) \right]$$
$$\times \left( \kappa_1\sqrt{m_3} + C_1 + \sqrt{m_3}\|W^{\rho,2}\|_F + \kappa_2\sqrt{m_2 m_3} + C_2 + \|V^{\rho,2}\|_F \right) + (O(C_1 C_2) + B)\Psi_1. \tag{309}$$

Now we define the following events

$$\Xi_1 := \{\|W^{\rho_1}\|_F \geq \chi\sqrt{d}\beta_1 \vee \|W^{\rho_2}\|_F \geq \chi\sqrt{d}\beta_1\},$$
$$\Xi_2 := \{\|V^{\rho_1}\|_F \geq \chi\sqrt{m_3}\beta_2 \vee \|V^{\rho_2}\|_F \geq \chi\sqrt{m_3}\beta_2\},$$

where recall we assume $\chi >> 1$. Then, as we know the variable $\|W^\rho\|_F^2$ has mean $d\beta_1^2$ and is subexponential with parameters $(d\beta_1^4/m_1, \beta_1^2/m_1)$. Hence, by a union bound and Bernstein (Note that $W^{\rho,1}, W^{\rho,2}$ are independent):

$$\mathbb{P}(\Xi_1)$$
$$\leq 2\mathbb{P}(\|W^\rho\|_F \geq \chi\beta_1\sqrt{d})$$
$$= 2\mathbb{P}(\|W^\rho\|_F^2 \geq \chi^2\beta_1^2 d)$$
$$\leq 4\max\left( \exp\{-(\chi^2-1)^2\beta_1^4 d^2/(4d\beta_1^4/m_1)\}, \exp\{-(\chi^2-1)\beta_1^2 d/(4\beta_1^2/m_1)\} \right)$$
$$= 4\max\left( \exp\{-(\chi^2-1)^2 dm_1/4\}, 2\exp\{-(\chi^2-1)dm_1/4\} \right) \leq 2\exp\{-(\chi^2-1)dm_1/4\}.$$

Similarly for $\|V^\rho\|_F$:

$$\mathbb{P}(\Xi_2) = \mathbb{P}(\|V^\rho\|_F \geq \chi\beta_2\sqrt{m_3}) \leq 4\exp\{-(\chi^2-1)m_3 m_2/4\}.$$

Moreover, because of the subexponential tails of $\|W^\rho\|_F^2$ and $\|V^\rho\|_F^2$, for each of $W^{\rho,1}$ or $W^{\rho_2}$, $V^{\rho_1}$, or $V^{\rho_2}$:

$$\mathbb{E}(\|W^\rho\|_F| \Xi_1) \lesssim \chi\sqrt{d}\beta_1, \quad \mathbb{E}(\|W^\rho\|_F^2| \Xi_1) \lesssim \chi^2 d\beta_1^2.$$
$$\mathbb{E}(\|V^\rho\|_F| \Xi_2) \lesssim \chi\sqrt{m_3}\beta_2, \quad \mathbb{E}(\|V^\rho\|_F^2| \Xi_2) \lesssim \chi^2 m_3\beta_2^2.$$

Now Defining $\Xi = \Xi_1 \cup \Xi_2$ and combining the above equations:

$$\mathbb{E}(\|W^\rho\|\mathbb{1}\{\Xi\}) \leq \mathbb{E}\|W^\rho\|(\mathbb{1}\{\Xi_1\} + \mathbb{1}\{\Xi_2\}) = \mathbb{E}(\|W^\rho\|| \Xi_1)\mathbb{P}(\Xi_1) + \mathbb{E}(\|W^\rho\|)\mathbb{P}(\Xi_2)$$
$$\lesssim \chi\sqrt{d}\beta_1 2\exp\{-(\chi^2-1)dm_1/4\} + \sqrt{d}\beta_1 2\exp\{-(\chi^2-1)m_3 m_2/4\}$$
$$= 2\sqrt{d}\beta_1(\chi\exp\{-(\chi^2-1)dm_1/4\} + \exp\{-(\chi^2-1)m_3 m_2/4\}),$$

and

$$\mathbb{E}\|W^\rho\|^2 \mathbb{1}\{\Xi_1\} = \mathbb{E}(\|W^\rho\|^2| \Xi_1)\mathbb{P}(\Xi_1)$$
$$\leq 2d\beta_1^2\chi^2 \exp\{-(\chi^2-1)dm_1/4\}.$$

Similarly

$$\mathbb{E}(\|V^\rho\|\mathbb{1}\{\Xi\}) \leq 2\sqrt{m_3}\beta_2(\exp\{-(\chi^2-1)dm_1/4\} + \chi\exp\{-(\chi^2-1)m_3m_2/4\}),$$

and

$$\mathbb{E}(\|V^\rho\|^2\mathbb{1}\{\Xi_2\}) \leq 2m_3\beta_2^2\chi^2 \exp\{-(\chi^2-1)m_3m_2/4\}.$$

Applying these equations to (309) with Cauchy-Schwartz to get the upper bounds $\mathbb{E}\mathbb{1}\{\Xi_1\}\|W^{\rho,1}\|\|W^{\rho,2}\| \leq \mathbb{E}\mathbb{1}\{\Xi_1\}\|W^\rho\|^2$ and $(\mathbb{E}_{W^\rho}\|W^\rho\|)^2 \leq \mathbb{E}_{W^\rho}\|W^\rho\|^2$ (for terms with only one $W^{\rho,i}$ or $V^{\rho,i}$, we simply write them as $W^\rho$ and $V^\rho$):

$$\mathbb{E}_{W^\rho,V^\rho}\|\tilde{\nabla}_{W',V'} - \nabla_{W',V'}\mathbb{E}_{(x_i,y_i)\sim\mathcal{Z}}\ell(f'_{W',V'}(x_i),y_i)\|_F \mathbb{1}\{\Xi\}$$

$$\leq \mathbb{E}_{W^\rho,V^\rho,(x_i,y_i)}\Big[B + (\kappa_2\sqrt{m_3} + \|V^{\rho,1}\|_F)\big(\sqrt{m_3}\kappa_1 + C_1 + \sqrt{m_3}\|W^\rho\|\big) + C_2(C_1 + \sqrt{m_3}\|W^\rho\|)\Big]$$

$$\times \Big(\kappa_1\sqrt{m_3} + C_1 + \sqrt{m_3}\|W^\rho\|_F + \kappa_2\sqrt{m_2 m_3} + C_2 + \|V^\rho\|_F\Big) + \alpha(O(C_1C_2) + B)\Psi_1$$

$$= \Big[B + (\kappa_2\sqrt{m_3})(\sqrt{m_3}\kappa_1 + C_1) + C_1C_2\Big]$$

$$\times \Big(\kappa_1\sqrt{m_3} + C_1 + \kappa_2\sqrt{m_2 m_3} + C_2 + \sqrt{m_3}\mathbb{E}_{W^\rho}\mathbb{1}\{\Xi\}\|W^\rho\|_F + \mathbb{E}_{V^\rho}\mathbb{1}\{\Xi\}\|V^\rho\|_F\Big)$$

$$+ \Big(B + (\sqrt{m_3}\kappa_1 + C_1)\sqrt{m_3} + (\kappa_2 m_3 + C_2\sqrt{m_3}) + \sqrt{m_3}(\kappa_1\sqrt{m_3} + C_1 + \kappa_2\sqrt{m_2 m_3} + C_2)\Big)$$

$$\times (\mathbb{E}_{W^\rho}\mathbb{1}\{\Xi_1\}\|W^\rho\|_F\mathbb{E}_{V^\rho}\|V^\rho\|_F + \mathbb{E}_{W^\rho}\|W^\rho\|_F\mathbb{E}_{V^\rho}\mathbb{1}\{\Xi_2\}\|V^\rho\|_F)$$

$$+ (B + \sqrt{m_3}\kappa_1 + C_1)(E_{V^\rho}\mathbb{1}\{\Xi_2\}\|V^\rho\|^2 + \mathbb{P}(\Xi_1)\mathbb{E}\|V^\rho\|^2)$$

$$+ C_2 m_3(\mathbb{E}_{W^\rho}\mathbb{1}\{\Xi_1\}\|W^\rho\|^2 + \mathbb{P}(\Xi_2)\mathbb{E}\|W^\rho\|^2)$$

$$+ m_3(\mathbb{E}_{W^\rho}\mathbb{1}\{\Xi_1\}\|W^\rho\|^2\mathbb{E}_{V^\rho}\|V^\rho\| + \mathbb{E}_{W^\rho}\|W^\rho\|^2\mathbb{E}_{V^\rho}\mathbb{1}\{\Xi_2\}\|V^\rho\|)$$

$$+ \sqrt{m_3}(\mathbb{E}_{V^\rho}\mathbb{1}\{\Xi_2\}\|V^\rho\|^2\mathbb{E}_{W^\rho}\|W^\rho\| + \mathbb{E}_{V^\rho}\|V^\rho\|^2\mathbb{E}_{W^\rho}\mathbb{1}\{\Xi_1\}\|W^\rho\|)$$

$$+ (O(C_1C_2) + B)\Psi_1\mathbb{P}(\Xi)$$

$$\leq (\exp\{-(\chi^2-1)dm_1/4\} + \chi\exp\{-(\chi^2-1)m_3m_2/4\})\text{poly}(m_1,m_2,m_3) = \text{negligible}. \tag{310}$$

But note that

$$\hat{\nabla}_{W',V'} = \tilde{\nabla}_{W',V'} + \nabla_{W',V'}(\psi_1\|W'\|^2 + \psi_2\|V'\|^2),$$
$$\nabla_{W',V'}L(W',V') = \nabla_{W',V'}\mathbb{E}_{(x_i,y_i)\sim\mathcal{Z}}\ell(f'_{W',V'}(x_i),y_i) + \nabla_{W',V'}(\psi_1\|W'\|^2 + \psi_2\|V'\|^2).$$

Applying this to Equation (310), we get that if we define

$$\Lambda := \hat{\nabla}_{W',V'} - \nabla_{W',V'}L(W',V'),$$

then

$$\mathbb{E}\|\Lambda\mathbb{1}\{\Xi\}\| \leq (\exp\{-(\chi^2-1)dm_1/4\} + \chi\exp\{-(\chi^2-1)m_3m_2/4\})\text{poly}(m_1,m_2,m_3).$$

On the other hand, note that using again Equation (309), we have the following a.s. bound:

$$\|\Lambda\mathbb{1}\{\Xi^c\}\| = \text{poly}(m_1,m_2,m_3,C_1,C_2,\chi).$$

Defining

$$\Lambda_1 = \Lambda\mathbb{1}\{\Xi^c\},$$
$$\Lambda_2 = \mathbb{1}\{\Xi\}\mathbb{E}(\Lambda|\Xi),$$
$$\bar{\Lambda} = \Lambda_1 + \Lambda_2,$$

we get that with probability at least $1 - \mathbb{P}(\Xi)$:

$$\hat{\nabla}_{W',V'} = \nabla_{W',V'}L(W',V') + \bar{\Lambda}$$

and also note that

$$\mathbb{E}\bar{\Lambda} = \mathbb{E}\Lambda = 0.$$

Finally by the a.s. bound for $\Lambda_1$, we have a.s.:

$\|\bar{\Lambda}\| \leq \mathbb{E}_{W^\rho, V^\rho, (x_i, y_i)}\|\Lambda \mathbb{1}\{\Xi\}\| + \|\Lambda \mathbb{1}\{\Xi^c\}\|$

$\leq (\exp\{-(\chi^2 - 1)dm_1/4\} + \chi \exp\{-(\chi^2 - 1)m_3 m_2/4\})\mathrm{poly}(m_1, m_2, m_3) + \mathrm{poly}(m_1, m_2, m_3)$

$= \mathrm{poly}(m_1, m_2, m_3, C_1, C_2, B, \chi),$

which completes the proof.

**Corollary 9.1** *It is easy to check that running* PSGD *with unbiased gradient estimate* $\hat{\nabla}_{W',V'}$ *is equivalent to running SGD after our change of coordinates, with unbiased gradient estimate* $\hat{\nabla}_{w',v'} := \bar{\Upsilon}\nabla_{W',V'}$, *where* $\bar{\Upsilon}$ *is the matrix for our change of coordinate, which is equal to* $\Upsilon$ *defined in Appendix A.10 for the coordinates in* $V'$ *and simply identity for the coordinates in* $W'$. *Therefore, projecting both sides in Equation (311) of Lemma 43 onto* $\Phi^\perp$ *by multiplying* $\Upsilon$ *implies that with high probability for all iterations of the algorithm*

$$\hat{\nabla}_{w',v'} = \bar{\Upsilon}\nabla_{W',V'}L^\Pi(W',V') + \bar{\Upsilon}\bar{\Lambda}$$
$$= \nabla_{w',v'}L^\Pi(w',v') + \bar{\Upsilon}\bar{\Lambda}, \tag{311}$$

*where* $\bar{\mathcal{L}} := \bar{\Upsilon}\bar{\Lambda}$ *(using the properties of* $\bar{\Lambda}$ *in Lemma 43) is a mean zero noise vector with almost surely bounded norm, i.e.* $\|\bar{\mathcal{L}}\| \leq Q'$ *for some* $Q' = poly(m_1, m_2, m_3, C_1, C_2)$. *(we dropped the* $\chi$ *parameters by considering constant high probability argument).*

*Finally, note that injecting noise* $(\Xi_1/\|\Xi_1\|, \Xi_2/(\sqrt{m_1}\|\Xi_2\|))$ *by* PSGD *results in adding an extra zero mean noise* $(\tilde{\Xi}_1, \tilde{\Xi}_2) := (\bar{\Upsilon}\Xi_1/\|\Xi_1\|, \bar{\Upsilon}\Xi_2/(\sqrt{m_1}\|\Xi_2\|))$ *to the gradient* $\nabla_{w',v'}L^\Pi(w',v')$. *Therefore, overall running SGD on* $L^\Pi$ *(which is equivalent to* PSGD *on L) observe an unbiased noise vector defined as* $\mathcal{L} := \bar{\mathcal{L}} + (\tilde{\Xi}_1, \tilde{\Xi}_2)$. *Now it is easy to check that the moment matrix of* $\tilde{\Xi}_1$ *and* $\tilde{\Xi}_2$ *are* $\sigma_1'^2 I$ *and* $\sigma_2'^2 I$ *for*

$$\sigma_1'^2 := \frac{1}{m_1^2 d}, \tag{312}$$

$$\sigma_2'^2 := \frac{m_2(m_3 - n)}{m_2^m 3}, \tag{313}$$

*which implies the moment matrix of* $\mathcal{L}$ *is upper bounded by*

$$\sigma_2^2 I := (Q'/(m_2 m_3 + m_1 d) + \max\{\sigma_1'^2, \sigma_2'^2\})I,$$

*and lower bounded by*

$$\sigma_2^2 I := \min\{\sigma_1'^2, \sigma_2'^2\}I,$$

*i.e.*

$$\sigma_1^2 I \mathbb{E}\mathcal{L}\mathcal{L}^T \leq \sigma_2^2 I.$$

*(Note that we look at the new coordinates* $(w', v')$ *as a vectors, so the term* $\mathbb{E}\mathcal{L}\mathcal{L}^T$ *makes sense.)*

*Moreover,* $\|\tilde{\Xi}_1\| = 1/\sqrt{m_1}, \|\tilde{\Xi}_2\| \leq 1$ *almost surely, which implies the following almost surely bound for* $\mathcal{L}$:

$$\|\mathcal{L}\| \leq Q := Q' + 1 + 1/\sqrt{m_1}.$$

**Lemma 44** *Let $g(x)$ be a second order differentiable function over $\mathbb{R}^N$ such that at point $x$, there exist a random direction $y$ and deterministic direction $z$ and fixed positive real $r$ with:*

$$\mathbb{E}y = 0,$$
$$\mathbb{E}_y g(x + \eta z + \sqrt{\eta}y) \leq g(x) - \eta r.$$

*Then, for the gradient and Hessian at point $x$, we have either*

$$\|\nabla g(x)\| \geq \frac{r}{4\|z\|},$$

*or*

$$\lambda_{min}\left(\nabla^2 g(x)\right) \leq -\frac{r}{2\|y\|^2}.$$

**Proof of Lemma 44**

We write the second order tailor approximation of $g$ around $x$:

$$g(x + w) = g(x) + \nabla g(x)^T w + \frac{1}{2}w^T \nabla^2 g(x)w + o(\|w\|^2).$$

Now substituting $w$ with $\eta z + \sqrt{\eta}y$ and taking expectation with respect to $y$, as we send $\eta \to 0$ and using the fact that $\mathbb{E}y = 0$:

$$\mathbb{E}_y g(x + \eta z + \sqrt{\eta}y) = g(x) + \mathbb{E}_y \nabla g(x)^T(\eta z + \sqrt{\eta}y) + \frac{1}{2}(\eta z + \sqrt{\eta}y)^T \nabla^2 g(x)(\eta z + \sqrt{\eta}y) + o(\|\eta z + \sqrt{\eta}y\|^2)$$

$$= \mathbb{E}_y g(x) + \eta \nabla g(x)^T z + \frac{1}{2}\eta^2 z^T \nabla^2 g(x)z + \eta\frac{1}{2}y^T \nabla^2 g(x)y + o(\eta\|y\|^2)$$

$$= \mathbb{E}_y g(x) + \eta \nabla g(x)^T z + \eta\frac{1}{2}y^T \nabla^2 g(x)y + o(\eta).$$

Combining the assumption with the above Equation, we get that for small enough $\eta$, we have

$$\eta \nabla g(x)^T z + \eta\frac{1}{2}\mathbb{E}_y y^T \nabla^2 g(x)y \leq -\eta r/2,$$

i.e.

$$\nabla g(x)^T z + \frac{1}{2}\mathbb{E}_y y^T \nabla^2 g(x)y \leq -r/2,$$

which means we should either have

$$\nabla g(x)^T z \leq -r/4,$$

which implies

$$\|\nabla g(x)\| \geq \frac{r}{4\|z\|},$$

or

$$\mathbb{E}_y y^T \nabla^2 g(x)y \leq -r/2,$$

which implies

$$\lambda_{min}\left(\nabla^2 g(x)\right) \leq -\frac{r}{2\max_{\tilde{y}\in\text{support}(y)}\|\tilde{y}\|^2}.$$

### B.4 HANDLING THE INJECTED NOISE BY PSGD

In this section, we prove that having SGD injecting noise into our gradient estimates mostly does not change the sign pattern of the first layer, namely among the set of rows in $P$ defined in Lemma 1.

**Lemma 45** *Having enough overparameterization, with high probability, at every iteration of the PSGD for $W'^{(2)}$ defined in the proof of Lemma 1, we have for every $j \in [m_1]$:*

$$\|W'^{(2)}_j\| \le c_2/(4\sqrt{m_1}).$$

**Proof of Lemma 45**

Let $\Phi'$ be the subspace of the first layer weight matrices which is zero in rows $j \in P$ ($P$ is defined in Lemma 1), while in other rows it is the span of $Z_i^k$'s, i.e. using our notation $\tilde{Z}_k^i$ introduced in the proof of Lemma 1, we can write $\Phi'$ is span$(Z_i^k)_{i,k}$.

Recall from Lemma 1 that we decompose the first layer weight $W'$ as $W'^{(1)} + W'^{(2)}$, namely the parts in the subspace $\Phi'$ and subspace $\Phi'^{\perp}$ respectively. Moreover, let $\Xi_1/(\sqrt{m_1}\|\Xi_1\|) = \Xi^{(1)} + \Xi^{(2)}$ be the decomposition of the injected noise at some iteration of PSGD into subspaces $\Phi'$ and $\Phi'^{\perp}$ respectively.

Now recall that the current $W'$ is the value of the previous iteration moved by the gradient plus the injected noise:

$$
\begin{aligned}
W' &= W' - \eta(\hat{\nabla}_{W'} + \Xi^{(1)} + \Xi^{(2)}) \\
&= W' - \eta\Big(\tilde{\nabla}_{W'} + 2\psi_1 W'^{-,(1)} + 2\psi_1 W'^{-,(2)} + \Xi^{(1)} + \Xi^{(2)}\Big),
\end{aligned}
$$

where $W'$ is the weight of the previous iteration and $W'^{-,(1)}, W'^{-,(2)}$ are again its decomposition to $\Phi'$ and $\Phi'^{\perp}$, where $\tilde{\nabla}_{W',V'}$ is defined in Lemma 43. Applying Lemma 24 for the previous iteration of the algorithm, we get $\tilde{\nabla}_{W'} \in \Phi'$ since the bad events $E$ defined in Lemma 33 occurs only with probability exponentially small (hence union bound across all the iterations rules it out). Hence, the decomposition for the current iteration becomes

$$W'^{(1)} = W'^{-,(1)} - \eta(\hat{\nabla}_{W'} + 2\psi_1 W'^{-,(1)} + \Xi^{(1)}), \tag{314}$$

$$W'^{(2)} = (1 - 2\eta\psi_1)W'^{-,(2)} + \eta\Xi^{(2)}. \tag{315}$$

We handle the $W'^{(1)}$ part in Lemma 1 and prove that as long as $\|W'^{(1)}\|^2 \le \|W'\|^2$ remains bounded by $C_1^2$, then the sign pattern of the first layer, when only considering the $W'^{(1)}$ part, is specified by the initialization except within set $P$; here we handle the $W'^{(2)}$ part as well.

Note that for every row $j \in [m_1]$, the variable $\|\big(\Xi_1/(\sqrt{m_1}\|\Xi_1\|)\big)_j\|^2$ is $(O(1/(m_1^4 d)), O(1/(m_1^2 d)))$-subexponential with mean $1/m_1$. Therefore, with probability that is exponentially small in $m_1$, $\|\big(\Xi_1/(\sqrt{m_1}\|\Xi_1\|)\big)_j\|$ is bounded by $O(1/m_1)$. It is not hard to see the same argument holds for the projection of $\Xi_1/(\sqrt{m_1}\|\Xi_1\|)$ onto $\Phi^{\perp}$, i.e. $\Xi^{(2)}$. Applying a union bound for all iterations, again using the fact that we run PSGD for poly iterations while the chance of error is exponentially small in $m_1$, we can then argue that with high probability over the noise of gradients, at every iteration and for every $j \in [m_1]$:

$$\|\Xi_j^{(2)}\| = \tilde{O}(1/m_1). \tag{316}$$

But applying trinagle inequality to Equation (315) and writing it in a telescope form, particularly for the $j$th row, and further using the assumption in 316, we get that $\|W'^{(2)}_j\|$ grows at most to $O(1/(m_1\psi_1))$; as we set $1/\psi_1 = O(poly(n))$, assuming polynomially large enough $m_1$ concludes the claim.

### B.4.1 BOUNDING THE NORM OF THE FIRST LAYER'S OUTPUT IN THE WORST CASE

**Lemma 46** *Suppose $W'$ satisfies the assumption of Lemma 1, i.e. $\|W'\| \le C_1$, and $\|W'_j\| \le c_2/(2\sqrt{m_1})$ except possible for indices in $P$, also defined in 1. Then, with high probability over initialization*

$$\sup_{x, \|x\|=1} \|\phi'(x)\| \lesssim (1 + O(m_3^2 d^2 \log(m_1)^2/m_1))C_1 + \sqrt{m_3} \frac{C_1^{3/2}}{\sqrt{\kappa_1}} \left(\frac{n^3 m_3}{m_1 \lambda_0}\right)^{1/4},$$

*which is $O(C_1)$ for large enough overparameterization.*

**Proof of Lemma 46**

Note the because the VC-dimension of the class of binary functions with respect to halfspaces in $\mathbb{R}^d$ is $d+1$, the number of different sign patterns $D_{W^{(0)},x}$ for different $x$ can be at most $m_1^{d+1}$. Now similar to Equation (317), for $k \in [m_3]$ define

$$Z_k(x) = 1/\sqrt{m_1}\left(W_{k,j}^s \mathbb{1}\{W_j^{(0)T}x\}x\right)_{j=1}^{m_1}. \tag{317}$$

Then, for $k_1 \ne k_2$, as $\|x\| = 1$:

$$\langle Z_{k_1}(x), Z_{k_2}(x)\rangle = \frac{1}{m_1} \sum_{j=1}^{m_1} W_{k_1,j}^s W_{k_2,j}^s \mathbb{1}\{W_j^{(0)T}x\}$$

$$\le \frac{1}{m_1} \sup_x \sum_{j=1}^{m_1} D_{W^{(0)},xj,j} W_{k_1,j}^s W_{k_2,j}^s.$$

But for each fixed $D_{W^{(0)},x}$, using Hoeffding bound, we have with probability $1 - \delta$:

$$\frac{1}{\sqrt{m_1}} \sum_{j=1}^{m_1} D_{W^{(0)},xj,j} W_{k_1,j}^s W_{k_2,j}^s \lesssim \sqrt{\frac{\log(1/\delta)}{m_1}}.$$

Applying the above for all possible sign patterns with $\delta < O(1/m_1^{d+1})$ and a union bound, we have with high probability

$$\sup_{x, \|x\|=1} \langle Z_{k_1}(x), Z_{k_2}(x)\rangle \le \frac{1}{m_1} \sup_x \sum_{j=1}^{m_1} D_{W^{(0)},xj,j} W_{k_1,j}^s W_{k_2,j}^s \lesssim d\log(m_1)/\sqrt{m_1}.$$

We can even state the following stronger bound with respect to two adversarially picked vectors $x, x'$:

$$\sup_{\|x\|=1, \|x'\|=1} \langle Z_{k_1}(x), Z_{k_2}(x')\rangle \le \frac{1}{m_1} \sup_x \sum_{j=1}^{m_1} D_{W^{(0)},xj,j} D_{W^{(0)},x'j,j} W_{k_1,j}^s W_{k_2,j}^s \lesssim d\log(m_1)/\sqrt{m_1}, \tag{318}$$

because each $D_{W^{(0)},x'j,j}$ has at most $m_1^{d+1}$ cases as we discussed above, then $D_{W^{(0)},xj,j} D_{W^{(0)},x'j,j}$ has at most $m_1^{d+1}$ possible cases, and applying a similar Hoeffding bound for each of them and a union bound as we did will imply (318). We will use this generalized version in another section.

Now combining Equation (318) with the fact that $\|W'\| \le C_1$ and applying Lemma 40:

$$\sup_{x, \|x\|=1} \sum_{k=1}^{m_3} \langle W', Z_k(x)\rangle^2 \le (1 + O(m_3^2 d^2 \log(m_1)^2/m_1))C_1^2. \tag{319}$$

On the other hand, setting $m_2 = m_1$, $m_3 = d$, and $R = c_2/(2\sqrt{m_1}\kappa_1)$ in Lemma 29, we get with high probability

$$\#\left(j \in [m]: |V_j^{(0)}x| \le c_2/(2\sqrt{m_1})\right) \le m_1 c_2/(2\sqrt{m_1}) = \sqrt{m_1}c_2/(2\kappa_1).$$

Noting that $\|W'_j\| \le c_2/(2\sqrt{m_1})$ for $j \notin P$, we conclude that with high probability, for any $x$, $\mathbb{1}\{(W^{(0)} + W')_j^T x \ge 0\}$ and $\mathbb{1}\{W_j^{(0)T} x \ge 0\}$ can be different in at most $\sqrt{m_1}c_2/(2\kappa_1)$ of the $j$'s outside of $[m_1] \setminus P$. Therefore, as we have also $|P| \lesssim nc_2\sqrt{m_1}/\kappa_1$ from Lemma 1, we conclude that with high probability, for any $x$, there are at most $O(nc_2\sqrt{m_1}/\kappa_1)$ sign changes by adding $W'$ to $W^{(0)}$. This further implies:

$$|\phi'_k(x) - \langle W', Z_k(x) \rangle| \le 2/\sqrt{m_1} \sum_{j:\, \text{Sgn}(W_j^{(0)T}x) \ne \text{Sgn}((W^{(0)}+W')_j^T x)} |W'_j x|$$

$$\le \|W'\| 2 \sqrt{\left| \{j|\, \text{Sgn}(W_j^{(0)T}x) \ne \text{Sgn}((W^{(0)} + W')_j^T x)\} \right|} / \sqrt{m_1}$$

$$\lesssim C_1 \sqrt{nc_2\sqrt{m_1}/\kappa_1} / \sqrt{m_1}$$

$$= \frac{C_1^{3/2}}{\sqrt{\kappa_1}} \left( \frac{n^3 m_3}{m_1 \lambda_0} \right)^{1/4}.$$

Combining this with (319), we conclude with high probability:

$$\sup_{x, \|x\|=1} \|\phi'(x)\| \lesssim (1 + O(m_3^2 d^2 \log(m_1)^2/m_1))C_1 + \sqrt{m_3} \frac{C_1^{3/2}}{\sqrt{\kappa_1}} \left( \frac{n^3 m_3}{m_1 \lambda_0} \right)^{1/4},$$

which completes the proof.

