# OpenReview forum: "Optimization and Adaptive Generalization of Three layer Neural Networks"
_ICLR.cc/2022/Conference — ICLR 2022 Poster_

### Official Review · Reviewer_aHSb · 2021-10-26

**Correctness:** 3
**Technical Novelty And Significance:** 4
**Empirical Novelty And Significance:** 1
**Recommendation:** 8
**Confidence:** 3

**Main Review:**

The theory of NTK provides a beautiful theoretical tools to analyze the convergence of generalization of neural network models in highly over-parameterized regime. Yet, essentially a linear analysis, NTK cannot characterize feature learning of neural networks, and practice neural networks work out of the NTK regime. Hence, theoretical study of neural network dynamics beyond the NTK regime is crucial for the understanding of the success of neural networks. This work tries to explore the beyond-NTK regime in a special case. Though both the model and the algorithm are specifically designed for theoretical analysis, the work done in this paper is still meaningful, because it provides an approach to analyze the feature learning effect of neural networks.

The major comment from the reviewer is about the comparison of the setting with NTK. The reviewer hope the authors can make it clearer why the studied setting is different from NTK. Is it because the neural network is less wide than required by NTK, or because the network structure makes go beyond NTK, or because the project SGD algorithm prevent the network from entering the NTK regime?

**Summary Of The Paper:**

This paper studies the behavior of a three-layer neural network trained by a projected SGD algorithm. The authors theoretically show that, in this setting the neural network has more complicated behavior than in the NTK regime. Instead of sticking at one kernel, now the neural network explores different kernels decided by the features learned by the first layer. Therefore, the generalization error is bounded by a minimum of many RKHS norms, and is proven to be smaller than the single NTK norm. The proposed generalization error bound outperforms the NTK bound. For target functions in the NTK, the proposed bound is no bigger than the NTK bound. The model and the algorithm are close to commonly used ones, while there are components specially designed for the theoretical analysis.

**Summary Of The Review:**

The paper studies the optimization and generalization of a three-layer neural network trained by a projected SGD algorithm. In this setting, the network runs beyond the NTK regime. Hence, it can achieve feature learning, and better generalization performance than NTK. Specifically, the generalization error is bounded by a complexity depending on the minimum of a family of RKHS norms. This paper explores the behavior of neural networks out of the NTK regime, which is important due to 1) the existence of feature learning, and 2) closer settings to practice.

---

> ### Author Response · Authors · 2021-11-19
> **Difference between our regime and that of NTK**
>
> Regarding the difference between our regime and that of NTK:
> As the reviewer has alluded to, our regime is achieved by a combination of the particular choice of the network widths $m_1, m_2, m_3$ and initialization variances $\kappa_1, \kappa_2$ which should satisfy specific polynomial inequalities with respect to one  another, stated in Appendix A.3. as well as the constraint $V’_j \perp \phi^{(0)}(x_i)$ that the algorithm imposes by projecting out some random directions each time. We will elaborate more on this in the main paper.

---

### Official Review · Reviewer_nxBT · 2021-10-31

**Correctness:** 4
**Technical Novelty And Significance:** 4
**Empirical Novelty And Significance:** Not applicable
**Recommendation:** 8
**Confidence:** 1

**Main Review:**

The paper present quite an interesting analysis and packs a lot of material in 9 page of main text and 101 pages of Appendix material. While it is generally well written, it is not very accessible for a casual ICLR reader with no background in theoretical computer science. Since I'm not an expert in this, I'll defer to the other reviewers to check the significance and correctness of this paper. Below I provide only some superficial comments that hopefully help authors improve the paper a little.

$K^{\infty}$, $\tilde K^{\infty}$ is never properly defined and can be implied only from the context. Generally, the paper defines _quite a lot_ of notation during its length, I think it would benefit a lot from proper definition of the terms being used, at least in the Appendix

Features $g_k$ in section 2 appear without definition and it is not clear its relation with $G$. When it is used in eq.9 it is not clear what is the number of intermediate features and how it is defined.
nit: some of the acronyms might not be familiar to a typical ICLR reader
- a.s.
- ' symbols are used as vars (e.g. \phi'), but are confusing as they have same notation as a partial derivatives. This can be easily avoid by ~ or \hat symbols.
- population risk

**Summary Of The Paper:**

The paper presents a theoretical analysis of a 3 layer neural network with ReLU activations. The main result is introduction of adaptive generalization bounds that are defined for a modified SGD algorithm. These bounds are robust to noise and data dependent.

The idea is to define a product kernel for the second layer of the form $K^{\infty}\odot G$, where $$K^{\infty}$ is fixed and $G$ is adaptive and can be modified so that $g_k$ has minimum possible norms. This leads to a definition of a complexity measure that together with a simple modification to the SGD algorithm defines a bound on the population risk.

**Summary Of The Review:**

Theoretical paper defining a a novel bound for a 3-layer network. Since I'm not an expert in this field, I can provide only superficial comments to the paper.

---

> ### Author Response · Authors · 2021-11-19
> **Clarification in the revision**
>
> We thank the reviewer for their appreciation of the paper and for the comments on clarity; we will address them in the revision.
>
> Further clarification regarding the comments:
> $\tilde K^\infty$ is defined at the bottom of page 3 and $K^\infty$ is defined in Equation (8).

---

### Official Review · Reviewer_MU3r · 2021-11-04

**Correctness:** 4
**Technical Novelty And Significance:** 3
**Empirical Novelty And Significance:** Not applicable
**Recommendation:** 8
**Confidence:** 3

**Main Review:**

Main review:
1. Strength:
    1. The results obtained here, especially on the proposed function norm (minimum RKHS norm) is interesting. It would be better if the authors can provide further comments on the key difference between the data-dependent complexity measure bound and the function-norm based bound.
    2. The paper is well written. I appreciate the author’s efforts in making the presentation clear, consider the paper itself is fairly technical in nature.
    3. The discussion with related work is sufficient and difference with prior work are pointed out clearly, as section 3.3 and 3.4 the authors discuss in details how the results differs with kernel learning and NTK regime results.
2. Weakness:
    1. The training objective in (1) is clearly not the standard loss in practice, considering the additional $W^{(0)}, V^{(0)}, W^s$ that  are not optimized during training, the authors should comment more on why this additional terms are introduced and what’s their implication for analysis.
    2. The proposed algorithm has an additional projection step that makes $V_j’ \perp \phi^{(0)}(x_i)$, it is unclear to me why this projection step is necessary (is it essential in establishing convergence beyond NTK regime?). This technical requirement is not sufficiently motivated in the current version.



**Summary Of The Paper:**

This paper makes contributions into two main category:

1. Algorithmically, it propose a projected SGD algorithm that can, in polynomial number of iteration, reaches a solution that generalizes.
2. In terms of generalization, it proposes a new generalization bound with data-dependent complexity measure that goes beyond NTK regime. Another generalization bound, based on a new function norm (minimum RKHS norm w.r.t. a family of kernels) is proposed. The proposed bound shows *adaptivity* nature of the generalization results.



**Summary Of The Review:**

Summary of Review:
1. The paper makes a fairly interesting technical contribution: the proposed generalization bound poses adaptive nature, it improves the current kernel based bound (NTK) by selecting the best kernel that tradeoff data fitting and complexity.
2. The presentation is mostly clear, and the author clearly make efforts to compliment technical results with rich discussions. However, some technical conditions imposed in the loss and algorithm are not sufficiently motivated or explained.

---

> ### Author Response · Authors · 2021-11-19
> **Clarifying reviewer's points**
>
> Loss function and $W^{(0)}, V^{(0)}$:
> The loss is standard square loss, the difference is in how we state it. We explicitly decompose the weights into the (Gaussian) initialization $W^{(0)}, V^{(0)}$ and the difference $W’,V’$ from the initialization. The standard weights would be $W = W^{(0)}+W’, V = V^{(0)}+V’$.
>
>
> Role of $W^{s}$:
> The matrix $W^s$ acts as a random projection of the output of the first layer onto a low-dimensional subspace. As the reviewer has mentioned, we use $W^s$ in our analysis to achieve the adaptivity, we will elaborate more on the role of this random projection in the proof outline.
>
> Projection step:
> As we have explained in section 5 (High-level idea of the PSGD analysis), this assumption is important both in our Rademacher complexity analysis as well as the convergence of our algorithm. Intuitively, it prevents the network from overfitting and makes our generalization bounds possible. We will clarify that description and reemphasize it earlier in section 3 for the revised version.

---

### Official Review · Reviewer_NvLr · 2021-11-07

**Correctness:** 4
**Technical Novelty And Significance:** 4
**Empirical Novelty And Significance:** Not applicable
**Recommendation:** 6
**Confidence:** 3

**Main Review:**

The motivation of the paper is very clear and convincing. The analysis of neural networks using the NTK is known to have its limits, and the effort to provide generalization and optimization results beyond this regime is one of the main challenges of the theoretical machine learning community when studying deep learning. Furthermore, the techniques used in the paper can be of independent interest, and can benefit others trying to study optimization of neural networks. However, I have a few concerns regarding the results introduced in the paper.

The paper suggests three possible improvements on NTK analysis:
1. Faster rates, with 1/n dependence on the number of examples instead of 1/sqrt(n) as in e.g. Arora et al. 2019.
2. Relaxation of the realizability assumption that is sometimes made in NTK results, allowing for generalization bounds in the presence of noise.
3. A better complexity measure of the data/target, that allows the rate to be adaptive to some data-dependent complexity.

It seems to me that the first two improvements can be achieved by a more careful analysis of the NTK case (specifically, the authors acknowledge that using a smoothed loss the same fast-rate can be achieved in Arora et al 2019a). So, the main reason to go through the effort of analyzing the network beyond the NTK regime (and also analyzing a somewhat non-standard version of SGD), is to get the third point. Namely, the main contribution of the paper is in presenting a complexity measure that gives generalization bounds that are better than those that can be obtained by the complexity measure of the NTK (the RKHS norm of the NTK).

However, it is not clear to me why the new complexity measure offered in the paper improves over the NTK. Indeed, the authors show that it is upper bounded by the NTK-based complexity, but showing that this measure is "better" requires some case where the error of NTK can be lower-bounded, to show a gap between the two methods. In other words, can you show some example of data/function/distribution where your method yields an upper bound on the loss that is significantly smaller than the best possible bound obtained by the NTK? or even better, by any kernel method? If so, this should be emphasised in the paper.

Related to this point, I believe that the discussion on adaptivity of the kernel is a little misleading. The authors claim that the difference between their analysis and the NTK analysis is that in the later the kernel is fixed while in the former the kernel can be chosen depending on the data. However, the kernel cannot be chosen freely, but from some restricted family of kernels. But it is not clear exactly how this family of kernels is restricted, and whether this does not end up being equivalent to learning with some fixed kernel that is different from the NTK. So, please clarify whether this adaptivity truly goes beyond the regime of learning with kernels, or just improves by using a "better" (but fixed) kernel.

Another point that could be improved is the technical introduction of the complexity measure. I found it hard to follow all the different kernels that were introduced, and how these were derived from the architecture. I think that working with some concrete example of a function (possibly some function that is "beyond the NTK regime") and calculating its complexity, will make it easier to understand the exact notion of complexity.

Overall, I believe the paper makes worthy contributions, and I am willing to raise my score if my concerns are answered.

===========================================================

After discussion with the authors, I still believe the results could be much stronger if the authors would show a clear gap between NTK and their analysis. However, the paper does offer new bounds and novel analysis techniques which are important contributions, and therefore I am raising my score.

**Summary Of The Paper:**

The paper introduces novel results on the optimization and generalization of three-layer networks, optimized with a variant of gradient-descent. The goal of the analysis introduced in the paper is to obtain bounds that improve over the bounds obtained using the well-studied NTK framework. Namely, the aim is to go beyond the "lazy training" regime to allow for better bounds on learning more complex functions/distributions.

**Summary Of The Review:**

The paper provides novel results analyzing optimization and generalization of three-layer neural networks. However, my main concern is that the paper does not provide theoretical evidence that the bounds are indeed strictly and significantly stronger than what can be obtained by learning with the NTK, or more generally using kernels. I am willing to raise my score if my concerns are answered.

---

> ### Author Response · Authors · 2021-11-19
> **Clarification on the contributions and addressing the reviewer's points**
>
> However, we clarify a couple of things regarding the reviewer’s note about the contribution of our work. First, we clarify the main contributions of this paper:
>
> 1. We introduce a novel regime beyond NTK which we show remains computationally tractable for learning under a first-order method. Despite the nonconvexity of the loss, we show that the algorithm outputs a network that generalizes by an adaptive data-dependent complexity measure, which also improves the result for NTK in Arora et al. [1] (A concrete example is provided in Appendix A.1)
>
> 2. Despite the current non-asymptotic NTK analysis by Arora et al. [1] for gradient descent which is not noise robust, our bounds can handle noisy distributions as well.
>
> 3. Beyond these, we believe a major contribution of our work are the proof techniques for proving convergence as well as effectively bounding the Rademacher complexity.
>
> Second, we address the reviewer’s points specifically.
>
> 1. Regarding the first point of the reviewer, the fast rate $1/n$ is the result of employing a more refined Rademacher complexity bound, hence not the main contribution of our work, as we also mention in the paper.
>
>
> 2. Regarding the second point, to the best of our knowledge, the current known non-asymptotic analysis of NTK by Arora et al. [1] analyzes gradient descent in the regime where it fits the data perfectly; this makes their NTK analysis crucially dependent on the noiseless assumption. This is because, if gradient descent fits the data perfectly with noise, then it is possible that the norm of the network weights grows proportional to $\sqrt n$, which makes the generalization bound trivial. In particular, the noiseless assumption underlies the bound on the network, and with noise, the norm of the gradients can blow up.
> It is possible that our proof strategy can be adopted for the pure NTK case, however, we are not aware of any work doing this, and hence, this remains a contribution of this paper. We would be happy to look at any reference that we may have missed.
>
> 3. Regarding the reviewer’s third point, in Appendix A.1, we provide explicit bounds for the general class of polynomial functions which improve upon the related bounds for the NTK regime stated in Arora et al. [1]. However, as we have stated in the paper, we believe this improvement over NTK is still not fully exploiting the flexibility of our complexity measure, as it only plugs in low-rank matrices $G$. Investigating the possibility of stronger improvements over the NTK bound and deriving other explicit analytic upper bounds on our complexity (whether the data-dependent or the functional version) in this regard, by exploiting perhaps full rank $G$’s, is left as an interesting question for future work. We believe that stronger explicit gaps may be possible in other, e.g., deeper regimes, with similar techniques. That said, even the adaptive form of the bounds may be viewed as a general conceptual contribution in understanding learning in neural networks, and inspire follow-up works using such adaptive analyses.
>
>
> 4. Besides the mentioned points, we believe that a major contribution of our work is the proof techniques that we use to obtain such adaptive bounds with a polynomially efficient algorithm. We hope that such techniques can be generalized to deeper networks, with possibly different regimes than those we introduced in this work, to achieve stronger adaptive bounds and algorithmic depth separation.
> Clarity of an example for the complexity measure and kernels:
> We have provided an explicit example for an explicit function class in Appendix A.1.
> We will make these descriptions clearer.
>
> [1] Sanjeev Arora, Simon Du, Wei Hu, Zhiyuan Li, and Ruosong Wang. A fine-grained analysis of optimization and generalization for overparameterized two-layer neural networks. In International Conference on Machine Learning, pp. 322–332. PMLR, 2019

---

> > ### Comment · Reviewer_NvLr · 2021-11-21
> > **Response to clarification**
> >
> > Thank you for the clarification, I believe I now understand your contribution better.
> >
> > I tried to read Appendix A.1 more carefully, but I'm still not completely sure what exactly is your result there.
> > You calculate the induced norm for the function s, and the bound on the norm for this function implies a generailzation bound via Theorem 1. Do you claim that this upper bound on the loss is better than the upper-bound that can be derived using Arora et. al, 2019? Do you state somewhere what exactly the later bound is? How big is the gap?
> >
> > Also, just to make sure I understand - you show an upper-bound on the loss that is lower than the upper-bound shown in Arora et. al, 2019 (in some case), is that correct? However, you **do not** show a lower-bound on the loss that can be achieved using the NTK for the function introduced in Appendix A.1, correct? Do you believe showing such bound is possible?
> >
> > Minor comment: there are some typos in Appendix A.1, for example the last equation in page 15 seems incomplete, and also the reference right after Equation 31 is missing.

---

> > > ### Author Response · Authors · 2021-11-22
> > > **Further clarification**
> > >
> > > We thank the reviewer for the additional comments. The Equation at the bottom of page 15 is actually in the middle of the sentence, we omitted the unnecessary $=$ from it to remove the ambiguity.
> > >
> > > Regarding the example in Appendix A.1: Yes, taking into account the behavior of the tailor series sequence $(\gamma_p)$ in Appendix A.1 is like $\gamma_p = \Theta(p\sqrt{p})$, our bound improves upon that of Arora et al. [1] in Corollary 6.2. The main improvement is that our bound supports all monomial powers $p$ while the bound in Arora et al. [1] does not work for odd powers larger than one. Note that regarding this example, we only plug in a low-rank matrix $G$ in our complexity measure, which means the power of this complexity measure can possibly be beyond the bound provided in this example.
> > >
> > > In other words, our explicit upper bounds for polynomial functions improve upon that of Arora et al. [1], which is stated in their Corollary 6.2. Regarding your second question, our improvement is indeed only regarding these upper bounds. For the NTK setting considered by Arora et al. [1], we are unaware of specific lower bounds that might prevent one from achieving improved generalization.

---

> > > > ### Comment · Area_Chair_22vT · 2021-11-22
> > > > **Question regarding lower bound**
> > > >
> > > > Let me try and further clarify the question from the reviewer: the question is whether you can show a case (i.e., a distribution) where "standard" NTK training has provably worse sample complexity than your approach (e.g., as in https://arxiv.org/abs/1810.05369).

---

> > > > > ### Author Response · Authors · 2021-11-24
> > > > > **Further Clarification**
> > > > >
> > > > > Currently, we do not have such an explicit example, albeit we think it may well exist. Nevertheless, our bounds strictly improve over the NTK bounds in Arora et al by being more general, e.g., our bounds hold for all monomials, and noisy distributions. Again, we emphasize that we are only plugging in low rank $G$’s in our polynomial functions example, to get a benefit over the NTK result of Arora et al. [1], while $G$ is allowed to be any matrix in our complexity measure.
> > > > >
> > > > > Directly comparing the lower bound in the paper you mention and our generalization bounds is nontrivial, since the lower and upper bounds there depend on dimension. Constructing an explicit special distribution for the lower bound is an interesting question for future work.
> > > > >
> > > > > That said, our results, as they are, provide many contributions. First, the adaptive bounds have conceptual value too, and we do show some relation to NTK. They provide a better understanding of adaptivity, and may also serve as a basis to extend adaptivity to larger classes of kernels, and we hope that this line of work will eventually help results on algorithmic depth separation for deeper networks. Second, our adaptivity result is actually algorithmically realizable (which is not the case in some previous works). Third, our proof techniques may be of more general interest, as we argued above.

---

### Decision · Program_Chairs · 2022-01-20

**Decision:**

Accept (Poster)

**Comment:**

This paper goes beyond the NTK setting in analyzing optimization and generalization in ReLU networks. It nicely generalizes NTK by showing that generalization depends on a family of kernels rather than the single NTK. The reviewers appreciated the results. One thing that is missing is a clear separation between NTK results and the ones proposed here. Although it is ok to defer this to future work, a discussion of this point in the paper would be helpful.